# Aerosol Optical Depth comparison between GAW-PFR and AERONET-Cimel radiometers from long-term (2005-2015) 1-minute synchronous measurements

Emilio Cuevas[1], Pedro Miguel Romero-Campos[1], Natalia Kouremeti[2], Stelios Kazadzis[2], Petri Räisänen[3], Rosa Delia García[4,1], Africa Barreto[5,1,4], Carmen Guirado-Fuentes[4,1], Ramón Ramos[1], Carlos Toledano[4], Fernando Almansa[5,1,4], and Julian Gröbner[2]

[1]Izaña Atmospheric Research Center (IARC), State Meteorological Agency (AEMET), Spain
[2]Physikalisch-Meteorologisches Observatorium Davos, World Radiation Center (PMOD/WRC), Davos, Switzerland
[3]Finnish Meteorological Institute, Helsinki, Finland
[4]Atmospheric Optics Group, Valladolid University, Valladolid, Spain
[5]Cimel Electronique, Paris, France

**Correspondence:** Emilio Cuevas
(ecuevasa@aemet.es)

**Abstract.** A comprehensive comparison of more than 70000 synchronous 1-minute aerosol optical depth (AOD) data from three Global Atmosphere Watch-Precision Filter Radiometers (GAW-PFR), traceable to the World AOD reference, and 15 Aerosol Robotic Network-Cimel (AERONET-Cimel) radiometers, calibrated individually with the Langley plot technique, was performed for four common or near wavelengths 380 nm, 440 nm, 500 nm and 870 nm in the period 2005-2015. The goal of this study is to assess whether, despite the marked technical differences between both networks (AERONET, GAW-PFR) and the number of instruments used, their long-term AOD data are comparable and consistent. The percentage of data meeting the WMO traceability requirements (95 % of the AOD differences of an instrument compared to the WMO standards lie within specific limits) is > 92 % at 380 nm, > 95 % at 440 nm and 500 nm, and 98 % at 870 nm, with the results being quite similar for both AERONET V2 and V3 versions. For the data outside these limits the contribution of calibration and differences in the calculation of the optical depth contribution due to Rayleigh scattering, and O3 and NO2 absorption have a negligible impact. For AOD > 0.1, a small but non-negligible percentage ($\sim$ 1.9 %) of the AOD data outside the WMO limits at 380 nm can be partly assigned to the impact of dust aerosol forward scattering on the AOD calculation due to the different field of view of the instruments. Due to this effect the GAW-PFR provides AOD values which are $\sim$3 % lower at 380 nm, and $\sim$2 % lower at 500 nm, compared with AERONET-Cimel. The comparison of the Angström exponent shows that under non-pristine conditions (AOD > 0.03 and AE < 1) the AE differences remain < 0.1. This long-term comparison shows an excellent traceability of AERONET-Cimel AOD with the World AOD reference at 440 nm, 500 nm and 870 nm channels and a fairly good agreement at 380 nm, although AOD should be improved in the UV range.

# 1 Introduction

In recent decades there has been a growing interest in the role played by atmospheric aerosols in the radiation budget and the Earth's hydrological cycle, mainly through their physical and optical properties (IPCC, 2013). The most comprehensive and important parameter that accounts for the optical activity of aerosols in the atmospheric column is the aerosol optical depth (AOD) (WMO, 2003, 2005). This is also a key parameter used in atmospheric column aerosol modelling (e.g. Basart et al. (2012); Benedetti et al. (2018); Cuevas et al. (2015); Huneeus et al. (2016)) and in satellite observations (e.g. Sayer et al. (2012, 2013); Kahn and Gaitley (2015); Amiridis et al. (2015)). The second aerosol optical parameter in importance is the Angström exponent (AE) (Angstrom, 1929) that accounts for the spectral dependency of the AOD. Since the AE is inversely related to the average size of the aerosol particles, it is a qualitative indicator of the atmospheric aerosol particle size and therefore a useful parameter to assess the aerosol type (WMO, 2003). At present, two global ground-based radiometer networks provide aerosol optical properties of the atmospheric column using centralized data processing procedures based on their respective standard criteria and also centralized protocols for calibration and quality control, linking all network instruments. These are GAW-PFR (Global Atmosphere Watch - Precision Filter Radiometer; http://www.pmodwrc.ch/worcc/; last access: 05 September 2018) and AERONET-Cimel (AErosol RObotic NETwork - Cimel Electronique radiometer; https://aeronet.gsfc.nasa.gov; last access: 01 September 2018) networks. AERONET is, in fact, a federation of ground-based remote sensing aerosol networks established by NASA (National Aeronautics and Space Administration) and PHOTONS (PHOtométrie pour le Traitement Opérationnel de Normalisation Satellitaire; University of Lille- Service d'Observation de l'INSU, France; Goloub et al. (2007)), being complemented by other sub-networks, such as, AEROCAN (Canadian sunphotometry network; Bokoye et al. (2001)), AeroSibnet (Siberian system for Aerosol monitoring; Sakerin et al. (2005)), AeroSpan (Aerosol characterisation via Sun photometry: Australia Network; Mitchell et al. (2017)), CARSNET (China Aerosol Remote Sensing NETwork; Che et al. (2015)), and RIMA (The Iberian network for aerosol measurements; Toledano et al. (2011)). There are other radiometer networks that in recent years have incorporated centralized protocols for data evaluation and databases, and performed regular intercomparisons with GAW-PFR and AERONET-Cimel. These include, for example, SKYNET (SKYradiometer NETwork), and its seven associated sub-networks, that uses the Prede-POM sky radiometer to investigate aerosol-cloud-solar radiation interactions (e.g. Campanelli et al. (2004); Nakajima et al. (2007); Takamura and Nakajima (2004)).

The World Optical Depth Research Calibration Center (WORCC) was established in 1996 at the Physikalisch Meteorologisches Observatorium Davos / World Radiation Center (PMOD / WRC). The GAW-PFR network (Wehrli, 2005) was initiated within PMOD/WRC for global and long-term atmospheric aerosol monitoring and accurate detection of trends. Aerosol data series measured at 12 core sites away from local and regional pollution sources, representative of atmospheric background conditions in different climates and environments of the planet, in addition to another 20 associated stations are included in this global network (Kazadzis et al., 2018a). For this reason, GAW-PFR uses the PFR, an accurate and reliable instrument regarding its absolute response stability over time that was designed for long-term AOD measurements (Wehrli, 2008a). The GAW-PFR was specifically designed by WORCC for this goal following the technical specifications defined by WMO (2003, 2016). In 2006, the Commission for Instruments and Methods of Observation (CIMO) of WMO (WMO, 2007) recommended

that the WORCC at the PMOD/WRC should be designated as the primary WMO Reference Centre for AOD measurements (WMO, 2005).

The AERONET-Cimel network (Holben et al., 1998) was, in principal, designed to validate satellite products and to characterize the spatial-temporal distribution of atmospheric aerosols based on their optical properties. It is the largest surface-based global aerosol network with more than 84 sites with measurement series longer than 10 years and more than 242 sites having data sets > 5 years. Cimel radiometer data, part of AERONET, are processed centrally and freely delivered in near real time by the NASA Goddard Space Flight Center. Both networks, although designed to meet different objectives, are now global bench-

marks for the study and characterization of aerosol optical properties worldwide, and for the evaluation of aerosol observations on board satellites and simulations with models. Multiple studies have proliferated in recent years to obtain aerosol climatology and to determine AOD trends in different parts of the world (e.g. Nyeki et al. (2012); Klingmüller et al. (2016); Chedin et al. (2018)). However, these networks use radiometers with significant technical differences. Moreover, calibration methodologies, AOD calculation algorithms and data evaluation methods are also relatively different between the two networks. Consequently,

the objective of this study is to assess whether, despite the marked differences between both networks, including the different day-to-day maintenance and operation procedures of the respective instruments during the study period, the long-term AOD provided by the two networks is comparable and consistent.

The WMO has defined the GAW-PFR Triad (three Master PFR instruments) as the world-wide reference for AOD measurements (WMO, 2005). Based on this concept, an instrument provides traceable measurements of AOD to this WMO reference,

when this instrument can demonstrate an unbroken chain of calibrations between itself and the GAW-PFR Triad with AOD measurements within specified limits of the GAW-PFR reference. This can either be achieved by a direct comparison to the GAW-PFR Triad (Kazadzis et al., 2018a), or by using a portable transfer standard radiometer as presented in this study. Several comparisons between AERONET-Cimel, GAW-PFR and other radiometers have been carried out in different places (Barreto et al., 2016; Kazadzis et al., 2014, 2018b; Kim et al., 2008; McArthur et al., 2003; Mitchell and Forgan, 2003; Nyeki et al.,

2015; Schmid et al., 1999; Toledano et al., 2012). However, these comparisons have been performed during field intercomparison campaigns or during relatively short periods of time, so they are not representative of a large variety of atmospheric conditions. In addition, the type of instrument maintenance and the number and qualifications of staff serving them during campaigns is generally of a higher quality compared to that of the instrument daily operation in unattended mode. This might cause an improvement of the instrument performance during intensive campaigns compared to the operational mode.

The growing interest in the analysis of long-term AOD and AE data series for climatological purposes requires an assessment of their quality assurance and long-term intercomparability. This is the first study to analyse the long-term traceability of AERONET-Cimel with respect to GAW-PFR, and therefore to assess the validity of the long AOD and AE AERONET-Cimel data series for climatological and climate change studies under specific quality control requirements.

GAW-PFR has a comprehensive calibration system (Kazadzis et al., 2018a; Schmid and Wehrli, 1995) that is transferred by

a worldwide suite of reference instruments. AERONET-Cimel does not have a CIMO-WMO linked reference and, as described by Holben et al. (1998); Eck et al. (1999); Toledano et al. (2018), is based on maintaining Reference AERONET radiometers based on the Langley calibration technique at Izaña, Spain and Mauna Loa, USA. Calibration of all other instruments are based

on raw voltage ratios comparisons with Reference instruments at dedicated sites (Carpentras-France, Washington DC-USA, Valladolid-Spain). There are few places in the world where synchronous observations of these two networks are available for long time periods and variable AOD conditions. The Izaña Observatory (IZO; Tenerife, Canary Islands) is one of them. The GAW-PFR measurements started at Izaña Observatory in 2001 (Wehrli, 2005) while AERONET-Cimel started in 2003 (Goloub et al., 2007). Since 2005, synchronous measurements (1-minute values) that have been evaluated following the calibration procedures of each of the networks, are available.

In addition, the Izaña Observatory is one of the two places in the world (the other is Mauna Loa - Hawaii, USA) where sun-calibrations are performed using the Langley plot technique for both AERONET-Cimel and GAW-PFR reference instruments (Toledano et al., 2018) because of stable and very low AOD conditions during many days per year. Consequently, the instruments compared at the Izaña Observatory have been calibrated under the same environmental conditions, and therefore AOD differences can be directly linked with calibration principles, AOD post-processing and other instrumental differences. In this work, we analyse and evaluate the comparison of 11 years (2005-2015) of 1-minute synchronous observations of AOD with AERONET-Cimel and GAW-PFR in four common or near wavelengths, assessing the results and explaining the possible causes of these differences. Some preliminary technical details on the traceability between GAW-PFR and AERONET-Cimel were reported in a technical report by Romero-Campos et al. (2017).

In Section 2 the facility in which this long-term comparison has been carried out is described. The technical characteristics of the AERONET-Cimel and GAW-PFR instruments are shown in Section 3, with special emphasis on the technical and methodological differences of both networks. Section 4 describes the methodology followed in this intercomparison based on the concept of WMO-GAW traceability. Results are given in Section 5. A summary and conclusions are provided in Section 6. The Supplement contains case analyses of inaccurate calibration and cloud contamination, some additional results of the comparison between PFR and Cimel with AERONET V3, complementary information of the very short natural AOD variability, and the simulations performed with the Monte Carlo model to evaluate the impact of dust forward scattering radiation on AOD determination.

## 2 Site Description

Izaña Observatory (28.3° N, 16.5° W; 2373 m a.s.l.) is located in Tenerife (Canary Islands, Spain) and is managed by the Izaña Atmospheric Research Center (IARC), which is part of the State Meteorological Agency of Spain (AEMET). It is a suitable place for long-term studies of aerosol optical properties under contrasting atmospheric and meteorological conditions. This is because IZO is located in the free troposphere (FT) above the temperature inversion caused by the trade wind regime in lower levels and general subsidence associated with the branch of the decay of Hadley's cell aloft (Carrillo et al., 2016). This meteorological feature favours, during most of the year, the presence of pristine skies and clean air representative of atmospheric background conditions (Cuevas et al., 2013; Rodríguez et al., 2009). On the other hand, its proximity to the African continent makes it a privileged site for observing and characterizing the Saharan Air Layer (SAL) that normally presents a high burden of desert mineral dust, especially during the summer months (Basart et al., 2009; Cuevas et al., 2015; Rodríguez et al.,

2011). At this time of the year, the SAL impacts the subtropical free troposphere over the North Atlantic with large interannual (Rodríguez et al., 2015) and sharp intraseasonal (Cuevas et al., 2017a) variability. The contrasting atmospheric conditions that occur at IZO allow the comparison of the two networks, which can be performed under a wide range of AOD values; mostly for pristine conditions (AOD $\leq$ 0.03) but also for relatively high turbidity (AOD > 0.6) linked with dust aerosol related intrusions. In addition, the location offers the possibility of observing rapid changes in AOD, going from pristine conditions to dusty skies, and vice versa, in a matter of a few hours, especially in the summer period. The periodical presence of a dust laden SAL allows us to evaluate the impact that the dust forward scattering into the field of view has on AOD retrieval. All this defines IZO as an excellent atmospheric aerosol natural laboratory to compare the performance of different radiometers measuring AOD. One of the first international AOD intercomparison campaigns was carried out at IZO in April 1984 (WMO, 1986) promoted and coordinated by PMOD / WRC.

The privileged conditions of pristine skies that characterize IZO during many days a year have allowed this observatory to become a calibration site for the GAW-PFR and AERONET-Cimel networks since 2001 and 2003, respectively, where the extraterrestrial constants are determined with direct sun observations using the Langley plot technique (Toledano et al., 2018). Note that the extraterrestrial constant (calibration constant) is the signal the instrument would read outside the atmosphere at a normalized earth-sun distance. In addition, since July 2014, IZO has also been designated by the WMO as a CIMO (WMO, 2014) testbed for aerosols and water vapour remote sensing instruments. IZO is a station of the Baseline Surface Radiation Network (BSRN) (Driemel et al., 2018; García et al., 2019). Details of IZO facilities, measurement programmes and main research activities can be found in Cuevas et al. (2017b).

## 3 GAW-PFR and AERONET-Cimel radiometers

The two types of radiometers intercompared in this study are Cimel CE318-N (Holben et al., 1998), hereinafter referred to as Cimel, the standard instrument of AERONET until the recent appearance of CE318-T (Barreto et al., 2016), and the PFR (Wehrli, 2005) standard instrument of the GAW-PFR network. The main features of these two radiometers are described in Table 1. The Cimel (Holben et al., 1994, 1998) is a radiometer equipped with a 2-axis robot that performs two types of basic radiation measurements: direct solar irradiance and sky (radiance) observations, thanks to an automatic pointing robot that executes the observation sequences that have been scheduled. The robot performs automatic pointing to the sun by stepping azimuth and zenith motors using ephemeris based on time, latitude and longitude. Additionally, a four-quadrant detector is used to improve the sun tracking before each scheduled measurement sequence. This sensor guides the robot to the point where the intensity of the signal channel is maximum. Diffuse-sky measurements are also performed by Cimel to infer aerosol optical and microphysical properties. Two different routines are executed: almucantar (varying the azimuth angle keeping constant the zenith angle) and principal plane (varying the zenith angle keeping constant the azimuth angle). The ability of Cimel to perform both direct and diffuse-sky measurements makes it necessary to use a specific robot rather than a simple sun tracker. The field of view angle (FOV) of the instrument is 1.29° ($\sim$1.3° from now on) (Torres et al., 2013). The wavelengths in which the measurements are sequentially made by a single detector depend on the interference filters that each version of the radiometer

has installed in the filter wheel, which is located inside the sensor head and which is moved by a stepper motor. The Cimel versions used in this study have at least eight interference filters centred at 340 nm, 380 nm, 440 nm, 500 nm, 675 nm, 870 nm, 940 nm, and 1020 nm and 10 nm full width at half maximum (FWHM) bandwidth, except for 340 nm and 380 nm which have 2 nm and 4 nm FWHM, respectively. Solar irradiance is measured with a Silicon detector in these channels. The possible deterioration of the interference filters is reduced since they are only sun-exposed during three consecutive 1-second direct-sun measurements per channel, this cycle being scheduled every ∼15 minutes. The rest of the time the Cimel is taking sky radiance measurements, or at rest position, looking downwards.

The PFR (Wehrli, 2000, 2005, 2008a, b) is designed for continuous and automated operation under a broad range of weather conditions. It accurately measures direct solar radiation transmitted in four independent narrow wavelength channels centred at 368 nm, 412 nm, 500 nm and 862 nm, with 5 nm FWHM bandwidth. The FOV of the instrument is 2.5° and the slope angle is 0.7°. Dielectric interference filters manufactured by the ion-assisted-deposition technique are used to assure significantly larger stability in comparison to the one manufactured by classic soft-coatings. The PFR was designed for long-term stable measurements, therefore the instrument is hermetically sealed with an internal atmosphere slightly pressurized (2000 hPa) with dry nitrogen, and is stabilized in temperature with a Peltier-type thermostatic system maintaining the temperature of the detector head at 20° C $\pm$ 0.5° C. This system makes corrections of the sensitivity for temperature unnecessary, and also prevents accelerated ageing of filters, ensuring the high stability of the PFR. The PFR is mounted on a sun tracker, pointing always at the sun without any active optimization of the sun-pointing. The detectors are only exposed for short time periods, since an automated shutter opens every minute for 10s for sun measurements, minimizing degradation related to the filters exposure.

The expected uncertainty of AOD in the four channels of the PFR radiometer is from 0.004 (862 nm) up to 0.01 (368 nm) (Wehrli, 2000). For the Cimel radiometer, the expected uncertainty of level 2-AOD product is between 0.002 and 0.009, for reference instruments, larger for shorter wavelengths, and between 0.01 and 0.02 for field instruments, larger in the UV, under conditions of clear skies (Eck et al., 1999; Barreto et al., 2016). It should be taken into account that, in general, in the ultraviolet range the AOD uncertainty is higher (Carlund et al., 2017).

In relation to the calibration of both networks, GAW and AERONET, they use measurements at high mountain stations with very stable and low AOD over a day in which consecutive measurements can be performed over a wide range of optical air mass (approximately between 2 and 5) in the shortest possible time, in order to calibrate Reference instruments using the Langley plot technique. In case of AERONET-Cimel these calibrations are subsequently transferred to the field instruments of the network in other sites through regular intercomparison campaigns. In case of the GAW-PFR, the calibration system is more complex in order to ensure traceability with the WORCC world reference. The maintenance of the AOD standard by the WORCC Calibration Central laboratory is described in Kazadzis et al. (2018b). It consists of a triad of instruments that measure continuously, and three additional portable transfer standard radiometers located at Mauna Loa (one instrument) and Izaña (two instruments) observatories. Every six months, one of the portable transfer standard radiometers visits the reference triad based at PMOD/WRC (Davos) and compares the calibration constants defined by the 6-month Langley calibrations in the two high mountain stations (Table 1 of Kazadzis et al. (2018b)) with the one defined by the triad. The comparison is based on the signals

(voltages) and not on AOD values. The differences between the Izaña GAW-PFR radiometers and the reference triad have been always lower than 0.5 %, being within the uncertainty of the Langley method plus the small possible instrument degradations that can be detected in a 6-12 month period. Such degradations are quite small and are accounted for in the calibration analysis since extraterrestrial constants are linearly interpolated between two triad visits or every 6-month periods. Additionally, the Izaña GAW-PFR "field" radiometers are calibrated on a routine basis using the Langley-plot technique for double checking quality assurance. Therefore, these radiometers cannot be considered as simple "field" instruments, but as regularly calibrated radiometers with assured traceability with the WORCC triad reference.

IZO is one of the two sites of Langley-plot calibration of both networks, which represents an advantage when comparing the two instruments, eliminating, to a large extent, errors caused by the calibration transfer. However, there are differences between the calibration methodologies used by both networks. AERONET obtains the calibration by means of the average of a few extraterrestrial constants ($V_0$), obtained from Langleys, performed in a relatively short time (the time needed to collect data from at least 10 morning Langley plots). However, PFR related Langleys are calculated by temporal linear fit to a larger number of extra-terrestrial constants $V_0$ obtained from Langley plots performed during 6 months (Wehrli, 2000; Kazadzis et al., 2018a). Details on requirements for performing Langley calibrations of reference instruments by GAW-PFR and AERONET, and their uncertainties, are analysed in detail by Toledano et al. (2018).

## 4 Data and methodology used in this study

The AOD at each wavelength is obtained from the Beer-Bouguer-Lambert law (Thomason et al., 1982; WMO, 2003) for radiometers collecting spectral direct sun measurements.

$$I(\lambda) = I_0(\lambda) \exp(-\tau m) \tag{1}$$

where $I(\lambda)$ is the direct sun signal at ground level at wavelength $\lambda$, $I_0(\lambda)$ is the extraterrestrial signal of the instrument corrected by the Earth-Sun distance, and $m$ is the optical air mass in the measurement path (Kasten and Young, 1989). A detailed description of how AOD is obtained and the determination of extraterrestrial constants by GAW-PFR and AERONET are provided by Holben et al. (1994, 1998, 2001); Toledano et al. (2018); Wehrli (2000, 2008b).

### 4.1 GAW-PFR and AERONET-Cimel data

GAW-PFR provides AOD values every 1 minute as an average of 10 sequential measurements of total duration less than 1 second (20 ms for each channel), then dark current is measured, going to the sleep mode until the next minute. AERONET-Cimel takes a sequence of three separate measurements (1-second per filter) in one minute interval (each one every 30 seconds). This sequence of measurements is called "triplet" and it is performed every $\sim 15$ minutes for air masses lower than 2, and with higher frequency for lower solar elevations. Therefore, AERONET-Cimel provides AOD values for each triplet, at least, every $\sim 15$ minutes. Note that AERONET-Cimel performs AOD measurements interspersed with sky radiance measurements, whose duration varies throughout the day, and therefore the AOD measurements are not necessarily provided at full minutes. We

consider the 1-minute data as synchronous when GAW-PFR and AERONET-Cimel AOD data were obtained with a difference of $\sim 30$ s.

GAW-PFR and AERONET-Cimel instruments use the same time reference. The synchronization between PC and GAW-PFR data-logger was performed every 12 hours since 2005, and improved to 6 hours after 2013 using NTP servers via Internet. From 2005 to 2012 the time of the AERONET-Cimel reference instruments was checked manually once per day using a handheld GPS. From 2012 onwards, the time was adjusted automatically three times per day using the ASTWIN Cimel software. In turn, the PC time is adjusted through the AEMET internal time server every 15 minutes. The AOD comparison has been performed using 1-minute synchronous data from the four closest channels of both instruments in the period 2005-2015 (more than 70000 data-pairs in each channel). Thus, in the case of GAW-PFR, the four available channels of 368 nm, 412 nm, 500 nm and 862 nm were analysed, while in the case of AERONET-Cimel, only the 380 nm, 440 nm, 500 nm and 870 nm channels were considered (Table 1). For the 500 nm channel, the nominal wavelengths of the two networks differ by a maximum of 1.8 nm. However, the nominal wavelengths in the rest of the compared channels present higher differences. Therefore, the AOD values of the original GAW-PFR 368 nm, 412 nm and 862 nm channels have been interpolated or extrapolated to the corresponding AERONET-Cimel channels (380 nm, 440 nm and 870 nm) using the Angström power law, and the GAW-PFR AE calculated from the four PFR AOD measurements.

Synchronous AE data provided by both instruments have also been compared (see Sect. 5.5). GAW-PFR determines AE using all four PFR wavelengths (Nyeki et al., 2015), while AERONET-Cimel uses different wavelength ranges (340-440 nm, 380-500 nm, 440-675 nm, 440-870 nm, 500-870 nm) (Eck et al., 1999). As a consequence, we have calculated a new AE for the Cimel radiometer using the four channels equivalent to those of the PFR.

In this study we have used the two versions of the AERONET database. Version 2 (V2) has been used so far in many scientific publications in high impact journals, and Version 3 (V3) has been released just recently (Giles et al., 2019). In Sect. 5.1, a comparison of V2 and V3 is presented. A total of three GAW-PFR and 15 AERONET-Cimel instruments have participated in this intercomparison study covering the period 2005-2015. Their corresponding reference numbers are shown in Table 2.

## 4.2 Cloud filtering

The data matching in our comparison analysis was performed with synchronous 1-minute AOD values of both networks labelled with quality control (QC) flags that guarantee proven quality data not affected by the presence of clouds. In the case of the AERONET-Cimel network, the selected AOD data are Level 2 data from both V2 and V3 AERONET databases, which have been cloud filtered by the Smirnov algorithm (Smirnov et al., 2000), based on the triplet method, with a second-order temporal derivative constraint (McArthur et al., 2003), and visually screened in V2. The cloud screening in AERONET V3 has been completely automated, and notably improved, especially by refining the triplet variability and cirrus cloud detection and removal (Giles et al., 2019). These two cloud screening methods are able to detect rapid changes in the atmosphere and remove those measurements in which AOD variability within the triplet is higher than the following criteria:

**Table 1.** Main features of the GAW-PFR (PFR; Wehrli (2000, 2005, 2008a, b)) and AERONET-Cimel (Holben et al., 1994, 1998; Torres et al., 2013) radiometers used in this study.

| | GAW-PFR | AERONET-Cimel |
|---|---|---|
| Type of instrument | Standard version | Standard version<br>Reference instrument |
| Type of observation | Automatic continuous direct sun irradiance | Automatic sun-sky tracking |
| Available standar channels | 368nm, 412nm, 500nm, 862nm | 340nm, 380nm, 440nm, 500nm, 675nm, 870nm, 1020nm, 1640nm |
| FWHM | 5 nm | 2 nm(340 nm), 4 nm(380 nm), 10 nm(VIS-NIR), 25 nm(1640 nm) |
| AOD uncertainty | ± 0.01 | 0.002-0.009 spectrally dependent<br>with the higher errors in the UV<br>(Reference instruments)(Eck et al., 1999) |
| FOV (FWHM) | 2.5°<br>(1.2° plateau, 0.7° slope) | 1.3° (slope angle unknown) |
| Sun Tracker | Any sun tracker with a<br>resolution of at least 0.08° | Robot specifically designed by CIMEL<br>and controlled in conjunction with the radiometer |
| Temperature control/correction | Temperature controlled 20°C±0.5°C | Temperature correction to 1020 nm is applied in V2. Corrections<br>from filter-specific temperature characterization<br>in V3 for VIS and NIR spectral bands (Giles et al., 2019) |
| Power | Grid | Solar panels/grid |
| Data transmission | Local PC/FTP | Local PC/FTP<br>Satellite transmission |
| Calibration | Comparison with reference triad.<br>Additional in situ long-term Langleys | At least 10 good morning Langleys plots |

**Table 2.** GAW-PFR and AERONET-Cimel instrument numbers used in this study in the period 2005-2015. Data from Reference Cimel #398 was not upgraded to Level 2 in V3 during the period 12 July 2008 - 15 September 2008.

| Instruments used in this study | Period 2005-2009 | Period 2010-2015 |
|---|---|---|
| GAW-PFR | 2 instruments: #6,#25 | 2 instruments: #6,#21 |
| AERONET-Cimel | 13 instruments: #25,#44,#45,#79,#117,#140<br>#244,#245,#380,#382,#383,#398,#421 | 5 instruments: #244, #347, #380<br>#421, #548 |

– AOD triplet variability > MAX { 0.02 or 0.03∗AOD } at all wavelengths (V2)

– AOD triplet variability > MAX { 0.01 or 0.015∗AOD } at 675 nm, 870 nm, and 1020 nm simultaneously (V3)

The selection of these thresholds ensures the triplet average does not exceed 0.02 (V2) or 0.01 (V3) within 1 min in case of low AOD conditions.

GAW-PFR cloud screening algorithms also use the Smirnov triplet measurement, and the second-order derivative check, but add a test for optically thick clouds with $AOD_{500nm} > 2$ (Kazadzis et al., 2018b). In the case of the GAW-PFR network (Wehrli, 2008a) the flags take the value 0 for cloudless conditions, no wavelength crossings, and sun pointing within certain limits. More details in Kazadzis et al. (2018a) for all those selected records.

### 4.3 WMO traceability criteria

The criterion for traceability used in this study follows the recommendation of the WMO (WMO, 2005) which states that 95 % of the AOD measurements fall within the specified acceptance limits, taking the PFR as a reference:

$$U_{95} = \pm(0.005 + 0.010/m) \tag{2}$$

where $m$ is the optical air mass. Note that the $U95$ range is larger for smaller optical air mass. The acceptance limits proposed by WMO take into account, on the one hand, the uncertainty inherent in the calculations of the AOD, and on the other hand, the uncertainty associated with the calibration of the instrument. The latter, for the case of instruments with finite field of view direct transmissions, such as the PFR and the Cimel, is dominated by the influence of the top-of-the-atmosphere signal determined by Langley plot measurements, divided by the optical air mass. The first term of Eq. 2 (0.005) represents the maximum tolerance for the uncertainty due to the atmospheric parameters used for the AOD calculation (additional atmospheric trace gas corrections, and Rayleigh scattering). The second term describes the calibration related relative uncertainties. The WMO recommends an upper limit for the calibration uncertainty of 1 %.

### 4.4 Modelling the impact of near-forward scattering on the AOD measured by the PFR and Cimel radiometers

In order to study the impact of near-forward scattering on the irradiance measured by the PFR and Cimel instruments, a forward Monte Carlo model (Barker, 1992, 1996; Räisänen et al., 2003) was employed. For the present work, the model was updated to account for the finite width of the solar disk (Räisänen and Lindfors, 2019). The starting point of each photon was selected randomly within the solar disk, assuming a disk half-width of $0.267°$ and the impact of limb darkening on the intensity distribution was included following Böhm-Vitense (1989). Some diagnostics were also added to keep track of the distribution of downwelling photons at the surface with respect to the angular distance from the centre of the sun. Gaseous absorption was accounted for following Freidenreich and Ramaswamy (1999), while the Rayleigh scattering optical depth was computed using Bodhaine et al. (1999).

## 5 Results

### 5.1 Comparison of long-term AERONET V2 and V3 datasets at Izaña site

Since V3 has been released recently (Giles et al., 2019), we present a comparison between V2 and V3 for the Cimel channels 380 nm, 440 nm, 500 nm and 870 nm for the period 2005-2015. The results indicate that for the Izaña site the agreement and

consistency between the two AERONET versions is very high for the four channels ($R^2 > 0.999$) in full agreement with the results of the V2-V3 comparison reported by Giles et al. (2019). It follows that the results of the AOD comparison between GAW-PFR and the two versions of AERONET are very similar as shown throughout this work. A detailed description of AERONET V3 and its improvements with respect to V2 is given in Giles et al. (2019). As such improvements depend on aerosol type, according to the changes introduced in V3, for the high mountain site such as Izaña characterized by low background AOD values or, alternatively, by the presence of dust (no pollution or biomass burning aerosols), the AOD differences between V2 and V3 are expected to be minimum as is confirmed in this study (Figure 1).

However, it should be noted that AERONET V3 does not restrict the calculation of AOD to optical air masses less than 5.0 (Giles et al., 2019), as V2 does. This results in an increase in the number of solar measurements occurring in the early morning and the late evening. Consequently, the GAW-PFR comparisons with AERONET V3 consisted of ∼ 9000 more data pairs than the GAW-PFR comparison with V2 (see Supplement S.1.).

## 5.2 AERONET-Cimel AOD comparison with GAW-PFR data

The comparison with GAW-PFR AOD shows that the AOD from AERONET-Cimel radiometers meet the WMO traceability criteria ("traceable AOD data" from now on) at 440 nm, 500 nm and 870 nm channels. The lowest agreement is found in the UV channel (380 nm) with 92.7 % of the data, and the highest in the infrared channel (870 nm) with 98.0 % for V2 (Figure 2; Table 3) because this channel is less affected by trace gases absorption. Almost identical results are obtained for V3 (Supplement S1 and S2.1). However, in the first half of the comparison period (2005-2009) there were some mechanical problems in the solar tracker where the GAW-PFR was mounted on, which caused sporadic problems of sun pointing. This finding was confirmed with data from the four–quadrant silicon detector (Wehrli, 2008a) that showed diurnal variation of the PFR sensors position up to $0.3°$. From 2010 onwards, the PFR was mounted on an upgraded solar tracker of higher performance and precision. This reduced problems in sun pointing, that were the main cause of most of the AOD discrepancies between PFR and Cimel, and therefore not attributable to the instruments themselves. In addition, since 2010, Cimel #244 has been in continuous operation for most of the time at the Izaña Observatory, greatly simplifying calibration procedures and the corresponding data evaluation, and minimizing errors of calibration uncertainties introduced by the use of a high number of radiometers in the intercomparison. During the 2010-2015 period, the fraction of traceable AOD measurements of the total between the AERONET-Cimel radiometer and the GAW-PFR improves to 93.46 % in the 380 nm channel, and this percentage rises to 99.07 % for the 870 nm channel. Despite the technical differences between both radiometers, described above, and the different calibration protocols, cloud screening and data processing algorithms, the data series of both instruments, can be considered as equivalent, except for 380 nm, according to the WMO traceability criteria defined previously (Eq. 2). This explains the excellent agreement in the long-term AOD climatology shown for GAW-PFR and AERONET-Cimel in Toledano et al. (2018).

We have compared the percentages of AERONET-Cimel AOD V2 data meeting the WMO criteria for the four interpolated GAW-PFR channels with those of AERONET V3 (Table 3). A more detailed statistical evaluation for different scenarios of aerosol loading (three ranges of AOD) and aerosol size (three ranges of AE) for each compared channel has been performed

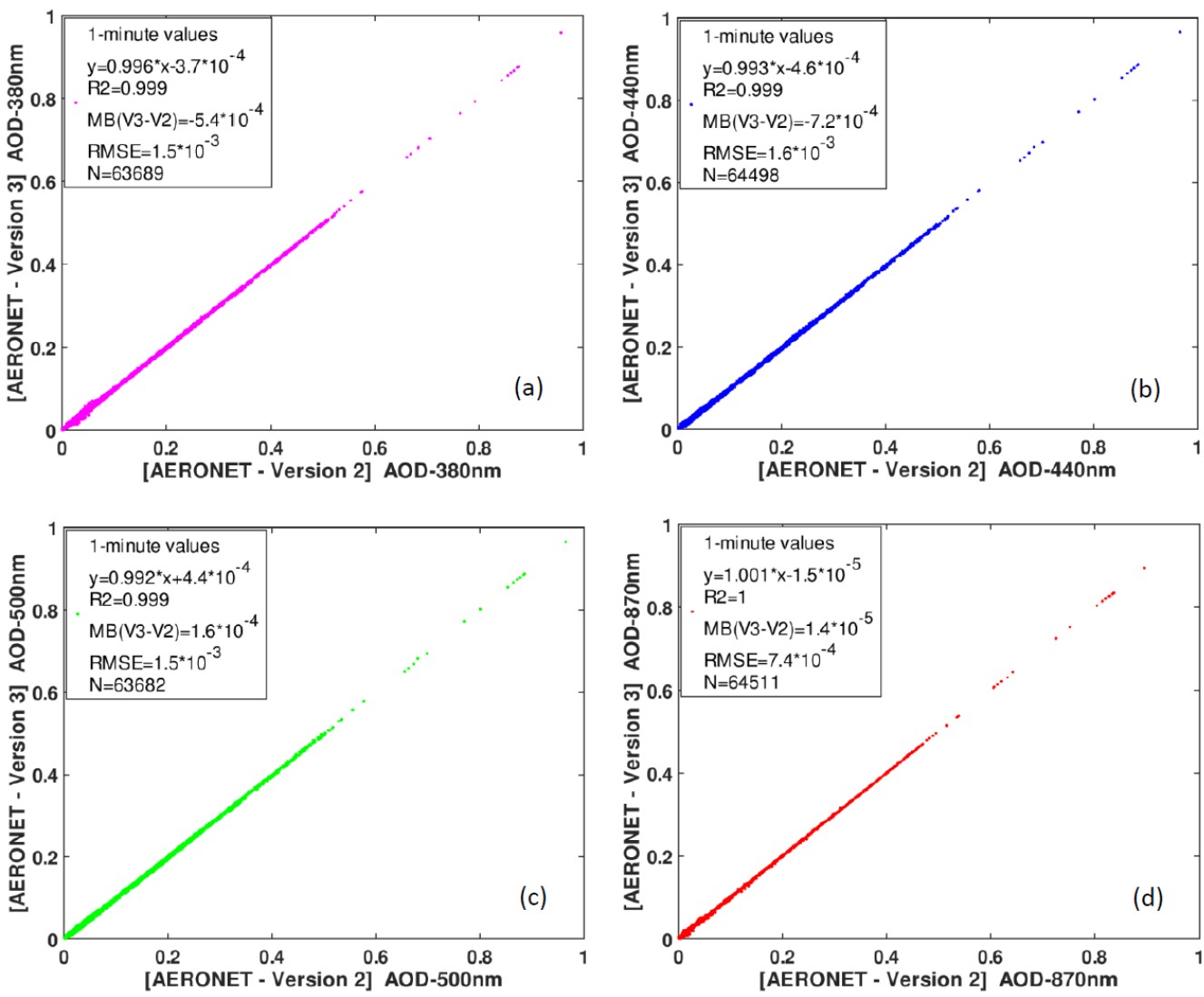

**Figure 1.** AERONET Version 3 (V3) vs Version 2 (V2) AOD 1-minute data scatterplot at Izaña Observatory for the period 2005-2015: (a) 380 nm; (b) 440 nm; (c) 500 nm and (d) 870 nm. The corresponding equations of the linear fits, the coefficients of determination ($R^2$), Mean Bias (MB), Root Mean Square Error (RMSE) and the number of data pairs (N) used are included in each legend.

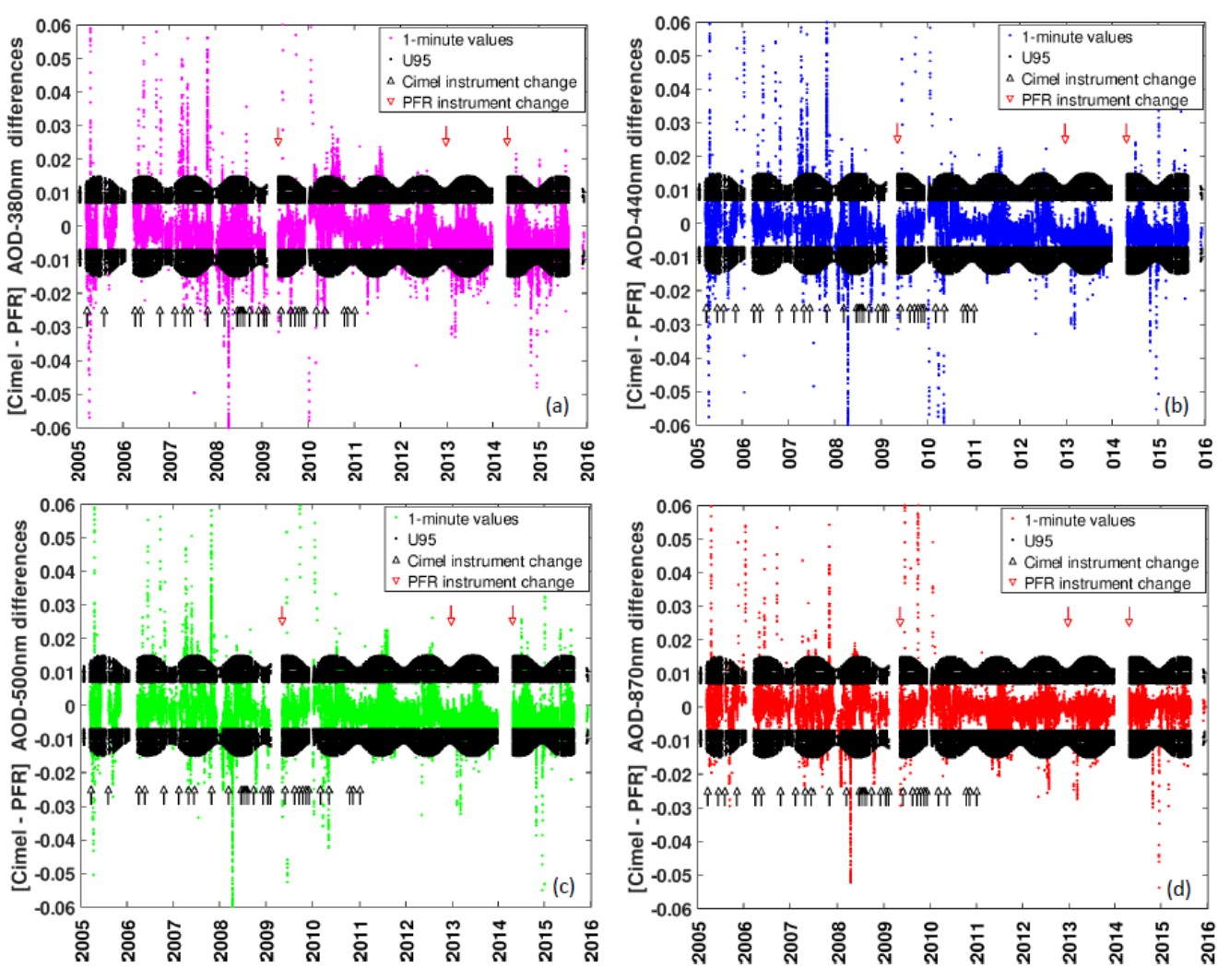

**Figure 2.** One-minute AOD data differences between AERONET-Cimel (V2) and GAW-PFR for (a) 380 nm (70838 data-pairs), (b) 440 nm (71645 data-pairs), (c) 500 nm (70833 data-pairs) and (d) 870 nm (71660 data-pairs) for the period 2005-2015. Black dots correspond to the $U95$ limits. A small number of outliers are out of the $\sim 0.06$ AOD differences range. Black arrows indicate a change of Reference AERONET-Cimel radiometer and red arrows indicate a change of the GAW-PFR instrument.

**Table 3.** Percentage of AERONET-Cimel (V2 and V3) 1-minute AOD data meeting the WMO criteria for the four interpolated GAW-PFR channels for the period 2005-2015.

| Channel (nm) | V2 (%) | V3 (%) |
|:---:|:---:|:---:|
| 380 | 92.7 | 92.3 |
| 440 | 95.7 | 95.2 |
| 500 | 95.8 | 95.7 |
| 870 | 98.0 | 97.8 |

(see Table 4). We observe that the poorest agreement is obtained at the shorter wavelength channels (440 nm, and especially 380 nm). Kazadzis et al. (2018b) also found a decrease in the percentage of AOD meeting the WMO criteria for 368 nm and 412 nm spectral bands during the Fourth WMO Filter Radiometer Comparison for aerosol optical depth measurements. As these authors pointed out, the shorter the wavelength, the poorer the agreement because of several reasons: AOD in the UV suffers from out-of-band or at least different blocking of the filters, small differences in central wavelength or FWHM have a larger impact, the Rayleigh correction is more critical, and $NO_2$ absorptions are treated differently. Regarding the effect of

the aerosol load and particle size on the AOD differences, our results confirm the decrease of agreement between the two instruments for very large particles coincident with almost pure dust (AE $\leq$ 0.3), and high turbidity conditions (AOD > 0.1). However, it should be noted that the percentage of data pairs in these situations is relatively low (e.g., 6 % for AOD > 0.1, and 3.2 % for AE < 0.25 at 380 nm) with respect to the total data (Table 4). A similar result was reported by Kim et al. (2008), who

attributed these discrepancies to the possible spatial and temporal variability of aerosols under larger optical depths in addition to the effect of the different FOV of both radiometers. In our case, and according to previous studies on AOD climatology at IZO (Barreto et al., 2014), the presence of high mineral dust burden when the station is within the SAL, does not necessarily imply lower atmospheric stability conditions resulting in daily AOD means with greater standard deviation. For these reasons, we assumed that the different FOV of these instruments is the main cause of part of the AOD 1-minute differences outside the

U95 limits, under high AOD conditions. This issue is specifically addressed in Sect. 5.3.

In general, the agreement obtained with the 1-minute AOD data is slightly lower than that obtained during short campaigns, such as those reported by Barreto et al. (2016) at IZO (5566 data-pairs), with agreements > 99 % for $AOD_{870nm}$ and $AOD_{500nm}$. However, our results for $AOD_{500nm}$ (> 95 % of 70833 data-pairs) are significantly better that that observed by Kazadzis et al. (2014) ($\sim$ 48 % of 468 data-pairs) covering a relatively narrow range of AOD. In addition, short-term cam-

340 paigns usually cover a small range of AOD, and instruments are carefully and frequently supervised. On the contrary, during our intercomparison over a period of 11 years, the operation of the instruments can be considered as the normal operation of such a system. An additional interesting aspect of this study is that it is not a simple intercomparison exercise between two instruments but a comparison of a number of instruments that acted as reference instruments for the AERONET/Europe Network.

In the first period (2005-2009), a total of 13 Cimel radiometers were used, while in the second period (2010-2015), five Cimel radiometers have participated, and for much of this period, the Cimel #244 was operating as the permanent AERONET

**Table 4.** Percentage of AERONET-Cimel 1-minute AOD data (V2) meeting the WMO criteria for the four compared channels, and different AOD and AE scenarios for the period 2005-2015. The last row corresponds to the total percentages for the sub-period 2010-2015. AOD and AE traceability > 95 % are marked in bold. Number of data pairs are in brackets.

| % of data within WMO limits | 380 nm | 440 nm | 500 nm | 870 nm |
|---|---|---|---|---|
| AOD≤0.05 | 94.4 (57008) | **96.8 (59130)** | **97.0 (58572)** | **98.5 (60191)** |
| 0.05<AOD≤0.10 | 91.0 (4723) | 93.1 (4850) | 92.8 (4817) | 94.2 (4908) |
| AOD>0.10 | 75.0 (3938) | 86.5 (4615) | 85.1 (4466) | **95.9 (5118)** |
| AE≤0.25 | 73.1 (2145) | 82.3 (2417) | 80.1 (2351) | **96.2 (2824)** |
| 0.25<AE≤0.6 | 91.2 (5407) | **96.2 (5810)** | **96.0 (5691)** | **97.9 (5911)** |
| AE>0.6 | 94.6 (55114) | **96.9 (57089)** | **97.0 (56504)** | **98.7 (58146)** |
| Total 2005-2015 | 92.7 (65669) | **95.7 (68595)** | **95.8 (67855)** | **98.0 (70217)** |
| Total 2010-2015 | 93.5 (41977) | **97.4 (43745)** | **97.2 (43627)** | **99.1 (44498))** |

**Table 5.** Basic skill-scores from the AOD intercomparison between GAW-PFR and AERONET-Cimel V2 for the period 2005-2015. The skill scores definitions are found in Huijnen and Eskes (2012).

| Period | 2005-2015 | | | |
|---|---|---|---|---|
| Wavelengths (nm) | 380 | 440 | 500 | 870 |
| Mean Bias (MB) | -0.0026 | -0.0018 | -0.0021 | -0.0001 |
| Modified Normalized Mean Bias (MNMB) | -0.1301 | -0.1046 | -0.1474 | 0.0129 |
| Fractional Gross Error (FGE) | 0.1727 | 0.1546 | 0.1918 | 0.1837 |
| Root Mean Square Error (RMSE) | 0.0081 | 0.0070 | 0.0064 | 0.0049 |
| Pearson's correlation coefficient (r) | 0.9910 | 0.9925 | 0.9939 | 0.9949 |
| Number of data-pairs | 70838 | 71645 | 70833 | 71660 |

reference instrument at IZO. Once the most important causes of non-traceability in the first period, which were associated with a poor pointing of GAW-PFR due to problems in the sun-tracker, were discounted, we can conclude that there are no significant differences in the percentages of traceable data between the two periods. This means that the continuous change of Reference Cimel instruments used in the 2005-2009 period did not have a significant impact on AOD data comparison differences. This provides proof of the consistency and homogeneity of the long AERONET-Cimel AOD data series, and their comparability with the GAW-PFR AOD data series, regardless of the number of instruments used to generate these data series. In our study, with a number of comparison data-pairs one or two orders of magnitude higher than those used in short campaigns, the results shown in Table 4 can be considered as fairly good.

In addition to the traceability scores, we have introduced some basic skill scores corresponding to the AOD intercomparison between GAW-PFR and AERONET-Cimel for the period 2005-2015 (Table 5) to be in line with previous studies that have performed short-term comparisons between these two instruments. The definitions of the used skill scores can be found in

Huijnen and Eskes (2012). The Pearson's correlation coefficient (r) values of the PFR-Cimel 1-minute AOD data-pairs are higher than 0.99 in all channels. Concerning Mean Bias (MB) and Root Mean Square Error (RMSE) associated with AOD

differences, our results show quite similar skill scores to those found at Mauna Loa, USA for $AOD_{500nm}$ (Kim et al., 2008), although the number of data pairs used at IZO ($\sim$ 71000) is much higher, and the AOD range of our study is much larger than that of the comparison performed in Mauna Loa. Kim et al. (2008) summarize results of previous short-term intensive studies (McArthur et al., 2003; Mitchell and Forgan, 2003; Kim et al., 2005; Schmid et al., 1999) carried out in stations where the radiometers were calibrated by intercomparison with Reference instruments. These results show MB values to be within 0.01

bias, one order of magnitude larger than in Mauna Loa and Izaña Observatories, highlighting the importance of having well calibrated instruments to carry out these type of comparisons. For the period 2010-2015 (not shown here), as expected, the RMSE and the Pearson's correlation improve slightly compared with the whole period 2005-2015.

In relation to the comparison between GAW-PFR and AERONET-Cimel V3, we have calculated the percentage of AERONET-Cimel 1-minute AOD data (V3) meeting the WMO criteria for optical air mass > 5.0 for the period 2005-2015 (Supplement

S2.2). The results are somewhat poorer than for optical air mass < 5.0, since the solar elevation is very low. Only for the 870 nm channel 95 % of the data meet with the WMO criteria, although the percentages of data in the 440 nm and 500 nm channels are close to this value. This would be the main reason to find slightly poorer traceability results with all V3 data compared to those found with V2 for which the AOD data are limited to optical air mass < 5.0.

## 5.3   Non-traceability assessment

As presented in Table 3, data outside the WMO traceability criteria vary from 2 % for 870 nm up to 7.3 % for 380 nm. In this section, the different possible causes of non-traceability in AOD are evaluated and, if possible, quantitatively estimated. In order to assess the relevance and quantitative impact of these causes, and estimate errors derived from a non-perfect AOD data synchronization, we first made an analysis on the natural variability of AOD in a very short time period (1 minute) shown below.

### 5.3.1   Short-time AOD variability

In order to determine the variability of AOD within one minute, we have performed two independent analyses with AOD data from the PFR and Cimel for the 368/380 nm and 501/500nm channels during one year (2013). On the one hand, and taking into account that GAW-PFR provides AOD every minute, we have calculated all the AOD differences for each channel in the successive minutes. So we have the variation of AOD from one minute to the next one during a whole year. On the other hand,

for AERONET-Cimel, we have taken advantage of the triplets, since each triplet consists of three successive measurements made in one minute. In this case, the strategy has been to calculate the standard deviation of the triplet AOD measurements during a whole year.

The results obtained on the AOD variability in 1-minute from PFR data are very similar and consistent to those obtained with Cimel. Less than $\sim$ 0.8% of the AOD data show variability higher than 0.005 in all wavelength ranges. It should be noted that

the possible instrumental noise is included in this variability, so that the actual natural AOD variability would be, in any case,

**Table 6.** Percentage of AOD data with variability within 1 minute less than 0.01 and 0.005, respectively, using AOD data from GAW-PFR (at 368nm and 501nm) and AERONET-Cimel (at 380nm and 500 nm) for 2013. A total of $\sim$ 32000 data-pairs per channel have been used from GAW-PFR, and 20117 triplets (60351 individual AOD measurements) from the Cimel #244 to calculate the AOD variability.

| GAW-PFR | | |
|---|---|---|
| Percentage of data with 1-minute AOD variability (%) | | |
| | 368 nm | 501 nm |
| <0.01 | 99.8 | 99.91 |
| <0.005 | 99.21 | 99.35 |
| AERONET-Cimel | | |
| Percentage of data with 1-minute AOD variability (%) | | |
| | 380 nm | 500 nm |
| <0.01 | 99.87 | 99.99 |
| <0.005 | 99.82 | 99.42 |

lower than that expressed in Table 6. The percentage of data with 1-minute AOD variability for all four GAW-PFR channels are given in Supplement S3.1.

We have also determined the percentage of 1-minute AOD data from the Cimel triplets (year 2013) whose diurnal range of variation $AOD_{max}$-$AOD_{min}$ > 0.015, for several AOD intervals. Note that this value is half of the WMO traceability interval

when m = 1 (maximum possible interval) (see Eq. 2). The results shown in S3.2 indicate that the 1-minute AOD variability is responsible for only 0.11 % (0.01 %) of 1-minute Cimel AOD values outside the WMO limits in the [0-0.03] AOD range (pristine conditions) for 380 nm (500 nm). The AOD variability maximizes in the 0.1-1 AOD range causing 2.31 % and 1.69 % of the AOD data outside WMO limits for 380 and 500 nm, respectively. This last scenario corresponds, as expected, to changes of air masses, such as transitions from pristine to dusty conditions, and vice versa, or to sharp onset and disappearance of very

sporadic biomass burning plumes. In any case, the AOD data with 1-minute variability exceeding 0.02 (V2) or 0.01 (V3) are filtered by AERONET (see Sect. 4.2) and therefore are not included in the GAW-PFR and AERONET-Cimel comparison.

These results indicate that the natural AOD variability is very low thus the non-ideal measurement synchronization cannot explain the percentages of non-traceable AOD cases shown in Tables 3 and 4.

### 5.3.2 Uncertainties of GAW-PFR channel interpolation to AERONET-Cimel channels

The interpolation of the CIMEL AODs to the PFR AOD wavelengths can be one of the sources of uncertainty in this comparison assessment. The greatest uncertainty arises in the extrapolation of the $AOD_{412nm}$ of the PFR to the Cimel wavelength 440 nm. Using the Angström formula we have calculated that for an uncertainty of $\pm$ 0.5 in the AE the introduced uncertainty in the AOD extrapolation from 412 nm to 440 nm is $\sim$ 5 % (i.e., 0.005 for $AOD_{412nm}$ = 0.1). The introduced uncertainty in AOD extrapolation is reduced to $\sim$ 2 % for an uncertainty of $\pm$ 0.3 in AE. For all other AOD interpolations the errors are smaller.

### 5.3.3 Calibration related errors

As described in Sect. 3, the calibration procedures of the AERONET-Cimel and GAW-PFR radiometers are different. While in the case of GAW-PFR, frequent calibrations are established throughout the year and the calibration value is linearly interpolated in time, in AERONET- Cimel a constant calibration value is assumed in the intermediate period between two consecutive calibrations carried out on an annual basis. The typical calibration uncertainty for a single Langley plot is 0.7-0.9 % (at the 95 % confidence level), and it is reduced to 0.4 % in the case of IZO when averaging at least 10 Langley-derived extraterrestrial constants (which is the normal procedure) (Toledano et al., 2018). Regarding the GAW-PFR radiometers operated at IZO, a direct yearly comparison of the Langley based $V_0$'s with the reference triad at PMOD/WRC showed differences lower than 1 % for all channels for the 2005-2015 period.

A not sufficiently accurate determination of the calibration constant results in a fictitious AOD diurnal evolution presenting a concave or convex characteristic curve due to the calibration error dependence on solar air mass. The largest error occurs in the central part of the day (lower air masses), mainly, in clean days with very low aerosol load (< 0.02 in 500 nm) as reported by Romero and Cuevas (2002) and Cachorro et al. (2004), and as can be derived from Equation 2. According to Cachorro et al. (2004, 2008) fictitious differences of up to 0.06 between the minimum and the maximum AOD can be recorded in a day with constant AOD as a result of a non-accurate calibration or non-cleaned instruments. However, these fictitious differences in AOD depend on the related calibration magnitude errors.

We have represented the AOD differences between GAW-PFR and AERONET-Cimel versus optical air mass for the four channels for pristine conditions ($AOD_{500nm} \leq 0.03$) for both V2 and V3 (See Supplement S4). It should be noted that although the few outliers are evenly distributed throughout the whole airmass range, they are not equally distributed with respect to the zero of the AOD difference, but there is a bias with positive large outliers (higher Cimel AOD), already reported by Nyeki et al. (2013), and small negative outliers for optical air mass lower than 2.

The total percentage of AOD traceable data pairs under pristine conditions ($AOD_{500nm} \leq 0.03$) is very high for all wavelengths (> 97.7 %) falling within the $U95$ limits (Table 7), except for 380 nm. There is no dependence on 1-minute AOD differences with optical air mass for 440, 500 and 870 nm, and a slight dependence for 380 nm (Table 7) with higher percentage of AOD differences outside the $U95$ limits at lower optical air masses. For the extended range of optical air mass > 5 in V3, the AOD differences do not increase with optical air mass (Supplement S5). The lower traceability at 380 nm for low air masses is especially clear in V3 with 92.9 % of traceable data (See Supplement S5). This result is consistent with the fact that the highest uncertainty in the determination of the calibration constants is observed in the UV range, and the lowest uncertainty in the near-infrared channel (Eck et al., 1999; Jarosławski et al., 2003; Toledano et al., 2018). This is attributable to an imperfect calibration, or to very small changes in the filters' transmittance, that can only be detectable in extreme conditions: UV range, very low optical air mass, and pristine conditions. According to (Toledano et al., 2018), the greatest variance in the extraterrestrial constant in the UV channel could be due to a number of factors: 1) higher AOD variability at the shorter wavelengths; 2) filter blocking issues; and 3) temperature effects affecting AERONET-Cimel instruments that have not been accounted for in the UV range.

**Table 7.** Percentage of 1-minute AOD data (V2) meeting the WMO criteria for each wavelength for different optical air mass intervals under pristine conditions (AOD$_{500nm}$ ≤ 0.03) in the period 2005-2015. See Supplement S5 for equivalent results with V3.

| Percentage of AOD differences within the $U$95 limits AOD$_{500nm}$ ≤ 0.03 | Total (%) | $1 \leq m < 2$ (%) | $2 \leq m < 3$ (%) | $3 \leq m < 4$ (%) | $4 \leq m < 5$ (%) |
|---|---|---|---|---|---|
| 380 nm | 95.8 | 94.5 | 96.0 | 97.4 | 97.2 |
| 440 nm | 97.9 | 97.9 | 97.7 | 98.2 | 97.7 |
| 500 nm | 98.3 | 98.4 | 98.1 | 98.6 | 98.4 |
| 870 nm | 99.2 | 99.4 | 99.3 | 99.2 | 98.6 |

The correct cause attribution of each outlier would require manual inspection and additional specific information on instrumental checking and maintenance information that is not always available. We have investigated in more detail the origin of the outliers and whether one of the two instruments predominantly caused them. Thus, we have calculated for the non-traceable AOD data the diurnal range of AOD variation (maximum value minus minimum value of AOD in one day) at 380 nm for each instrument under pristine conditions (Figure 3) using Cimel AOD$_{500nm}$ daily mean < 0.03 to select the pristine days. According to this approach, the instrument that shows the highest daytime AOD range is the one that is responsible for the outlier. As the wavelength increases both the number of outliers and the magnitude thereof decreases significantly (Supplement S6). Then, we identified those outliers with a diurnal AOD range higher than 25 % of the mean daily AOD value and investigated their possible causes. A total of 51 cases for GAW-PFR and 81 cases for AERONET Cimel V3 were obtained and analysed in detail, using auxiliary information, such as 1-minute in-situ meteorological data, 5-minute all-sky images, 1-minute BSRN data, and satellite imagery (not shown here). We obtained the percentage of AOD outliers of GAW-PFR and AERONET Cimel (V3) for which a certain cause has been identified, such as calibration uncertainties, cloud screening algorithm failures, mixture of the two previous causes, poor sun pointing, or not well-defined causes (electronic problems, humidity inside the lenses, filter dirtiness, obstruction of the lenses collimators, insects on the optics outside, etc.) (see Supplement S7).

From the analysis of these cases, under the conditions described above, it should be noted that ∼ 44 % of the cases with fictitious AOD diurnal cycles were due to small uncertainties in the calibration of AERONET-Cimel (V3), while for this same cause ∼ 8 % of cases were identified in GAW-PFR. Some examples of AOD non-traceability for both AERONET-Cimel and GAW-PFR in the ∼ 380 nm channel are shown in Supplement S8. The fictitious diurnal AOD cycle is mainly visible in the UV channels as shown in the examples reported in Supplement S9, where the convex or concave diurnal AOD curvature symmetrical at noon provides a hint of calibration inaccuracies. Note that the fictitious diurnal AOD can be more easily identified under very low AOD conditions. We should emphasize that the rare finding of small calibration inaccuracies in a high mountain site with pristine skies and stable atmosphere does not detract from the quality of any instrument as they often measure near or below the detection limit. Simply, these small inaccuracies are the result of limitations in the photometric measurement technique.

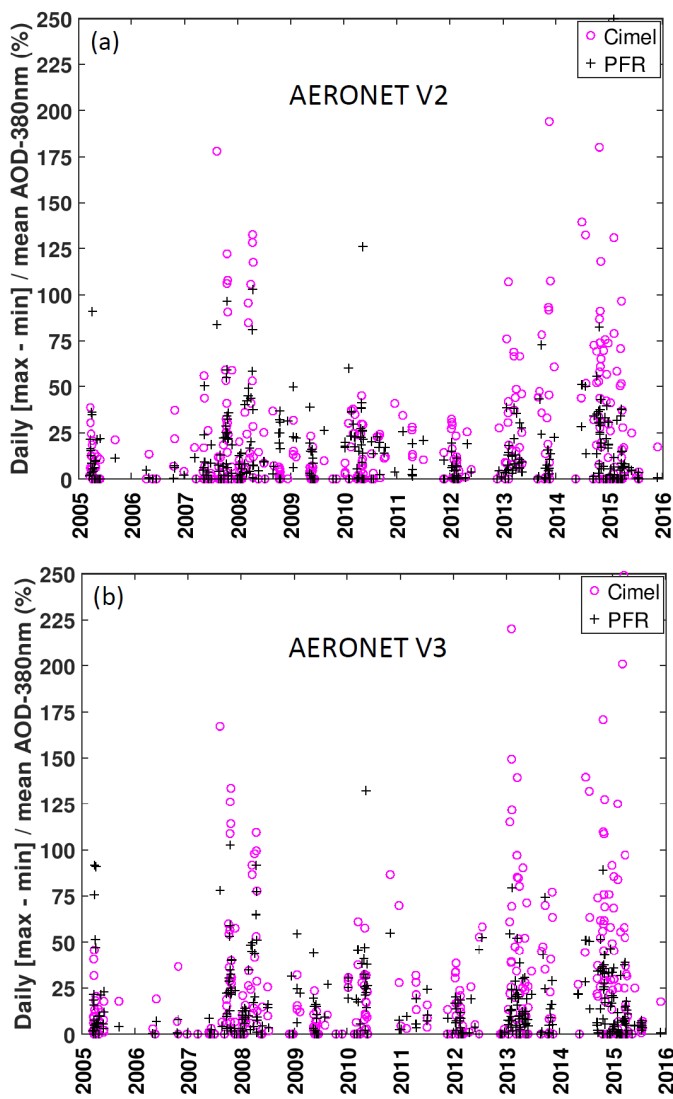

**Figure 3.** AOD diurnal range variation (maximum value minus minimum value of AOD in one day) at 380nm corresponding to AOD outliers (non-traceable AOD) under pristine conditions ($\text{AOD}_{Cimel-500nm} \leq 0.03$) in the period 2005-2015 for AERONET V2 (a) and V3 (b).

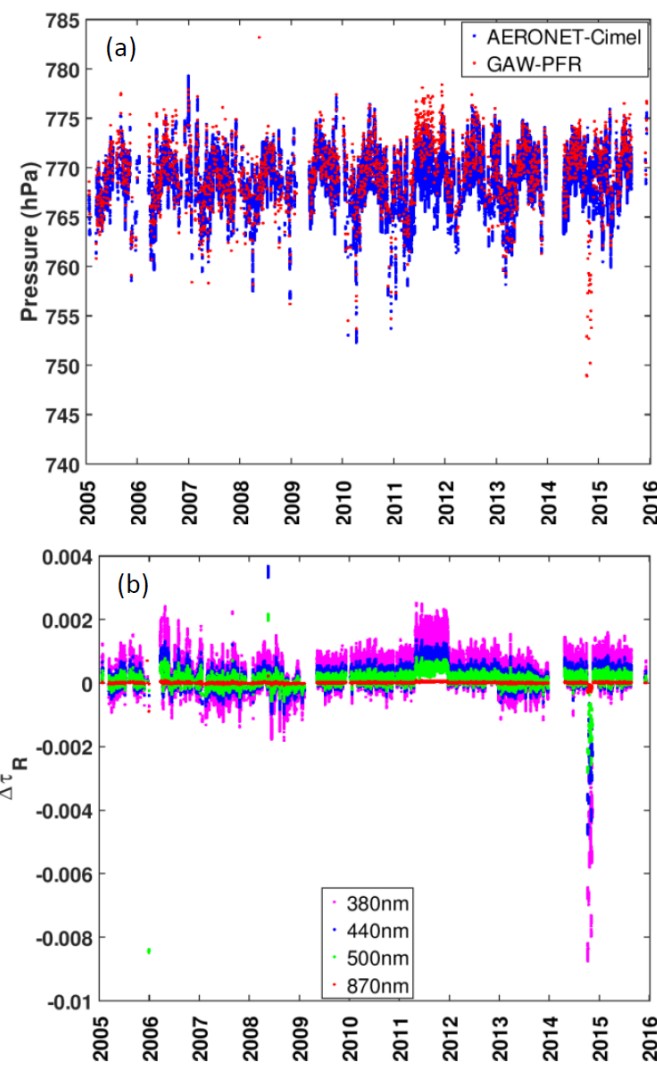

**Figure 4.** (a) 1-minute pressure data (hPa) from GAW-PFR and 6-hour pressure data at Izaña Observatory altitude from the National Centers for Environmental Prediction (NCEP) and the National Center for Atmospheric Research (NCAR) reanalysis for the case of AERONET-Cimel, and (b) corresponding 1-minute $\triangle\tau_R$ caused by pressure differences in the period 2005-2015.

**Table 8.** Percentage of AOD data within the $U95$ limits for each channel and five daily fractions of clear sky (FCS) intervals. In brackets, relative frequency of each FCS interval for AERONET V2 and V3, respectively. In bold, the percentages of V3 that are greater than those of V2.

| | 380 nm | | 440 nm | | 500 nm | | 870 nm | |
|---|---|---|---|---|---|---|---|---|
| | V2 | V3 | V2 | V3 | V2 | V3 | V2 | V3 |
| 0%≤FCS<20% (0.03%) (0.04%) | 47.6 | 44.4 | 43.5 | 44.4 | 47.6 | 44.4 | 87.0 | **92.6** |
| 20%≤FCS<40% (0.22%) (0.22%) | 69.3 | **76.6** | 73.3 | **82.2** | 73.6 | **80.8** | 86.3 | **94.1** |
| 40%≤FCS<60% (1.08%) (1.09%) | 79.1 | 77.5 | 87.8 | 84.8 | 88.8 | 87.2 | 91.9 | **92.0** |
| 60%≤FCS<80% (7.10%) (7.17%) | 88.4 | **89.6** | 93.9 | 93.9 | 93.4 | **94.4** | 97.8 | 97.6 |
| FCS≥80% (91.6%) (91.5%) | 93.3 | 92.8 | 96.2 | 95.6 | 96.2 | 96.1 | 98.3 | 98.1 |

### 5.3.4 Differences in cloud-screening and sun tracking

We have examined the effect that the presence of clouds might have on AOD differences and the percentage of cases outside the $U95$ limits. The impact of clouds on AOD differences only occurs when both GAW-PFR and AERONET-Cimel cloud screening algorithms fail to identify clouds in the direct sun path. In order to assess the impact that cloud conditions might cause on AOD traceability, we have used the concept of daily fractions of clear sky (FCS) that has been applied before to solar radiation data at IZO (García et al., 2014). FCS represents the percentage of observed sunshine hours in a day with respect to the maximum possible sunshine hours in that day. The higher the daily FCS, the higher the clear sky percentage we have on that day. The percentages of traceable and non-traceable AOD data versus FCS values grouped into five intervals are shown in Table 8. It should be emphasized that the number of cases linked with FCS between 0 % and 60 % are less than 2 % of the total cases. As the fraction of clear sky increases, the percentage of traceable AOD data significantly exceeds the number of non-traceable AOD data. The percentage of traceable data is especially large (> 90 %) when FCS > 80 % (almost clear skies).

This is the FCS range in which a significant percentage of days with cases presenting scattered clouds are recorded, which qualitatively confirms that V3 has introduced more efficient cloud screening than V2. However, the real impact of clouds on AOD traceability at IZO is very low due to its special characteristics of a high mountain station with very little cloudiness. Therefore, in practice, the possible impact of clouds on the non-traceability of AOD data-pairs is insignificant at IZO. GAW-PFR and AERONET-Cimel cloud screening algorithms provide successful identification on clear direct-sun conditions during cloudy skies (FCS < 40 %) for 99.75 % of the cases, excluding those with very thin clouds.

In the particular case of Izaña there are two specific cloud scenarios in which cloud screening algorithms could fail resulting in non-AOD traceability: 1) Cirrus clouds (see Supplement S10); and 2) low clouds (stratocumulus) that sometimes exceed

the observatory height level (see Supplement S11). As can be deduced from the analysis of these cloud cases, the impact of the different types of clouds on AOD retrieval is very complex and further specific investigations are required in order to understand the reasons behind failures in the GAW-PFR and AERONET-Cimel cloud screening algorithms.

### 5.3.5 Rayleigh scattering, absorption by $O_3$ and $NO_2$ corrections

In this section, we evaluate the possible impact on the 1-minute AOD data outside the $U95$ limits due to the different processing of each network regarding the correction by Rayleigh scattering and by the light absorption of column $O_3$ and $NO_2$. Although GAW-PFR and AERONET-Cimel use spectral channels with weak absorption by atmospheric gases, AOD can only be determined if optical depth contributions from those gases are well estimated and subtracted from the total optical depth ($\tau$). GAW-PFR and AERONET-Cimel separate the contributions of the molecules (Rayleigh scattering, $\tau_R$), aerosols ($\tau_a$; in this study referred to as AOD) and absorbing gases: total column ozone ($\tau_{O3}$) and nitrogen dioxide ($\tau_{NO2}$) due to their different optical air masses at low solar elevation:

$$I(\lambda) = I_0(\lambda) \exp -(\tau_R m_R + \tau_a m_a + \tau_{O3} m_{O3} + \tau_{NO2} m_{NO2}) \tag{3}$$

So, AOD can be derived from:

$$AOD = \frac{1}{m_a} (ln \frac{I_0(\lambda)}{I(\lambda)} - \tau_R m_R - \tau_{O3} m_{O3} - \tau_{NO2} m_{NO2}) \tag{4}$$

**Rayleigh scattering**

The Rayleigh scattering contribution to total optical depth would be:

$$\tau_R = \delta_R \frac{m_R}{m_a} \tag{5}$$

where, $m_R$ is written, according to Kasten and Young (1989), as:

$$m_R = \frac{1}{sin\theta + 0.50572(\theta + 6.07995)^{(-1.6364)}} \tag{6}$$

and $m_a$, according to Kasten (1966), has the following expression:

$$m_a = \frac{1}{sin\theta + 0.0548(\theta + 2.65)^{(-1.452)}} \tag{7}$$

where $\theta$ is the sun elevation, and $\delta_R$ can be expressed as (Bodhaine et al., 1999):

$$\delta_R(\lambda) = 0.00864\lambda^{-(3.916 + 0.074\lambda + \frac{0.050}{\lambda})} \frac{P}{P_o} \tag{8}$$

where $P_o$ = 1013.25 hPa, $\lambda$ is the wavelength in microns ($\mu$) and $P$ is the pressure in hPa at the measurement site. The depolarization factor recommended by (Young, 1980) is already included in Eq. 8. From Eq. 8, we can derive the differences in $\tau_R$ contribution ($\triangle \tau_R$):

$$\triangle \tau_R = (0.00864\lambda^{-(3.916 + 0.074\lambda + \frac{0.050}{\lambda})} \frac{1}{1013.25} \frac{m_R}{m_a})(P_{PFR} - P_{Cimel}) \tag{9}$$

Accordingly, the main $\tau_R$ from GAW-PFR and AERONET-Cimel can arise from the different way the two instruments obtain the atmospheric pressure ($P_{PFR}$ and $P_{Cimel}$, respectively). While AERONET-Cimel obtains the site station pressure from the National Centers for Environmental Prediction (NCEP) and the National Center for Atmospheric Research (NCAR) reanalysis at standard levels, GAW-PFR has a solid-state pressure transducer in the control box to read barometric pressure simultaneously with each PFR measurement. As Giles et al. (2019) have stated, the expected error in the station pressure $P_{Cimel}$ is generally < 2 hPa provided the elevation of the station is well-known and the weather conditions are stable. In order to assess this possible difference, we have compared the 1-minute synchronous pressure data of both instruments, and the corresponding 1-minute $\triangle\tau_R$ from Eq. 9. Note that, in practice, this comparison is performed at 6-hour intervals since the NCEP/NCAR reanalysis data are available routinely with this temporal resolution (Kalnay et al., 1996). The results are depicted in Figure 4.

The results indicate that most of the 1-minute pressure differences are within ± 5 hPa (Figure 4a), resulting in 1-minute $\triangle\tau_R$ data within ± 0.001. However, when pressure differences are significantly higher, such as those registered at the end of 2014 (> 30 hPa) (Figure 4a), $\triangle\tau_R$ increases significantly ($\sim$ 0.01) (Figure 4b). However, it should be noted that only 99 AOD data pairs have been registered for which the pressure difference between PFR and Cimel is greater than 20 hPa at 870nm and 440nm, and one AOD data pair at 500nm and 380nm channels. Taking into account that the accuracy of the new barometers built into new radiometers is $\sim$ 3 hPa only dramatic barometer malfunctioning could cause $\tau_R$ > 0.01. As stated by Kazadzis et al. (2018b), the use of erroneous pressure values can lead to wavelength-dependent AOD errors and to large errors in AE. However, these flagrant barometer malfunctions are quickly detected and easily corrected if there are other pressure measurements at the station, as is the case in Izaña.

### Differences in $O_3$ absorption

The $O_3$ optical depth is determined with the following expression:

$$\tau_{O3}(\lambda) = \sigma_{O3}(\lambda)\frac{O_3}{1000}\frac{m_{O3}}{m_a} \tag{10}$$

Where $O_3$ is expressed in Dobson units (DU), and the absorption coefficients ($\sigma_{O3}(\lambda)$) take the following values (Gueymard, 1995): 0.0026 cm$^{-1}$ (440 nm), 0.03150 cm$^{-1}$ (500 nm), and 0.00133 cm$^{-1}$ (870 nm). The ozone absorption is maximum in the 500 nm channel and zero in the 380 nm channel. GAW-PFR uses for $m_{O3}$ the following expression (Komhyr, 1980):

$$m_{O3} = \frac{R+h}{\sqrt{(R+h)^2 - (R+r)^2(cos\theta)^2}} \tag{11}$$

where $R$ = 6370 km is the mean radius of the Earth, $r$ = 2.370 km is the altitude of the station, $h$ = 22 km is the estimated height of the ozone layer, and $\theta$ is the solar elevation. However, AERONET-Cimel uses an updated expression (Komhyr et al., 1989) in which h is not fixed and takes a value in function of the latitude, and the absorption coefficients are obtained for each particular filter using the spectral response provided by the manufacturer.

For most of the period covered in this study, measured total ozone values from the GAW Izaña station were used to calculate $\tau_{O3}$ (Wehrli, 2008a). If no Brewer data is available, data retrieved from the Total Ozone Mapping Spectrometer (TOMS) satellite-sensor was used. Nowadays, GAW-PFR uses ozone data from AURA satellite overpass observations with the Ozone Monitoring Instrument (OMI) (McPeters et al., 2015) for daily operations (Kazadzis et al., 2018b). Concerning AERONET-

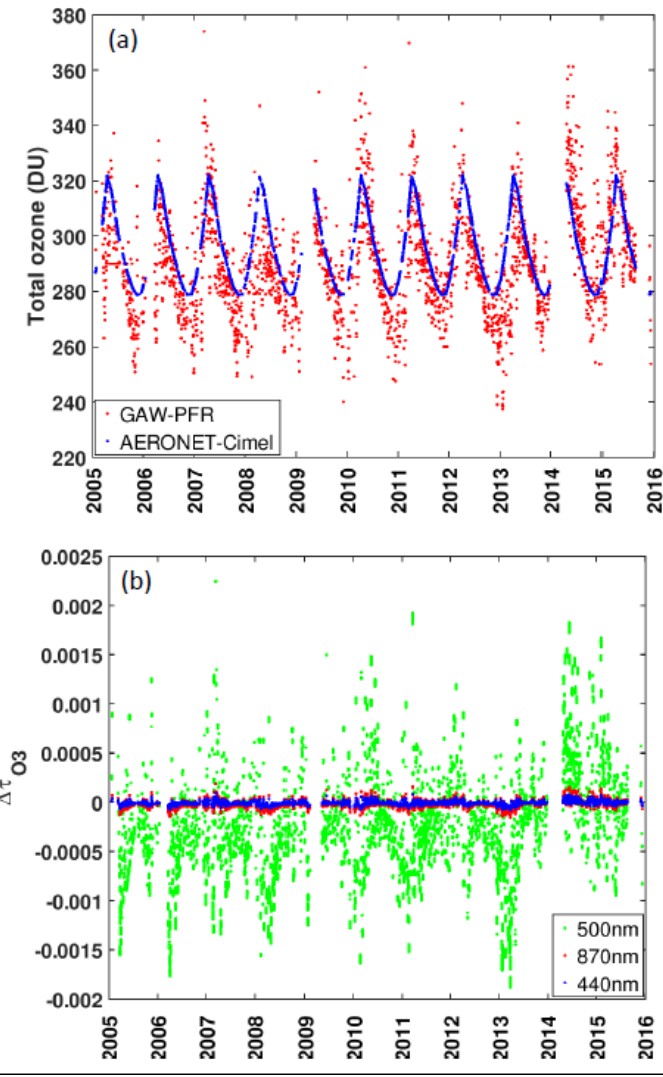

**Figure 5.** (a) Total $O_3$ used by GAW-PFR (measured Brewer $O_3$ values from IZO, OMI $O_3$ overpass or Brewer $O_3$ climatology) and AERONET-Cimel (TOMS $O_3$ climatology), and (b) $\triangle \tau_{O3}$ ($\lambda$) caused by differences in daily total $O_3$ between the two instruments in the period 2005-2015.

Cimel V2, a NASA TOMS $1° \times 1.25°$ resolution $O_3$ climatology is used. From Eq. 10, the differences in $O_3$ optical depth $\triangle\tau_{O3}$ can be derived:

$$\triangle\tau_{O3} = \sigma_{O3}(\lambda)\frac{1}{1000}\frac{m_{O3}(O_{3PFR} - O_{3Cimel})}{m_a} \tag{12}$$

The largest influence of total ozone data uncertainty in O3 occurs at 500 nm (Figure 5). According to Wehrli (2008b) and Kazadzis et al. (2018b), total ozone needs to be determined to $\pm$ 30 DU or 10 % of typical values, to ensure an uncertainty of $\sim$ 0.001 in $\tau_{O3}$ at 500 nm. In the case of the GAW-PFR / AERONET-Cimel comparison, and due to the very different method in which both networks obtained O3 values for their corresponding corrections, the ozone differences found on some days (1761 out of 71965 days; 2.4 %) are very large (> 40 DU), exceeding a difference in the ozone optical depth of 0.001. Even so, the potential contribution to AOD differences outside the $U95$ limits between the two networks is negligible. Total $O_3$ over over IZO shows a relatively small amplitude throughout the year, but both surface ozone concentrations and column ozone amount could sharply increase under the influence of cut-off lows injecting air from the high-mid troposphere into the lower subtropical troposphere, which is not uncommon in spring and the first half of summer (Cuevas et al., 2015; Kentarchos et al., 2000). In addition through exchange processes in the Upper Troposphere Lower Stratosphere (UTLS) due to the presence of the subtropical jet (mainly from February to April) (Rodriguez-Franco and Cuevas, 2013). However, if we wanted to repeat this traceability study of 1-minute AOD data in mid or high latitude stations where sharp $O_3$ variations (several tens of DU) could be registered in a few hours, the correction of 1-minute AOD measurements by $\tau_{O3}$ might be a challenging issue.

### Differences in NO$_2$ absorption

AERONET-Cimel applies a correction by absorption of NO$_2$, but GAW-PFR does not include this correction. AERONET-Cimel V2 obtains daily total NO$_2$ data from a $0.25° \times 0.25°$ resolution NO$_2$ monthly climatology obtained from the ESA Scanning Imaging Absorption SpectroMeter for Atmospheric CHartographY (SCIAMACHY) (Eskes and Boersma, 2003). AERONET-Cimel V3 uses a geographic and temporally dependent multiyear monthly climatology from the Ozone Monitoring Instrument (OMI) NO2 concentration (Giles et al., 2019). In order to assess the contribution to AERONET-Cimel 1-minute AOD data non-traceability by NO$_2$ absorption we have to estimate the NO$_2$ optical depth ($\tau_{NO2}(\lambda)$) of AERONET-Cimel since GAW-PFR does not perform this correction. Analogously to $\triangle\tau_{O3}$, the differences in nitrogen dioxide optical depth $\Delta NO2$ can be obtained from:

$$\triangle\tau_{NO2} = \sigma_{NO2}(\lambda)\frac{1}{1000}\frac{m_{NO2}}{m_a}(-NO_{2Cimel}) \tag{13}$$

Where $m_a$ is given by Eq. 7, $NO_{2Cimel}$ (DU) is the daily total NO$_2$ used by AERONET-Cimel, $\sigma_{NO2}(\lambda)$ is the NO$_2$ absorption (Gueymard, 1995) weighted by the specific filter response: 15.6 cm$^{-1}$ (380 nm), 12.3 cm$^{-1}$ (440 nm), and 4.62 cm$^{-1}$ (500 nm). Finally $m_{NO2}$ has the following expression (Gueymard, 1995):

$$m_{NO2} = \frac{1}{sin\theta + 602.30(90-\theta)^{0.5}(27.96 + \theta)^{-3.4536}} \tag{14}$$

In Figure 6a the total NO$_2$ used by AERONET-Cimel to evaluate $\sigma_{NO2}(\lambda)$ is depicted. Figure 6b shows the $\triangle\tau_{NO2}$ caused by differences in daily total NO2 between GAW-PFR and AERONET-Cimel. $\triangle\tau_{NO2}$ is of the order of $10^{-3}$ for 380 and

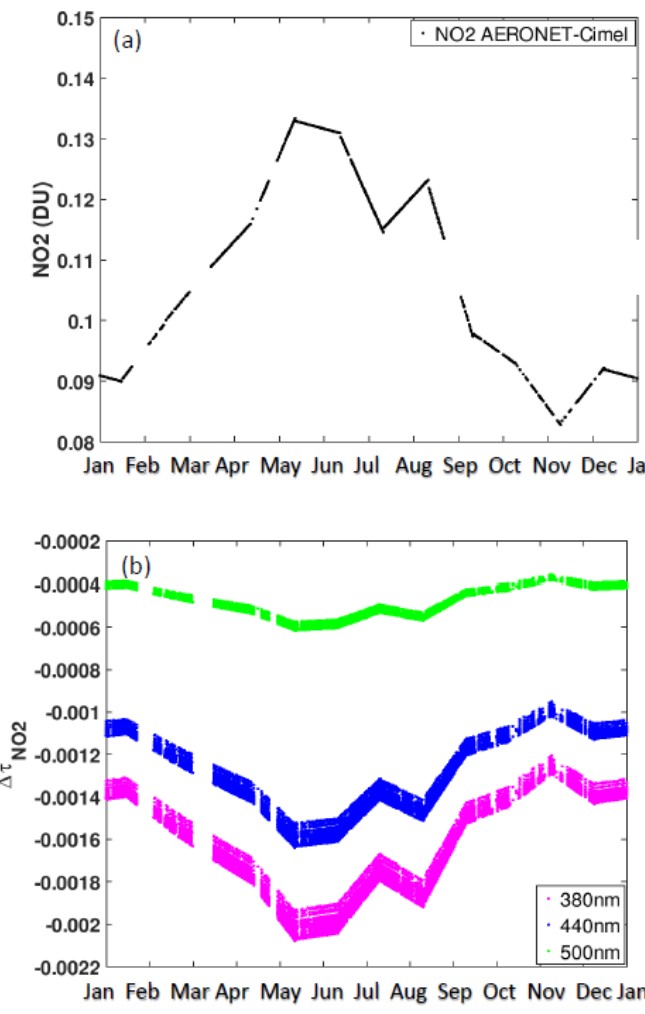

**Figure 6.** (a) $NO_2$ monthly climatology obtained from the ESA SCanning Imaging Absorption SpectroMeter for Atmospheric CHartographY (SCIAMACHY), used by AERONET-Cimel at IZO, and (b) $\triangle\tau_{NO2}$ caused by differences in daily total $NO_2$ between GAW-PFR and AERONET-Cimel in the period 2005-2015. Note that GAW-PFR does not take into account the correction for the $NO_2$ absorption.

**Table 9.** Percentage (%) of additional traceable AERONET-Cimel AOD 1-minute data (V2 and V3) after correcting by pressure, and total column $O_3$ and $NO_2$ for the period 2005-2015.

| Channel | Increment (%) of traceable AOD data after P, $O_3$ and $NO_2$ corrections | |
|---|---|---|
| | V2 | V3 |
| 380 nm | 1.3 | 1.7 |
| 440 nm | 0.2 | 0.3 |
| 500 nm | 0.3 | 0.1 |
| 870 nm | $\sim 0.0$ | $\sim 0.1$ |

**Table 10.** Percentage of AERONET V2 AOD data outside the U95 limits at 380nm, 440nm, 500nm and 870nm channels and for three AOD$_{500nm}$ thresholds respect to all data and respect to all data for each AOD interval (in brackets).

| | Percentage of AOD data outside the $U95$ limits (%) | | |
|---|---|---|---|
| | AOD$_{500nm} > 0.1$ | AOD$_{500nm} > 0.2$ | AOD$_{500nm} > 0.3$ |
| 380 nm | 1.9 (25.0) | 1.2 (47.2) | 0.5 (59.8) |
| 440 nm | 1.0 (13.5) | 0.8 (32.0) | 0.5 (57.6) |
| 500 nm | 0.6 (8.0) | 0.5 (18.7) | 0.3 (39.3) |
| 870 nm | 0.3 (4.1) | 0.2 (6.4) | 0.1 (14.0) |

440 nm channels, while, for 500 nm channel, it is of the order of $10^{-4}$. However, it should be noted that an impact on AOD calculation is expected when replicating similar analysis in highly NO$_2$ polluted regions. Such cases include large industrial cities from East Asia and Central and Eastern Europe (e.g., Chubarova et al. (2016)).

Taking into account the corrections for Rayleigh scattering and for the absorptions by O$_3$ and NO$_2$, we have calculated the additional traceable AOD data (Table 9). This percentage is maximum at 380 nm with 1.3 % (V2) and 1.7 % (V3) of the whole dataset. The 870 nm channel is only affected by the Rayleigh correction component and therefore the increment of traceable data after the mentioned corrections is minimal.

## 5.4 GAW PFR and AERONET-Cimel comparison under high AOD conditions: the impact of dust forward scattering for different FOVs

When we present the AOD differences between AERONET-Cimel and GAW-PFR versus AOD (GAW-PFR) for AOD > 0.1 (dusty conditions), we note that AERONET-Cimel shows slightly higher AOD values than GAW-PFR (Figure 7). Similar results for V3 are shown in S12. In fact, the percentage of data outside the $U95$ limits increases as AOD increases (Table 10), so for dust-related aerosol conditions (AOD$_{500nm} > 0.3$) the percentage of AOD data outside the U95 limits is > 50 % for 380 nm and 440 nm (Table 10, percentages in brackets). Similar results are found when using AERONET V3 (see Supplement S13). Taking into account the number of data compared with the total cases, these results show a small but non-negligible percentage of AOD differences outside the U95 limits for AOD > 0.1, ranging from $\sim 0.3$ % at 870 nm to $\sim 1.9$ % at 380 nm (Table 10).

Aerosol forward scattering within the FOV of various instruments and calculated AOD was investigated some decades ago by Grassl (1971) who determined that at AOD = 1 the circumsolar radiation increases by > 10 % the incoming radiation. Russell et al. (2004), using dust and marine aerosols data, quantified the effect of diffuse light for common sun photometer FOV. They reported that the correction to AOD is negligible (< 1 % of AOD) for sun photometers with narrow FOV (< 2°), which is greater than the Cimel FOV and slightly smaller than the PFR FOV (2.5°). Sinyuk et al. (2012) assessed the impact of the forward scattering aerosol on the uncertainty of the AERONET AOD, concluding that only dust aerosol with high AOD and low solar elevation could cause a significant bias in AOD (> 0.01).

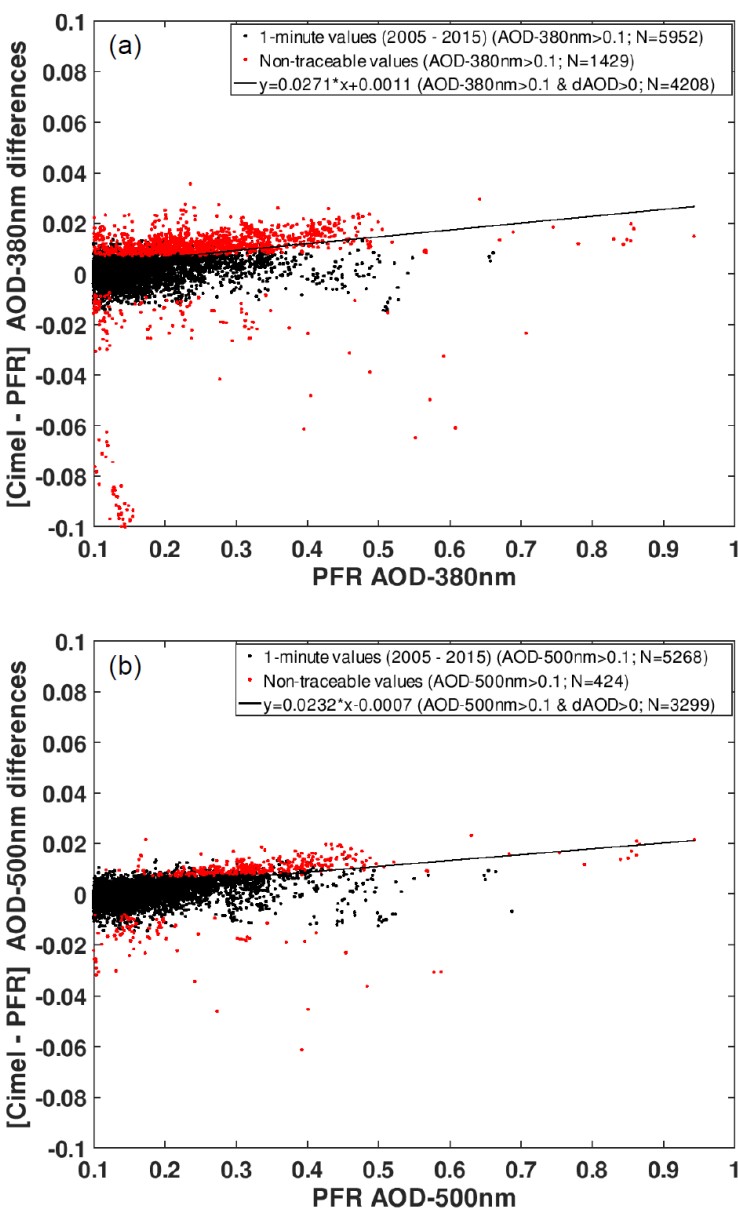

**Figure 7.** Actual AOD differences between AERONET-Cimel V2 and GAW-PFR vs AOD$_{PFR}$ at (a) 380 nm (b) and 500 nm for the period 2005-2015. The fitting line has been calculated with those data points with AOD > 0.1 and Cimel-PFR AOD difference > 0. The number of data used in the plots are indicated in the legend. The percentage of non-traceable AOD data with these conditions is $\sim 24$ % for 380 nm, and $\sim 8$ % for 500 nm. Note that some traceable (black) points show larger AOD differences than non-traceable (red) points because of the air mass dependence of the WMO traceability criterion.

GAW-PFR has double the FOV (2.5°; Wehrli (2000)) compared to the AERONET-Cimel (1.3° $\pm$ 4.8 %; Torres et al. (2013)), so it is reasonable to expect that it is more affected by the circumsolar radiation than the AERONET-Cimel radiometer. Taking advantage of the fact that Saharan dust intrusions regularly affect IZO, we provide a detailed analysis on the impact that dust forward scattering causes on the AOD retrieval of the two radiometers with different FOV, explaining the AOD differences under moderate-to-high dust load (AOD > 0.1) conditions. For this purpose we have used a forward Monte Carlo model (see Sect. 4.4) with which we perform simulations that include accurate dust aerosol near-forward scattering effects.

Dust aerosol single-scattering properties were computed using Mie theory, assuming a refractive index of 1.47+0.0025i at the wavelengths 380 nm, 440 nm and 500 nm and 1.46+0.012i at 870 nm, based on AERONET measurements at IZO. Seven values of aerosol effective radius ($r_e$) in the range 0.2 to 3.0 μm were considered, and a lognormal size distribution with a geometric standard deviation of 2 was assumed. A mid-latitude summer atmospheric profile starting from the Izaña altitude (2.4 km a.s.l.) was assumed, with the aerosol layer located at 5-6 km a.s.l. (typical of summertime). A spectrally uniform surface albedo of 0.11 was employed. Computations were performed for nine AOD values (AOD = 0, 0.1, 0.2, 0.3, 0.4, 0.5, 0.6, 0.8, and 1.0) and for five solar elevation angles ($\theta$ = 80°, 60°, 45°, 30° and 20°). The Monte Carlo model assumes a plane-parallel atmosphere, so the air mass factor is m = $1/\sin\theta$. Ten million photons were used for each case and wavelength.

Supplement S14 shows the ratio of scattered to direct radiation for cases with AOD up to 0.5. The ratio increases with increasing $r_e$, as the aerosol forward-scattering peak grows stronger. In the case of Saharan dust intrusions at IZO, the median $r_e$ determined from both AERONET data inversion and the in-situ aerodynamic particle sizer (APS) analyzer is $\sim$ 1.5 m. This value agrees with the dust size distribution found during SAMUM-2 during long-range transport regime (Weinzierl et al., 2011). For this particle size, the ratio of scattered to direct radiation is $\sim$ 3 times larger for FOV of 2.5° than FOV of 1.3°.

The error in the retrieved AOD due to scattered radiation within the instrument FOV was evaluated by comparing the apparent AODs, defined as:

$$AOD_{app,PFR} = -\frac{1}{m} ln \frac{F_{PFR}}{F_{PFR}(AOD=0)} \tag{15}$$

$$AOD_{app,Cimel} = -\frac{1}{m} ln \frac{F_{Cimel}}{F_{Cimel}(AOD=0)} \tag{16}$$

with the true AOD

$$AOD_{true} = -\frac{1}{m} ln \frac{F_{dir}}{F_{dir}(AOD=0)} \tag{17}$$

Here, $F_{dir}$ is the irradiance due to direct (i.e., non-scattered) radiation, and $F_{PFR}$ ($F_{Cimel}$) is the total irradiance that would be measured by the PFR (Cimel) radiometer, considering the instrument FOV and the FOV angular function. The relative error in AOD depends strongly on the particle size but it is fairly constant for each $r_e$ value considered (see Supplement S15). For $r_e$ $\sim$ 1.5 $\mu$m, the relative error in AOD at 380 nm (500 nm) is $\sim$ 1.6 % (1.0 %) for Cimel, and $\sim$ 5 % ( $\sim$ 3 %) for PFR. These errors are in good agreement with those estimated by Russell at al. (2004), and slightly higher than the relative AOD error of 0.7 % due to coarse dust aerosol forward scattering reported by Eck et al. (1999).

The Monte-Carlo-simulated relative differences in retrieved AOD (in %) that would result from the scattered radiation within the FOV of the PFR and Cimel instruments, and the difference in retrieved AOD between PFR and Cimel as a function of the AOD retrieved with PFR, for 380 nm and 500 nm, are shown in Figure 8. The main results of these simulations are: 1) the higher FOV of the PFR, compared to that of the Cimel, results in lower AOD values for the PFR; 2) the fractional AOD difference related to the different FOVs of PFR and Cimel is fairly constant for any aerosol effective radius, but increases with

increasing the effective radius; and 3) this fact might explain at least some of the systematic differences seen in Figure 7. Note that, as lower AOD values derived from the PFR are expected based on its larger FOV, the linear fits in Figure 7 have been calculated for those data points with the Cimel-PFR AOD differences> 0. In this way, we discard those pairs of AOD data whose difference is not only due to the different FOV between both instruments, obtaining in this way a better approximation to quantify this effect.

The slopes of the fitting lines of the Cimel-PFR AOD differences vs. PFR AOD for AOD> 0.1 (dusty conditions) are 2.7 % for 380 nm and 2.3 % for 500 nm (Figure 7), which are quite consistent with the percentage differences of AOD between Cimel and PFR for an effective radius of 1.5 $\mu$m (Figures 8a and 8b). These percentages correspond to absolute AOD differences of 0.016 at 380 nm, and 0.011 at 500 nm for AOD = 0.5 (Figure 8c and 8d), that are of sufficient magnitude to cause an appreciable number of 1-minute AOD data outside the $U95$ limits, as indicated in Table 10.

If we apply the corresponding corrections to the 1-minute AOD PFR data > 0.1 assuming an effective radius of 1.5 $\mu$m, + 3.3 % at 380 nm and + 2.2 % at 500 nm, it turns out that the slopes of the fitting lines of the Cimel-PFR AOD differences vs. PFR AOD become practically zero (Figure 9). Moreover, the number of AOD data outside the U95 limits is reduced by approximately 53 % for 380 nm and by 13 % for 500 nm. It must be taken into account that the percentage of AOD data for AOD > 0.1 outside the $U95$ limits, before the corrections, is only 8 % at 500 nm, while at 380 nm it is a significant value (24

655    %).

This AOD "correction" reduces the Cimel-PFR AOD differences substantially but does not eliminate them completely. The main reason is the inherent limitation of data correction using the percentage difference in AOD obtained by model simulation for a fixed effective radius. We have assumed an effective radius of 1.5 $\mu$m but, in reality, the radius of dust particles varies. A reasonable range of dust particle radius is between 0.1 and 3 $\mu$m (Balkanski et al., 1996; Denjean et al., 2016; Mahowald et al.,

2014). So, depending on the distance from the dust source to IZO and the size of the emitted dust, the effective radius could vary slightly between dust episodes. As can be seen in Figures 8a and 8b, the percentage differences in AOD between Cimel and PFR for a 1-2 $\mu$m effective radius interval, the PFR-Cimel AOD relative difference at 380 nm (500 nm) might change between ~ -1.8 % (-1.1 %) to -4.9 % (3.3 %).

A similar analysis has been carried out for AERONET V3 (see Supplement S16), where we observe that the corrections

obtained are not as good as those obtained for V2. The effect of FOV on AOD retrieval should be taken into account for those radiometers with a relatively high FOV (> 3°) measuring in regions with relatively high AOD (> 0.2) for most of the year, as is the case in many sites of Northern Africa, the Middle East and East Asia (Basart et al., 2009; Cuevas et al., 2015; Eck et al., 1999; Kim et al., 2007). This effect leads to AOD underestimation, and the variable number of high AOD episodes in each

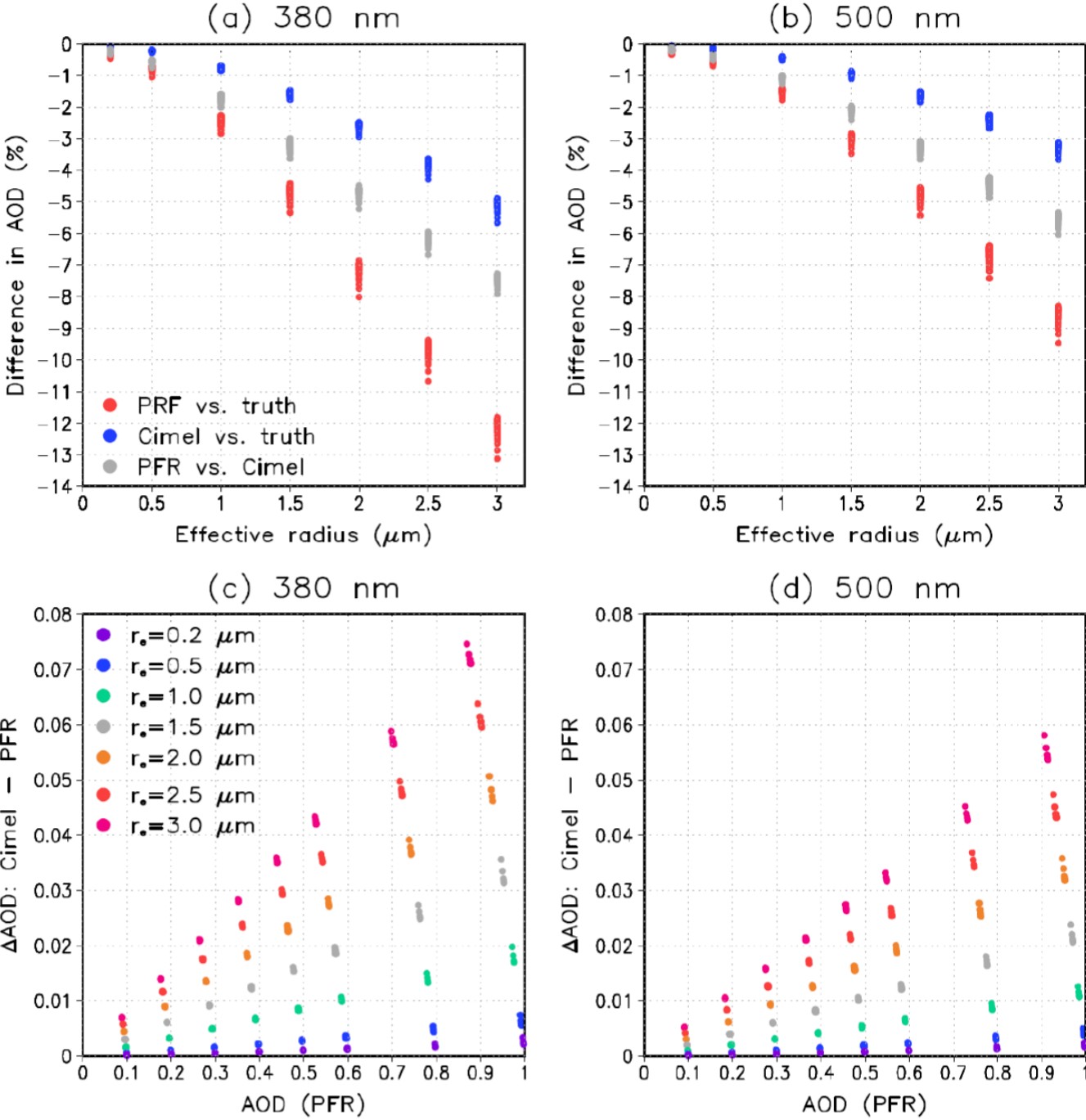

**Figure 8.** Panels a) and b): the simulated relative differences in retrieved AOD (in %) that would result from the scattered radiation within the FOV of the PFR and Cimel instruments. The red (blue) dots show the differences between the AOD that would be retrieved using PFR (Cimel) and the actual AOD, and the grey dots the difference between PFR and Cimel, at wavelengths (a) 380 nm and (b) 500 nm. Panels c) and d): the difference in retrieved AOD between PFR and Cimel, plotted as a function of the AOD retrieved with PFR, for seven values of aerosol effective radius between 0.2 and 3.0 $\mu$m, at (c) 380 nm and (d) 500 nm.

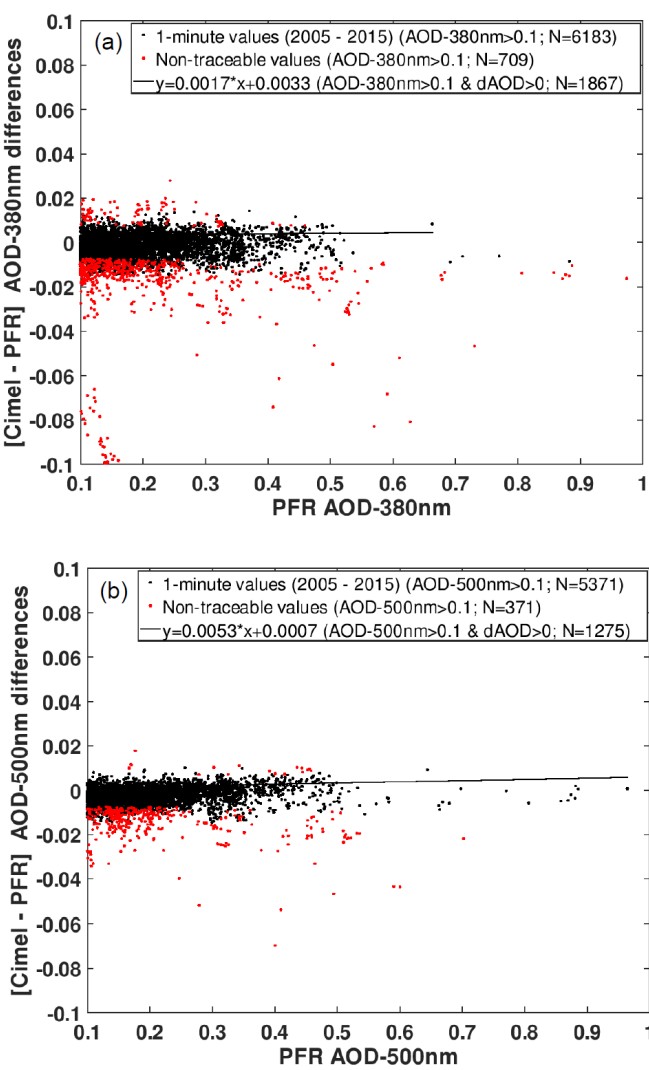

**Figure 9.** The same as Figure 7 after "correcting" the PFR AOD data by adding + 3.3 % at 380nm and + 2.2 % at 500 nm to the 1-minute PFR AOD data > 0.1.

season of the year might affect the AOD long-term trends. AOD measurements under these conditions would be especially affected for optical air mass < 3.

## 5.5 Angström exponent comparison

We have performed a comparison of the AE provided by GAW-PFR and AERONET-Cimel using in both cases the AOD data obtained from the four common channels (380 nm, 440 nm, 500 nm and 870 nm) with a total of 70716 data-pairs. The PFR-AOD values have been ordered from lowest to highest by grouping them in intervals of 500 values for which the averages (and

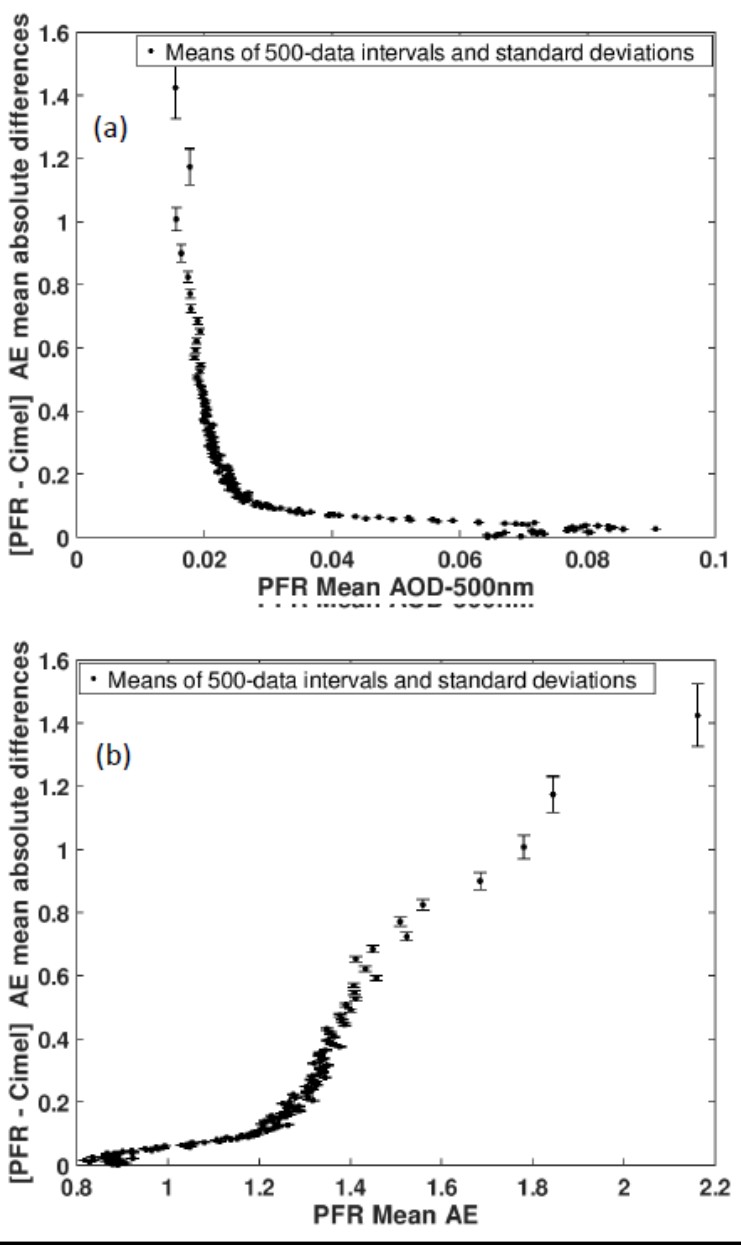

**Figure 10.** (a) PFR-Cimel AE mean absolute differences (and corresponding standard deviations) vs PFR mean $AOD_{500nm}$ in 500 data intervals (b) and vs PFR mean AE in 500 data intervals. AE has been computed for both PFR and Cimel using the four common channels (380nm, 440nm, 500nm and 870nm).

corresponding standard deviations) of the PFR-Cimel AE differences have been calculated (Figure 10a). In a similar way we proceeded with the PFR-AE values (Figure 10b).

**Table 11.** Uncertainty in AE determination for three typical atmospheric situations.

| | Uncertainty in AE |
|---|---|
| Normal pristine conditions<br>$AOD_{500nm} = 0.03$ and AE = 1.4 | $\geq 1$ |
| Hazy conditions<br>$AOD_{500nm} = 0.14$ and AE = 1.15 | $\geq 0.2$ |
| Strong dust intrusion<br>$AOD_{500nm} = 0.3$ and AE = 0.3 | $\sim 0$ |

AE differences > 0.2 increase exponentially for AOD < 0.02, reaching AE differences of up to 1.6 under pristine conditions (Figure 10a). For very low AOD the provided instruments uncertainty is the source of the sharp increase in AE, and at the same time AE becomes very sensitive to slight AOD changes. However, for AOD < 0.02 the atmospheric aerosol load is practically zero and so, its characterization with AE have in practice relatively minor importance.

In addition, the AE differences remain < 0.1 when $AE_{PFR}$ values are < 1 (Figure 10b), which shows that these differences are small in most of the possible atmospheric scenarios. For $1 < AE_{PFR} < 1.2$ the AE differences increase slightly to values < 0.2, and for $AE_{PFR} > 1.2$ (very fine particles or pristine conditions) the AE differences increase sharply to reach values of $\sim 1.2$. In our case, the non-pristine conditions, or those with a high content of mineral dust, have associated AOD > 0.03 and AE < 1, where the AE differences remain < 0.1. In case of pristine conditions AOD $\leq 0.03$ and AE $\geq 1$ the AE differences can reach a maximum of 1.6. Wagner and Silva (2008) estimated the usual maximum AE error by error propagation using a pair of spectral channels in which AOD is measured. Their results show that for clean optical conditions ($AOD_{440nm} = 0.06$) the maximum AE error is 1.17, and for hazy conditions ($AOD_{440nm} = 0.17$) the error is 0.17, assuming an underlying AE of 1.5. These values decrease to 0.73 and 0.11, respectively, if AE = 0. The AE differences found between GAW-PFR and AERONET-Cimel lie within the estimated errors reported by Wagner and Silva (2008).

In any case, as in our study the AE has been determined from AOD measured in the four common channels of GAW-PFR and AERONET-Cimel, we estimated the uncertainty in the calculation of the AE for three typical aerosol scenarios at Izaña. Following the methodology shown by Wagner and Silva (2008), the AE uncertainty estimations have been calculated using AOD measurements at four wavelengths and AOD uncertainty error propagation (Table 11). The AE derived from more than two wavelengths is less affected by AOD uncertainties than AE calculated with pairs of wavelengths, since the latter are calculated from the ratio of AOD at two channels (Cachorro et al., 2008).

The AE differences of our study (Figure 10) are within the AE uncertainty estimated for each type of atmospheric condition (pristine, hazy and heavily dust loaded). However, although AE is a quantitative parameter, it is only used in a qualitative way to estimate the range of sizes (fine, medium, coarse) of the predominant aerosol in the inevitable mixture of aerosols that we observe. With this parameter, and together with the information that is available in the measurement site about the most frequent types of aerosols and their concentration, we can estimate the type of aerosols that are being measured. There are many publications with different thresholds of AE and AOD in order to classify different types of aerosols (e.g. Basart et al. (2009);

Cuevas et al. (2015); Dubovik et al. (2002); Guirado et al. (2014); Holben et al. (2001); Kim et al. (2007); Todd et al. (2007); Toledano et al. (2007); Wang et al. (2004)). However, there is no consensus on these thresholds since at each site there are different mixtures of aerosols, and each type of aerosol shows specific frequencies of appearance and different concentrations. An alternative way of analyzing the degree of agreement in AE between GAW-PFR and AERONET-Cimel is to verify to what extent both networks provide the same information regarding the type of aerosol they observe in a certain site.

Considering the AE criteria established by Cuevas et al. (2015) and Berjón et al. (2019), we have identified the following four main categories according to the $AE_{PFR}$ and $AE_{Cimel}$ values:

1. $AE_{PFR}$ & $AE_{Cimel} > 0.6$: Pristine conditions.

2. $0.25 < AAE_{PFR}$ & $AE_{Cimel} \leq 0.6$: Hazy, mineral dust being the main aerosol component.

3. $AE_{PFR}$ & $AE_{Cimel} \leq 0.25$: Pure dust.

4. $AE_{PFR}$ & $AE_{Cimel}$ do not fit any of the previous categories.

In 94.9 % of the cases, GAW-PFR and AERONET-Cimel V2 match the AE intervals of each aerosol scenario. Similar results (93.4 %) were obtained when comparing with AERONET V3. Most of the agreement (>80 %) occurs in the predominant scenario of pristine conditions despite the AE uncertainty under pristine conditions being $\geq 1$. See Supplement S17 for more details. Note that the choice of these categories is not relevant since this is only used to examine the long-term agreement in AE between GAW-PFR and AERONET-Cimel in different atmospheric conditions.

## 6 Summary and Conclusions

While GAW-PFR is the WMO-defined global AOD reference, being directly linked to WMO / CIMO, and was specifically designed to detect long-term AOD trends, AERONET-Cimel is the densest AOD measurement network globally, and the network most frequently used for aerosol characterization and for model and satellite observation evaluation.

An AERONET-Cimel 11-year AOD data series at IZO was obtained using a large number of radiometers. A total of 13 Reference instruments were used in the period 2005-2009 which means that every 4 and a half months, approximately, an instrument was replaced by another one to be calibrated. Their calibrations were performed during their respective measurement time periods at IZO. Therefore, these calibrations were not in any way linked with those of the instruments that preceded or replaced them, nor with GAW-PFR reference. These facts led us to investigate the homogeneity of the AERONET-Cimel AOD data series and their intercomparability with the much more homogeneous AOD data series from GAW-PFR (3 instruments in 11 years). The traceability concept for AOD suggested by WMO consists in determining whether the AOD difference of the AERONET CIMELs vs the GAW PFRs lie within the $U95$ limits. We have used uncertainty limits for AOD traceability established by WMO (2005) for these type of instruments with finite FOV. The acceptable traceability is when 95 % of the absolute AOD differences lie within these limits, in which case both data populations are considered equivalent. It should be clarified that "traceability" is not used in a strict metrological sense. This study has addressed the comparison of the GAW-PFR

dataset with the two versions of AERONET (V2 and V3) in the period 2005-2015. An excellent agreement between V2 and V3 for the four analysed channels ($R^2 > 0.999$) has been obtained.

More than 70000 synchronous GAW-PFR (PFR) and AERONET-Cimel (Cimel) 1-minute data-pairs in each channel in the period 2005-2015 were analysed. An excellent traceability of AOD from the AERONET-Cimel (V2 and V3) is found for 440 nm, 500 nm and 870 nm, and fairly good results for 380 nm. The lowest percentage of traceable AOD data is registered in 380 nm with 92.7 % of the 1-minute data within the WMO limits, and the highest in 870 nm with 98.0 % of the data.

The different possible causes of non-traceability in AOD were investigated as follows:

- Absolute AOD measurements synchronization.

  Analyzing 1-minute AOD variability we concluded that its impact on the AOD differences is quite small as only $\sim 0.8$ % of the AOD data has a variability larger than 0.005 in all spectral ranges.

- Sun tracking misalignments.

  Sun tracking misalignments constitute a serious problem and a major cause of non-traceability of AOD data-pairs as demonstrated by the AOD data outside the U95 limits from the period 2005-2009 as a consequence of episodic problems with the sun-tracker of the GAW-PFR radiometer. For the 2010-2015 period the percentage of traceable data-pairs improves to 93.5 % (380 nm), 97.4 % (440 nm), 97.2 % (500 nm) and 99.1 % (870 nm). However, most of these cases could be identified and excluded from the analysis.

- Cloud screening failure by both network algorithms.

  According to our observations, the simultaneous failure of both cloud screening algorithms might occur only under the presence of large and stable cirrus. In these cases, the radiometers interpret these clouds as aerosol layers and might provide values very different from the real AOD. For the comparison at IZO, however, this effect is negligible since GAW-PFR and AERONET-Cimel cloud screening algorithms provide successful cloud identification on clear direct-sun conditions during cloudy skies (FCS < 40 %) for 99.75 % of the cases.

- Pressure measurements related errors.

  Since the accuracy of the new barometers built into new radiometers is about 3 hPa, and only errors in atmospheric pressure > 30 hPa might produce an impact on Rayleigh scattering, the AOD non-traceability due to errors in Rayleigh scattering is negligible.

- Total column ozone input uncertainty.

  The largest influence of total ozone data uncertainty on ozone absorption occurs mainly at 500 nm. Total ozone needs to be determined to $\pm$ 30 DU or 10 % of typical values, to ensure an uncertainty of ±0.001 ozone absorption at 500 nm. In the case of the GAW-PFR / AERONET-Cimel comparison, despite the very different methods in which both networks obtained values for their corresponding corrections, large ozone differences were found (> 40 DU) only on 2.4 % of the days, resulting in a difference in the ozone optical depth slightly above $\sim$ 0.001. The potential contribution

to non-traceable AOD values between the two networks is negligible. However, in mid or high latitude stations where fast $O_3$ variations of several tens of DU might be registered, the correction of 1-minute AOD measurements by ozone absorption might be an issue to be considered.

- Total column $NO_2$ input uncertainty.

The differences in $NO_2$ absorption caused by differences in daily total $NO_2$ between GAW-PFR and AERONET-Cimel is of the order of $10^{-3}$ for 380 nm and 440 nm channels, while, for 500 nm channel, it is even lower, of the order of $10^{-4}$. So, differences in $NO_2$ absorption are negligible in the 1-minute AOD non-traceability of our study. However, $NO_2$ absorption might have some impact on AOD in highly polluted regions, such as in large industrial cities, where column $NO_2$ values are much larger than the climatological ones.

Taking into account the corrections for Rayleigh scattering and for the absorptions by $O_3$ and $NO_2$, we have calculated the combined effect of all of them on the non-traceability of the 1-minute AOD values. The highest impact occurs in the 380 nm channel, in which 25 % of the AOD data outside the U95 limits ($\sim$ 2 % of the total compared data) are due to significant differences in pressure, and in $O_3$ and $NO_2$ absorption. The 1-minute AOD data outside the $U95$ limits by these corrections is negligible in the 870 nm channel.

- Impact of dust forward scattering in AOD retrieval uncertainty for different instrument FOVs.

Since GAW-PFR has almost double the FOV ($\sim 2.5°$) compared to the AERONET-Cimel ($\sim 1.3°$), and direct solar irradiance measurements are biased by the amount of aureole radiation that is assumed to be direct solar radiation, it is reasonable to expect that GAW-PFR is more affected by the circumsolar irradiance than AERONET-Cimel radiometer when AOD is relatively high. Modelling the dust forward scattering we have shown that a non-negligible percentage of the non-traceable 1-minute AOD data for AOD > 0.1, ranging from $\sim$ 0.3 % at 870 nm to $\sim$ 1.9 % at 380 nm is caused by the different FOV. Due to this effect, the GAW-PFR provides AOD values which are $\sim$ 3% lower at 380 nm, and $\sim$ 2 % lower at 500 nm, compared with AERONET-Cimel. However, AOD underestimation could only have some relevance in dusty regions if radiometers with relatively large FOV are used.

A comparison of the AE provided by GAW-PFR and AERONET-Cimel has been performed using in both cases AOD data obtained from the four nearby common channels with a total of 70716 data-pairs. This is a very strict AE calculation since it is necessary that AOD be accurately measured by the four channels simultaneously. AE differences > 0.2 increase exponentially under very pristine conditions (AOD $\leq$ 0.03 and AE $\geq$ 1), reaching AE differences of up to 1.6. However, for these conditions the atmospheric aerosol load is practically zero and so, its characterization with AE does not have any importance in practice. Under non-pristine conditions or those with a high mineral dust content (associated AOD > 0.03 and AE < 1), the AE differences remain < 0.1.

Summarizing, we have presented for the first time a long-term (2005-2015) 1-minute AOD comparison among different types of radiometers belonging to different aerosol global networks. This comparison is a very demanding test of both GAW-PFR and AERONET-Cimel validated AOD datasets since aerosol scenarios correspond to extreme conditions: either very low

aerosol loading, a "pristine" scenario that reveals small uncertainties in the calibration and in the cloud screening, or large dust load, which leads to a significant increase in the forward scattering aerosol with AOD, resulting in a slightly higher AOD underestimation by the GAW-PFR. From this comprehensive comparison, we can conclude that both AOD datasets are representative of the same AOD population, which is a remarkable fact for the global aerosol community. It should be noted that AOD traceability at 380 nm (92.7 %) does not reach 95 % of the common data, the percentage recommended by WMO $U95$ criterion, so more efforts should be made to improve AOD in the UV range. In this study we have also investigated the data that are outside of the WMO $U95$ limits in order to understand their causes and to be eventually able to correct the small inconsistencies detected in instrumental and methodological aspects in the future.

Our results suggest that WMO/CIMO traceability limits could be redefined as a function of wavelength, and the recommended radiometer FOV range should be reconsidered. The widely deployed AERONET-Cimel and GAW-PFR datasets play a crucial role in understanding long-term AOD changes and detecting trends, so it would be desirable for both networks to be linked to the same GAW-WMO related reference.

*Code availability.* TEXT

*Data availability.* TEXT

The AERONET V2 and V3 1-minute AOD data from the Izaña station ("Izana") are available in the AERONET data repository at https://aeronet.gsfc.nasa.gov (last access: 15 July 2019). The Izaña's GAW-PFR 1h resolution AOD data are available in the World Data Center for Aerosols (WDCA) through http://ebas.nilu.no (last access: 15 July 2019). Evaluated GAW-PFR 1-minute AOD data has been processed at PMOD/WRC. For further information, please contact Dr. Stelios Kazadzis and Dr. Natalia Kouremeti.

*Code and data availability.* TEXT

*Author contributions.* TEXT

Emilio Cuevas, Pedro M. Romero-Campos, Natalia Kouremeti and Stelios Kazadzis conceived and designed the structure and methodology of the paper and wrote the main part of the manuscript, being coordinated by Emilio Cuevas. Pedro M. Romero-Campos evaluated the AERONET V2 and V3 1-minute AOD and AE data and performed the data comparison with GAW-PFR. Natalia Kouremeti processed the assured quality GAW-PFR 1-minute AOD and AE data, and analyzed the short term AOD natural variability from the GAW-PFR 1-minute AOD data, and was in charge of the GAW-PFR radiometers calibrations. Petri Räisänen performed and discussed the modelling results with the Monte Carlo forward model. Rosa D. García

contributed to the dust forward scattering section and prepared the paper edition in Latex. Africa Barreto obtained the results of the 1-minute AOD natural variability from the Cimel Triplets. Carmen Guirado-Fuentes was responsible for the careful calibration of Cimel Masters. Rosa D. García, Africa Barreto, Carmen Guirado-Fuentes, and Pedro M. Romero-Campos participated actively in the analysis of the case studies shown in the Supplement material. Ramón Ramos coordinated the maintenance
and daily checks of the PFR and Cimel radiometers since 2005, resolving technical and logistical problems. Carlos Toledano contributed to the GAW-PFR and AERONET-Cimel calibration aspects addressed in the paper. Fernando Almansa provided detailed information of the Cimel radiometers, taking care of their complex technical problems. Julian Gröbner contributed with concepts on traceability and comparability used in the paper. Emilio Cuevas, Julian Gröbner, Stelios Kazadzis and Petri Räisänen provided financial support and/or resources. All authors discussed the results and contributed to the final paper.
Emilio Cuevas and Stelios Kazadzis supervised this research activity.

*Competing interests.* The authors declare that they have no conflict of interest

*Disclaimer.* TEXT

*Acknowledgements.* The authors thank Luc Blarel and Philippe Goloub (LOA, CNRS-University of Lille, France) for supervising the periodic calibrations of the Cimel Reference instruments. This study has been performed in the frame of the WMO CIMO Izaña Testbed for
Aerosols and Water Vapour Remote Sensing Instruments. SK and NK would like to thank the Federal Office of Meteorology and Climatology MeteoSwiss International Affairs Division, Swiss GCOS Office for financing the project "The Global Atmosphere Watch Precision Filter Radiometer (GAW-PFR) Network for Aerosol Optical Depth long term measurements". Part of the AERONET-Cimel radiometers have been calibrated at Izaña Observatory by AERONET- EUROPE Calibration Service, financed by the European Community specific programs for Integrating Activities: Research Infrastructure Action under the Seventh Framework Programme (FP7/2007-2013), ACTRIS grant agree-
ment No. 262254. This research has received funding from the European Union's Horizon 2020 Research and Innovation Programme under grant agreement No. 654109 (ACTRIS-2). The funding by MINECO (CTM2015-66742-R) and Junta de Castilla y León (VA100P17) is also gratefully acknowledged. We thank the staff of the Izaña Observatory for their effort and dedication in maintaining the instruments. We acknowledge the constructive comments of the anonymous referees. Our colleague Celia Milford has improved the English language of the paper. In memory of Prof. Klaus Fröhlich, former director of PMOD-WRC, who initiated the AOD measurements programme at the Izaña
Observatory in 1984 within the WMO Background Atmospheric Pollution Monitoring Network (BAPMoN).

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
