# Peer review of "Aerosol Optical Depth comparison between GAW-PFR and AERONET-Cimel radiometers from long-term (2005-2015) 1-minute synchronous measurements"

_Atmospheric Measurement Techniques, 2018_

## Referee Comment (RC1) · Anonymous Referee #1 · 8 Jan 2019

This is a very thorough and overall quite clearly written manuscript of great use to anyone relying on PFR or Aeronet derived AODs. It establishes how well the master instruments used in both networks compare over the long term and where and why minute differences arise. It does not address how well these calibrations are transferred across the networks and what the resulting uncertainties are for various stations. However, such questions have been addressed by many others as evidenced by citations in this manuscript. This reviewer only has a very small number of relatively minor comments

Page 1, Line 4. Suggest to say "wavelength near" instead of "nearby wavelength"

Page 2, Line 15. Suggest mentioning some other sunphotometer networks, PHOTON, Japanese network etc.

Page 2, Line 30. Suggest inserting "NASA" before "Goddard"

Page 3, Line 6. Numerous is an understatement.

Page 4, Line 5. The use of "absolute" is misleading here. A Langley calibration alone is never absolute (i.e. the calibration value is just a signal in an engineering unit not W/m2. The beauty is that an absolute calibration is not needed to derive AOD.

Page 5, Line 5. Suggest replacing "constant" with "signal" here and everywhere else in the manuscript. Or maybe just explain once that in sunphotometry the calibration "constant" is the signal the instrument would read outside the atmosphere (extraterrestrial) at a normalized earth-sun distance?

Page 6, Line 24. Font issue.

Page 6, Line 6. Incorrect grammar: "were" not "was".

Page 14: Line 14. This is misleading as the error in AOD due to error in the calibration "constant" is independent of AOD.

---

## Referee Comment (RC2) · Anonymous Referee #2 · 10 Jan 2019

The paper "Aerosol optical Depth comparison between GAW-PFR and AERONET-Cimel radiometers from long term ( 2005-2015) 1-minute synchronous measurements" is very interesting for the scientific community working on photometry. It is really well written, very accurate in the analysis of all the aspects affecting the comparison and it is pleasing to read. I recommend the publication on this journal also because homogenization of international networks of photometers is an important issue at this stage of research.

I have few specific comments for the authors: 1. Introduction, lines 12-15. It

is not true that only two global ground-based radiometer networks exist, that is AERONET and GAW. SKYNET ( https://www.skynet-isdc.org/) and its regional sub-network ESR ( www.eroroskyrad.net) provide centralized AOD and other optical and physical aerosol parameters, on a daily bases and downloadable from the website. SKYNET is also attending several intercomparison campaigns against PFR as example www.eroroskyrad.net/quatram.html, and the Fourth WMO Filter Radiometer Comparison for aerosol optical depth measurements. So I'd suggest the authors to mention their existence.

3.GAW-PFR and AERONET-Cimel radiometers, line 21: specify if these Cimel models have one sensor for both direct and diffuse solar radiation ( new models) or two different sensors ( old version). Lines 21-24, the deterioration of the filter is however "not minimized" by the absence of a system for keeping the temperature constant inside the optics. Page 6 line 1-3, it is better stating also here that the final measurement is an average of the 10-s measurements, even if declared later. Line 12, is not the triad of PFR calibrated by lamps, but using Langley plot?

4. Data and methodology used in this study Line 24, format problem Line 33, state here that in the section 5.4 AE will be compared

5.1 AERONET-Cimel AOD traceability Line 5, I don't know if it is easy to represent, but in Figure1 it would be interesting to highlight the changes of equipment during the time.

5.2.1 Calibration related errors line 16: are all the involved Cimel "Maters"?; "in-situ absolute calibration": with absolute do you mean by lamp? Lines 20-21: It is not clear why you transfer calibration among Cimels if they are all masters, and therefore calibrated separately. Figure 2: why there is a hole of data for optical mass at about 4.2?

5.2.3 Did you consider a possible influence of WV absorption at 500 nm? Small differences came out during the Fourth WMO Filter Radiometer Comparison. Pag 19 line 11 Cimel doesn't measure pressure, so better saying " from the different way the two

instruments assume the atmospheric pressure".

Pag 24 line 9 : Figure 6 is not about the combined effect of the 3 components, but about NO2.

---

## Referee Comment (RC3) · Anonymous Referee #3 · 11 Jan 2019

Three major data interpretation issues discourage publication. They are about natural variability, sampling bias and instrument field of view (FOV).

First, the paper compares AOD observations between 60s averages and less-than-1s averages (Page 8, line 1-2). The true AOD in the atmosphere generally changes over the 59s differentials. Yet the paper neglects the natural changes when making inferences on calibration.

For example, the paper shows that Cimel observes generally wider AOD differences

between daily minimum and maximum than does PFR (page 15, line 21-24; Figure 3). This must be at least partly because Cimel samples with 30s intervals and captures natural changes while PFR sleeps. Yet the paper interprets the wide diurnal ranges as a sign of "an imperfect calibration" of Cimel (page 15, line 24-25). The comparison is fairer if PFR data are paired up over 61s and, better still, if the second of the three consecutive Cimel samples is excluded.

Misinterpretation is evident on a multi-year basis too. The paper attributes poor AOD agreements at 380 nm under pristine conditions to "insufficiently accurate calibration of AERONET-Cimel" (page 15, line 33-34). This is unsubstantiated. Because AOD is generally higher at shorter wavelengths, so is its natural variability in the absolute term. This would make the AOD discrepancies greatest at the shortest wavelength, even if calibration were perfect for both instruments.

Second, the paper finds "a bias with positive large outliers (higher Cimel AOD)" (page 15, line 5; Figure 2). This results from the intentional exclusion of high PFR values, an attempt to assess "pristine conditions (AOD500nm <= 0.03)" (page 13, line 26, 28). It is misleading to use this assessment to suggest contaminations on Cimel (page 15, line 6, 7). It is only fair to explicitly state that many negative outliers (higher PFR AOD) exist for PFR AOD just above 0.03 (shown in Figure 7).

Third, FOV is not adequately appreciated as a significant source of AOD uncertainty under dust. The forward scattering by aerosols into FOV is definitely the primary cause for the poor agreement at 380 nm away from pristine conditions, not just "might be" (page 33, line 30). A support for this statement partly comes from the theory that the forward scattering is greater at shorter wavelengths.

A more definitive support is given in Figure 7. It shows that the observed AOD differences is about 3% of AOD at 380 nm. Russell et al. [2004] explain why the FOV effect increases with AOD. And their calculation predicts a ∼4% error for 1.25 degree half FOV and a ∼1% error for 0.60 degree half FOV, both at 380 nm (their Figure 3). The
difference, ∼3%, is similar to Figure 7. It is incorrect to assign "an insufficient[ly] accurate calibration of AERONET-Cimel in this channel" as "the most likely cause" (page 32, line 9) of the observed differences (page 32, line 9). Such errors would not increase with AOD.

FOV will remain as a source of uncertainty even if the adjustments are made for it. Because the FOV is much wider, the adjustments are greater for PFR than for Cimel. So is the uncertainty in them. Thus, other things being equal, the PFR is destined to be more erroneous than Cimel. It is incorrect to hold both instruments equally responsible for the resulting AOD discrepancies (page 34, line 8-9).

Also, the FOV-related uncertainty leads to a question as to why "WMO defines the PFR FOV as the recommended one for sun radiometers" (page 33, line 24-25). The quoted statement is not explained. Forward scattering into the FOV constitutes a deviation from the condition to which Beer's law pertains – a deviation that should be minimized, not recommended. Nor is the statement supported by a citation. To be sure, WMO (2007) recommends that the WORCC be designated the primary WMO Reference Centre for OD measurements, as referenced in page 2, line 24-26. But the WMO report does not mention a specific instrument, let alone support the PFR-specific statement.

Meanwhile, there are a few reasons to encourage resubmission. The topic is important, given how widely ground-based sunphotometers are used in climate science and satellite validation. The data are abundant. Careful and useful pieces of discussion are provided regarding the impact of unattended operations (page 3, line 18-21) and the implications for other measurement sites (page 24, line 4).

Authors are encouraged to consider the following suggestions and comments, in addition to the major issues raised above, prior to resubmission.

Page 2, line 16. Skynet with PREDE instruments is worth mentioning as another ground-based sunphotometer/sky-radiometer network. [Takamura, T., and T. Nakajima, Overview of SKYNET and its activities, Opt. Pura Apl. 37, 3303-3308, 2004.,

http://atmos3.cr.chiba-u.jp/skynet/index.html]. Skynet has about as many stations as GAW-PFR does.

Page 4, line 32. Separate vice from versa.

Page 5, line 1. Insert something like "Dust provides a good test on the treatment of the forward scattering into the field of view."

Page 5, line 14. The precise sun-tracking enabled by a quad detector should be mentioned somewhere in this paragraph for Cimel. Sun tracker is only described for PFR (page 6, line 1).

Page 5, line 27. Is there a reference on the instrument response over the field of view, especially over the 0.7 degree "slope" (Table1)?

Page 5, line 28. Replace significant with significantly.

Page 6, line 1. Replace the second comma with a period.

Page 6, line 4-7. The air mass dependence of uncertainty is worth mentioning here, since radiometric calibration is the primary concern of this paper.

Page 6, line17. Make 0 subscript, as in line 16.

Table 1. What does "No specific Sun tracker" mean? What does "Sun tracker robot" refer to? Replace "long-term" with 6 months, as described in page 6, line 18. Replace "2-3 months" with 3-4 months as described in page 6, line 16.

Page 6, line 21. Provide references that detail the data processing protocol, preferably including Langley plots, for each instrument.

Page 6, line 24. Break down the second word.

Page 7, line 8. Drop "s" from corresponds, since the word data is used as plural in the preceding sentence.

Page 7, line 5. Replace criterium with criterion or criteria. Note that the plural, if

chosen, requires changes in the rest of the sentence.

Page 9, line 3. Revise this sentence, as it contradicts with the fact that the agreement is achieved for less than 95% at 380 nm.

Page 9, line 5. Table 4 is expected to appear after Table 3. Revise numbering.

Page 9, line 6. Is the first period "2005-2010"? Table 2 has "2005-2009", as in page 12, line 1 and page 34, line 14.

Page 9, line 11. Explain how it is determined that problems in sun pointing were "the main cause" of AOD discrepancies between PFR and Cimel. Table 4 indicates that the fail rate decreased merely by one third - from 4.2% to 2.8% - at 500 nm upon the tracker update.

Page 9, line 11. This sentence implies that a sun tracker is not considered a part of the PFR instrument. That is surprising to those who perceive the tracker as an essential, fully-integrated component of a sun radiometer. It is like saying the steering wheel is not part of a car. Consider dropping the comma and the subsequent eight words.

Page 11, line 9. Replace "can be one of the main causes of part of" with "is the main cause of".

Page 11, line 10. Replace 5.4 with 5.3.

Page 11, line 16. Bring Kazadzis et al. before the first parenthesis, and drop the comma.

Page 12, line 3. Again it is not clear what supports the notion "the most important cause [. . .], which were associated with a poor pointing of GAW-PFR".

Page 12, line 5. Replace "of both periods" with "between the two periods".

Page 12, line 21. Elaborate on "MB values to be within 0.01 bias".

Page 12, line 23. Drop the first "and".

Page 13, line 2. This paragraph feels misplaced. It deserves to be a stand-alone subsection under Section 5.2. Combine it with the sixth paragraph of section 5.1.

Page 15, line 3. Remove the comma.

Page 15, line 19. "maximum value minus minimum value of AOD in one day" is less logical a metric for evaluating the calibration than the difference between the measurements at minimum and maximum air mass factor of the day.

Page 15, line 28. Insert a comma after "causes".

Page 15, line 29. Replace "worst" with worse.

Page 15, line 30. Replace the comma.

Page 17, line 10. Replace "a" with an.

Page 24, line 17. Explain what exactly is "more clearly" shown at 500 nm.

Figure 7. If this figure is to remain on the paper, note in the caption that an identical data set, PFR AOD, appears in both x and y axes, a practice generally discouraged. Also state that the numbers in the legend are rounded to the significant digits. This is to forestall questions as to why the black lines do not reach exactly (x,y)=(0,0).

Page 26, line 7. "first" is misleading. Refer to previous studies such as Russell et al. [2004]. Replace "might play" with plays.

Table 9. "(14.9)" for 500 nm, AOD>0.1 is not consistent with the "AOD>0.10" row of Table 4.

Figure 8. Revise the caption, yaxis labels and legends to better clarify what are plotted.

Page 28, line 13. Replace "could lead" with leads.

Table 10. Complete the right-most vertical line.

Page 31, line 8. Explain in what way GAW-PFR is "the AOD reference globally" and

"directly linked to WMO/CIMO".

Page 31, line 14, remove the first comma and "which".

Page 31, line 28. Does the judgment made here with the word "excellent" hold even while the 95% criteria are not met at 380 nm?

Page 31, line 30. Remove the last five words.

Page 32, line 5. Provide more details or references regarding the Langley plots so that their quality can be verified.

Page 32, line 29. Replace "in" with on.

Page 32, line 33. Remove "Even so,".

Page 33, line 13. Explain why the traceability metric should be redefined based on actual performance.

Page 33, line 14. Replace "shouted".

Page 33, line 15. The purpose of this paragraph is unclear. If it is a disclosure that low AOD cases are removed from the present analysis, it should be noted much earlier. If this paragraph is a suggestion to exclude negative AOD values from future data analyses, it is misguided. While such values are not physical, their exclusion would artificially bias the remaining data high.

Page 33, line 16. Replace "absolute" with relative.

Page 34, line 7 says "both [PFR and Cimel] are representative of the same AOD population" over the 11 years, except for 380 nm. Similarly, page 12, line 7 says "[The agreement in AOD] proves the consistency and homogeneity of the long AERONET-Cimel AOD data series". These conclusions imply that Cimel's stability is adequate and that PFR's features for greater stability (page 5, line 27-31) are, while remarkable, not a significant advantage. It is, then, logical to favor Cimel for its much narrower FOV,

a clear advantage over PFR. Arguments like this, or ones against it if any, would be a good addition.

Page 34, line 15. Revise the sentence. The paper does not directly address calibration transfer. Rather, the paper reveals that special attention should be paid to natural variability, sampling bias and PFR's wide FOV.

References

Russell, P. B., Livingston, J. M., Dubovik, O., Ramirez, S. A., Wang, J., Redemann, J., Schmid, B., Box, M., and Holben, B. N.: Sunlight transmission through desert dust and marine aerosols: Diffuse light corrections to Sun photometry and pyrheliometry, J. Geophys. Res., 109, D08207, 10.1029/2003JD004292, 2004.

---

## Referee Comment (RC4) · Anonymous Referee #4 · 11 Jan 2019

Review of Aerosol Optical Depth comparison between GAW-PFR and AERONET-Cimel radiometers from long term (2005-2015) 1-minute synchronous measurements

General Comments: The manuscript provides a comparison of AERONET-Cimel and GAW-PFR at IZO. The work is scientifically relevant given the analyzed data volume compared to previous studies. The manuscript identifies issues in the comparison of the AOD from the two different radiometers that are independently calibrated and processed in different networks. One major issue is the comparison of AERONET-Cimel to potentially suspect GAW-PFR field instrument (not reference) data (e.g., 2005-2010).

[Figure]

The authors present results in the abstract and conclusion with percentage agreement of 92.7% to 98.0% spectrally; however, these values are not from the apparently more optimal data set comparison for GAW-PFR (2010-2015), which show 1% to 2% improvement. Further, description of the causes of anomalies (i.e., not meeting the WMO standards) tend to be difficult to follow (e.g. reasoning for calibration and FOV) and it is not clear any corrections are actually applied to a final GAW-PFR data set. Also, the time matching criteria of 30 seconds is quite large for an instrument that performs measurements every minute and the AOD could change up to 0.01 per minute or potentially higher for dust. Some comparison results appear to be repeated in the tables. For example, the same "traceability" results (i.e., 92.7%, 95.6%, 95.8%, and 98.0%) appear to be repeated in three different tables (Tables 3, 4, and 8). Another major issue is that the study focuses more on the long-term and does not present any specific cases in pristine and dust events to specifically show examples of the differences in each instrument AOD measurements with subsequent analysis. A further issue is that the AERONET version 3 data are not included in the analysis. While these data are referenced as being available, the new product has some significant changes in regards to cloud screening and corrections made to the data that may impact agreement between the AERONET-Cimel and GAW-PFR instruments. Utilizing these AERONET V3 data would provide an added element of importance in AERONET/GAW-PFR comparisons. Last, the presentation of the document was difficult to follow at times. For example, the study objective statement is first encountered in the summary and conclusions section and this is not easy to follow. The "traceability" criteria tend to indicate that AERONET-Cimel is compared to the GAW-PFR but, in this case, GAW-PFR is not a reference instrument but a field instrument, which has higher uncertainty. Also, description in the text of the Figures and Tables needs further elaboration. Some specific comments are provided below on organization and other issues. The authors should take care to correct and address the issues here and below before resubmission.

Specific Comments: Abstract, Page 1, Line 6-9: "Traceability" as described only relates the precision of these two instruments. The result of the measurement may not

[Figure]

be accurate and but both measurements may be precise. What do you mean by "WMO standard?" In the abstract, the authors need to state that GAW-PFR is considered a ground-based WMO standard for AOD measurements and the field GAW-PFR and AERONET Cimel tend to have strong agreement. The use of "traceable" is an ambiguous term in defining more specifically the "agreement" between the two instruments. For example, the field PFR has a "traceable" calibration to the PFR triad. Please consider changing instances of "traceable" and "traceability" through the document. Page 2, Lines 28-29: How did you determine these totals? The AERONET web site provides partitioning of sites by equivalent data year and not the actual years (which can be greater). >10 years (data equivalent) is 84 sites (https://aeronet.gsfc.nasa.gov: last accessed 1/8/2019) >5 years (data equivalent) is 242 sites (https://aeronet.gsfc.nasa.gov last accessed 1/8/2019).

Page 3: Line 9: Please define "Triad" for PFR; perhaps the mean of three master PFR instruments?

Page 3, Line 13: What is a portable transfer standard radiometer? Is it a "reference" PFR?

Page 3, Lines 26-29: Additional reference is needed (e.g., Holben 1998, Eck 1999, Toledano 2018)

Page 4, Line 6: Change "thanks" to "has to be"

Page 4, Line 18: Need to spell-out "IZO" since it starts the sentence.

Page 5, Lines 10-11: Briefly discuss the differences between the two Cimels that affect the optical characteristics (i.e., why is this important?).

Page 5: Line 32: What type of filters?

Page 5: Line 33: Need citation and further explanation.

Page 6, Line 1: Place period after "position"

Page 6, Lines 26-27: How is the time matching performed between the AERONET-Cimel and the PFR? Is the closest PFR value used or an average of the two PFR values?

Page 7, Table 1: What is a GAW-PFR "field instrument?" What other can there be if it is the "world standard?" Also, AERONET-Cimel temperature control is different between AERONET Version 2 and Version 3.

Page 7, Lines 9-10: The results of the present study should be using AERONET Version 3, which has been available since early 2018 as discussed in Giles et al 2018.

Page 8, Line 1-4: Most instruments should be collecting data every 3 minutes. Why was the Cimel instrument at IZO collecting every 15 minutes? Why is 30 second difference used for time matching (the farthest away the PFR can be from the Cimel measurement? Should not the PFR be within say 10 seconds of the Cimel measurement to be matched?

Page 8, Line 19: AERONET data provides wavelength pairs and also computation of 440-870nm using all of the wavelengths in the range.

Page 9, Lines 1-31: The paragraphs are fragmented and they should be reorganization. Please revise and condense.

Page 9, Lines 10-12: As a result, it seems the PFR data from 2005-2010 should not be used in the study?

Page 10, Figure 1 Caption, Line 3: Please discuss here and in the text providing at least one specific case to analyze what is causing the large outliers.

Page 10, Table 3: The "Original GAW-PFR channel (%)" column is not correct if since you are expecting the AERONET values at different wavelengths to meet the AOD at PFR wavelengths. This column should be removed unless you interpolate AERONET AOD to PFR wavelengths.

Page 11, Table 4: What are the number of measurements for each cell used to compute the "traceability?"

Page 11, Line 15: Data-pairs should be 468 for AOD500nm as stated in Kazadzis et al (2014) plots.

Page 11, Line 16: Which instrument had poor calibration in the 500nm channel during the 9-day analysis period? Were both instruments field PFR and field Cimel instrument and not reference instruments or triad? It is important to draw this distinction in relation to the present study.

Page 11, Line 21: Change "among" to "between"

Page 12, Lines 3-5: It should be made clear that "non-traceability" is referring to PFR instrument and not the AERONET Cimel reference.

Page 12, Lines 9-13: More analysis and interpretation of Table 5 is needed with respect to the statistics presented in Table 5.

Page 12, Line 20: Better to use "reference" instead of "Master" throughout manuscript.

Page 13, Lines 8-33: The paragraphs are fragmented and they should be reorganized. Please revise and condense paragraphs.

Page 13, Lines 8-11: AERONET reference instrument should obtain Langley calibration coefficients every 3 to 8 months. (e.g., Giles et al., 2019). Please check with AERONET calibration center on the calibration interval.

Page 13, Lines 24-27; Page 14, Figure 2 Caption: Is the PFR AOD500nm used for the limitation "AOD-500nm <= 0.03" shown in Figure 2? The outliers in Figure 2 appear to be independent of air mass. Are these due to the PFR or Cimel?

Page 13, Lines 30-31: Uncertainty of field PFR (0.01) and uncertainty of reference AERONET-Cimel (0.005) at 380nm are maximum at low optical air mass and therefore the agreement between the two instruments will be inherently lower.

[Figure]

Page 14, Figure 2: Please change in the y-axis ticks to show zero. What is the optical air mass limitation for PFR measurements? The AERONET Version 3 processing is available up to optical air mass of up to 7 and AERONET Version 2 is up to 5.

Page 14, Table 6: What are the number of matched measurements in each optical air mass interval?

Page 15: Line 12: Also, include reference to Eck et al. 1999.

Page 15: Lines 29-32: AERONET-Cimel reference Langley is performed more frequently than once a year?

Page 16, Figure 3: It is difficult to quantify the relative significant difference between the Cimel and PFR in the logarithmic scale? Can you show a plot of the relative difference between the two instruments? Some significant variability in the differences exist and it could be due to differences in cloud screening, for example.

Page 16, Lines 3-12; Page 17 Lines 20-22: Authors should also utilize AERONET Version 3 with improved cloud screening techniques (Giles et al., 2019). It is not clear why the authors do not investigate AERONET Version 3 for this study since these data are available.

Page 17, Lines 10-22: The "traceability" here is difficult to interpret since the instruments do not use the same cloud screening which is evidenced by the lower "T(%)" numbers in the Table 7. More importantly, perhaps, are the percentage numbers in the column with FCS ranges in Table 7; however, although the solar radiation data may indicate a cloud, the sun photometers have the ability to find gaps in the clouds to perform measurements.

Page 17, Lines 29-35: A discussion of the COT effect on apparent optical depth due to different instrument FOV should be discussed (Kinne et al., 1997).

Page 18: Table 7: Define "FCS" in the caption. What are the number of measurements in each interval?

Page 19, Lines 22-25: Is this the PFR instrument with erroneous pressure reading in late 2014 at shown in Figure 4? Are the values in Table 4 computed without the malfunctioned barometer?

Page 20, Figure 4: What do you mean by 1-minute pressure data for AERONET-Cimel? Did Cimel have a pressure sensor in 2005? Please clarify.

Page 21, Lines 7-8: However, OMI O3 data are problematic due to sampling issues (McPeters et al, 2015).

Page 21, Lines 14-17: Are the discrepancies when using the OMI O3 for PFR?

Page 22, Figure 5: Need to state GAW-PFR uses OMI O3 and AERONET-Cimel uses climatological monthly average of TOMS O3.

Page 23, Figure 6: What is NO2 "annual course" in caption?

Page 24, Table 8: What do these data represent? What is the date period? Please provide the total number of match measurements for each wavelength.

Page 24, Lines 8-11: Where is it shown that 25% of the data outside U95 are from P, O3, and NO2?

Page 24, Lines 11-13: What corrections were applied and to which instrument?

Page 24, Lines 19-21: The AOD 870nm is only affected by Rayleigh component and therefore has the highest agreement as well as the lowest midday uncertainty of the four wavelengths analyzed.

Page 25, Figure 7: Are these data now all PFR (or AERONET-Cimel) corrected values to remove anomalies due to Rayleigh (due to pressure sensor issues) and trace gas corrections? I notice now that 2010-2015 is only presented. What are the total number of measurements in these plots? The limit of AOD>0.1 seems arbitrary as a threshold to use a fit of the non-traceable values. Note that many points appear to be very close to the "traceability" boundary limit. Also, black "traceable" points have larger difference

than red "untraceable" points with a smaller difference (somewhat counterintuitive). The reason (e.g., air mass dependence impact on the traceability criterion) for this affect should be stated in the caption for clarity.

Page 25, Table 9: Provide the number of measurements for each wavelength and AOD range.

Page 26, Line 12-13: Do you vary the surface albedo spectrally?

Page 26, Lines 20-24: AERONET-Cimel and PFR are simulated? Please state.

Page 27, Figure 8: What are the outliers between 1% and 2% PFR Circumsolar/Direct radiation in (a) and (b)? I cannot see the blue dots very well on panels (c) and (d); perhaps you can include on another plot? Please state that these results are simulated from the radiometers.

Page 28, Lines 17-22: Include the analyzed wavelength range for AE in the text, Figure 9, and Figure 9 caption.

Page 29, Table 10: Provide additional context on how these values were determined in the text.

Page 29, Lines 21-25: Please provide more background on how these values were determined. For example, it is not clear why AE is provided differently to AE PFR and AE Cimel in the number list.

Page 30, Figure 9: Please correct formatting problem with x-axis label of panel (a). Legend state "500-data intervals and standard deviations"; what does this mean? How do the plots change with "relative" difference since all differences are taken as absolute value in this plot? Does the relative difference show any trend in AE for Cimel lower than PFR or vice versa?

Page 31, Lines 3-4: This comment needs to be substantiated.

Page 31, Line 26: "traceability" is a confusing term and should be changed to agreement or something similar.

Page 31, Lines 17-18: What is the "concern?"

Page 31, Lines 21-26: What are these "limits?"

Page 31, Line 30: Fragmented sentence.

Page 32, Lines 8-10: The fact that 380nm channel is more divergent is not a new finding since it is known to have higher uncertainty than other channels. Please state relevant citations.

Page 32, Lines 18-20: Also, optically thin cirrus clouds can also produce a difference in the measured values (Chew et al. 2011, Huang et al., 2011).

Page 32, Lines 23-28: Which instrument? PFR or Cimel?

Page 32, Lines 30-33: What causes large change in O3 concentration at Izana high altitude site?

Page 33, Line 14: Change "shouted" to "should be"

Page 33, Lines 15-18: Please elaborate.

Page 34, Line 15: Change "paid" to "made"

References:

Kinne et al., 1997: https://doi.org/10.1175/1520-0469(1997)054<2320:CCRAMP>2.0.CO;2

Eck et al., 1999: https://agupubs.onlinelibrary.wiley.com/doi/abs/10.1029/1999JD900923

Giles et al., 2019: https://www.atmos-meas-tech.net/12/169/2019/

---

## Author Comment (AC1) · 7 Apr 2019

**Anonymous Referee #1 (amt-2018-438-RC1):**

**G.1. This is a very thorough and overall quite clearly written manuscript of great use to anyone relying on PFR or Aeronet derived AODs. It establishes how well the master instruments used in both networks compare over the long term and where and why minute differences arise. It does not address how well these calibrations are transferred across the networks and what the resulting uncertainties are for various stations. However, such questions have been addressed by many others as evidenced by citations in this manuscript. This reviewer only has a very small number of relatively minor comments**

Authors:

We appreciate the positive comments of the Referee, and his/her specific corrections and comments that are addressed below.

The Referee must take into account that, according to the requirements and comments of the Referees # 3 and # 4, new relevant analysis have been incorporated in the paper what has required large additional workload, resulting in a significant improvement of the paper.

The most relevant new analyses have been:

1) Comparison of GAW-PFR with version V3 of AERONET and previous comparison between V2 and V3.

2) New study on modelling the impact of near-forward scattering on the AOD measured by the PFR and Cimel radiometers (with different FOVs).

3) Study of AOD variability in 1 minute to rule out possible differences in AOD due to a non-perfect synchronization of the PFR and Cimel sampling data.

4) Incorporation of case studies on the impact on AOD due to small inaccuracies in the calibrations or the clouds contamination.

Much more information about the above points has been included in a new Supplement material document of 24 pages.

**S.1. Page 1, Line 4. Suggest to say "wavelength near" instead of "nearby wavelength"**

Authors:

Rephrased.

**S.2. Page 2, Line 15. Suggest mentioning some other sunphotometer networks, PHOTON, Japanese network etc.**

**Authors:**

**References to other networks have been requested by the four Referees. Although we referred to global networks with centralized data processing and databases, as well as standard calibration procedures in each of the networks, we fully agree to include a reference to other similar sunphotometer networks of global regional scope.**

**The new paragraphs introduced in the manuscript are as follows:**

*"…These are GAW-PFR (Global Atmosphere Watch - Precision Filter Radiometer; http://www.pmodwrc.ch/worcc/; last access: 05 September 2018) and AERONET-Cimel (AErosol RObotic NETwork - Cimel Electronique radiometer; http://aeronet.gsfc.nasa.gov; last access: 01 September 2018) networks. AERONET is, in fact, a federation of ground-based remote sensing aerosol networks established by NASA (National Aeronautics and Space Administration) and PHOTONS (PHOtométrie pour le Traitement Opérationnel de Normalisation Satellitaire; University of Lille- Service d'Observation de l'INSU, France; Goloub et al., 2007), being complemented by other sub-networks, such as, AEROCAN (Canadian sunphotometry network; Bokoye et al., 2001), AeroSibnet (Siberian system for Aerosol monitoring; Sakerin et al., 2005), AeroSpan (Aerosol characterisation via Sun photometry: Australia Network; Mitchell et al., 2017), CARSNET (China Aerosol Remote Sensing NETwork; Che et al., 2015), and RIMA (The Iberian network for aerosol measurements; Toledano et al., 2011).*

*There are other radiometer networks that in recent years have incorporated centralized protocols for data evaluation and databases, and performed regular intercomparisons with GAW-PFR and AERONET-Cimel. These include, for example, SKYNET (SKYradiometer NETwork), and its seven associated sub-networks, that uses the Prede-POM sky radiometer to investigate aerosol-cloud-solar radiation interactions (e.g. Campanelli et al., 2004; Nakajima et al., 2007; Takamura et al., 2004)."*

**The corresponding references have been added.**

**S.3. Page 2, Line 30. Suggest inserting "NASA" before "Goddard"**

**Authors:  Done**

**S.4. Page 3, Line 6. Numerous is an understatement.**

**Authors:  Replaced "numerous" by "many".**

**S.5. Page 4, Line 5. The use of "absolute" is misleading here. A Langley calibration alone is never absolute (i.e. the calibration value is just a signal in an engineering unit not W/m$^2$. The beauty is that an absolute calibration is not needed to derive AOD.**

**Authors:  We agree. "Absolute" has been removed.**

**S.6. Page 5, Line 5. Suggest replacing "constant" with "signal" here and everywhere else in the manuscript. Or maybe just explain once that in sunphotometry the calibration "constant" is the signal the instrument would read outside the atmosphere (extraterrestrial) at a normalized earth-sun distance?**

**Authors:  We have added:**

*"Note, that the extraterrestrial constant (calibration constant) is the signal the instrument would measure outside the atmosphere at a normalized earth-sun distance."*

**S.7. Page 6, Line 24. Font issue.**

**Authors:  Corrected**

**S.8. Page 6, Line 6. Incorrect grammar: "were" not "was".**

**Authors:  We have not found this mistake. The paper has now been corrected throughout for English.**

**S.9. Page 14: Line 14. This is misleading as the error in AOD due to error in the calibration "constant" is independent of AOD.**

**Authors:**

**We agree, but we guess the misleading statement was in Page 13, Line 14. So, the paragraph:**

[revised manuscript text omitted]

Considering the AE criteria established by Cuevas et al. (2015) and Berjón et al. (2019), we have identified the following four main categories according to the $AE_{PFR}$ and $AE_{Cimel}$ values:

1. $AE_{PFR}$ & $AE_{Cimel} > 0.6$: Pristine conditions.
2. $0.25 < AE_{PFR}$ & $AE_{Cimel} \leq 0.6$: Hazy, mineral dust being the main aerosol component.
3. $AE_{PFR}$ & $AE_{Cimel} \leq 0.25$: Pure dust.
4. $AE_{PFR}$ and $AE_{Cimel}$ do not fit any of the previous categories.

In 94.9 % of the cases, GAW-PFR and AERONET-Cimel V2 match the AE intervals of each aerosol scenario. Similar results (93.4%) were obtained when comparing with AERONET V3. Most of the agreement (>80 %) occurs in the predominant scenario of pristine conditions despite the AE uncertainty under pristine conditions being $\geq 1$. See Supplement S18 for more details. Notice that given the special characteristics of the Izaña Observatory, and according to Cuevas et al. (2015) and Berjón et al. (2019), AE is a self-sufficient parameter to define different types of aerosol scenarios without the need to combine its information with AOD.

**6. Summary and Conclusions**

While GAW-PFR is the WMO-defined global AOD reference, being directly linked to WMO / CIMO, and was specifically designed to detect long-term AOD trends, AERONET-Cimel is the densest AOD measurement network globally, and the network most frequently used for aerosol characterization and for model and satellite observation evaluation.

An AERONET-Cimel 11-year AOD data series at IZO was obtained using a large number of radiometers. A total of 13 Reference instruments were used in the period 2005-2009 which means that every 4 and a half months, approximately, an instrument was replaced by another one to be calibrated. Their calibrations were performed during their respective measurement time periods at IZO. Therefore, these calibrations were not in any way linked with those of the instruments that preceded or replaced them, nor with GAW-PFR reference. These facts led us to investigate the homogeneity of the AERONET-Cimel AOD data series and their intercomparability with the much more homogeneous AOD data series from GAW-PFR (3 instruments in 11 years). The traceability concept for AOD suggested by WMO consists in determining whether the AOD difference of the AERONET CIMELs vs the GAW PFRs lie within the U95 limits. We have used uncertainty limits for AOD traceability established by WMO (2005) for these type of instruments with finite FOV. The acceptable traceability is when 95 % of the absolute AOD differences lie within these limits, in which case both data populations are considered equivalent. It should be clarified that "traceability" is not used in a strict metrological sense. This study has addressed the comparison of the GAW-PFR dataset with the two versions of AERONET (V2 and V3) in the period 2005-2015. An excellent agreement between V2 and V3 for the four analysed channels ($R^2 > 0.999$) has been obtained.

More than 70000 synchronous GAW-PFR (PFR) and AERONET-Cimel (Cimel) 1-minute data-pairs in each channel in the period 2005-2015 were analysed. An excellent traceability of AOD from the AERONET-Cimel (V2 and V3) is found for 440 nm, 500 nm and 870 nm, and fairly good results for 380 nm. The lowest percentage of traceable AOD data is registered in 380 nm with 92.7 % of the 1-minute data within the WMO limits, and the highest in 870 nm with 98.0 % of the data.

The different possible causes of non-traceability in AOD were investigated as follows:

- Absolute AOD measurements synchronization.

Analyzing 1-minute AOD variability we concluded that its impact on the AOD differences is negligible as only ~0.8% of the AOD data has a variability larger than 0.005 in all spectral ranges.

- Sun tracking misalignments.

Sun tracking misalignments constitute a serious problem and a major cause of non-traceability of AOD data-pairs as demonstrated by the AOD data outside the U95 limits from the period 2005-2009 as a consequence of episodic problems with the sun-tracker of the GAW-PFR radiometer. For the 2010-2015 period the percentage of traceable data-pairs improves to 93.5% (380 nm), 97.4% (440 nm), 97.2% (500 nm) and 99.1% (870 nm). However, most of these cases could be identified and excluded from the analysis.

- Cloud screening failure by both network algorithms.

According to our observations, the simultaneous failure of both cloud screening algorithms might occur only under the presence of large and stable cirrus, or altostratus ($\sim$ 6000 m a.s.l.) on the top of a heavily dust loaded Saharan air layer, hiding very wide and stable clouds. 
[revised manuscript text omitted]

Blanc, P., Espinar, B., Geuder, N., Gueymard, C., R., M., Pitz-Paal, R., Reinhardt, B., Renné, D., M., S., Wald, L., and Wilbert, S.: Direct normal irradiance related definitions and applications: The circumsolar issue, Solar Energy, 110, 561 – 577, https://doi.org/https://doi.org/10.1016/j.solener.2014.10.001, 2014.

Bodhaine, B. A., Wood, N. B., Dutton, E. G., and Slusser, J. R.: On Rayleigh optical depth calculations, Journal of Atmospheric and Oceanic Technology, 16, 1854–1861, 1999.

Böhm-Vitense, E.: Introduction to stellar astrophysics, volume 2: stellar atmospheres, Cambridge University Press, 260 pp., 1989.

Bokoye, A. I., Royer, A., O'Neill, N. T., Cliche, P., Fedosejevs, G., Teillet, P. M., and McArthur, L. J. B.: Characterization of atmospheric aerosols across Canada from a ground-based sunphotometer network:

AEROCAN, Atmosphere-Ocean, 39, 429–456, https://doi.org/10.1080/07055900.2001.9649687, 2001.

[revised manuscript text omitted]

Eissa, Y., Blanc, P., Wald, L., and Ghedira, H.: Can AERONET data be used to accurately model the monochromatic beam and circumsolar irradiances under cloud-free conditions in desert environment?, Atmospheric Measurement Techniques, 8, 5099–5112, https://doi.org/DOI = 10.5194/amt-8-5099-2015, 2015.

Eissa, Y., Blanc, P., Ghedira, H., Oumbe, A., and Wald, L.: A fast and simple model to estimate the contribution of the circumsolar irradiance to measured broadband beam irradiance under cloud-free conditions in desert environment, Solar Energy, 163, 497–509, https://doi.org/https://doi.org/10.1016/j.solener.2018.02.015, 2018.

Eskes, H. J. and Boersma, K. F.: Averaging kernels for DOAS total-column satellite retrievals, Atmospheric Chemistry and Physics, 3, 1285–1291, https://doi.org/10.5194/acp-3-1285-2003, https://www.atmos-chem-phys.net/3/1285/2003/, 2003.

Freidenreich, S. M., and Ramaswamy, V.: A new multiple-band solar radiative parameterization for general circulation models, J. Geophys. Res., 104, 31389-31409, doi: 10.1029/1999JD900456, 1999.

García, R. D., Cuevas, E., García, O. E., Cachorro, V. E., Pallé, P., Bustos, J. J., Romero-Campos, P. M., and de Frutos, A. M.: Reconstruction of global solar radiation time series from 1933 to 2013 at the Izaña Atmospheric Observatory, Atmospheric Measurement Techniques, 7, 3139–3150, https://doi.org/10.5194/amt-7-3139-2014, https://www.atmos-meas-tech.net/7/3139/2014/, 2014.

García, R. D., Barreto, A., Cuevas, E., Gröbner, J., García, O. E., Gómez-Peláez, A., Romero-Campos, P. M., Redondas, A., Cachorro, V. E., and Ramos, R.: Comparison of observed and modeled cloud-free longwave downward radiation (2010–2016) at the high mountain BSRN Izaña station, Geoscientific Model Development, 11, 2139–2152, https://doi.org/10.5194/gmd-11-2139-2018, https://www.geosci-model-dev.net/11/2139/2018/, 2018.

García, R. D., Cuevas, E., Ramos, R., Cachorro, V. E., Redondas, A., and Moreno-Ruiz, J. A.: Description of the Baseline Surface Radiation Network (BSRN) station at the Izaña Observatory (2009–2017): measurements and quality control/assurance procedures, Geosci. Instrum. Method. Data Syst., 8, 77-96, https://doi.org/10.5194/gi-8-77-2019, 2019.

[revised manuscript text omitted]

[3]Finnish Meteorological Institute, Helsinki, Finland

[4]Atmospheric Optics Group, Valladolid University, Valladolid, Spain

[5]Cimel Electronique, Paris, France

Correspondence: Emilio Cuevas (ecuevasa@aemet.es)

**Supplement S1. One-minute AOD data differences between AERONET-Cimel (V3) and GAW-PFR.**

One-minute AOD data differences between AERONET-Cimel (V3) and GAW-PFR for (a) 380 nm (75303 data-pairs), (b) 440 nm (76290 data-pairs), (c) 500 nm (75335 data-pairs) and (d) 870 nm (76307 data-pairs) for the period 2005-2015. Black dots correspond to the *U*95 limits. A small number of outliers are out of the ±0.06 AOD differences range. Black arrows indicate a change of Reference AERONET-Cimel radiometer and red arrows indicate a change of the GAW-PFR instrument.

[Figure]

**Supplement S2. Percentage of [Cimel (V3)-PFR] 1-minute AOD differences meeting the WMO criteria for the four compared channels.**

Percentage of AERONET-Cimel 1-minute AOD data (V3) meeting the WMO criteria for the four compared channels, and different AOD and AE scenarios for the period 2005-2015, number of data pairs are shown in brackets. The last row corresponds to the total percentages for the sub-period 2010-2015. AOD and AE traceability > 95% are marked in bold.This Table is equivalent to Table 4 of the manuscript for AERONET V2.

| % of data within WMO limits | 380 nm | 440 nm | 500 nm | 870 nm |
|---|---|---|---|---|
| AOD≤0.05 | 93.6 (60264) | **96.3 (62836)** | **97.1 (62545)** | **98.4 (64213)** |
| 0.05<AOD≤0.10 | 91.0 (5138) | 92.0 (5217) | 92.6 (5222) | 94.7 (5372) |
| AOD>0.10 | 77.1 (4085) | 84.1 (4537) | 81.6 (4326) | 93.3 (5034) |
| AE≤0.25 | 78.7 (2472) | 82.3 (2588) | 79.0 (2483) | 92.9 (6530) |
| 0.25<AE≤0.6 | 90.2 (5941) | 94.3 (6321) | 94.9 (6255) | **97.4 (6530)** |
| AE>0.6 | 94.1 (56952) | **96.5 (59181)** | **97.1 (58793)** | **98.7 (60514)** |
| Total 2005-2015 | 92.3 (69487) | **95.2 (72590)** | **95.7 (72093)** | **97.8 (74619)** |
| Total 2010-2015 | 92.8 (42463) | **96.8 (44328)** | **96.8 (44329)** | **98.8 (44329)** |

**Supplement S3. AOD variability 1 minute interval.**

Standard deviation values of 20117 AOD triplets measured with the Cimel#244 in 2013 for 380 and 500nm.

[Figure]

The percentage of data with 1-minute AOD variability for the four GAW-PFR channels are shown in the next figure.

**Supplement S4. One-minute AOD differences between AERONET-Cimel (V2 and V3) GAW- and PFR versus optical air mass (m).**

One-minute AOD differences between AERONET-Cimel (V2 and V3) and GAW-PFR versus optical air mass (m) under pristine conditions (AOD$_{500nm}$≤ 0.03) in the period 2005-2015 for (a) 380 nm, (b) 440 nm, (c) 500 nm and (d) and 870 nm.

[Figure]

**Supplement S5. Percentage of [Cimel (V3)-PFR] 1-minute AOD differences meeting the WMO criteria for each wavelength and for different optical air mass.**

Percentage of 1-minute AOD data (V3) meeting the WMO criteria for each wavelength for different optical air mass intervals under pristine conditions ($AOD_{500nm} \leq 0.03$) in the period 2005-2015. This Table is equivalent to Table 7 in the manuscript for AERONET V2.

| Percentage of AOD differences within the U95 limits $AOD500nm \leq 0.03$ | Total | $1 \leq m < 2$ | $2 \leq m < 3$ | $3 \leq m < 4$ | $4 \leq m < 5$ | $5 \leq m < 6$ |
|---|---|---|---|---|---|---|
| | (%) | (%) | (%) | (%) | (%) | (%) |
| 380 nm | 94.9 | 92.9 | 95.5 | 96.7 | 96.6 | 96.6 |
| 440 nm | 97.5 | 97.2 | 97.3 | 98.0 | 97.6 | 97.7 |
| 500 nm | 98.3 | 98.2 | 98.2 | 98.5 | 98.2 | 98.3 |
| 870 nm | 99.0 | 99.1 | 99.1 | 99.1 | 98.6 | 98.7 |

**Supplement S6. AOD diurnal range corresponding to AOD outliers under pristine conditions.**

AOD diurnal range variation (maximum value minus minimum value of AOD in one day) corresponding to AOD outliers (non-traceable AOD) under pristine conditions (AOD$_{Cimel-500nm}$≤ 0.03) in the period 2005-2015 for AERONET V2 and V3 and for 440 nm, 500 nm and 870 nm: a) 440 nm V2; b) 440 nm V3; c) 500 nm V2; d) 500 nm V3; e) 870 nm V2; and f) 870 nm V3.

This Figure is equivalent to Figure 3 of the manuscript for 380 nm.

[Figure]

**Supplement S7. Percentage of AOD$_{380nm}$ outliers of GAW-PFR and AERONET Cimel (V3).**

Percentage of cases with AOD$_{380nm}$ outliers of both GAW-PFR and AERONET Cimel (V3) under pristine conditions (Cimel AOD$_{500nm}$≤0.03). In these cases the diurnal AOD range was higher than 25% of the daily mean AOD value for which a certain cause has been determined: calibration inaccuracies, cloud screening algorithm failures, mixture of the two previous causes, poor sun pointing, or unknown causes.

|  | PFR 51 cases | Cimel 81 cases |
|---|---|---|
| Calibration inaccuracies | 7.8% | 44.4% |
| Cloud screening failures | 29.4% | 21.0% |
| Calibration+ cloud screening errors | 9.8% | 11.1% |
| Sun misalignments | 17.6% | 0% |
| Unknown | 35.3% | 33.5% |

**Supplement S8. Examples of fictitious AOD diurnal variation in both GAW-PFR and AERONET-Cimel.**

Examples of fictitious AOD diurnal variation in both GAW-PFR and AERONET-Cimel V3 due to small calibration inaccuracies in the UV channel (368 nm for GAW- PFR and 380 nm for AERONET-Cimel). The date is indicated in the x-axis. In all these cases a clear fictitious AOD diurnal cycle is observed in AERONET-Cimel V3, normally less than 0.01. In cases d), e), and f) an anomalous diurnal cycle is also observed, but in the opposite direction (convex curve), in the case of the GAW-PFR. These cases reflect a non-perfect calibration in the UV channel and are a cause of non-traceability.

[Figure]

**Supplement S9. Examples of AOD diurnal variation of all chanels from AERONET-Cimel Level 2 V3.**

The screenshots of AERONET V3 level 2 show that the fictitious diurnal cycle is accentuated, or only clearly observed, in the 340 and 380nm channels.

Screenshots from http://aeronet.gsfc.nasa.gov (last access: 1 february 2019). Izana AERONET station Level 2 Version 3.

[Figure]

**Supplement S10.** **Case analysis of altostratus above the Saharan Air Layer top.**

In this section of the supplement some case analyses are shown analysed in order to highlight the complexity of properly filtering some types of clouds during moderate dust intrusions.

Case analysis S10.1 (18 July 2012): The range corrected backscattering signal vertical cross section of the Micropulse lidar (MPL) (a) shows the presence of altostratus just above the top of the Saharan Ar Layer (SAL), around 6 km height. The total-sky camera images show the presence of middle clouds and dust at 09UTC (b), 11UTC (c) and 16UTC (d). The AERONET V2 AOD records are filtered correctly at those three hours (e), but the AOD values "recovered" by AERONET V3 at those times are very high unreal AOD values, greater than 1.

[Figure]

Case analysis S10.2 (8 March 2015): The range corrected backscattering signal vertical cross section of the Micropulse lidar (MPL) (a) shows the presence of thick altostratus just above the top of the Saharan Ar Layer (SAL) that increase altitude throughout the day from 5 to 8 km height to gradually disappear after 15:30 UTC. The total-sky camera image shows the presence of middle clouds and dust at 15:00 UTC (b). Both GAW-PFR and AERONET-V3 have successfully filtered contaminated data by altostratus until 15:00 UTC but do not do so after that time and until they disappear. During that time the clouds are between 7 and 8 km height and give a weaker signal. AOD contaminated by clouds (AOD500nm = 0.35) is substantially higher than at the end of the day (AOD500nm = 0.35) (c) when the SAL shows a greater thickness.

[Figure]

**S11. Case analysis of cirrus clouds.**

A second type of clouds that cause problems in AOD retrieval are the cirrus clouds, usually being present at Izaña between January and April, associated with the presence of the subtropical jet that is normally found in the vicinity of the Canary Islands at this time of year (Rodríguez-Franco and Cuevas, 2013). A constant cloud optical thickness (COT) corresponding to a cloud of a certain horizontal extension would cause the successive measurements within a minute to correspond to the same cloud stage, and therefore it would not be discernible from the extinction caused by aerosols. In the case of very thin cirrus clouds, AOD could increase up to 0.03 (Chew et al., 2011; Giannakaki et al., 2007) with small fluctuations, that cloud-screening algorithms could interpret as the presence of an aerosol layer. Huang et al. (2012) evaluated the impact on AERONET level 2.0 AOD retrievals from cirrus contamination highlighting the difficulties to remove completely their signature, mainly from those subvisual thin cirrus. According to Kinne et al. (1997), optical depth estimates from cirrus derived with sunphotometers have to include forward-scattering effects. Their results show that for cirrus, and instruments with 2.0° and 2.4° FOV, the correction factors vary between 1.6 and 2.5 depending on the crystal size. Taking into account that the FOV of the GAW-PFR is 2.5°, while that of the AERONET-Cimel is 1.3°, such cases will affect the comparison results.

Three case analyses on cirrus clouds are shown below.

Case analysis S11.1 (September 23, 2015): The range corrected backscattering signal vertical cross section of the Micropulse lidar (MPL) (a) shows scattered cirrus clouds throughout the day (a), and one in particular around 17:45UTC that affects the Cimel AOD measurements. Unfortunately, we do not have measurements for the PFR at this time. The all-sky camera confirms the presence of cirrus clouds at that time (d). The AERONET V2 snapshot registers the impact of the cirrus (b), punctually increasing the AOD values by two. AERONET V3 (c) does not filter these values.

[Figure]

Case analysis S11.2 (February 12, 2015): The range corrected backscattering signal vertical cross section of the Micropulse lidar (MPL) (a) shows the presence of cirrus clouds at around 11 km height between 17:30 and 19:00UTC (a), this is confirmed by the all-sky camera image (b). These cirrus clouds affected the AERONET V2 AOD, increasing the AOD values between 2 and 5 times, depending on the channel (c). AERONET V3 cloud screening correctly filtered these anomalous AOD values (d).

[Figure]

Case analysis S11.3 (March 27, 2010): The cirrus cloud observed by the all-sky camera around 18:30UTC affected both GAW-PFR and AERONET V3, giving AOD values about 8 times higher than those observed early in the morning. The erroneous AOD values of the GAW-PFR are slightly lower than those of AERONET V3. The cause could be a greater forward-scattering effect of the cirrus cloud on the GAW-PFR due to its higher FOV (compared with that of the Cimel).

[Figure]

**Supplement S12. Impact of low stratocumulus on AOD retrieval.**

Another cloud scenario that can affect AOD traceability is the presence of low clouds (stratocumulus) that sometimes exceed the observatory height level because the temperature inversion is around 2400 m height. Sometimes the fog can affect the radiometers as shown in the following case analysis in which the GAW-PFR cloud screening algorithm fails giving a few erroneous 1-minute AOD data around 09:00UTC.

Case analysis February 12, 2015: The range corrected backscattering signal vertical cross section of the Micropulse lidar (MPL) (a) shows the presence low stratocumulus very close to the Izaña level in the early morning as confirmed by the all-sky camera (b) and the webcam (c) images. In the all-sky camera it is possible to appreciate the presence of ice due to recent freezing fog. These clouds exceed the level of the observatory in some moments, slightly hiding the sun with mist. The result in this case are very high AOD values from GAW-PFR (one order of magnitude) (d) due to a failure of its cloud screening algortithm. The rest of the day the agreement between GAW-PFR and AERONET-Cimel AOD (V3) measurements was very good.

[Figure]

February 12, 2015, 09:05UTC
Low stratocumulus passing over Izana

**Supplement S13. Actual AOD differences between AERONET-Cimel V3 and GAW-PFR vs PFR AOD**

Actual AOD differences between AERONET-Cimel V3 and GAW-PFR vs PFR AOD at (a) 380 nm (b) and 500 nm for the period 2005-2015. The fitting line has been calculated with AOD data > 0.1 and Cimel-PFR AOD difference > 0. Number of data used in the plots are indicated in the legend. The percentage of non-traceable AOD data with these conditions is ~22% for 380nm, and ~13% for 500nm. Note that some traceable (black) points show larger AOD differences than non-traceable (red) points because of air mass dependence of the WMO traceability criterion.

[Figure]

**Supplement S14. Percentage of AERONET V3 AOD data outside the U95 limits for high AOD conditions**

Percentage of AERONET V3 AOD data outside the U95 limits at 380 nm, 440 nm, 500 nm and 870 nm channels and for three $AOD_{500nm}$ thresholds with respect to all data and with respect to all data for each AOD interval (in brackets).

| | Percentage of AOD data outside the $U$95 limits (%) | | |
|---|---|---|---|
| | $AOD_{500nm} > 0.1$ | $AOD_{500nm} > 0.2$ | $AOD_{500nm} > 0.3$ |
| 380 nm | 1.6 (22.9) | 1.1 (42.0) | 0.4 (54.4) |
| 440 nm | 1.1 (15.9) | 0.9 (32.5) | 0.4 (49.0) |
| 500 nm | 1.3 (18.4) | 1.0 (37.6) | 0.5 (55.7) |
| 870 nm | 0.5 (6.7) | 0.4 (13.4) | 0.2 (19.0) |

Comparing versions V2 and V3, we can see that, except for the 380 nm channel, in V3 the non-AOD traceability increases with respect to that found in V2.

**Supplement S15. Simulations of scattered to direct radiation simulations.**

Scattered to direct radiation simulations made with a forward Monte Carlo model (Barker 1992, Barker 1996, Räisänen et al. 2003) for FOVs of 2.5° and 1.2° for seven values of effective radius ($r_e$=0.2, 0.5, 1.0, 1.5, 2.0, 2.5, and 3.0 μm), for five AOD values (AOD= 0.1, 0.2, 0.3, 0.4, and 0.5), and for five solar zenith angles (θ = 20°, 30°, 45°, 60° and 80°).

[Figure]

**Supplement S16. Relative error in AOD for PFR and Cimel.**

Relative error in AOD for PFR (x-axis) and Cimel (y-axis) for seven values of effective radius (r_e=0.2, 0.5, 1.0, 1.5, 2.0, 2.5 and 3.0 μm), for five AOD values (AOD= 0.1, 0.2, 0.3, 0.4, and 0.5), and for five solar zenith angles (θ = 20°, 30°, 45°, 60° and 80°).

[Figure]

**Supplement S17. Actual AOD differences between AERONET-Cimel V3 and GAW-PFR vs PFR AOD after AODPFR correction.**

The same as the Figure of Supplement S13 (AERONET V3) after the PFR AOD data were "corrected" by adding + 3.3% at 380nm and + 2.2.% at 500 nm to the 1-minute AOD PFR data > 0.1.

[Figure]

**Supplement S18. Ångström exponent comparison**

Percentage of cases in which GAW PFR and AERONET V2 (a) and V3 (b) coincide in each AE scenario (period 2005-2015).

[revised manuscript text omitted]

*4.2. Cloud filtering*

The data matching in our comparison analysis was performed with synchronous 1-minute AOD values of both networks labelled with quality control (QC) flags that guarantee proven quality data not affected by the presence of clouds.  In the case of the AERONET-Cimel network, the selected AOD data are Level 2 data from both V2 and V3 AERONET databases, which have been cloud filtered by the Smirnov algorithm (Smirnov et al., 2000), based on the triplet method, with a second-order temporal derivative constraint (McArthur et al., 2003), and visually screened in V2. The cloud screening in AERONET V3 has been completely automated, and notably improved, especially by refining the triplet variability and cirrus cloud detection and removal (Giles et al., 2019). GAW-PFR cloud screening algorithms also use the Smirnov triplet measurement, and the second-order derivative check, but add a test for optically thick clouds with $AOD_{500nm} > 2$ (Kazadzis et al., 2018b). In the case of the GAW-PFR network (Wehrli, 2008a) the flags take the value 0 (cloudless conditions, no wavelength crossings and sun pointing within certain limits, more details in Kazadzis et al. (2018a)) for all those selected records.

*Table 2. GAW-PFR and AERONET-Cimel instrument numbers used in this study in the period 2005-2015. Data from Reference Cimel #398 was not upgraded to Level 2 in V3 during the period 12 July 2008 - 15 September 2008.*

| Instruments used in this study | Period 2005-2009 | Period 2010-2015 |
|---|---|---|
| GAW-PFR | 2 instruments: #6,#25 | 2 instruments: #6,#21 |
| AERONET-Cimel | 13 instruments: #25,#44,#45,#79,#117,#140 #244,#245,#380,#382,#383,#398,#421 | 5 instruments: #244, #347, #380 #421, #548 |

~~GAW-PFR provides AOD values every 1 minute as an average of 10 sequential measurements of total duration less than 1 second, while AERONET-Cimel provides AOD values every 15 minutes (from 3 measurements separated by 30 seconds). We consider synchronous 1-minute data when GAW-PFR and AERONET-Cimel AOD data were obtained with a difference of ~ 30 s.~~

**4.3. WMO traceability criteria.**

The criterion for traceability used in this study follows the recommendation of the WMO (WMO, 2005) which states that 95% of the AOD measurements fall within the specified acceptance limits, taking the  PFR as a reference:

$$U_{95} = \pm(0.005 + 0.010/m) \tag{2}$$

where $m$ is the optical air mass. Note that the U95 range is larger for smaller optical mass.

The acceptance limits proposed by WMO take into account, on the one hand, the uncertainty inherent in the calculations of the AOD, and on the other hand, the uncertainty associated with the calibration of the instrument. The latter, for the case of instruments with finite field of view direct transmissions, such as the PFR and the Cimel, is dominated by the influence of the top-of-the-atmosphere signal determined by Langley plot measurements, divided by the optical air mass.

The first term of Eq. 2 (0.005) represents the maximum  tolerance for the uncertainty due to the atmospheric parameters used for the AOD calculation (additional atmospheric trace gas corrections, and Rayleigh scattering). The second term describes the calibration related relative uncertainties . The WMO recommends an upper limit for the calibration uncertainty of 1%.

~~A first simple approach to calculate the circumsolar radiation of each radiometer taking into account their respective FOVs, AOD, total $O_3$ and pressure values, has been performed with the SMARTS (Simple Model of the Atmospheric Radiative Transfer of Sunshine) model version 2.9.5 (Gueymard, 1995). This spectral model, that covers the UVA, UVB, Visible and Near-Infrared bands, can be used to simulate the spectral irradiance that would be measured by a spectroradiometer (Gueymard, 2001). This model, which has been used and compared with LibRadtran for determining circumsolar radiation (Eissa et al., 2015) is used in~~

this study to estimate, in a first approximation, the differences in AOD caused by the different FOV of PFR and Cimel radiometers.

**4.4. Modelling the impact of near-forward scattering on the AOD measured by the PFR and Cimel radiometers**

In order to study the impact of near-forward scattering on the irradiance measured by the PFR and CimelIMEL instruments, a forward Monte Carlo model (Barker 1992, Barker 1996, Räisänen et al. 2003) was employed. For the present work, the model was updated to account for the finite width of the solar disk. The starting point of each photon was selected randomly within the solar disk, assuming a disk half-width of 0.267° and the impact of limb darkening on the intensity distribution was included following Böhm-Vitense (1989). Some diagnostics were also added to keep track of the distribution of downwelling photons at the surface with respect to the angular distance from the centre of the sun. Gaseous absorption was accounted for following Freidenreich and Ramaswamy (1999), while the Rayleigh scattering optical depth was computed using Bodhaine et al. (1999) below..

**5. Results.**

**5.1. CPreliminary comparison of long-term AERONET V2 and V3 data basesdatasets at Izaña site.**

Since V3 has been released recently (Giles et al., 2019), we present a comparison between V2 and V3 for the Cimel channels 380 nm, 440 nm, 500 nm and 870 nm for the period 2005-2015.

The results indicate that for the Izaña site the agreement and consistency between the two AERONET versions is very high for the four channels ($R^2$>0.999) in full agreement with the results of the V2-V3 comparison reported by Giles et al. (2019). So, we can advanceIt follows that the results of the comparison between GAW-PFR and the two versions of AERONET are very similar as shown throughout this work. A detailed description ofn AERONET V3 and its improvements with respect to V2 is given in Giles et al. (2019). A detailed description of the improvements introduced in V3 are given in Giles et al. (2019). As such Logically, these improvements depend on aerosol type, are not homogeneous in terms of theirfor the different types of aerosols and their nature and variable impact at a global level. A according to the changes introduced in V3, for thea high mountain site such as Izaña characterized by low background AOD values or, alternatively, by the presence of dust presence of dust (, but with no pollution or biomass burning aerosols), the expected AOD differences between V2 and V3 are expected to be minimum as it is confirmed in this studyhas been shown. (Figure 1).
the

However, it should be noted  that  AERONET V3 does not restrict the calculation of AOD to optical masses less than 5.0 (Giles et al., 2019), as V2 does. This results in an increase in the number of solar measurements occurring in the early morning and the late evening. Consequently, the GAW-PFR comparisons with AERONET V3 consisted of ~ 5000 more data pairs than  the GAW-PFR comparison with V2 (see Supplement S.1.1.).

[Figure]

*Figure 1. AERONET Version 3 (V3) vs Version 2 (V2) AOD 1-minute data scatterplot at Izaña Observatory  for the period 2005-2015: a) 380 nm; b) 440 nm; c) 500 nm and d) 870 nm. The corresponding equations of the linear fits, the coefficients of determination (R2), Mean Bias (MB), Root Mean Square Error (RMSE) and the number of data pairs (N) used are included in each legend.*

*5.2.- AERONET-Cimel AOD comparison with GAW-PFR data.*

The comparison with GAW-PFR AOD  shows that the AOD from AERONET-Cimel radiometers meet the WMO traceability criteria ("traceable AOD data" from now on) at 440 nm, 500 nm and 870 nm  channels. The lowest agreement is found in the UV channel (380 nm) with 92.7 % of the  data, and  the highest in the infrared channel (870 nm) with 98.0 % for V2 (Figure 2; Table 3). Almost identical results are obtained for V3 (Supplement S1 and S2).

However, in the first half of the comparison period (2005-20010) there was a some  mechanical problems in the solar tracker where the GAW-PFR was mounted on, which caused sporadic problems of sun pointing. This finding was confirmed with data from the four–quadrant silicon detector (Wehrli, 2008a) that showed diurnal variation of the PFR sensors position up to 0.3°. From 2010 onwards, the PFR was mounted on an upgraded solar tracker of higher performance and precision. This reduced problems in sun pointing, that were the main cause of  most of the AOD discrepancies between PFR and Cimel, and therefore not attributable to the instruments themselves.

In addition, since 2010, Cimel #244 has been in continuous operation for most of the time at the Izaña Observatory, greatly simplifying calibration procedures and the corresponding data evaluation, and minimizing errors of calibration uncertainties introduced by the use of a high number of radiometers in the intercomparison. During the 2010-2015 period, the fraction of traceable AOD measurements of the total between the AERONET-Cimel radiometer and the GAW-PFR improves to 93.46 % in the 380 nm channel, and this percentage rises to 99.07 % for the 870 nm channel.

Despite the technical differences between both radiometers, described above, and the different calibration protocols, cloud screening  and data processing procedures, the data series of both instruments, can be considered as equivalent, except for 380 nm, according to the WMO traceability criteria defined previously (Eq. 2). This explains the excellent agreement in the long-term AOD climatology shown for GAW-PFR and AERONET-Cimel in Toledano et al. (2018).

We have compared the percentages of AERONET-Cimel AOD V2 data meeting the WMO criteria for the four interpolated GAW-PFR channels with those of AERONET V3 (Table 3).

~~For shorter wavelengths, the percentage of data within the WMO limits decreases when the original GAW-PFR channels are used as a reference (not shown here), mainly, and as expected, in the 412 nm channel as this differs considerably from the nominal value of the corresponding AERONET-Cimel channel (440 nm). For 500 nm and 870/862 nm there are no significant differences. Hereinafter, in this study the interpolated GAW-PFR channels are used.~~

A more detailed statistical evaluation for different scenarios of aerosol loading (three ranges of AOD) and aerosol size (three ranges of AE) for each compared channel has been performed (see Table 4). We

observecan see that the poorest agreement is obtained at the shorter wavelength channels (440 nm, and especially 380 nm).

Kazadzis et al. (2018b) also found a decrease in the percentage of AOD meeting the WMO criteria for 368 nm and 412 nm spectral bands during the Fourth WMO Filter Radiometer Comparison for aerosol optical depth measurements. As these authors pointed out, the shorter the wavelength, the poorer the agreement because of several reasons: AOD in the UV suffers from out--of-band or at least different blocking of the filters, small differences in central wavelength or FWHM have a larger impact, the Rayleigh correction is more critical, and $NO_2$ absorptions are treated differently. Regarding the effect of the aerosol load and particle size on the AOD differences, our results confirm the decrease of agreement between the two instruments for very large particles coincident with almost pure dust (AE ≤ 0.3), and high turbidity conditions (AOD > 0.1). However, it should be noted that the percentage of data pairs in these situations is relatively low (e.g., 6% for AOD > 0.1, and 3.2% for AE > 0.25 at 380nm) with respect to the total data (Table 4). A similar result was reported by Kim et al. (2008), who attributed these discrepancies to the possible spatial and temporal variability of aerosols under larger optical depths in addition to the effect of the different FOV of both radiometers. In our case, and according to previous studies on AOD climatology at IZO (Barreto et al., 2014), the presence of high mineral dust burden when the station is within the SAL, does not necessarily imply lower atmospheric stability conditions resulting in daily AOD means with greater standard deviation. For these reasons, we assumed that the different FOV of these instruments is can be one of the main causes of part of the AOD 1-minute differences outside the U95 limits, under high AOD conditions. This issue is specifically addressed in Section 5.34.

[Figure]

*Figure 2. One-minute AOD data differences between AERONET-Cimel (V2) and GAW-PFR for (a) 380 nm (70838 data-pairs), (b) 440 nm (71645 data-pairs), (c) 500 nm (70833 data-pairs) and (d) 870 nm (71660 data-pairs) for the period 2005-2015. Black dots correspond to the U95 limits. A small number of  outliers are out of the ±0.06 AOD differences range. Black arrows  indicate a change of Reference AERONET-Cimel radiometer and red arrows indicate a change of the GAW-PFR instrument.*

*Table 3. Percentage of AERONET-Cimel (V2 and V3) 1-minute AOD data meeting the WMO criteria for the four interpolated GAW-PFR channels for the period 2005-2015.*

| Interpolated GAW-PFR channel (%) | Original GAW-PFR channel (%) |
|---|---|
|  |  |
|  |  |
|  |  |
|  |  |

| Channel | V2 (%) | V3 (%) |
|---|---|---|

| | | |
|---|---|---|
| 380 nm | 92.7 | 92.3 |
| 440 nm | 95.7 | 95.2 |
| 500 nm | 95.8 | 95.7 |
| 870 nm | 98.0 | 97.8 |

*Table 4. Percentage of AERONET-Cimel 1-minute AOD data (V2) meeting the WMO criteria for the four compared channels, and different AOD and AE scenarios for the period 2005-2015, number of data pairs are shown in brackets. The lLast row corresponds to the total percentages for the sub-period 2010-2015. In bold, AOD and AE traceability is > 95% are marked in bold. Number of data pairs are in brackets.*

| % of data within WMO limits | 380 nm | 440 nm | 500 nm | 870 nm |
|---|---|---|---|---|
| AOD ≤ 0.05 | 94.4 (57008) | **96.8 (59130)** | **97.0 (58572)** | **98.5 (60191)** |
| 0.05 < AOD ≤ 0.10 | 91.0 (4723) | 93.1 (4850) | 92.8 (4817) | 94.2 (4908) |
| AOD > 0.10 | 75.0 (3938) | 86.5 (4615) | 85.1 (4466) | **95.9 (5118)** |
| AE ≤ 0.25 | 73.1 (2145) | 82.3 (2417) | 80.1 (2351) | **96.2 (2824)** |
| 0.25 < AE ≤ 0.6 | 91.2 (5407) | **96.2 (5810)** | **96.0 (5691)** | **97.9 (5911)** |
| AE > 0.6 | 94.6 (55114) | **96.9 (57089)** | **97.0 (56504)** | **98.7 (58146)** |
| Total 2005-2015 | 92.7 (65669) | **95.7 (68595)** | **95.8 (67855)** | **98.0 (70217)** |
| Total 2010-2015 | 93.5 (41977) | **97.4 (43745)** | **97.2 (43627)** | **99.1 (44498)** |

In general, the agreement obtained with the 1-minute AOD data is slightly lower than that obtained during short campaigns, such as those reported by (Kazadzis et al. (, 2014) at Athens observatory (4685 data-pairs), and Barreto et al. (2016) at Izaña ObservatoryIZO (5566 data-pairs), with agreements > 99 % for $AOD_{870nm}$ and $AOD_{500nm}$ in case of Barreto et al. (2016). However, our results for $AOD_{500nm}$ (> 95 % of 70833 data-pairs) areis significantly better that that observed by Kazadzis et al. (2014) (~ 48 % of 4685 data-pairs) covering a relatively narrowshort range of AOD. The probable cause for the poor agreement found by (Kazadzis et al., 2014) was a poor calibration in the 500 nm channel in at least one of the instruments operating at Athens.

In addition, short-term campaigns usually cover a small range of AOD, normally with low AOD, and instruments are carefully and frequently supervised. On the contrary, during our intercomparison over a period of 11 years, the operation of the instruments can be described much moreconsidered as the normal operation of such a system. for a long term period of measurements, 20 than that of intensively attended instrumentation during short period intercomparison campaigns.

An additional interesting aspect of this study is that it is not a simple intercomparison exercise between two instruments but a comparison of a number of instruments that acted as reference instruments  for the AERONET/Europe Network.

*Table 5. Basic skill-scores from the AOD intercomparison between GAW-PFR and AERONET-Cimel V2 for the period 2005-2015. The skill scores definitions are found in Huijnen and Eskes (2012).*

| Period | 2005-2015 | | | |
|---|---|---|---|---|
| Wavelengths (nm) | 380 | 440 | 500 | 870 |
| Mean Bias (MB) | -0.0026 | -0.0018 | -0.0021 | -0.0001 |
| Modified Normalized Mean Bias (MNMB) | -0.1301 | -0.1046 | -0.1474 | 0.0129 |
| Fractional Gross Error (FGE) | 0.1727 | 0.1546 | 0.1918 | 0.1837 |
| Root Mean Square Error (RMSE) | 0.0081 | 0.0070 | 0.0064 | 0.0049 |
| Pearson's correlation coefficient (r) | 0.9910 | 0.9925 | 0.9939 | 0.9949 |
| Number of data-pairs | 70838 | 71645 | 70833 | 71660 |

In the first period (2005-2009), a total of 13 Cimel radiometers were used, while in the second period (2010-2015), five Cimel radiometers have participated, and for much of this period, the Cimel #244 was operating as the permanent AERONET reference instrument at IZO. Once the most important causes of non-traceability in the first period, which were associated with a poor pointing of GAW-PFR due to problems in the sun-tracker, were discounted, we can conclude that there are no significant differences in the percentages of traceable data between the two  periods. This means that the continuous change of Reference Cimel instruments used in the 2005-200910 period did not have a significant impact on AOD data comparison differences. This provides proof of  the consistency and homogeneity of the long AERONET-Cimel AOD data series, and their comparability with the GAW-PFR AOD data series, regardless of the number of instruments used to generate these data series.

In our study, with a number of comparison data-pairs one or two orders of magnitude higher than those used in short campaigns, the results shown in Table 4 can be considered as fairly good.

In addition to the traceability scores, we have introduced some basic skill scores corresponding to the AOD intercomparison between GAW-PFR and AERONET-Cimel for the period 2005-2015 (Table 5) to be in line with previous studies that have performed short-term comparisons between these two instruments. The definitions of the used skill scores can be found in Huijnen and Eskes (2012).

The Pearson's correlation coefficient (r) values of the PFR-Cimel 1-minute AOD data-pairs are higher than 0.99 in all channels. Concerning Mean Bias (MB) and Root Mean Square Error (RMSE) associated with AOD differences, our results show quite similar skill scores to those found at Mauna Loa, USA for $AOD_{500nm}$ (Kim et al., 2008), although the number of data pairs used at IZO (~71000) is

much higher ~~than that of Mauna Loa (~9700)Master or R~~reference instruments. These results show MB values to be within 0.01 bias, one order of magnitude lower than in Mauna Loa and Izaña Observatories, highlighting the importance of having well calibrated instruments to carry out these type of comparisons.

For the period 2010-2015 (not shown here),  as expected, the RMSE and the Pearson's correlation improve slightly compared with the whole period 2005-2015.
* * *
**5.3. Non-traceability assessment**

As presented in the able 3, data outside the WMO traceability criteria vary from 2% for 870 nm up to 7.3% for 380 nm. In this section, the different possible causes of non-traceability in AOD are evaluated and, if possible, quantitatively estimated. In order to assess the relevance and quantitative impact of these causes, and estimate errors derived from a non-perfect AOD data synchronization, we first made an analysis on the natural variability of AOD in a very short time period (1 minute) shown below.

**5.3.1. Short-time AOD variability**

In order to determine the variability of AOD within one minute, we have performed two independent analyses with AOD data from the PFR and Cimel for the  368/380 nm and 501/500nm channels during one year (2013). On the one hand, and taking into account that GAW-PFR provides AOD every minute, we have calculated all the AOD differences for each channel in the successive minutes. So we have the variation of AOD from one minute to the next one during a whole year. On the other hand, for AERONET-Cimel, we have taken advantage of the triplets, since each triplet consists of three successive measurements made in one minute . In this case, the strategy has been to calculate the standard deviation of the triplet AOD measurements during a whole year. We have verified that the AOD variability in 1 minute is independent of AOD (see Supplement S3).

*Table 6. Percentage of AOD data with variability within 1 minute less than 0.01 and 0.005, respectively, using AOD data from GAW-PFR (at 368 and 501nm) and AERONET-Cimel (at 380 and 500 nm) for 2013. A total of ~32000 d data-pairs per channel have been used from GAW-PFR, and 20117 triplets (60351 individual AOD measurements) from the Cimel#244 to calculate the AOD variability.*

| GAW- PFR |
|---|

| Percentage of data with 1-minute AOD variability (%) | | |
|---|---|---|
| | 368 nm | 501 nm |
| < 0.01 | 99.88 | 99.91 |
| < 0.005 | 99.21 | 99.35 |
| AERONET-Cimel | | |
| Percentage of data with 1-minute AOD variability (%) | | |
| | 380 nm | 500 nm |
| < 0.01 | 99.87 | 99.99 |
| < 0.005 | 99.82 | 99.42 |

The results obtained on the AOD variability in 1-minute from PFR data are very similar and consistent to those obtained with Cimel. Less than ~ 0.8% of the AOD data show variability higher than 0.005 in all wavelength ranges. It should be noted that the possible instrumental noise is included in this variability, so that the actual natural AOD variability would be, in any case, lower than that expressed in Table 6. The percentage of data with 1-minute AOD variability for allthe four GAW-PFR channels are given in Supplement S3. These results indicate that the natural AOD variability is very low thus the non-ideal measurement synchronization cannot explain the percentages of non-traceable AOD cases shown in Tables 3 and 4.

**5.3.2. Uncertainties ofby GAW-PFR channel interpolation to AERONET-Cimel channels**

The interpolation of the CIMEL AODs to the PFR AOD wavelengths can be one of the sources of uncertainty in this comparison assessment. The greatest uncertainty arises in the extrapolation of the $AOD_{412 nm}$ of the PFR to the Cimel wavelength 440 nmretrieve AODs at the CimelCIMEL $AOD_{440nm}$.

Using the Angström formula we have calculated that for an uncertainty of ±0.5 in the Angström exponentAE and for AOD of 0.1 at 412 nm, the introduced uncertainty in the AOD extrapolation from 412 nm to 440 nm is ~5% (i.e., 0.005 for $AOD_{412nm}$=0.1). The introduced uncertainty in AOD extrapolation is reduced to ~2% for an uncertainty of ±0.3 in AE. of the order of ±0.003, while for an $AOD_{412nm}$ of 0.5 and an AE uncertainty of ±0.3, the introduced uncertainty is ±0.008. For all other AOD interpolations the errors are smaller.

**5.32.31 Calibration related errors**

[revised manuscript text omitted]

The correct cause attribution of each outlier would require manual inspection and additional specific information on instrumental checking and maintenance information that is not always available. We have investigated in more  detail the origin of the outliers and whether one of the two instruments predominantly caused them.

Thus, we have calculated for the non-traceable AOD data the diurnal range of AOD variation (maximum value minus minimum value of AOD in one day) at 380 nm for each instrument under pristine conditions (Figure 3) using Cimel AOD$_{500nm}$ daily mean < 0.03 to select the pristine days. According to this approach, the instrument that shows the highest daytime AOD range is the one that is responsible for the outlier. As the wavelength increases both the number of outliers and the magnitude thereof decreases significantly. (Supplement S6). Then, we identified those outliers with a diurnal AOD range higher  than 25% of the mean daily AOD value and investigated their possible causes. A total of 51 cases for GAW-PFR and 81 cases for AERONET Cimel V3 were obtained and  analysed in detail, using  auxiliary information, such as 1-minute in-situ meteorological data, 5-minute all-sky images, 1-minute BSRN data, and satellite imagery (not shown here). We obtained the percentage of AOD outliers of GAW-PFR and AERONET Cimel (V3) for which a certain cause has been identified, such as calibration uncertainties, cloud screening algorithm failures, mixture of the two previous causes, poor sun pointing, or not well-defined  causes (electronic problems, humidity inside the lenses, filter dirtiness, obstruction of the lenses collimators, insects on the optics outside, etc.) (see Supplement S7).

From the analysis of these cases, under the conditions described above, it should be noted that ~ 44% of the cases with fictitious AOD diurnal cycles were due to small uncertainties in the calibration of AERONET-Cimel (V3), while for this same cause  ~ 8% of cases were identified in GAW-PFR.

Some examples of  AOD non-traceability for both AERONET-Cimel and GAW-PFR in the ~380 nm channel are shown in Supplement S8. The fictitious diurnal AOD cycle is mainly visible in the UV channels as shown in the examples reported in  Supplement S9.  that the fictitious diurnal AOD  can be more easily identified  under very low AOD conditions.

[Figure]

*Figure 3. AOD diurnal range  variation (maximum value minus minimum value of AOD in one day) at 380nm corresponding to AOD outlier (non-traceable AOD)  under pristine conditions (AOD$_{Cimel-500nm}$ ≤ 0.03) in the period 2005-2015 for AERONET V2 (a) and V3 (b).*

*5.32.4.2      Differences in cloud-screening and sun tracking.*

In this section wWe have examined the effect that the presence of clouds might have on AOD differences and the number percentage of cases outside the U95 limits. The impact of clouds on AOD differences only occurs when both GAW-PFR and AERONET-Cimel cloud screening algorithms fail to identify clouds in the direct sun path. AERONET-Cimel Version 2 data uses the so-called "triplet check" cloud screening algorithm developed by Smirnov et al. (2000) and a second order temporal derivative constraint (McArthur et al., 2003) to rule out AOD measurements potentially contaminated by clouds. GAW-PFR algorithms also use the Smirnov triplet measurement, and the second-order derivative check, but add a test for optically thick clouds with $AOD_{500nm} > 2$ (Kazadzis et al., 2018b). This algorithm, used by both networks with certain variants, assumes a transitory character in the presence of a cloud, which causes a sudden change of AOD. This sharp change would be detected by measuring the stability of three successive optical depth measurements, so that, when a cloud totally or partially blocks the sun, the standard deviation associated with the average of the triplets increases enormously. Note that if either one or both cloud screening algorithms (GAW-PFR and AERONET-Cimel) are flagged as cloudy, then the corresponding AOD data pair does not take part in the comparison. However, in the case of stratiform and very stable clouds or in the case of very thin clouds such as cirrus clouds, the algorithm could erroneously interpret that there are no clouds since there would be no appreciable changes in the stability of the triplets. A hint that cloud flagging failure could lead to large ADO calculated differences is coming from an analysis of AOD differences for days with different cloudy sky fractions. We do not have precise ancillary information to verify in each 1-minute data the influence that a certain cloud could cause in the non-traceability found. As a first approach forIn order to assess assessing the impact that cloud conditions might cause on AOD traceability, we have used the concept of daily fractions of clear sky (FCS) that has been applied before to solar radiation data at Izaña ObservatoryIZO (García et al., 2014). FCS represents the percentage of observed sunshine hours in a day with respect to the maximum possible sunshine hours in that day. The higher the daily FCS, the higher the clear sky percentage we have on that day.

The percentages of traceable and non-traceable AOD data versus FCS values grouped into 5five intervals are shown in Table 7Table 8. The results indicate that with a FCS lower than 20 % (almost overcast skies), and for wavelengths lower than 870 nm, data outside the U95 limits comprises, at least, 50 % of the total AOD data. It should be emphasized that the number of cases linked with FCS between 0% and 60% are less than 2% of the total cases. There, ~8% (870nm) to 24% (380nm) of the data are outside the WMO limits (maximum of 0.5% of the total data for 380nm outside the WMO limits). As the fraction of clear

sky increases, the percentage of traceable AOD data significantly exceeds the number of non-traceable AOD data. The percentage of traceable data is especially large (> 90 %) when FCS > 80 % (almost clear skies).

This is the FCS range in which a significant percentage of days with cases presenting scattered clouds are recorded, which qualitatively confirms that V3 has introduced more efficient cloud screening than V2. However, the real impact of clouds on AOD traceability at IZO is very low due to its special characteristics of a high mountain station with very little cloudiness.  Therefore, in practice, the possible impact of clouds on the non-traceability of AOD data-pairs is insignificant at IZO .  GAW-PFR and AERONET-Cimel cloud screening algorithms provide successful identification on clear direct-sun conditions during cloudy skies (FCS < 40 %) for  99.75 % of the cases, excluding those with very thin clouds.  of AERONET V3 (Giles et al., 2018).

However, in the case of stratiform and very stable clouds or in the case of very thin clouds such as cirrus clouds, the algorithm could erroneously interpret that there are no clouds since there would be no appreciable changes in the stability of the triplets.

In the particular case of Izaña there are some very specific cloud scenarios in which cloud screening algorithms could fail resulting in non-AOD traceability: 1) Altostratus above the top of the SAL, at ~6 Km altitude (see Supplement S10); 2) Cirrus clouds (see Supplement S11); and 3) low clouds (stratocumulus) that sometimes exceed the observatory height level (see Supplement S11).

~~A more detailed analysis of more rare atmospheric conditions, such as those of strati-form and homogeneous cirrus clouds, or when altostratus are present above the Saharan Air Layer (SAL), around 6 Km altitude, and thus masked by a heavy mineral dust layer below needs further investigation. A constant cloud optical thickness (COT) corresponding to a cloud of a certain horizontal extension would cause the successive measurements within a minute to correspond to the same cloud stage, and therefore it would not be discernible from the extinction caused by aerosols. In the case of very thin cirrus clouds, the fluctuations in AOD would be very small and could be interpreted as the presence of a light layer of aerosols. Another factor that must be taken into account is that the FOV of the instruments is different. Thus, GAW-PFR (FOV = 2.5°) could detect the entry of a constant COT cloud in part of its~~

 shown reported1-minute

[revised manuscript text omitted]

climatology. Concerning AERONET Cimel Version 2, a NASA TOMS 1°_x_1.25° resolution $O_3$ climatology is used. From Eq. 10, the differences in $O_3$ optical depth $\Delta 4 \tau_{O3}$ can be derived:

$$\triangle \tau_{O3} = \sigma_{O3}(\lambda) \frac{1}{1000} \frac{m_{O3}(O_{3PFR} - O_{3Cimel})}{m_a}$$

(12)

The largest influence of total ozone data uncertainty in $\tau_{O3}$ occurs , by far, at 500 nm (Figure 5). According to Wehrli (2008b) and Kazadzis et al. (2018b), total ozone needs to be determined to ± 30 DU or 10 % of typical values, to ensure an uncertainty of ± 0.001 in $\tau_{O3}$ at 500 nm. In the case of the GAW-PFR / AERONET-Cimel comparison, and due to the very different method in which both networks obtained $O_3$ values for their corresponding corrections, the ozone differences found on some days (1761 out of 71965 days; 2.4 %) are very large (> 40 DU), exceeding a difference in the ozone optical depth of 0.001. Even so, the potential contribution toof AOD differences outside the $U$95 limits between the two networks is negligible. Total $O_3$ over the Izaña ObservatoryIZO is quite stable, showingshows a relatively small amplitude throughout the year, but both surface ozone concentrations and column ozone amount could sharply increase under the influence of cut-off lows injecting air from the high-mid troposphere into the lower subtropical troposphere, which is not uncommon in spring and the first half of summer (Cuevas et al., 2015; Kentarchos et al., 2000). In addition through exchange processes in the Upper Troposphere Lower Stratosphere (UTLS) due to the presence of the subtropical jet (mainly from February to April) (Rodríguez-Franco and Cuevas, 2013). However, if we wanted to repeat this traceability study of 1-minute AOD data in mid or high latitude stations where sharp $O_3$ variations (several tens of DU) could be registered in a few hours, the correction of 1-minute AOD measurements by $\tau_{O3}$ might be a challenging issue.

[Figure]

*Figure 5. (a) Total O₃ used by GAW-PFR (measured Brewer O₃ values from IZO, OMI O₃ overpass or Brewer O₃ climatology) and AERONET-Cimel (TOMS O₃ climatology), and (b) Δτ$_{O3}$ (λ) caused by differences in daily total O₃ between the two instruments in the period 2005-2015.*

5.3.54.3 Differences in NO₂ absorption

AERONET-Cimel applies a correction by absorption of NO₂, but GAW-PFR does not include this correction. AERONET--Cimel obtains daily total NO₂ data from a 0.25° x 0.2° resolution NO₂ monthly climatology obtained from the ESA Scanning Imaging Absorption SpectroMeter for Atmospheric CHartographY (SCIAMACHY) (Eskes and Boersma, 2003). In order to assess the contribution toin AERONET-Cimel 1-minute AOD data non-traceability by NO₂ absorption what we havereally to estimate is the NO₂ optical depth ($τ_{NO2}$(λ)) of AERONET-Cimel since GAW-PFR does not perform this correction. Analogously to Δτ$_{O3}$, the differences in nitrogen dioxide optical depth Δ4τ$_{NO2}$ can be obtained from:

$$\triangle \tau_{NO2} = \sigma_{NO2}(\lambda)\frac{1}{1000}\frac{m_{NO2}}{m_a}(-NO_{2Cimel})$$

(13)

Where $m_a$ is given by Eq. 7, $NO_{2Cimel}$ (DU) is the daily total NO$_2$ used by AERONET-Cimel, $\sigma_{NO2}(\lambda)$ is the NO$_2$ absorption  (Gueymard, 1995)  weighted by the specific filter response: 15.6 cm$^{-1}$ (380 nm), 12.3 cm$^{-1}$ (440 nm), and 4.62 cm$^{-1}$ (500 nm). Finally,  $m_{NO2}$ has the following expression (Gueymard, 1995):

$$m_{NO2} = \frac{1}{sin\theta + 602.30(90 - \theta)^{0.5}(27.96 + \theta)^{-3.4536}}$$

(14)

[Figure]

*Figure 5. (a) Total O₃ used by GAW-PFR (OMI O₃ overpass or Brewer O₃ climatology) and AERONET-Cimel (TOMS O₃ climatology), and (b) Δτ₀₃(λ) caused by differences in daily total O₃ between the two instruments in the period 2005-2015.*

[Figure]

*Figure 6. (a)*  *NO₂ monthly climatology obtained from the ESA SCanning Imaging Absorption SpectroMeter for Atmospheric CHartographY (SCIAMACHY), used by AERONET-Cimel at* *IZO, and (b)* Δ4τ₆NO₂ *(λ) caused by differences in daily total NO₂ between GAW-PFR and AERONET-Cimel in the period 2005-2015. Note that GAW-PFR does not take into account the correction for the NO₂ absorption.*

*Table 9. Percentage (%) of* additional *traceable AERONET-Cimel AOD 1-minute data* (V2 and V3)  *after correcting by pressure, and total column O₃ and NO₂* for the period 2005-2015.

| Channel | Increment (%) of traceable data after P, O₃ and NO₂ corrections |
|---------|------------------------------------------------------------------|
| 380 nm | 1.3 |
| 440 nm | 0.2 |
| 500 nm | 0.3 |
| 870 nm | ~ 0.0 |

| Channel | Increment (%) of traceable AOD data after P, $O_3$ and $NO_2$ corrections | |
|---------|------|------|
|         | V2   | V3   |
| 380 nm | 1.3 | 1.7 |
| 440 nm | 0.2 | 0.3 |
| 500 nm | 0.3 | 0.1 |
| 870 | ~0.0 | ~0.1 |

In Figure 6a the total $NO_2$ used by AERONET-Cimel to evaluate $\tau_{NO2}(\lambda)$ is depicted. Figure 6b shows the $\Delta\tau_{NO2}(\lambda)$ caused by differences in daily total $NO_2$ between GAW-PFR and AERONET-Cimel. $\Delta\tau_{NO2}$ is of the order of $10^{-3}$ for 380 and 440 nm channels, while, for 500 nm channel, it is of the order of $10^{-4}$. As for O₃, the absorption due to total NO₂ is negligible in the 1-minute AOD non-traceability in our study. However, it should must be notedtaken into account that an impact on AOD calculation is expected when replicating similar analysis in if this type of traceability analysis is replicated in highly $NO_2$ polluted regions where the NO₂ absorption might have somean impact on AOD calculation is expected. Such cases include, such as in large industrial cities from East Asia and Central and Eastern Europe , in which tropospheric NO₂ adds to the natural stratospheric NO₂ resulting in column values much larger than the climatological ones (e.g., Chubarova et al., 2016).

Taking into account the corrections for Rayleigh scattering and for the absorptions by $O_3$ and $NO_2$, we have calculated the additional traceable AOD of datacombin that lie within the U95 AOD limits ed effect of all of them on percentage the non-traceability of the 1-minute AOD values (Figure 6; Table 8Table 9). This percentage is maximum at 380 nm with 1.3% (V2) and 1.7% (V3) of the whole dataset. At most (in the 380 nm channel), 25 % (1.3 % of total common measurements) of data outside the $U95$ limits are due to significant differences in pressure, and in O₃ and NO₂ absorption. Most of the AOD data outside the $U95$ limits that becomes traceable data after corrections are applied, had errors in the pressure measurement and therefore in the Rayleigh scattering correction. The 870 nm channel is only affected by the Rayleigh correction component and therefore the increment of traceable data after the mentioned corrections is

minimal.

4 *GAW PFR and AERONET-Cimel comparison under high AOD conditions: the impact of dust forward scattering  for different FOVs*

When we present the AOD differences between AERONET-Cimel and GAW-PFR versus AOD (GAW-PFR) for AOD > 0.1 (dusty  conditions), we not that AERONET-Cimel shows slightly higher AOD values than GAW-PFR (Figure 7).

In fact, the percentage of data outside the U95 limits increases as AOD increases (Table 10), so  for dust-related aerosol conditions (AOD$_{500nm}$ > 0.3) the percentage of AOD data outside the *U*95 limits is > 50 % for 380 nm and 440 nm (Table 10, percentages in brackets). Similar results are found when using AERONET V3 (see Supplement S13).  Taking into account the number of data compared with the total cases, these results show a small but non-negligible percentage of AOD differences outside the *U*95 limits for AOD > 0.1, ranging from ~0.3 % at 870 nm to ~ 1.9 % at 380 nm (Table 10).

[Figure]

*Figure 7. Actual AOD differences between AERONET-Cimel V2 and GAW-PFR vs PFR AOD$_{PFR}$ at (a) 380 nm (b) and 500 nm for the period 200510-2015. The fitting line has been calculated with those data points with AOD data > 0.1 and whose Cimel-PFR AOD difference > 0. The nNumber of data used in the plots are indicated in the legend. The percentage of non-traceable AOD data with these conditions is ~24% for 380 nm, and ~8% for 500 nm. Note that some traceable (black) points show larger AOD differences than non-traceable (red) points because of the air mass dependence of the WMO traceability criterion.*

*Table 10. Percentage of  AOD data outside the U95 limits at 380, 440, 500 and 870 nm channels and for three AOD$_{500nm}$ thresholds respect to all data and respect to all data for each AOD interval (in brackets).*

| | Percentage of AOD data outside the $U$95 limits (%) | | |
|---|---|---|---|
| | AOD$_{500nm}$>0.1 | AOD$_{500nm}$>0.2 | AOD$_{500nm}$>0.3 |
| 380 nm | 1.9 (25.0) | 1.2 (47.2) | 0.5 (59.8) |
| 440 nm | 1.0 (13.5) | 0.8 (32.0) | 0.5 (57.6) |
| 500 nm | 0.6 (8.0) | 0.5 (18.7) | 0.3 (39.3) |
| 870 nm | 0.3 (4.1) | 0.2 (6.4) | 0.1 (14.0) |

aerosol forward scattering within the FOV of various instruments and calculated AOD was investigated some decades ago by Grassl (1971) who determined that at AOD=1 the circumsolar radiation increases by >10% the incoming radiation. Russell et al (2004), using dust and marine aerosols data, quantified the effect of diffuse light for common sun photometer FOV. They reported that the correction to AOD is negligible (<1% of AOD) for sun photometers with narrow FOV (< 2°), which is greater than  the Cimel FOV and slightly smaller than the PFR FOV  (2.5°). Sinyuk et al. (2012) assessed the impact of the forward scattering aerosol on the uncertainty of the AERONET AOD, concluding that only dust aerosol  with high AOD and low solar elevation could cause a significant bias in AOD (> 0.01). ~~Torres et al. (2013) investigated the uncertainty of the FOV in the AERONET-Cimel measurements indicating that direct solar irradiance measurements are biased by the amount of aureole radiation that is assumed to be direct solar radiation. The solar aureole, also known as the circumsolar region, is the bright region surrounding the solar disc, which becomes especially visible when there is a burden of moderate-high aerosols in the atmosphere.~~

[revised manuscript text omitted]
 completelyis fairly good but it is not complet, e-mainly for two reasons. The first one it is the inherent limitation of data correction using the percentage difference in AOD obtained by model simulation for a fixed effective radius.

We have assumed an effective radius of 1.5 μm but, in reality, the radius of dust particles varies. A reasonable range of dust particle radiussize is between 0.1 and 3 μm (Balkanski et al., 1996; Denjean et al., 2016; Mahowald et al., 2014). So, depending on the distance from the dust source to Izaña ObservatoryIZO and the size of the emitted dust, the effective radius could vary slightly between dust episodes. As can be seen in Figures 8a and 8b, the percentage differences in AOD between Cimel and PFR for a 1-2[1,2 μum] effective radius interval, the PFR-Cimel AOD relative difference at 380 nm (500 nm) might change between (around) ~-1.8% (-1.1%) to -4.9% (3.3%).

The second reason is a possible cloud contamination in AOD retrieval when altostratus are present above the SAL, as discussed in Section 5.3.4.

A similar analysis has been carried out for AERONET V3 (see Supplement S17), where we observe that the corrections obtained are not as good as those obtained for V2. This may be due to the very high AOD data retention in V3 which could include more cases in which altostratus clouds and dust are present. The effect of FOV on AOD retrieval should be taken into account for those radiometers with a relatively high FOV (>3°) measuring in regions with relatively high AOD (> 0.2) for most of the year, as is the case in many sites of Northern Africa, the Middle East and East Asia (Basart et al., 2009; Cuevas et al., 2015; Eck et al., 1999; Kim et al., 2007). This effect leads to AOD underestimation, and the variable number of high AOD episodes in each season of the year might affect the AOD long-term trends. AOD measurements under these conditions would be especially affected for optical air mass < 3.

[Figure]

*Figure 9. The same as Figure 7 after "correcting" the PFR AOD data by adding + 3.3% at 380nm and + 2.2.% at 500 nm to the 1-minute PFR AOD data > 0.1.*

Furthermore, it should be taken into account that, as discussed in section 5.2.2., under relatively high AOD conditions, the presence of altostratus above SAL is not infrequent, and they could also cause non-traceability in AOD when the cloud screening algorithms

fail. Note that a graphic equivalent to Figure 9 is shown in Supplement S17 but for AERONET V3, observing that the corrections obtained are not as good as those obtained for V2. This may be due to the very high AOD data retention in V3 which could include more cases in which altostratus clouds and dust are present. Therefore, the FOV study should be done once the dust events with presence of clouds over the SAL have been ruled out.

The effect of FOV on AOD retrieval should be taken into account for those radiometers with a relatively high FOV (>3°) measuring in regions with relatively high AOD (> 0.2) for most of the year (Basart et al., 2009; Cuevas et al., 2015; Eck et al., 1999; Kim et al., 2007), as is the case in many places sites of Northern Africa, the Middle East and East Asia (Basart et al., 2009; Cuevas et al., 2015; Eck et al., 1999; Kim et al., 2007). This effect could leads to AOD underestimation, and the variable number of high AOD episodes in each season of the year might affect the AOD long-term trends. AOD measurements under these conditions would be especially affected for optical air mass < 3.

**5.5.4 Angström exponent comparison**

We have performed a comparicomparison of the AE provided by GAW-PFR and AERONET-Cimel using in both cases the AOD data obtained from the four common channels (380 nm, 440 nm, 500 nm and 870 nm) with a total of 70716 data-pairs. The PFR-AOD values have been ordered from lowest to highest by grouping them in intervals of 500 values for which the averages (and corresponding standard deviations) of the PFR-Cimel AE differences have been calculated ( to produce Figure 10a). In a similar way we proceeded with the PFR-AE values (Figure 10b).

[revised manuscript text omitted]

4.

In 94.93.8 % of the cases, GAW-PFR and AERONET-Cimel V2 match the AE intervals of each aerosol scenario. Similar results (93.4%) were obtained when comparing with AERONET V3. Most of the agreement (>80 79 %) occurs in the predominant scenario of pristine conditions despite the AE uncertainty under pristine conditions being ≥ 1. See Supplement S18 for more details. Notice that given the special characteristics of the Izaña Observatory, and according to Cuevas et al. (2015) and Berjón et al. (2019), AE is a self-sufficient parameter to define different types of aerosol scenarios without the need to combine its information with AOD.

**6. Summary and Conclusions**

In this study, a long-term comparison of synchronous 1-minute AOD data from GAW-PFR and AERONET-Cimel was carried out in four wavelengths (380, 440, 500 and 870 nm) for an 11 year period (2005-2015).

While GAW-PFR is the WMO- defined global AOD reference , being directly linked to WMO / CIMO, and was specifically designed to detect long-term AOD trends, AERONET-Cimel is the densest AOD measurement network globally, and the network most frequently used for aerosol characterization and for model and satellite observation evaluation.

An AERONET-Cimel 11-year AOD data series at IZO was obtained using a large number of radiometers. A total of 13 Reference instruments were used  in the period 2005-2009 which means that every 4 and a half months, approximately, an instrument was replaced by another one to be calibrated. Their calibrations were performed during their respective measurement time periods at IZO. Therefore, these calibrations were not in any way linked with those of the instruments that preceded or replaced them, nor with GAW-PFR reference. These facts led us to investigate the homogeneity of the AERONET-Cimel AOD data series and their intercomparability with the much more homogeneous AOD data series from GAW-PFR (3 instruments in 11 years).

is comparable and consistent. The traceability concept for AOD suggested by WMO consists in determining whether the AOD difference of the AERONET CIMELs vs the GAW PFRs lie within the U95specific limits.

We have used uncertainty limits for AOD traceability established by WMO (2005) for theseis type of instruments with finite FOV. The acceptable traceability is when 95 % of the absolute AOD differences lie within these limits, in which case both data populations are considered equivalent. It should be clarified that "traceability" is not used in a strict metrological sense.

This study has addressed the comparison of the GAW-PFR dataset base with the two versions of AERONET (V2 and V3) in the period 2005-2015. An excellent agreement between V2 and V3 for the four analyzedanalysed channels ($R^2 > 0.999$) has been obtained.

More than 70000 synchronous GAW-PFR (PFR) and AERONET-Cimel (Cimel) 1- minute data-pairs in each channel in the period 2005-2015 were analysed. An excellent traceability of AOD from the AERONET-Cimel (V2 and V3) is found for 440 nm, 500 nm and 870 nm, and fairly good results for 380 nmin the four channels. The lowest percentage of traceable AOD data is registered in 380 nm with 92.7 % of the 1-minute data within the WMO limits, and the highest in 870 nm with 98.0 % of the data. The percentage of traceable data pairs

The different possible causes of non-traceability in AOD Trying to identify the reasons of the AOD data outside the WMO limits we havewere investigated as follows:

- Absolute AOD measurements synchronization.
  Analyzing 1-minute AOD variability we concluded that its impact on the AOD differences is negligible as only ~0.8% of the AOD data has a variability larger than 0.005 in all spectral ranges.

- Sun tracking misalignments.

The 1-minute AOD differences Mean Bias of this study is 0.001, an order of magnitude lower than those obtained from previous short-term PFR-Cimel comparison campaigns, in which the Cimel instruments were calibrated by transferring the calibration coefficients by comparison with co-located Master instruments. This indicates the importance of good calibration and maintenance of the Cimel instruments to obtain AOD data very similar to that of GAW-PFR.

In this study, since the AERONET-Cimel radiometers were calibrated using the Langley plot technique at the Izaña Observatory, and the calibrations of the GAW-PFRs are directly traceable to the WMO-GAW reference, being double checked by Langley plot calibrations at Izaña, we have the best possible calibrations in the instruments used by both networks.

The results confirm that the AOD data outside the $U$95 limits due to calibration related errors is quite small and not observable for 440, 500 and 870 nm since AOD non-traceability is < 2.1 % for pristine conditions ($AOD_{500nm} \leq 0.03$) in these channels. In addition, no dependence of the 1-minute AOD differences with the air optical mass is observed. However, for 380 nm the percentage of non-traceable values increases up to

Sun tracking misalignments  constitute a serious problem and a major cause of non-traceability of AOD data-pairs as demonstrated by the AOD data outside the U95 limits from the period 2005-20 as a consequence of episodic problems with the sun-tracker of the GAW-PFR radiometer. For the 2010-2015 period the percentage of traceable data-pairs improves to 93.5% (380 nm), 97.4% (440 nm), 97.2% (500 nm) and 99.1% (870 nm). However, most of these cases could be identified and excluded from the analysis.

- Cloud screening failure by both network algorithms.

~~Regarding AOD non-traceability due to the different cloud-screening algorithms of both networks, it must be said that both algorithms are very similar. GAW-PFR uses the same cloud screening as AERONET-Cimel but incorporates some additional controls. The only reason for AOD non-traceability comes from the simultaneous failure of both cloud screening algorithms because if one or both of them detect clouds, the data will not be part of the comparison.~~ According to our observations, the simultaneous failure of both cloud screening algorithms might occur only under the presence of large and stable cirrus, or altostratus (~ 6000 m a.s.l.) on the top of a heavily dust loaded Saharan air layer, hiding very wide and stable clouds. In these cases, the radiometers interpret these clouds as aerosol layers and might provide values very different from the real AOD. This effect, or the comparison at IZO, however, this effect is negligible since GAW-PFR and AERONET-Cimel cloud screening algorithms provide successful cloud identification on clear direct-sun conditions during cloudy skies (FCS < 40 %) for 99.75 % of the cases.

- Pressure measurements related errors.

-

 Since the accuracy of the new barometers built into new radiometers is about 3 hPa, and only errors in atmospheric pressure > 30 hPa might produce an impact on Rayleigh scattering, the AOD non-traceability due to errors in Rayleigh scattering is negligible.

- Total column ozone input uncertainty.

The largest influence of total ozone data uncertainty in ozone absorption occurs mainly at 500 nm. Total ozone needs to be determined to ±30 DU or 10 % of typical values, to ensure an uncertainty of ±0.001 ozone absorption at 500 nm. In the case of the GAW-PFR / AERONET-Cimel comparison, despite the very different methods in which both networks obtained $O_3$ values for their corresponding corrections, large ozone differences were found (> 40 DU) only on 2.4 % of the days , resulting in a

difference in the ozone optical depth slightly above ~0.001. The potential contribution to non-traceable AOD values between the two networks is negligible. However, in mid or high latitude stations where fast $O_3$ variations of several tens of DU might be registered, the correction of 1-minute AOD measurements by ozone absorption might be an issue to be considered.

- Total column $NO_2$ input uncertainty.

The differences in $NO_2$ absorption caused by differences in daily total $NO_2$ between GAW-PFR and AERONET-Cimel is of the order of $10^{-3}$ for 380 nm and 440 nm channels, while, for 500 nm channel, it is even lower, of the order of $10^{-4}$. So, differences in $NO_2$ absorption are negligible in the 1-minute AOD non-traceability of our study. However, $NO_2$ absorption might have some impact on AOD in highly polluted regions, such as in large industrial cities, where column $NO_2$ values are much larger than the climatological ones.

Taking into account the corrections for Rayleigh scattering and for the absorptions by $O_3$ and $NO_2$, we have calculated the combined effect of all of them on the non-traceability of the 1-minute AOD values. The highest impact occurs in the 380 nm channel, in which 25 % of the AOD data outside the $U95$ limits (~2% of the total compared data) are due to significant differences in pressure, and in $O_3$ and $NO_2$ absorption.  The 1-minute AOD data outside the $U95$ limits by these corrections is negligible in the 870 nm channel.

~~We have to note that the excellent results of this 11 year comparison and the small differences found under the strict U95 criterion cannot be linked with the relatively low AODs that can be found at IZO. This is because absolute calibration errors contribute to the AOD calculation in an absolute way so larger than 1 % calibration errors for a given period of time can lead to even negative AOD calculations for IZO site.~~

- Impact of dust forward scattering in AOD retrieval uncertainty for different instrument FOVs

 Since GAW-PFR has almost double the FOV (~2.5°) compared to the AERONET-Cimel (~1.3°), and direct solar irradiance measurements are biased by the amount of aureole radiation that is assumed to be direct solar radiation, it is reasonable to expect that GAW-PFR is more affected by the circumsolar irradiance than AERONET-Cimel radiometer when AOD is relatively high.  Modelling the dust forward scattering we have shown ~~We have explained part of the non-traceabilities found for relatively high AOD values, by analysing the relationship between the differences in circumsolar radiation measured by both instruments with the differences observed in AOD. We have observed a clear relationship between the Cimel PFR AOD differences and the PFR Cimel circumsolar radiation differences, with the slope of the fitted line greater for shorter wavelengths (380 nm). These results show~~ that a non-negligible percentage of

the non-traceable 1-minute AOD data for AOD > 0.1, ranging from ~0.3 % at 870 nm to ~1.9 % at 380 nm is,  caused by the different FOV.  Due to this effect, the GAW-PFR provides AOD values which are ~3% lower at 380 nm, and ~2% lower at 500 nm, compared with AERONET-Cimel. However, AOD underestimation  could only have  some relevance  in dusty regions if radiometers with relatively large FOV are used.

A comparison of the AE provided by GAW-PFR and AERONET-Cimel has been performed using in both cases AOD data obtained from the four nearby common channels with a total of 70716 data-pairs. This is a very strict AE calculation since it is necessary that AOD be accurately measured by the four channels simultaneously. AE differences > 0.2 increase exponentially under very pristine conditions (AOD ≤ 0.03 and AE ≥ 1), reaching AE differences of up to 1.6. However, for these conditions the atmospheric aerosol load is practically zero and so, its characterization with AE does not have any importance in practice. Under non-pristine conditions or those with a high mineral dust content (associated AOD > 0.03 and AE < 1), the AE differences remain <  0.1.

Summarizing, we have presented for the first time a long- term (2005-2015) 1-minute AOD comparison among different types of radiometers belonging to different aerosol global networks. This comparison is a very demanding test of both GAW-PFR and AERONET-Cimel validated AOD datasets  since aerosol scenarios correspond to extreme conditions: either very low aerosol loading, a "pristine" scenario that reveals small uncertainties in the calibration and in the cloud screening, or large dust load, which leads to a significant increase in the forward scattering aerosol with AOD, resulting in a slightly higher AOD underestimation by the GAW-PFR. From this comprehensive comparison,  we can conclude that  both AOD datasets  are representative of the same AOD population, which  is a remarkable fact for the global aerosol community. It should be noted that AOD traceability at 380 nm (92.7 %) does not reach 95 % of the common data, the percentage recommended by WMO U95 criterion, so more efforts should be made to improve AOD in the UV range. In this study we have also  investigated the data that are out the WMO U95 limits  in order to understand their causes and to be eventually able to correct the small inconsistencies detected in instrumental and methodological aspects in the future.

Our results suggest that,  WMO/CIMO traceability limits could be redefined as a function of wavelength, and the recommended radiometer FOV range  should be reconsidered.

The widely deployed AERONET-Cimel and GAW-PFR datasets play a crucial role in understanding  long-term AOD changes and detecting trends, so it would be desirable for both networks to be linked to the same GAW-WMO related reference.

~~also to provide additional insight on long-term AOD trend analysis, and its significance and validity, based on single instruments and their calibration and AOD processing procedure and uncertainty budget. Finally, special attention should be paid to the AERONET Cimel AOD data series used in trend detection in combination with the used data set homogeneity and their periodic calibration transfer from Master instruments.~~

*Competing interests.* The authors declare that they have no conflict of interest.

*Acknowledgements.* The authors thank Luc Blarel and Philippe Goloub (LOA, CNRS-University of Lille, France) for supervising the periodic calibrations of the Cimel Reference instruments. This study has been performed in the frame of the WMO CIMO Izaña Testbed for Aerosols and Water Vapour Remote Sensing Instruments. The work was supported by the project "The Global Atmosphere Watch Precision Filter Radiometer (GAW-PFR) Network for Aerosol Optical Depth long term measurements" funded by the Federal Office of Meteorology and Climatology MeteoSwiss International Affairs Division, Swiss GCOS Office. Part of the AERONET-Cimel radiometers have been calibrated at Izaña Observatory by AERONET- EUROPE Calibration Service, financed by the European Community specific programs for Integrating Activities: Research Infrastructure Action under the Seventh Framework Programme (FP7/2007-2013), ACTRIS grant agreement No. 262254, and Horizon 2020 Research and Innovation Program, ACTRIS-2 grant agreement No. 654109. This research has received funding from the European Union's Horizon 2020 Research and Innovation Programme under grant agreement No. 654109 (ACTRIS-2).The funding by MINECO (CTM2015-66742-R) and Junta de Castilla y León (VA100P17) is also gratefully acknowledged. We thank the staff of the Izaña Observatory for their effort and dedication in maintaining the instruments. We acknowledge the constructive comments of the anonymous referees. Our colleague Celia Milford has improved the English language of the paper. In memory of Prof Klaus Fröhlich, former director of PMOD-WRC, who initiated the AOD measurements programme at the Izaña Observatory in 1984 within the WMO Background Atmospheric Pollution Monitoring Network (BAPMoN).

---

## Author Comment (AC2) · 7 Apr 2019

**Anonymous Referee #2 (amt-2018-438-RC2):**

**G.1. The paper "Aerosol optical Depth comparison between GAW-PFR and AERONET Cimel radiometers from long term (2005-2015) 1-minute synchronous measurements" is very interesting for the scientific community working on photometry. It is really well written, very accurate in the analysis of all the aspects affecting the comparison and it is pleasing to read. I recommend the publication on this journal also because homogenization of international networks of photometers is an important issue at this stage of research.**

Authors: Thank you for the positive assessment of the paper. Specific comments and suggestions are addressed below.

The Referee must take into account that, according to the requirements and comments of the Referees # 3 and # 4, new relevant analysis have been incorporated in the paper what has required large additional workload, resulting in a significant improvement of the paper.

The most relevant new analyses have been:

1) Comparison of GAW-PFR with version V3 of AERONET and previous comparison between V2 and V3.

2) New study on modelling the impact of near-forward scattering on the AOD measured by the PFR and Cimel radiometers (with different FOVs).

3) Study of AOD variability in 1 minute to rule out possible differences in AOD due to a non-perfect synchronization of the PFR and Cimel sampling data.

4) Incorporation of case studies on the impact on AOD due to small inaccuracies in the calibrations or the clouds contamination.

Much more information about the above points has been included in a new Supplement material document of 24 pages.

 **I have few specific comments for the authors:**

**S.1. 1. Introduction, lines 12-15. It paper is not true that only two global ground-based radiometer networks exist, that is AERONET and GAW. SKYNET (https://www.skynet-isdc.org/) and its regional subnetwork ESR (www.eroroskyrad.net) provide centralized AOD and other optical and physical aerosol parameters, on a daily bases and downloadable from the website. SKYNET is also attending several intercomparison campaigns against PFR as example www.eroroskyrad.net/quatram.html, and the Fourth WMO Filter Radiometer Comparison for aerosol optical depth measurements. So I'd suggest the authors to mention their existence.**

**Authors:** References to other networks have been requested by the four Referees. Although we referred to global networks with centralized data processing and databases, as well as standard calibration procedures in each of the networks, we fully agree to include a reference to other similar sunphotometer networks of global and regional scope.

**The new paragraphs introduced in the manuscript are as follows:**

*"…These are GAW-PFR (Global Atmosphere Watch - Precision Filter Radiometer; http://www.pmodwrc.ch/worcc/; last access: 05 September 2018) and AERONET-Cimel (AErosol RObotic NETwork - Cimel Electronique radiometer; http://aeronet.gsfc.nasa.gov; last access: 01 September 2018) networks. AERONET is, in fact, a federation of ground-based remote sensing aerosol networks established by NASA (National Aeronautics and Space Administration) and PHOTONS (PHOtométrie pour le Traitement Opérationnel de Normalisation Satellitaire; University of Lille- Service d'Observation de l'INSU, France; Goloub et al., 2007), being complemented by other sub-networks, such as, AEROCAN (Canadian sunphotometry network; Bokoye et al., 2001), AeroSibnet (Siberian system for Aerosol monitoring; Sakerin et al., 2005), AeroSpan (Aerosol characterisation via Sun photometry: Australia Network; Mitchell et al., 2017), CARSNET (China Aerosol Remote Sensing NETwork; Che et al., 2015), and RIMA (The Iberian network for aerosol measurements; Toledano et al., 2011).*

*There are other radiometer networks that in recent years have incorporated centralized protocols for data evaluation and databases, and performed regular intercomparisons with GAW-PFR and AERONET-Cimel. These include, for example, SKYNET (SKYradiometer NETwork), and its seven associated sub-networks, that uses the Prede-POM sky radiometer to investigate aerosol-cloud-solar radiation interactions (e.g. Campanelli et al., 2004; Nakajima et al., 2007; Takamura et al., 2004)."*

**The corresponding references have been added.**

**S.2. 3.GAW-PFR and AERONET-Cimel radiometers, line 21: specify if these Cimel models have one sensor for both direct and diffuse solar radiation ( new models) or two different sensors ( old version). Lines 21-24, the deterioration of the filter is however "not minimized" by the absence of a system for keeping the temperature constant inside the optics. Page 6 line 1-3, it is better stating also here that the final measurement is an average of the 10-s measurements, even if declared later. Line 12, is not the triad of PFR calibrated by lamps, but using Langley plot?**

**Authors:** The Cimel radiometers used in this study are CE318-N, the classical standard version of AERONET before accepting the triple CE318-T. This information appears already in the manuscript.

**Concerning specific information on the sensors, the Cimel CE318-N models used in this study and classified by number of sensors, for both direct and diffuse solar radiation, are listed below:**

- One sensor for both direct and diffuse solar radiation: Cimel instruments #25, #44, #45, #140, #244, #347, #421, #548.
- Two different sensors for direct and diffuse solar radiation: Cimel instruments #79, #117, #245, #380, #382, #383, #398.

However, we have not included this detailed information in the paper since we consider it is irrelevant in terms of AERONET level 2 V2 and V3 quality data.

Concerning filter deterioration, we agree. We have replaced "minimized" by "reduced". The corresponding sentences were slightly modified as follows:

*"The possible deterioration of the interference filters is reduced since they are only sun-exposed during three consecutive 1-second direct-sun measurements per channel, this cycle being scheduled every ~15 minutes. The rest of the time the Cimel is taking sky radiance measurements, or at rest position, looking downwards."*

We have completed the Cimel measurement schedule in section 4.1 as follows:

*"AERONET-Cimel takes a sequence of three separate measurements (1-second per filter) in one minute interval (each one every 30 seconds). This sequence of measurements is called "triplet" and it is performed every ~15 minutes for air masses lower than 2, and with higher frequency for lower solar elevations."*

The GAW-PFR is calibrated routinely onsite with regular Langley plots, and additionally is periodically compared with the PMOD GAW-PFR reference triad.

**S.3.  4. Data and methodology used in this study Line 24, format problem Line 33, state here that in the section 5.4 AE will be compared**

**Authors:  Corrected the format problem in line 24.**

**In section 4.1 we have added the following text::**

*"Synchronous AE data provided by both instruments have also been compared (see section 5.5). GAW-PFR determines AE using all four PFR wavelengths (Nyeki et al., 2015), while AERONET-Cimel uses different wavelength ranges (340-440 nm, 380-500 nm, 440-675 nm, 440-870 nm, 500-870 nm) (Eck et al., 1999). As a consequence, we have calculated a new AE for the Cimel radiometer using the four channels equivalent to those of the PFR."*

**S.4.  5.1 AERONET-Cimel AOD traceability Line 5, I don't know if it is easy to represent, but in Figure1 it would be interesting to highlight the changes of equipment during the time.**

**Authors:  We agree. We have included the change of instruments for both GAW-PFR (red arrows pointing down) and AERONET-Cimel (black arrows pointing up). See the following example for 870 nm.**

[Figure]

**S.5.   5.2.1 Calibration related errors line 16: are all the involved Cimel "Maters"?; "in-situ absolute calibration": with absolute do you mean by lamp? Lines 20-21: It is not clear why you transfer calibration among Cimels if they are all masters, and therefore calibrated separately. Figure 2: why there is a hole of data for optical mass at about 4.2?**

**Authors:  The term "absolute" is not correct. We refer to Langely plot calibrations. Each Cimel master is individually calibrated at Izana using the Langley plot calibration. Calibration factors are not transferred from Master to Master instruments, but to field instruments, and this is done in other labs (LOA-Lille, GOA-Valladolid, CARSNET-Beijing).**

**Regarding the gap around 4.2 masses of air is, indeed, caused by the Teide volcano mountain since the sunset and previous moments take place behind the Teide during a part of the year around winter.**

**S.6.  5.2.3 Did you consider a possible influence of WV absorption at 500 nm? Small differences came out during the Fourth WMO Filter Radiometer Comparison. Pag 19 line 11 Cimel doesn't measure pressure, so better saying " from the different way the two instruments assume the atmospheric pressure".**

**Authors:   Neither GAW-PFR nor AERONET Cimel consider the absorption of PWV at 500 nm. It has not been taken into account in this work. In any case, and according to Kazadzis et al. (2018) (their Figure 9), differences in column optical thickness (when compared with the PFR triad) due to PWV is of the order of 10[-4], very small.**

**S.7. Pag 24 line 9: Figure 6 is not about the combined effect of the 3 components, but about NO2.**

Authors:  **The reference to Figure 6 has been removed.**

[revised manuscript text omitted]

Considering the AE criteria established by Cuevas et al. (2015) and Berjón et al. (2019), we have identified the following four main categories according to the $AE_{PFR}$ and $AE_{Cimel}$ values:

1. $AE_{PFR}$ & $AE_{Cimel}$ > 0.6: Pristine conditions.

2. $0.25 < AE_{PFR}$ & $AE_{Cimel} \leq 0.6$: Hazy, mineral dust being the main aerosol component.

3. $AE_{PFR}$ & $AE_{Cimel} \leq 0.25$: Pure dust.

4. $AE_{PFR}$ and $AE_{Cimel}$ do not fit any of the previous categories.

In 94.9 % of the cases, GAW-PFR and AERONET-Cimel V2 match the AE intervals of each aerosol scenario. Similar results (93.4%) were obtained when comparing with AERONET V3. Most of the agreement (>80 %) occurs in the predominant scenario of pristine conditions despite the AE uncertainty under pristine conditions being $\geq 1$. See Supplement S18 for more details. Notice that given the special characteristics of the Izaña Observatory, and according to Cuevas et al. (2015) and Berjón et al. (2019), AE is a self-sufficient parameter to define different types of aerosol scenarios without the need to combine its information with AOD.

**6. Summary and Conclusions**

While GAW-PFR is the WMO-defined global AOD reference, being directly linked to WMO / CIMO, and was specifically designed to detect long-term AOD trends, AERONET-Cimel is the densest AOD measurement network globally, and the network most frequently used for aerosol characterization and for model and satellite observation evaluation.

An AERONET-Cimel 11-year AOD data series at IZO was obtained using a large number of radiometers. A total of 13 Reference instruments were used in the period 2005-2009 which means that every 4 and a half months, approximately, an instrument was replaced by another one to be calibrated. Their calibrations were performed during their respective measurement time periods at IZO. Therefore, these calibrations were not in any way linked with those of the instruments that preceded or replaced them, nor with GAW-PFR reference. These facts led us to investigate the homogeneity of the AERONET-Cimel AOD data series and their intercomparability with the much more homogeneous AOD data series from GAW-PFR (3 instruments in 11 years). The traceability concept for AOD suggested by WMO consists in determining whether the AOD difference of the AERONET CIMELs vs the GAW PFRs lie within the U95 limits. We have used uncertainty limits for AOD traceability established by WMO (2005) for these type of instruments with finite FOV. The acceptable traceability is when 95 % of the absolute AOD differences lie within these limits, in which case both data populations are considered equivalent. It should be clarified that "traceability" is not used in a strict metrological sense. This study has addressed the comparison of the GAW-PFR dataset with the two versions of AERONET (V2 and V3) in the period 2005-2015. An excellent agreement between V2 and V3 for the four analysed channels ($R^2 > 0.999$) has been obtained.

More than 70000 synchronous GAW-PFR (PFR) and AERONET-Cimel (Cimel) 1-minute data-pairs in each channel in the period 2005-2015 were analysed. An excellent traceability of AOD from the AERONET-Cimel (V2 and V3) is found for 440 nm, 500 nm and 870 nm, and fairly good results for 380 nm. The lowest percentage of traceable AOD data is registered in 380 nm with 92.7 % of the 1-minute data within the WMO limits, and the highest in 870 nm with 98.0 % of the data.

The different possible causes of non-traceability in AOD were investigated as follows:

- Absolute AOD measurements synchronization.

Analyzing 1-minute AOD variability we concluded that its impact on the AOD differences is negligible as only ~0.8% of the AOD data has a variability larger than 0.005 in all spectral ranges.

- Sun tracking misalignments.

Sun tracking misalignments constitute a serious problem and a major cause of non-traceability of AOD data-pairs as demonstrated by the AOD data outside the U95 limits from the period 2005-2009 as a consequence of episodic problems with the sun-tracker of the GAW-PFR radiometer. For the 2010-2015 period the percentage of traceable data-pairs improves to 93.5% (380 nm), 97.4% (440 nm), 97.2% (500 nm) and 99.1% (870 nm). However, most of these cases could be identified and excluded from the analysis.

- Cloud screening failure by both network algorithms.

According to our observations, the simultaneous failure of both cloud screening algorithms might occur only under the presence of large and stable cirrus, or altostratus ($\sim$ 6000 m a.s.l.) on the top of a heavily dust loaded Saharan air layer, hiding very wide and stable clouds. 
[revised manuscript text omitted]

Blanc, P., Espinar, B., Geuder, N., Gueymard, C., R., M., Pitz-Paal, R., Reinhardt, B., Renné, D., M., S., Wald, L., and Wilbert, S.: Direct normal irradiance related definitions and applications: The circumsolar issue, Solar Energy, 110, 561 – 577, https://doi.org/https://doi.org/10.1016/j.solener.2014.10.001, 2014.

Bodhaine, B. A., Wood, N. B., Dutton, E. G., and Slusser, J. R.: On Rayleigh optical depth calculations, Journal of Atmospheric and Oceanic Technology, 16, 1854–1861, 1999.

Böhm-Vitense, E.: Introduction to stellar astrophysics, volume 2: stellar atmospheres, Cambridge University Press, 260 pp., 1989.

Bokoye, A. I., Royer, A., O'Neill, N. T., Cliche, P., Fedosejevs, G., Teillet, P. M., and McArthur, L. J. B.: Characterization of atmospheric aerosols across Canada from a ground-based sunphotometer network:

AEROCAN, Atmosphere-Ocean, 39, 429–456, https://doi.org/10.1080/07055900.2001.9649687, 2001.

[revised manuscript text omitted]

Eissa, Y., Blanc, P., Wald, L., and Ghedira, H.: Can AERONET data be used to accurately model the monochromatic beam and circumsolar irradiances under cloud-free conditions in desert environment?, Atmospheric Measurement Techniques, 8, 5099–5112, https://doi.org/DOI = 10.5194/amt-8-5099-2015, 2015.

Eissa, Y., Blanc, P., Ghedira, H., Oumbe, A., and Wald, L.: A fast and simple model to estimate the contribution of the circumsolar irradiance to measured broadband beam irradiance under cloud-free conditions in desert environment, Solar Energy, 163, 497–509, https://doi.org/https://doi.org/10.1016/j.solener.2018.02.015, 2018.

Eskes, H. J. and Boersma, K. F.: Averaging kernels for DOAS total-column satellite retrievals, Atmospheric Chemistry and Physics, 3, 1285–1291, https://doi.org/10.5194/acp-3-1285-2003, https://www.atmos-chem-phys.net/3/1285/2003/, 2003.

Freidenreich, S. M., and Ramaswamy, V.: A new multiple-band solar radiative parameterization for general circulation models, J. Geophys. Res., 104, 31389-31409, doi: 10.1029/1999JD900456, 1999.

García, R. D., Cuevas, E., García, O. E., Cachorro, V. E., Pallé, P., Bustos, J. J., Romero-Campos, P. M., and de Frutos, A. M.: Reconstruction of global solar radiation time series from 1933 to 2013 at the Izaña Atmospheric Observatory, Atmospheric Measurement Techniques, 7, 3139–3150, https://doi.org/10.5194/amt-7-3139-2014, https://www.atmos-meas-tech.net/7/3139/2014/, 2014.

García, R. D., Barreto, A., Cuevas, E., Gröbner, J., García, O. E., Gómez-Peláez, A., Romero-Campos, P. M., Redondas, A., Cachorro, V. E., and Ramos, R.: Comparison of observed and modeled cloud-free longwave downward radiation (2010–2016) at the high mountain BSRN Izaña station, Geoscientific Model Development, 11, 2139–2152, https://doi.org/10.5194/gmd-11-2139-2018, https://www.geosci-model-dev.net/11/2139/2018/, 2018.

García, R. D., Cuevas, E., Ramos, R., Cachorro, V. E., Redondas, A., and Moreno-Ruiz, J. A.: Description of the Baseline Surface Radiation Network (BSRN) station at the Izaña Observatory (2009–2017): measurements and quality control/assurance procedures, Geosci. Instrum. Method. Data Syst., 8, 77-96, https://doi.org/10.5194/gi-8-77-2019, 2019.

[revised manuscript text omitted]

[3]Finnish Meteorological Institute, Helsinki, Finland

[4]Atmospheric Optics Group, Valladolid University, Valladolid, Spain

[5]Cimel Electronique, Paris, France

Correspondence: Emilio Cuevas (ecuevasa@aemet.es)

**Supplement S1. One-minute AOD data differences between AERONET-Cimel (V3) and GAW-PFR.**

One-minute AOD data differences between AERONET-Cimel (V3) and GAW-PFR for (a) 380 nm (75303 data-pairs), (b) 440 nm (76290 data-pairs), (c) 500 nm (75335 data-pairs) and (d) 870 nm (76307 data-pairs) for the period 2005-2015. Black dots correspond to the $U$95 limits. A small number of outliers are out of the ±0.06 AOD differences range. Black arrows indicate a change of Reference AERONET-Cimel radiometer and red arrows indicate a change of the GAW-PFR instrument.

[Figure]

**Supplement S2. Percentage of [Cimel (V3)-PFR] 1-minute AOD differences meeting the WMO criteria for the four compared channels.**

Percentage of AERONET-Cimel 1-minute AOD data (V3) meeting the WMO criteria for the four compared channels, and different AOD and AE scenarios for the period 2005-2015, number of data pairs are shown in brackets. The last row corresponds to the total percentages for the sub-period 2010-2015. AOD and AE traceability > 95% are marked in bold.This Table is equivalent to Table 4 of the manuscript for AERONET V2.

| % of data within WMO limits | 380 nm | 440 nm | 500 nm | 870 nm |
|---|---|---|---|---|
| AOD≤0.05 | 93.6 (60264) | **96.3 (62836)** | **97.1 (62545)** | **98.4 (64213)** |
| 0.05<AOD≤0.10 | 91.0 (5138) | 92.0 (5217) | 92.6 (5222) | 94.7 (5372) |
| AOD>0.10 | 77.1 (4085) | 84.1 (4537) | 81.6 (4326) | 93.3 (5034) |
| AE≤0.25 | 78.7 (2472) | 82.3 (2588) | 79.0 (2483) | 92.9 (6530) |
| 0.25<AE≤0.6 | 90.2 (5941) | 94.3 (6321) | 94.9 (6255) | **97.4 (6530)** |
| AE>0.6 | 94.1 (56952) | **96.5 (59181)** | **97.1 (58793)** | **98.7 (60514)** |
| Total 2005-2015 | 92.3 (69487) | **95.2 (72590)** | **95.7 (72093)** | **97.8 (74619)** |
| Total 2010-2015 | 92.8 (42463) | **96.8 (44328)** | **96.8 (44329)** | **98.8 (44329)** |

**Supplement S3. AOD variability 1 minute interval.**

Standard deviation values of 20117 AOD triplets measured with the Cimel#244 in 2013 for 380 and 500nm.

[Figure]

The percentage of data with 1-minute AOD variability for the four GAW-PFR channels are shown in the next figure.

[Figure]

**Supplement S4. One-minute AOD differences between AERONET-Cimel (V2 and V3) GAW- and PFR versus optical air mass (m).**

One-minute AOD differences between AERONET-Cimel (V2 and V3) and GAW-PFR versus optical air mass (m) under pristine conditions (AOD$_{500nm} \leq 0.03$) in the period 2005-2015 for (a) 380 nm, (b) 440 nm, (c) 500 nm and (d) and 870 nm.

[Figure]

**Supplement S5. Percentage of [Cimel (V3)-PFR] 1-minute AOD differences meeting the WMO criteria for each wavelength and for different optical air mass.**

Percentage of 1-minute AOD data (V3) meeting the WMO criteria for each wavelength for different optical air mass intervals under pristine conditions ($AOD_{500nm} \leq 0.03$) in the period 2005-2015. This Table is equivalent to Table 7 in the manuscript for AERONET V2.

| Percentage of AOD differences within the U95 limits $AOD500nm \leq 0.03$ | Total (%) | $1 \leq m < 2$ (%) | $2 \leq m < 3$ (%) | $3 \leq m < 4$ (%) | $4 \leq m < 5$ (%) | $5 \leq m < 6$ (%) |
|---|---|---|---|---|---|---|
| 380 nm | 94.9 | 92.9 | 95.5 | 96.7 | 96.6 | 96.6 |
| 440 nm | 97.5 | 97.2 | 97.3 | 98.0 | 97.6 | 97.7 |
| 500 nm | 98.3 | 98.2 | 98.2 | 98.5 | 98.2 | 98.3 |
| 870 nm | 99.0 | 99.1 | 99.1 | 99.1 | 98.6 | 98.7 |

AOD diurnal range variation (maximum value minus minimum value of AOD in one day) corresponding to AOD outliers (non-traceable AOD) under pristine conditions (AOD$_{Cimel-500nm}\leq$ 0.03) in the period 2005-2015 for AERONET V2 and V3 and for 440 nm, 500 nm and 870 nm: a) 440 nm V2; b) 440 nm V3; c) 500 nm V2; d) 500 nm V3; e) 870 nm V2; and f) 870 nm V3.

This Figure is equivalent to Figure 3 of the manuscript for 380 nm.

[Figure]

**Supplement S7. Percentage of AOD$_{380nm}$ outliers of GAW-PFR and AERONET Cimel (V3).**

Percentage of cases with AOD$_{380nm}$ outliers of both GAW-PFR and AERONET Cimel (V3) under pristine conditions (Cimel AOD$_{500nm}$≤0.03). In these cases the diurnal AOD range was higher than 25% of the daily mean AOD value for which a certain cause has been determined: calibration inaccuracies, cloud screening algorithm failures, mixture of the two previous causes, poor sun pointing, or unknown causes.

| | PFR 51 cases | Cimel 81 cases |
|---|---|---|
| Calibration inaccuracies | 7.8% | 44.4% |
| Cloud screening failures | 29.4% | 21.0% |
| Calibration+ cloud screening errors | 9.8% | 11.1% |
| Sun misalignments | 17.6% | 0% |
| Unknown | 35.3% | 33.5% |

**Supplement S8. Examples of fictitious AOD diurnal variation in both GAW-PFR and AERONET-Cimel.**

Examples of fictitious AOD diurnal variation in both GAW-PFR and AERONET-Cimel V3 due to small calibration inaccuracies in the UV channel (368 nm for GAW- PFR and 380 nm for AERONET-Cimel). The date is indicated in the x-axis. In all these cases a clear fictitious AOD diurnal cycle is observed in AERONET-Cimel V3, normally less than 0.01. In cases d), e), and f) an anomalous diurnal cycle is also observed, but in the opposite direction (convex curve), in the case of the GAW-PFR. These cases reflect a non-perfect calibration in the UV channel and are a cause of non-traceability.

[Figure]

**Supplement S9. Examples of AOD diurnal variation of all chanels from AERONET-Cimel Level 2 V3.**

The screenshots of AERONET V3 level 2 show that the fictitious diurnal cycle is accentuated, or only clearly observed, in the 340 and 380nm channels.

Screenshots from http://aeronet.gsfc.nasa.gov (last access: 1 february 2019). Izana AERONET station Level 2 Version 3.

[Figure]

**Supplement S10. Case analysis of altostratus above the Saharan Air Layer top.**

In this section of the supplement some case analyses are shown analysed in order to highlight the complexity of properly filtering some types of clouds during moderate dust intrusions.

Case analysis S10.1 (18 July 2012): The range corrected backscattering signal vertical cross section of the Micropulse lidar (MPL) (a) shows the presence of altostratus just above the top of the Saharan Ar Layer (SAL), around 6 km height. The total-sky camera images show the presence of middle clouds and dust at 09UTC (b), 11UTC (c) and 16UTC (d). The AERONET V2 AOD records are filtered correctly at those three hours (e), but the AOD values "recovered" by AERONET V3 at those times are very high unreal AOD values, greater than 1.

[Figure]

Case analysis S10.2 (8 March 2015): The range corrected backscattering signal vertical cross section of the Micropulse lidar (MPL) (a) shows the presence of thick altostratus just above the top of the Saharan Ar Layer (SAL) that increase altitude throughout the day from 5 to 8 km height to gradually disappear after 15:30 UTC. The total-sky camera image shows the presence of middle clouds and dust at 15:00 UTC (b). Both GAW-PFR and AERONET-V3 have successfully filtered contaminated data by altostratus until 15:00 UTC but do not do so after that time and until they disappear. During that time the clouds are between 7 and 8 km height and give a weaker signal. AOD contaminated by clouds (AOD500nm = 0.35) is substantially higher than at the end of the day (AOD500nm = 0.35) (c) when the SAL shows a greater thickness.

[Figure]

**S11. Case analysis of cirrus clouds.**

A second type of clouds that cause problems in AOD retrieval are the cirrus clouds, usually being present at Izaña between January and April, associated with the presence of the subtropical jet that is normally found in the vicinity of the Canary Islands at this time of year (Rodríguez-Franco and Cuevas, 2013). A constant cloud optical thickness (COT) corresponding to a cloud of a certain horizontal extension would cause the successive measurements within a minute to correspond to the same cloud stage, and therefore it would not be discernible from the extinction caused by aerosols. In the case of very thin cirrus clouds, AOD could increase up to 0.03 (Chew et al., 2011; Giannakaki et al., 2007) with small fluctuations, that cloud-screening algorithms could interpret as the presence of an aerosol layer. Huang et al. (2012) evaluated the impact on AERONET level 2.0 AOD retrievals from cirrus contamination highlighting the difficulties to remove completely their signature, mainly from those subvisual thin cirrus. According to Kinne et al. (1997), optical depth estimates from cirrus derived with sunphotometers have to include forward-scattering effects. Their results show that for cirrus, and instruments with 2.0° and 2.4° FOV, the correction factors vary between 1.6 and 2.5 depending on the crystal size. Taking into account that the FOV of the GAW-PFR is 2.5°, while that of the AERONET-Cimel is 1.3°, such cases will affect the comparison results.

Three case analyses on cirrus clouds are shown below.

Case analysis S11.1 (September 23, 2015): The range corrected backscattering signal vertical cross section of the Micropulse lidar (MPL) (a) shows scattered cirrus clouds throughout the day (a), and one in particular around 17:45UTC that affects the Cimel AOD measurements. Unfortunately, we do not have measurements for the PFR at this time. The all-sky camera confirms the presence of cirrus clouds at that time (d). The AERONET V2 snapshot registers the impact of the cirrus (b), punctually increasing the AOD values by two. AERONET V3 (c) does not filter these values.

[Figure]

Case analysis S11.2 (February 12, 2015): The range corrected backscattering signal vertical cross section of the Micropulse lidar (MPL) (a) shows the presence of cirrus clouds at around 11 km height between 17:30 and 19:00UTC (a), this is confirmed by the all-sky camera image (b). These cirrus clouds affected the AERONET V2 AOD, increasing the AOD values between 2 and 5 times, depending on the channel (c). AERONET V3 cloud screening correctly filtered these anomalous AOD values (d).

[Figure]

Case analysis S11.3 (March 27, 2010): The cirrus cloud observed by the all-sky camera around 18:30UTC affected both GAW-PFR and AERONET V3, giving AOD values about 8 times higher than those observed early in the morning. The erroneous AOD values of the GAW-PFR are slightly lower than those of AERONET V3. The cause could be a greater forward-scattering effect of the cirrus cloud on the GAW-PFR due to its higher FOV (compared with that of the Cimel).

[Figure]

**Supplement S12. Impact of low stratocumulus on AOD retrieval.**

Another cloud scenario that can affect AOD traceability is the presence of low clouds (stratocumulus) that sometimes exceed the observatory height level because the temperature inversion is around 2400 m height. Sometimes the fog can affect the radiometers as shown in the following case analysis in which the GAW-PFR cloud screening algorithm fails giving a few erroneous 1-minute AOD data around 09:00UTC.

Case analysis February 12, 2015: The range corrected backscattering signal vertical cross section of the Micropulse lidar (MPL) (a) shows the presence low stratocumulus very close to the Izaña level in the early morning as confirmed by the all-sky camera (b) and the webcam (c) images. In the all-sky camera it is possible to appreciate the presence of ice due to recent freezing fog. These clouds exceed the level of the observatory in some moments, slightly hiding the sun with mist. The result in this case are very high AOD values from GAW-PFR (one order of magnitude) (d) due to a failure of its cloud screening algortithm. The rest of the day the agreement between GAW-PFR and AERONET-Cimel AOD (V3) measurements was very good.

[Figure]

February 12, 2015, 09:05UTC
Low stratocumulus passing over Izana

**Supplement S13. Actual AOD differences between AERONET-Cimel V3 and GAW-PFR vs PFR AOD**

Actual AOD differences between AERONET-Cimel V3 and GAW-PFR vs PFR AOD at (a) 380 nm (b) and 500 nm for the period 2005-2015. The fitting line has been calculated with AOD data > 0.1 and Cimel-PFR AOD difference > 0. Number of data used in the plots are indicated in the legend. The percentage of non-traceable AOD data with these conditions is ~22% for 380nm, and ~13% for 500nm. Note that some traceable (black) points show larger AOD differences than non-traceable (red) points because of air mass dependence of the WMO traceability criterion.

[Figure]

**Supplement S14. Percentage of AERONET V3 AOD data outside the U95 limits for high AOD conditions**

Percentage of AERONET V3 AOD data outside the U95 limits at 380 nm, 440 nm, 500 nm and 870 nm channels and for three $AOD_{500nm}$ thresholds with respect to all data and with respect to all data for each AOD interval (in brackets).

| | Percentage of AOD data outside the $U95$ limits (%) | | |
|---|---|---|---|
| | $AOD_{500nm} > 0.1$ | $AOD_{500nm} > 0.2$ | $AOD_{500nm} > 0.3$ |
| 380 nm | 1.6 (22.9) | 1.1 (42.0) | 0.4 (54.4) |
| 440 nm | 1.1 (15.9) | 0.9 (32.5) | 0.4 (49.0) |
| 500 nm | 1.3 (18.4) | 1.0 (37.6) | 0.5 (55.7) |
| 870 nm | 0.5 (6.7) | 0.4 (13.4) | 0.2 (19.0) |

Comparing versions V2 and V3, we can see that, except for the 380 nm channel, in V3 the non-AOD traceability increases with respect to that found in V2.

**Supplement S15. Simulations of scattered to direct radiation simulations.**

Scattered to direct radiation simulations made with a forward Monte Carlo model (Barker 1992, Barker 1996, Räisänen et al. 2003) for FOVs of 2.5° and 1.2° for seven values of effective radius ($r_e$=0.2, 0.5, 1.0, 1.5, 2.0, 2.5, and 3.0 µm), for five AOD values (AOD= 0.1, 0.2, 0.3, 0.4, and 0.5), and for five solar zenith angles (θ = 20°, 30°, 45°, 60° and 80°).

[Figure]

**Supplement S16. Relative error in AOD for PFR and Cimel.**

Relative error in AOD for PFR (x-axis) and Cimel (y-axis) for seven values of effective radius ($r_e$=0.2, 0.5, 1.0, 1.5, 2.0, 2.5 and 3.0 μm), for five AOD values  (AOD= 0.1, 0.2, 0.3, 0.4, and 0.5), and for five solar zenith angles (θ = 20°, 30°, 45°, 60° and 80°).

[Figure]

**Supplement S17. Actual AOD differences between AERONET-Cimel V3 and GAW-PFR vs PFR AOD after AOD$_{PFR}$ correction.**

The same as the Figure of Supplement S13 (AERONET V3) after the PFR AOD data were "corrected" by adding + 3.3% at 380nm and + 2.2.% at 500 nm to the 1-minute AOD PFR data > 0.1.

[Figure]

**Supplement S18. Ångström exponent comparison**

Percentage of cases in which GAW PFR and AERONET V2 (a) and V3 (b) coincide in each AE scenario (period 2005-2015).

[Figure]

[Figure]

*Table 1. Main features of the GAW-PFR (PFR (Wehrli, 2000, 2005, 2008a, b) and AERONET-Cimel (Holben et al., 1994, 1998; Torres et al., 2013) -radiometers used in this study.*

| | GAW-PFR | AERONET-Cimel |
|---|---|---|
| Type of instrument | Standard version  | Standard version Reference instrument |
| Type of observation | Automatic continuous direct sun irradiance | Automatic sun-sky tracking |
| Available standard channels | 368, 412, 500, 862 nm | 340, 380, 440, 500, 675, 870, 1020, nm |
| FWHM | 5 nm | 2 nm (340 nm), 4 nm (380 nm), 10 nm(VIS-NIR) |
| AOD uncertainty | ± 0.01 | ± 0.005 (Reference instruments) |
| FOV (FWHM) | 2.5° (1.2° plateau, 0.7° slope angle) | 1.3° (slope angle unknown) |
| Sun tracker | Any sun tracker with a resolution of at least 0.08° |  Robot specifically designed by Cimel and controlled in conjunction with the radiometer |
| Temperature control | Temperature controlled 20 °C ± 0.5 °C | Temperature correction is applied in V2. Corrections from filter-specific temperature characterization in V3 (Giles et al., 2019) |
| Power | Grid | Solar panels/grid |

| Data transmission | Local PC / FTP | Local PC / FTP Satellite transmission  |
|---|---|---|
| Calibration | Comparison with reference triad. Additional in situ long-term Langleys  | At least 10 good morning  Langleys plots |

*4.2. Cloud filtering*

The data matching in our comparison analysis was performed with synchronous 1-minute AOD values of both networks labelled with quality control (QC) flags that guarantee proven quality data not affected by the presence of clouds.  In the case of the AERONET-Cimel network, the selected AOD data are Level 2 data from both V2 and V3 AERONET databases, which have been cloud filtered by the Smirnov algorithm (Smirnov et al., 2000), based on the triplet method, with a second-order temporal derivative constraint (McArthur et al., 2003), and visually screened in V2. The cloud screening in AERONET V3 has been completely automated, and notably improved, especially by refining the triplet variability and cirrus cloud detection and removal (Giles et al., 2019). GAW-PFR cloud screening algorithms also use the Smirnov triplet measurement, and the second-order derivative check, but add a test for optically thick clouds with $AOD_{500nm} > 2$ (Kazadzis et al., 2018b). In the case of the GAW-PFR network (Wehrli, 2008a) the flags take the value 0 (cloudless conditions, no wavelength crossings and sun pointing within certain limits, more details in Kazadzis et al. (2018a)) for all those selected records.

*Table 2. GAW-PFR and AERONET-Cimel instrument numbers used in this study in the period 2005-2015. Data from Reference Cimel #398 was not upgraded to Level 2 in V3 during the period 12 July 2008 - 15 September 2008.*

| Instruments used in this study | Period 2005-2009 | Period 2010-2015 |
|---|---|---|
| GAW-PFR | 2 instruments: #6,#25 | 2 instruments: #6,#21 |
| AERONET-Cimel | 13 instruments: #25,#44,#45,#79,#117,#140 #244,#245,#380,#382,#383,#398,#421 | 5 instruments: #244, #347, #380 #421, #548 |

~~GAW-PFR provides AOD values every 1 minute as an average of 10 sequential measurements of total duration less than 1 second, while AERONET-Cimel provides AOD values every 15 minutes (from 3 measurements separated by 30 seconds). We consider synchronous 1-minute data when GAW-PFR and AERONET-Cimel AOD data were obtained with a difference of ~ 30 s.~~

*4.3. WMO traceability criteria.*

The criterion for traceability used in this study follows the recommendation of the WMO (WMO, 2005) which states that 95% of the AOD measurements fall within the specified acceptance limits, taking the  PFR as a reference:

$$U_{95} = \pm(0.005 + 0.010/m) \tag{2}$$

where $m$ is the optical air mass. Note that the U95 range is larger for smaller optical mass.
The acceptance limits proposed by WMO take into account, on the one hand, the uncertainty inherent in the calculations of the AOD, and on the other hand, the uncertainty associated with the calibration of the instrument. The latter, for the case of instruments with finite field of view direct transmissions, such as the PFR and the Cimel, is dominated by the influence of the top-of-the-atmosphere signal determined by Langley plot measurements, divided by the optical air mass.
The first term of Eq. 2 (0.005) represents the maximum  tolerance for the uncertainty due to the atmospheric parameters used for the AOD calculation (additional atmospheric trace gas corrections, and Rayleigh scattering). The second term describes the calibration related relative uncertainties . The WMO recommends an upper limit for the calibration uncertainty of 1%.
~~A first simple approach to calculate the circumsolar radiation of each radiometer taking into account their respective FOVs, AOD, total O₃ and pressure values, has been performed with the SMARTS (Simple Model of the Atmospheric Radiative Transfer of Sunshine) model version 2.9.5 (Gueymard, 1995). This spectral model, that covers the UVA, UVB, Visible and Near-Infrared bands, can be used to simulate the spectral irradiance that would be measured by a spectroradiometer (Gueymard, 2001). This model, which has been used and compared with LibRadtran for determining circumsolar radiation (Eissa et al., 2015) is used in~~

this study to estimate, in a first approximation, the differences in AOD caused by the different FOV of PFR and Cimel radiometers.

**4.4. Modelling the impact of near-forward scattering on the AOD measured by the PFR and Cimel radiometers**

In order to study the impact of near-forward scattering on the irradiance measured by the PFR and CimelIMEL instruments, a forward Monte Carlo model (Barker 1992, Barker 1996, Räisänen et al. 2003) was employed. For the present work, the model was updated to account for the finite width of the solar disk. The starting point of each photon was selected randomly within the solar disk, assuming a disk half-width of 0.267° and the impact of limb darkening on the intensity distribution was included following Böhm-Vitense (1989). Some diagnostics were also added to keep track of the distribution of downwelling photons at the surface with respect to the angular distance from the centre of the sun. Gaseous absorption was accounted for following Freidenreich and Ramaswamy (1999), while the Rayleigh scattering optical depth was computed using Bodhaine et al. (1999) below..

**5. Results.**

**5.1. CPreliminary comparison of long-term AERONET V2 and V3 data basesdatasets at Izaña site.**

Since V3 has been released recently (Giles et al., 2019), we present a comparison between V2 and V3 for the Cimel channels 380 nm, 440 nm, 500 nm and 870 nm for the period 2005-2015.

The results indicate that for the Izaña site the agreement and consistency between the two AERONET versions is very high for the four channels ($R^2$>0.999) in full agreement with the results of the V2-V3 comparison reported by Giles et al. (2019). So, we can advanceIt follows that the results of the comparison between GAW-PFR and the two versions of AERONET are very similar as shown throughout this work. A detailed description ofn AERONET V3 and its improvements with respect to V2 is given in Giles et al. (2019). A detailed description of the improvements introduced in V3 are given in Giles et al. (2019). As such Logically, these improvements depend on aerosol type, are not homogeneous in terms of theirfor the different types of aerosols and their nature and variable impact at a global level. A according to the changes introduced in V3, for thea high mountain site such as Izaña characterized by low background AOD values or, alternatively, by the presence of dust presence of dust (, but with no pollution or biomass burning aerosols), the expected AOD differences between V2 and V3 are expected to be minimum as it is confirmed in this studyhas been shown. (Figure 1).

the

However, it should be noted  that  AERONET V3 does not restrict the calculation of AOD to optical masses less than 5.0 (Giles et al., 2019), as V2 does. This results in an increase in the number of solar measurements occurring in the early morning and the late evening. Consequently, the GAW-PFR comparisons with AERONET V3 consisted of ~ 5000 more data pairs than  the GAW-PFR comparison with V2 (see Supplement S.1.1.).

[Figure]

*Figure 1. AERONET Version 3 (V3) vs Version 2 (V2) AOD 1-minute data scatterplot at Izaña Observatory  for the period 2005-2015: a) 380 nm; b) 440 nm; c) 500 nm and d) 870 nm. The corresponding equations of the linear fits, the coefficients of determination (R2), Mean Bias (MB), Root Mean Square Error (RMSE) and the number of data pairs (N) used are included in each legend.*

*5.2.- AERONET-Cimel AOD comparison with GAW-PFR data.*

The comparison with GAW-PFR AOD analysis shows that the AOD from AERONET-Cimel radiometers meet the WMO traceability criteria ("traceable AOD data" from now on) atin 440 nm, 500 nm and 870 nm all four common wavelength channels. The lowest agreement is found in the UV channel (380 nm) with 92.7 % of the minute average data, and 5 the highest in the infrared channel (870 nm) with 98.0 % for V2 (Figure 21; Table 34). Almost identical results are obtained for V3 (Supplement S1 and S2).

However, in the first half of the comparison period (2005-200910) there wereas a some number of mechanical problems in the solar tracker where the GAW-PFR was mounted on, which caused sporadicfrequent problems of sun pointing. This finding was confirmed with data from the four–quadrant silicon detector (Wehrli, 2008a) that showed diurnal variation of the PFR sensors position up to 0.3°°, and relatively poor long term stability. From 2010 onwards, the PFR was mounted on an upgraded solar tracker of higher performance and precision. This reduced problems in sun pointing, that were the main cause of the most of the AOD discrepancies between PFR and Cimel, and therefore not attributable to the instruments themselves.

In addition, since 2010, Cimel #244 has been in continuous operation for most of the time at the Izaña ObservatoryAERONET station, greatly simplifying calibration procedures and the corresponding data evaluation, and minimizing errors of calibration uncertainties introduced by the use of a high number of radiometers in the intercomparison. During the 2010-2015 period, the fraction of traceable AOD measurements of the total between the AERONET-Cimel radiometer and the GAW-PFR improves to 93.46 % in the 380 nm channel, and this percentage rises to 99.07 % for the 870 nm channel. We must clarify that this improvement is mostly due to the upgraded solar tracker used with the PFR since 2010.

Despite the technical differences between both radiometers, described above, and the different calibration protocols, cloud screening algorithms and data processing proceduresalgorithms, the data series of both instruments, can be considered as equivalent, except for 380 nm, according to the WMO traceability criteria defined previously (Eq. 2). This explains the excellent agreement in the long-term AOD climatology shown for GAW-PFR and AERONET-Cimel in Toledano et al. (2018).

In order to confirm the appropriateness of performing the AOD comparison in common channels by interpolating those of the GAW-PFR to those of AERONET-Cimel, wWe have compared the percentages of AERONET-Cimel AOD V2 data meeting the WMO criteria for the four interpolated GAW-PFR channels with those of AERONET V3using the original GAW-PFR channels (Table 3).

For shorter wavelengths, the percentage of data within the WMO limits decreases when the original GAW-PFR channels are used as a reference (not shown here), mainly, and as expected, in the 412 nm channel as this differs considerably from the nominal value of the corresponding AERONET-Cimel channel (440 nm). For 500 nm and 870/862 nm there are no significant differences. Hereinafter, in this study the interpolated GAW-PFR channels are used.

A more detailed statistical evaluation for different scenarios of aerosol loading (three ranges of AOD) and aerosol size (three ranges of AE) for each compared channel has been performed (see Table 4). We

observe that the poorest agreement is obtained at the shorter wavelength channels (440 nm, and especially 380 nm).

Kazadzis et al. (2018b) also found a decrease in the percentage of AOD meeting the WMO criteria for 368 nm and 412 nm spectral bands during the Fourth WMO Filter Radiometer Comparison for aerosol optical depth measurements. As these authors pointed out, the shorter the wavelength, the poorer the agreement because of several reasons: AOD in the UV suffers from out-‑of-band or at least different blocking of the filters, small differences in central wavelength or FWHM have a larger impact, the Rayleigh correction is more critical, and $NO_2$ absorptions are treated differently. Regarding the effect of the aerosol load and particle size on the AOD differences, our results confirm the decrease of agreement between the two instruments for very large particles coincident with almost pure dust (AE $\leq$ 0.3), and high turbidity conditions (AOD > 0.1). However, it should be noted that the percentage of data pairs in these situations is relatively low (e.g., 6% for AOD > 0.1, and 3.2% for AE > 0.25 at 380nm) with respect to the total data (Table 4). A similar result was reported by Kim et al. (2008), who attributed these discrepancies to the possible spatial and temporal variability of aerosols under larger optical depths in addition to the effect of the different FOV of both radiometers. In our case, and according to previous studies on AOD climatology at IZO (Barreto et al., 2014), the presence of high mineral dust burden when the station is within the SAL, does not necessarily imply lower atmospheric stability conditions resulting in daily AOD means with greater standard deviation. For these reasons, we assumed that the different FOV of these instruments is  the main cause of part of the AOD 1-minute differences outside the U95 limits, under high AOD conditions. This issue is specifically addressed in Section 5.3.

[Figure]

*Figure 2. One-minute AOD data differences between AERONET-Cimel (V2) and GAW-PFR for (a) 380 nm (70838 data-pairs), (b) 440 nm (71645 data-pairs), (c) 500 nm (70833 data-pairs) and (d) 870 nm (71660 data-pairs) for the period 2005-2015. Black dots correspond to the U95 limits. A small number of  outliers are out of the ~±0.06 AOD differences range. Black arrows  indicate a change of Reference AERONET-Cimel radiometer and red arrows indicate a change of the GAW-PFR instrument.*

*Table 3. Percentage of AERONET-Cimel (V2 and V3) 1-minute AOD data meeting the WMO criteria for the four interpolated GAW-PFR channels for the period 2005-2015.*

| Interpolated GAW-PFR channel (%) | Original GAW-PFR channel (%) |
|---|---|
|  |  |
|  |  |
|  |  |
|  |  |

| Channel | V2 (%) | V3 (%) |
|---|---|---|

| | | |
|---|---|---|
| 380 nm | 92.7 | 92.3 |
| 440 nm | 95.7 | 95.2 |
| 500 nm | 95.8 | 95.7 |
| 870 nm | 98.0 | 97.8 |

*Table 4. Percentage of AERONET-Cimel 1-minute AOD data (V2) meeting the WMO criteria for the four compared channels, and different AOD and AE scenarios for the period 2005-2015, ~number of data pairs are shown in brackets. The l~Last row corresponds to the total percentages for the sub-period 2010-2015. In bold, AOD and AE traceability is > 95% are marked in bold. Number of data pairs are in brackets.*

| % of data within WMO limits | 380 nm | 440 nm | 500 nm | 870 nm |
|---|---|---|---|---|
| AOD ≤ 0.05 | 94.4 (57008) | **96.8 (59130)** | **97.0 (58572)** | **98.5 (60191)** |
| 0.05 < AOD ≤ 0.10 | 91.0 (4723) | 93.1 (4850) | 92.8 (4817) | 94.2 (4908) |
| AOD > 0.10 | 75.0 (3938) | 86.5 (4615) | 85.1 (4466) | **95.9 (5118)** |
| AE ≤ 0.25 | 73.1 (2145) | 82.3 (2417) | 80.1 (2351) | **96.2 (2824)** |
| 0.25 < AE ≤ 0.6 | 91.2 (5407) | **96.2 (5810)** | **96.0 (5691)** | **97.9 (5911)** |
| AE > 0.6 | 94.6 (55114) | **96.9 (57089)** | **97.0 (56504)** | **98.7 (58146)** |
| Total 2005-2015 | 92.7 (65669) | **95.7 (68595)** | **95.8 (67855)** | **98.0 (70217)** |
| Total 2010-2015 | 93.5 (41977) | **97.4 (43745)** | **97.2 (43627)** | **99.1 (44498)** |

In general, the agreement obtained with the 1-minute AOD data is slightly lower than that obtained during short campaigns, such as those reported by ~(Kazadzis et al. (, 2014) at Athens observatory (4685 data-pairs), and~ Barreto et al. (2016) at ~Izaña Observatory~IZO (5566 data-pairs), with agreements > 99 % for $AOD_{870nm}$ and $AOD_{500nm}$~in case of Barreto et al. (2016).~ However, our results for $AOD_{500nm}$ (> 95 % of 70833 data-pairs) are~is~ significantly better that that observed by Kazadzis et al. (2014) (~ 48 % of 4685 data-pairs) covering a relatively narrow~short~ range of AOD. ~The probable cause for the poor agreement found by (Kazadzis et al., 2014) was a poor calibration in the 500 nm channel in at least one of the instruments operating at Athens.~

In addition, short-term campaigns usually cover a small range of AOD~, normally with low AOD~, and instruments are carefully and frequently supervised. On the contrary, during our intercomparison over a period of 11 years, the operation of the instruments can be ~described much more~considered as the normal operation of such a system. ~for a long term period of measurements, 20 than that of intensively attended instrumentation during short period intercomparison campaigns.~

An additional interesting aspect of this study is that it is not a simple intercomparison exercise between two instruments but a comparison of a number of instruments that acted as reference instruments s for the AERONET/Europe Network.

*Table 5. Basic skill-scores from the AOD intercomparison between GAW-PFR and AERONET-Cimel V2 for the period 2005-2015. The skill scores definitions are found in Huijnen and Eskes (2012).*

| Period | 2005-2015 | | | |
|---|---|---|---|---|
| Wavelengths (nm) | 380 | 440 | 500 | 870 |
| Mean Bias (MB) | -0.0026 | -0.0018 | -0.0021 | -0.0001 |
| Modified Normalized Mean Bias (MNMB) | -0.1301 | -0.1046 | -0.1474 | 0.0129 |
| Fractional Gross Error (FGE) | 0.1727 | 0.1546 | 0.1918 | 0.1837 |
| Root Mean Square Error (RMSE) | 0.0081 | 0.0070 | 0.0064 | 0.0049 |
| Pearson's correlation coefficient (r) | 0.9910 | 0.9925 | 0.9939 | 0.9949 |
| Number of data-pairs | 70838 | 71645 | 70833 | 71660 |

In the first period (2005-2009), a total of 13 Cimel radiometers were used, while in the second period (2010-2015), five Cimel radiometers have participated, and for much of this period, the Cimel #244 was operating as the permanent AERONET reference instrument at IZO. Once the most important causes of non-traceability in the first period, which were associated with a poor pointing of GAW-PFR due to problems in the sun-tracker, were discounted ruled out, we can conclude that there are no significant differences in the percentages of traceable data between the two of both periods. This means that the continuous change of Master Reference Cimel instruments used in the 2005-2009 10 period did not have a significant impact on AOD data comparison differences. This provides proof of proves the consistency and homogeneity of the long AERONET-Cimel AOD data series, and their comparability with the GAW-PFR AOD data series, regardless of the number of instruments used to generate these data series.

In our study, with a number of comparison data-pairs one or two orders of magnitude higher than those used in short campaigns, the results shown in Table 4 can be considered excellent as fairly good.

In addition to the traceability scores, we have introduced some basic skill scores corresponding to the AOD intercomparison between GAW-PFR and AERONET-Cimel for the period 2005-2015 (Table 5) to be in line with previous studies that have performed short-term comparisons between these two instruments. The definitions of the used skill scores can be found in Huijnen and Eskes (2012).

The Pearson's correlation coefficient (r) values of the PFR-Cimel 1-minute AOD data-pairs, are higher than 0.99 in all channels. Concerning Mean Bias (MB) and Root Mean Square Error (RMSE) associated with to AOD differences, our results show quite similar skill scores to those found at Mauna Loa, USA for $AOD_{500nm}$ (Kim et al., 2008), although the number of data pairs used at Izaña Observatory IZO (~71000) is

much higher ~~than that of Mauna Loa (~9700)Master or R~~reference instruments. These results show MB values to be within 0.01 bias, one order of magnitude lower than in Mauna Loa and Izaña Observatories, highlighting the importance of having well calibrated instruments to carry out these type of comparisons.

For the period 2010-2015 (not shown here),  as expected, the RMSE and the Pearson's correlation improve slightly compared with the whole period 2005-2015.
* * *
**5.3. Non-traceability assessment**

As presented in the able 3, data outside the WMO traceability criteria vary from 2% for 870 nm up to 7.3% for 380 nm. In this section, the different possible causes of non-traceability in AOD are evaluated and, if possible, quantitatively estimated. In order to assess the relevance and quantitative impact of these causes, and estimate errors derived from a non-perfect AOD data synchronization, we first made an analysis on the natural variability of AOD in a very short time period (1 minute) shown below.

**5.3.1. Short-time AOD variability**

In order to determine the variability of AOD within one minute, we have performed two independent analyses with AOD data from the PFR and Cimel for the  368/380 nm and 501/500nm channels during one year (2013). On the one hand, and taking into account that GAW-PFR provides AOD every minute, we have calculated all the AOD differences for each channel in the successive minutes. So we have the variation of AOD from one minute to the next one during a whole year. On the other hand, for AERONET-Cimel, we have taken advantage of the triplets, since each triplet consists of three successive measurements made in one minute . In this case, the strategy has been to calculate the standard deviation of the triplet AOD measurements during a whole year. We have verified that the AOD variability in 1 minute is independent of AOD (see Supplement S3).

_Table 6. Percentage of AOD data with variability within 1 minute less than 0.01 and 0.005, respectively, using AOD data from GAW-PFR (at 368 and 501nm) and AERONET-Cimel (at 380 and 500 nm) for 2013. A total of ~32000 d data-pairs per channel have been used from GAW-PFR, and 20117 triplets (60351 individual AOD measurements) from the Cimel#244 to calculate the AOD variability._

| GAW- PFR |
|---|

| Percentage of data with 1-minute AOD variability (%) | | |
|---|---|---|
| | 368 nm | 501 nm |
| < 0.01 | 99.88 | 99.91 |
| < 0.005 | 99.21 | 99.35 |
| AERONET-Cimel | | |
| Percentage of data with 1-minute AOD variability (%) | | |
| | 380 nm | 500 nm |
| < 0.01 | 99.87 | 99.99 |
| < 0.005 | 99.82 | 99.42 |

The results obtained on the AOD variability in 1-minute from PFR data are very similar and consistent to those obtained with Cimel. Less than ~ 0.8% of the AOD data show variability higher than 0.005 in all wavelength ranges. It should be noted that the possible instrumental noise is included in this variability, so that the actual natural AOD variability would be, in any case, lower than that expressed in Table 6. The percentage of data with 1-minute AOD variability for allthe four GAW-PFR channels are given in Supplement S3. These results indicate that the natural AOD variability is very low thus the non-ideal measurement synchronization cannot explain the percentages of non-traceable AOD cases shown in Tables 3 and 4.

**5.3.2. Uncertainties ofby GAW-PFR channel interpolation to AERONET-Cimel channels**

The interpolation of the CIMEL AODs to the PFR AOD wavelengths can be one of the sources of uncertainty in this comparison assessment. The greatest uncertainty arises in the extrapolation of the $AOD_{412\ nm}$ of the PFR to the Cimel wavelength 440 nmretrieve AODs at the CimelCIMEL $AOD_{440nm}$.

Using the Angström formula we have calculated that for an uncertainty of ±0.5 in the Angström exponentAE and for AOD of 0.1 at 412 nm, the introduced uncertainty in the AOD extrapolation from 412 nm to 440 nm is ~5% (i.e., 0.005 for $AOD_{412nm}$=0.1). The introduced uncertainty in AOD extrapolation is reduced to ~2% for an uncertainty of ±0.3 in AE. of the order of ±0.003, while for an $AOD_{412nm}$ of 0.5 and an AE uncertainty of ±0.3, the introduced uncertainty is ±0.008. For all other AOD interpolations the errors are smaller.

**5.32.31  Calibration related errors**

[revised manuscript text omitted]

The correct cause attribution of each outlier would require manual inspection and additional specific information on instrumental checking and maintenance information that is not always available. We have investigated in more  detail the origin of the outliers and whether one of the two instruments predominantly caused them.

Thus, we have calculated for the non-traceable AOD data the diurnal range of AOD variation (maximum value minus minimum value of AOD in one day) at 380 nm for each instrument under pristine conditions (Figure 3) using Cimel AOD$_{500nm}$ daily mean < 0.03 to select the pristine days. According to this approach, the instrument that shows the highest daytime AOD range is the one that is responsible for the outlier. As the wavelength increases both the number of outliers and the magnitude thereof decreases significantly (Supplement S6). Then, we identified those outliers with a diurnal AOD range higher  than 25% of the mean daily AOD value and investigated their possible causes. A total of 51 cases for GAW-PFR and 81 cases for AERONET Cimel V3 were obtained and  analysed in detail, using  auxiliary information, such as 1-minute in-situ meteorological data, 5-minute all-sky images, 1-minute BSRN data, and satellite imagery (not shown here). We obtained the percentage of AOD outliers of GAW-PFR and AERONET Cimel (V3) for which a certain cause has been identified, such as calibration uncertainties, cloud screening algorithm failures, mixture of the two previous causes, poor sun pointing, or not well-defined  causes (electronic problems, humidity inside the lenses, filter dirtiness, obstruction of the lenses collimators, insects on the optics outside, etc.) (see Supplement S7).

From the analysis of these cases, under the conditions described above, it should be noted that ~ 44% of the cases with fictitious AOD diurnal cycles were due to small uncertainties in the calibration of AERONET-Cimel (V3), while for this same cause  ~ 8% of cases were identified in GAW-PFR.

Some examples of  AOD non-traceability for both AERONET-Cimel and GAW-PFR in the ~380 nm channel are shown in Supplement S8. The fictitious diurnal AOD cycle is mainly visible in the UV channels as shown in the examples reported in  Supplement S9.  that the fictitious diurnal AOD  can be more easily identified  under very low AOD conditions.

[Figure]

*Figure 3. AOD diurnal range  variation (maximum value minus minimum value of AOD in one day) at 380nm corresponding to AOD outliers (non-traceable AOD)  under pristine conditions (AODCimel-500nm ≤ 0.03) in the period 2005-2015 for AERONET V2 (a) and V3 (b).*

*5.32.4.2   Differences in cloud-screening and sun tracking.*

In this section wWe have examined the effect that the presence of clouds might have on AOD differences and the number percentage of cases outside the U95 limits. The impact of clouds on AOD differences only occurs when both GAW-PFR and AERONET-Cimel cloud screening algorithms fail to identify clouds in the direct sun path. AERONET-Cimel Version 2 data uses the so-called "triplet check" cloud screening algorithm developed by Smirnov et al. (2000) and a second order temporal derivative constraint (McArthur et al., 2003) to rule out AOD measurements potentially contaminated by clouds. GAW-PFR algorithms also use the Smirnov triplet measurement, and the second-order derivative check, but add a test for optically thick clouds with $AOD_{500nm} > 2$ (Kazadzis et al., 2018b). This algorithm, used by both networks with certain variants, assumes a transitory character in the presence of a cloud, which causes a sudden change of AOD. This sharp change would be detected by measuring the stability of three successive optical depth measurements, so that, when a cloud totally or partially blocks the sun, the standard deviation associated with the average of the triplets increases enormously. Note that if either one or both cloud screening algorithms (GAW-PFR and AERONET-Cimel) are flagged as cloudy, then the corresponding AOD data pair does not take part in the comparison. However, in the case of stratiform and very stable clouds or in the case of very thin clouds such as cirrus clouds, the algorithm could erroneously interpret that there are no clouds since there would be no appreciable changes in the stability of the triplets. A hint that cloud flagging failure could lead to large ADO calculated differences is coming from an analysis of AOD differences for days with different cloudy sky fractions. We do not have precise ancillary information to verify in each 1-minute data the influence that a certain cloud could cause in the non-traceability found. As a first approach forIn order to assess assessing the impact that cloud conditions might cause on AOD traceability, we have used the concept of daily fractions of clear sky (FCS) that has been applied before to solar radiation data at Izaña ObservatoryIZO (García et al., 2014). FCS represents the percentage of observed sunshine hours in a day with respect to the maximum possible sunshine hours in that day. The higher the daily FCS, the higher the clear sky percentage we have on that day.

The percentages of traceable and non-traceable AOD data versus FCS values grouped into 5five intervals are shown in Table 7Table 8. The results indicate that with a FCS lower than 20 % (almost overcast skies), and for wavelengths lower than 870 nm, data outside the U95 limits comprises, at least, 50 % of the total AOD data. It should be emphasized that the number of cases linked with FCS between 0% and 60% are less than 2% of the total cases. There, ~8% (870nm) to 24% (380nm) of the data are outside the WMO limits (maximum of 0.5% of the total data for 380nm outside the WMO limits). As the fraction of clear

sky increases, the percentage of traceable AOD data significantly exceeds the number of non-traceable AOD data. The percentage of traceable data is especially large (> 90 %) when FCS > 80 % (almost clear skies).

This is the FCS range in which a significant percentage of days with cases presenting scattered clouds  are recorded, which qualitatively confirms that V3 has introduced more efficient cloud screening than V2. However, the real impact of clouds on AOD traceability at IZO is very low due to its special characteristics of a high mountain station with very little cloudiness.  Therefore, in practice, the possible impact of clouds on the non-traceability of AOD data-pairs is insignificant at IZO .  GAW-PFR and AERONET-Cimel cloud screening algorithms provide successful identification on clear direct-sun conditions during cloudy skies (FCS < 40 %) for  99.75 % of the cases, excluding those with very thin clouds.  of AERONET V3 (Giles et al., 2018).

However, in the case of stratiform and very stable clouds or in the case of very thin clouds such as cirrus clouds, the algorithm could erroneously interpret that there are no clouds since there would be no appreciable changes in the stability of the triplets.

In the particular case of Izaña there are some very specific cloud scenarios in which cloud screening algorithms could fail resulting in non-AOD traceability: 1) Altostratus above the top of the SAL, at ~6 Km altitude (see Supplement S10); 2) Cirrus clouds (see Supplement S11); and 3) low clouds (stratocumulus) that sometimes exceed the observatory height level (see Supplement S11).

~~A more detailed analysis of more rare atmospheric conditions, such as those of strati-form and homogeneous cirrus clouds, or when altostratus are present above the Saharan Air Layer (SAL), around 6 Km altitude, and thus masked by a heavy mineral dust layer below needs further investigation. A constant cloud optical thickness (COT) corresponding to a cloud of a certain horizontal extension would cause the successive measurements within a minute to correspond to the same cloud stage, and therefore it would not be discernible from the extinction caused by aerosols. In the case of very thin cirrus clouds, the fluctuations in AOD would be very small and could be interpreted as the presence of a light layer of aerosols. Another factor that must be taken into account is that the FOV of the instruments is different. Thus, GAW-PFR (FOV = 2.5°) could detect the entry of a constant COT cloud in part of its~~

field of view in a different way than AERONET-Cimel (FOV = 1.2°). In all these cases, the cloud-screening algorithms may fail simultaneously in both GAW-PFR and AERONET-Cimel, resulting in a different AOD measurement derived by the two instruments. shown reported1-minute

As can be deduced from the analysis of these cloud cases, the impact of the different types of clouds oin AOD retrieval is very complex and further specific investigations are required in order to understand, the reasons behind failures in the GAW-PFR and AERONET-Cimel cloud screening algorithms.

These type of rare situations should be the subject of future studies through measurement campaigns using ancillary observation systems (e.g. lidar, all sky camera).

*Table 7Table 8. Percentage of traceable (T) data and percentage of AOD data outside within the U95 limits (NT) for each channel and 5 daily fractions of clear sky (FCS) intervals. In brackets, relative frequency of each FCS interval for AERONET V2 and V3, respectively. In bold, the percentages of V3 that are greater than those of V2.*

[revised manuscript text omitted]

climatology. Concerning AERONET-Cimel Version 2, a NASA TOMS 1°_x_1.25° resolution $O_3$ climatology is used. From Eq. 10, the differences in $O_3$ optical depth $\Delta 4\tau_{O3}$ can be derived:

$$\triangle\tau_{O3} = \sigma_{O3}(\lambda)\frac{1}{1000}\frac{m_{O3}(O_{3PFR} - O_{3Cimel})}{m_a}$$

(12)

The largest influence of total ozone data uncertainty in $\tau_{O3}$ occurs , by far, at 500 nm (Figure 5). According to Wehrli (2008b) and Kazadzis et al. (2018b), total ozone needs to be determined to ± 30 DU or 10 % of typical values, to ensure an uncertainty of ± 0.001 in $\tau_{O3}$ at 500 nm. In the case of the GAW-PFR / AERONET-Cimel comparison, and due to the very different method in which both networks obtained $O_3$ values for their corresponding corrections, the ozone differences found on some days (1761 out of 71965 days; 2.4 %) are very large (> 40 DU), exceeding a difference in the ozone optical depth of 0.001. Even so, the potential contribution toof AOD differences outside the $U$95 limits between the two networks is negligible. Total $O_3$ over the Izaña ObservatoryIZO is quite stable, showingshows a relatively small amplitude throughout the year, but both surface ozone concentrations and column ozone amount could sharply increase under the influence of cut-off lows injecting air from the high-mid troposphere into the lower subtropical troposphere, which is not uncommon in spring and the first half of summer (Cuevas et al., 2015; Kentarchos et al., 2000). In addition through exchange processes in the Upper Troposphere Lower Stratosphere (UTLS) due to the presence of the subtropical jet (mainly from February to April) (Rodríguez-Franco and Cuevas, 2013). However, if we wanted to repeat this traceability study of 1-minute AOD data in mid or high latitude stations where sharp $O_3$ variations (several tens of DU) could be registered in a few hours, the correction of 1-minute AOD measurements by $\tau_{O3}$ might be a challenging issue.

[Figure]

*Figure 5. (a) Total O₃ used by GAW-PFR (measured Brewer O₃ values from IZO, OMI O₃ overpass or Brewer O₃ climatology) and AERONET-Cimel (TOMS O₃ climatology), and (b) ΔτO3 (λ) caused by differences in daily total O₃ between the two instruments in the period 2005-2015.*

-5.3.54.3 Differences in NO₂ absorption

AERONET-Cimel applies a correction by absorption of NO₂, but GAW-PFR does not include this correction. AERONET--Cimel obtains daily total NO₂ data from a 0.25° ᵒx 0.2°ᵒ resolution NO₂ monthly climatology obtained from the ESA Scanning Imaging Absorption SpectroMeter for Atmospheric CHartographY (SCIAMACHY) (Eskes and Boersma, 2003). In order to assess the contribution to̶i̶n̶ AERONET-Cimel 1-minute AOD data non-traceability by NO₂ absorption w̶h̶a̶t̶ we hav̶e̶r̶e̶a̶l̶l̶y̶ to estimate i̶s̶ the NO₂ optical depth ($\tau_{NO2}(\lambda)$) of AERONET-Cimel since GAW-PFR does not perform this correction. Analogously to ΔτO3, the differences in nitrogen dioxide optical depth Δ4τNO₂ can be obtained from:

$$\triangle \tau_{NO2} = \sigma_{NO2}(\lambda)\frac{1}{1000}\frac{m_{NO2}}{m_a}(-NO_{2Cimel})$$

(13)

Where $m_a$ is given by Eq. 7, $NO_{2Cimel}$ (DU) is the daily total NO$_2$ used by AERONET-Cimel, $\sigma_{NO2}(\lambda)$ is the NO$_2$ absorption  (Gueymard, 1995)  weighted by the specific filter response: 15.6 cm$^{-1}$ (380 nm), 12.3 cm$^{-1}$ (440 nm), and 4.62 cm$^{-1}$ (500 nm). Finally,  $m_{NO2}$ has the following expression (Gueymard, 1995):

$$m_{NO2} = \frac{1}{\sin\theta + 602.30(90-\theta)^{0.5}(27.96+\theta)^{-3.4536}}$$

(14)

[Figure]

*Figure 5. (a) Total O₃ used by GAW-PFR (OMI O₃ overpass or Brewer O₃ climatology) and AERONET-Cimel (TOMS O₃ climatology), and (b) Δτ₀₃(λ) caused by differences in daily total O₃ between the two instruments in the period 2005-2015.*

[Figure]

*Figure 6. (a)*  *NO₂ monthly climatology obtained from the ESA SCanning Imaging Absorption SpectroMeter for Atmospheric CHartographY (SCIAMACHY), used by AERONET-Cimel at* *IZO, and (b)* *τ**NO2 (λ) caused by differences in daily total NO₂ between GAW-PFR and AERONET-Cimel in the period 2005-2015. Note that GAW-PFR does not take into account the correction for the NO₂ absorption.*

*Table 9. Percentage (%) of* _additional_ *traceable AERONET-Cimel AOD 1-minute data* _(V2 and V3)_  *after correcting by pressure, and total column O₃ and NO₂* _for the period 2005-2015._

| Channel | Increment (%) of traceable data after P, O₃ and NO₂ corrections |
|---------|-----------------------------------------------------------------|
| 380 nm  | 1.3 |
| 440 nm  | 0.2 |
| 500 nm  | 0.3 |
| 870 nm  | ~ 0.0 |

| Channel | Increment (%) of traceable AOD data after P, $O_3$ and $NO_2$ corrections | |
|---------|:---:|:---:|
|         | V2 | V3 |
| 380 nm  | 1.3 | 1.7 |
| 440 nm  | 0.2 | 0.3 |
| 500 nm  | 0.3 | 0.1 |
| 870     | ~0.0 | ~0.1 |

In Figure 6a the total $NO_2$ used by AERONET-Cimel to evaluate $\tau_{NO2}(\lambda)$ is depicted. Figure 6b shows the $\Delta\tau_{NO2}(\lambda)$ caused by differences in daily total $NO_2$ between GAW-PFR and AERONET-Cimel. $\Delta\tau_{NO2}$ is of the order of $10^{-3}$ for 380 and 440 nm channels, while, for 500 nm channel, it is of the order of $10^{-4}$. As for O₃, the absorption due to total NO₂ is negligible in the 1-minute AOD non-traceability in our study. However, it should must be notedtaken into account that an impact on AOD calculation is expected when replicating similar analysis in if this type of traceability analysis is replicated in highly $NO_2$ polluted regions where the NO₂ absorption might have somean impact on AOD calculation is expected. Such cases include, such as in large industrial cities from East Asia and Central and Eastern Europe , in which tropospheric NO₂ adds to the natural stratospheric NO₂ resulting in column values much larger than the climatological ones (e.g., Chubarova et al., 2016).

Taking into account the corrections for Rayleigh scattering and for the absorptions by O₃ and NO₂, we have calculated the additional traceable AOD of datacombin that lie within the U95 AOD limits ed effect of all of them on percentage the non-traceability of the 1-minute AOD values (Figure 6; Table 8Table 9). This percentage is maximum at 380 nm with 1.3% (V2) and 1.7% (V3) of the whole dataset. At most (in the 380 nm channel), 25 % (1.3 % of total common measurements) of data outside the *U*95 limits are due to significant differences in pressure, and in O₃ and NO₂ absorption. Most of the AOD data outside the *U*95 limits that becomes traceable data after corrections are applied, had errors in the pressure measurement and therefore in the Rayleigh scattering correction. The 870 nm channel is only affected by the Rayleigh correction component and therefore the increment of traceable data after the mentioned corrections is

minimalmum.The 1minute AOD data outside the *U*95 limits by these corrections is negligible in the 870 nm channel.

*———5.4.3 —GAW PFR and AERONET-Cimel comparison under high AOD conditions: the impact of dust forward scattering role of the foron different FOVs.*

When we represent the AOD differences between AERONET-Cimel and GAW-PFR versus AOD (GAW-PFR) for AOD > 0.1, we observe a positive slope that increases when the AOD fitted data are > 0.05 (dusty non-pristine conditions), we noteing that AERONET-Cimel shows slightly higher AOD values than GAW-PFR, being higher than +0.01 for AOD > 0.15 (Figure 7). , and more clearly at 500 nm. The AOD data outside the *U*95 limits (in red) increases notably from AOD > 0.1.

In fact, the percentage of data outside the U95 limitsnon traceable AOD data increases as AOD increases (Table 9Table 10), so that for dust-related aerosol conditions ($AOD_{500nm}$ > 0.3) the percentage of AOD data outside the *U*95 limits is > 50 % for 380 nm and 440 nm (all channels except for 870 nm (Table 9Table 10, percentages in brackets). Similar results are found when using AERONET V3 (see Supplement S13). The increase in the percentage of AOD data outside the *U*95 limits is especially significant at the 380 nm channel. Taking into account the number of data compared with the total cases, Tthese results show a small but non-negligible percentage of AOD differences outside the *U*95 limits for AOD > 0.1, ranging from ~0.3 % at 870 nm to ~ 1.9 % at 380 nm. This especially affects the shorter wavelengths (Table 9Table 10).

[Figure]

*Figure 7. Actual AOD differences between AERONET-Cimel V2 and GAW-PFR vs  AODPFR at (a)
380 nm (b) and 500 nm for the period 2005-2015. The fitting line has been calculated with those
data points with AOD  > 0.1 and  Cimel-PFR AOD difference > 0. The number of data
used in the plots are indicated in the legend. The percentage of non-traceable AOD data with these
conditions is ~24% for 380 nm, and ~8% for 500 nm. Note that some traceable (black) points show
larger AOD differences than non-traceable (red) points because of the air mass dependence of the
WMO traceability criterion.*

*Table 10. Percentage of  AOD data outside the U95 limits at 380, 440, 500 and 870 nm channels and for three $AOD_{500nm}$ thresholds respect to all data and respect to all data for each AOD interval (in brackets).*

|  | Percentage of AOD data outside the *U*95 limits (%) | | |
|---|---|---|---|
|  | $AOD_{500nm}$>0.1 | $AOD_{500nm}$>0.2 | $AOD_{500nm}$>0.3 |
| 380 nm | 1.9 (25.0) | 1.2 (47.2) | 0.5 (59.8) |
| 440 nm | 1.0 (13.5) | 0.8 (32.0) | 0.5 (57.6) |
| 500 nm | 0.6 (8.0) | 0.5 (18.7) | 0.3 (39.3) |
| 870 nm | 0.3 (4.1) | 0.2 (6.4) | 0.1 (14.0) |

aerosol forward scattering within the FOV of various instruments and calculated AOD was investigated some decades ago by Grassl (1971) who determined that at AOD=1 the circumsolar radiation increases by >10% the incoming radiation. Russell et al (2004), using dust and marine aerosols data, quantified the effect of diffuse light for common sun photometer FOV. They reported that the correction to AOD is negligible (<1% of AOD) for sun photometers with narrow FOV (< 2°), which is greater than  the Cimel FOV and slightly smaller than the PFR FOV  (2.5°). Sinyuk et al. (2012) assessed the impact of the forward scattering aerosol on the uncertainty of the AERONET AOD, concluding that only dust aerosol  with high AOD and low solar elevation could cause a significant bias in AOD (> 0.01). ~~Torres et al. (2013) investigated the uncertainty of the FOV in the AERONET-Cimel measurements indicating that direct solar irradiance measurements are biased by the amount of aureole radiation that is assumed to be direct solar radiation. The solar aureole, also known as the circumsolar region, is the bright region surrounding the solar disc, which becomes especially visible when there is a burden of moderate-high aerosols in the atmosphere.~~

[revised manuscript text omitted]
 completelyis fairly good but it is not complet, e mainly for two reasons. The first one it is the inherent limitation of data correction using the percentage difference in AOD obtained by model simulation for a fixed effective radius.

We have assumed an effective radius of 1.5 μm but, in reality, the radius of dust particles varies. A reasonable range of dust particle radiussize is between 0.1 and 3 μm (Balkanski et al., 1996; Denjean et al., 2016; Mahowald et al., 2014). So, depending on the distance from the dust source to Izaña ObservatoryIZO and the size of the emitted dust, the effective radius could vary slightly between dust episodes. As can be seen in Figures 8a and 8b, the percentage differences in AOD between Cimel and PFR for a 1-2[1,2 μum] effective radius interval, the PFR-Cimel AOD relative difference at 380 nm (500 nm) might change between (around) ~-1.8% (-1.1%) to -4.9% (3.3%).

The second reason is a possible cloud contamination in AOD retrieval when altostratus are present above the SAL, as discussed in Section 5.3.4.

A similar analysis has been carried out for AERONET V3 (see Supplement S17), where we observe that the corrections obtained are not as good as those obtained for V2. This may be due to the very high AOD data retention in V3 which could include more cases in which altostratus clouds and dust are present. The effect of FOV on AOD retrieval should be taken into account for those radiometers with a relatively high FOV (>3°) measuring in regions with relatively high AOD (> 0.2) for most of the year, as is the case in many sites of Northern Africa, the Middle East and East Asia (Basart et al., 2009; Cuevas et al., 2015; Eck et al., 1999; Kim et al., 2007). This effect leads to AOD underestimation, and the variable number of high AOD episodes in each season of the year might affect the AOD long-term trends. AOD measurements under these conditions would be especially affected for optical air mass < 3.

[Figure]

*Figure 9. The same as Figure 7 after "correcting" the PFR AOD data by adding + 3.3% at 380nm and + 2.2.% at 500 nm to the 1-minute PFR AOD data > 0.1.*

Furthermore, it should be taken into account that, as discussed in section 5.2.2., under relatively high AOD conditions, the presence of altostratus above SAL is not infrequent, and they could also cause non-traceability in AOD when the cloud screening algorithms

~~fail. Note that a graphic equivalent to Figure 9 is shown in Supplement S17 but for AERONET V3, observing that the corrections obtained are not as good as those obtained for V2. This may be due to the very high AOD data retention in V3 which could include more cases in which altostratus clouds and dust are present. Therefore, the FOV study should be done once the dust events with presence of clouds over the SAL have been ruled out.~~

5.4 *Angström exponent comparison*

We have performed a comparison of the AE provided by GAW-PFR and AERONET-Cimel using in both cases the AOD data obtained from the four common channels (380 nm, 440 nm, 500 nm and 870 nm) with a total of 70716 data-pairs. The PFR-AOD values have been ordered from lowest to highest by grouping them in intervals of 500 values for which the averages (and corresponding standard deviations) of the PFR-Cimel AE differences have been calculated  Figure 10a). In a similar way we proceeded with the PFR-AE values (Figure 10b).

[Figure]

*Figure 10. (a) PFR-Cimel AE mean absolute differences (and corresponding standard deviations) vs PFR mean AOD$_{500nm}$ in 500 data intervals (b) and vs PFR mean AE in 500 data intervals. AE has been computed for both PFR and Cimel using the four common channels (380, 440, 500 and 870nm).*

AE differences > 0.2 increase exponentially for AOD < 0.02, reaching AE differences of up to 1.6 under pristine conditions (Figure 109a). For very low AOD the provided instruments uncertainty is the source of the sharp increase in AE, and at the same time AE becomes very sensitive to slight AOD changes. However, for AOD < 0.02 the atmospheric aerosol load is practically zero and so, its characterization with AE have in practice relatively minor importance, in practice.

In addition, the AE differences remain ≤below 0.1 when AE$_{PFR}$ values are ≤below 1 (Figure 109b), which shows that these differences are small in most of the possible atmospheric scenarios. For $1 < $ AE$_{PFR}$ $< 1.2$ the AE differences increase slightly to values ≤below 0.2, and for AE$_{PFR}$ > 1.2 (very fine particles or pristine conditions) the AE differences increase sharply to reach values of ~around -1.2. In our case, the non-pristine conditions, or those with a high content of mineral dust, have associated AOD > 0.03 and AE $< 1$, where the AE differences remain ≤below 0.1. In case of pristine conditions AOD ≤ 0.03 and AE ≥ 1 the AE differences can reach a maximum of 1.6.

Wagner and Silva (2008) estimated the usual maximum AE error by error propagation using a pair of spectral channels in which AOD is measured. Their results show that for clean optical conditions ($AOD_{440nm}$= 0.06) the maximum AE error is 1.17, and for hazy conditions ($AOD_{440nm}$=0.17) the error is 0.17, assuming an underlying AE of 1.5. These values decreaserop down to 0.73 and 0.11, respectively, if AE=0. The AE differences found between GAW-PFR and AERONET-Cimel lie within the estimated errors reported by Wagner and Silva (2008).

Table 10*Table 11*. *Uncertainty in AE determination for three typical atmospheric situations.*

|  | Uncertainty in AE |
|---|---|
| Normal pristine conditions $AOD_{500nm}$= 0.03 and AE = 1.4 | ≥ 1 |
| Hazy conditions $AOD_{500nm}$= 0.14 and AE = 1.15 | ≥ 0.2 |
| Strong dust intrusion $AOD_{500nm}$= 0.3 and AE = 0.3 | ~ 0 |

In any case, as in our study the AE has been determined from AOD measured in the four common channels of GAW-PFR and AERONET-Cimel, we have made an estimated of the uncertainty in the calculation of the AE for three typical aerosol scenarios that are typically recorded at Izaña. Ffollowing the the methodology shown by Wagner and Silva (2008), methodology but including the AOD uncertainty related with each of the two instruments and for different conditions. Tthe AE uncertaintyse estimations have been calculated using AOD measurements at four wavelengths and AOD uncertainty error propagation following the Wagner and Silva (2008) methodology but including the AOD uncertainty related with each of the two instruments and for different conditions. The uncertainty estimations are shown in Table 10(Table 11). The AE derived from more than 2 wavelengths is less affected by AOD inaccuracies uncertainties than AE calculated with pairs of wavelengths, since the latter are calculated from the ratio of AOD at two channels (Cachorro et al., 2008).

The AE differences of our study (Figure 109) are within the AE uncertainty estimated for each type of atmospheric condition (pristine, hazy and heavily dust loaded).

However, although AE is a quantitative parameter, it is only used in a qualitative way to estimate the range of sizes (fine, medium, coarse) of the predominant aerosol in the inevitable mixture of aerosols that we observe. With this parameter, and together with the information that is available in the measurement

site about the most frequent types of aerosols and their concentration, we can estimate the type of aerosols that are being measured.

There are many publications with different thresholds of AE and AOD in order to classify different types of aerosols (e.g. Basart et al., 2009; Cuevas et al., 2015; Dubovik et al., 2002; Guirado et al., 2014; Holben et al., 2001; Kim et al., 2007; Todd et al., 2007; Toledano et al., 2007; Wang et al., 2004). However, there is no consensus on these thresholds since at each site there are different mixtures of aerosols, and each type of aerosol shows specific frequencies of appearance and different concentrations.

An alternative way of analyzing the degree of agreement in AE between GAW-PFR and AERONET-Cimel is to verify to what extent both networks provide the same information regarding the type of aerosol they observe in a certain site.

Considering the AE criteria established by Cuevas et al. (2015) and Berjón et al. (2019),  we have  identified the following  four main  categories according to  the $AE_{PFR}$ and $AE_{Cimel}$ values :

1. $AE_{PFR}$ & $AE_{Cimel}$ > 0.6: Pristine conditions.
2. 0.23 < $AE_{PFR}$ & $AE_{Cimel}$ ≤ 0.6: Hazy,  mineral dust being the main aerosol component.
3.  $AE_{PFR}$ & $AE_{Cimel}$ ≤ 0.23: Pure dust.

$AE_{PFR}$ and $AE_{Cimel}$ do not fit any of the previous categories.

[Figure]

*Figure 109. (a) PFR Cimel AE mean absolute differences (and corresponding standard deviations) vs PFR mean AOD₅₀₀ₙₘ in 500 data intervals (b) and vs PFR mean AE in 500 data intervals. AE has been computed for both PFR and Cimel using the four common channels (380, 440, 500 and 870nm).*

4.

In 94.93.8 % of the cases, GAW-PFR and AERONET-Cimel V2 match the AE intervals of each aerosol scenario. Similar results (93.4%) were obtained when comparing with AERONET V3. Most of the agreement (>80 79 %) occurs in the predominant scenario of pristine conditions despite the AE uncertainty under pristine conditions being ≥ 1. See Supplement S18 for more details. Notice that given the special characteristics of the Izaña Observatory, and according to Cuevas et al. (2015) and Berjón et al. (2019), AE is a self-sufficient parameter to define different types of aerosol scenarios without the need to combine its information with AOD.

**6. Summary and Conclusions**

While GAW-PFR is the WMO-defined global AOD reference , being directly linked to WMO / CIMO, and was specifically designed to detect long-term AOD trends, AERONET-Cimel is the densest AOD measurement network globally, and the network most frequently used for aerosol characterization and for model and satellite observation evaluation.

An AERONET-Cimel 11-year AOD data series at IZO was obtained using a large number of radiometers. A total of 13 Reference instruments were used  in the period 2005-2009 which means that every 4 and a half months, approximately, an instrument was replaced by another one to be calibrated. Their calibrations were performed during their respective measurement time periods at IZO. Therefore, these calibrations were not in any way linked with those of the instruments that preceded or replaced them, nor with GAW-PFR reference. These facts led us to investigate the homogeneity of the AERONET-Cimel AOD data series and their intercomparability with the much more homogeneous AOD data series from GAW-PFR (3 instruments in 11 years).

 The traceability concept for AOD suggested by WMO consists in determining whether the AOD difference of the AERONET CIMELs vs the GAW PFRs lie within the U95 limits.

We have used uncertainty limits for AOD traceability established by WMO (2005) for these type of instruments with finite FOV. The acceptable traceability is when 95 % of the absolute AOD differences lie within these limits, in which case both data populations are considered equivalent. It should be clarified that "traceability" is not used in a strict metrological sense.

This study has addressed the comparison of the GAW-PFR dataset  with the two versions of AERONET (V2 and V3) in the period 2005-2015. An excellent agreement between V2 and V3 for the four analysed channels ($R^2 > 0.999$) has been obtained.

More than 70000 synchronous GAW-PFR (PFR) and AERONET-Cimel (Cimel) 1-minute data-pairs in each channel in the period 2005-2015 were analysed. An excellent traceability of AOD from the AERONET-Cimel (V2 and V3) is found for 440 nm, 500 nm and 870 nm, and fairly good results for 380 nm. The lowest percentage of traceable AOD data is registered in 380 nm with 92.7 % of the 1-minute data within the WMO limits, and the highest in 870 nm with 98.0 % of the data.

The different possible causes of non-traceability in AOD were investigated as follows:

- Absolute AOD measurements synchronization.
  Analyzing 1-minute AOD variability we concluded that its impact on the AOD differences is negligible as only ~0.8% of the AOD data has a variability larger than 0.005 in all spectral ranges.

- Sun tracking misalignments.
~~The 1-minute AOD differences Mean Bias of this study is 0.001, an order of magnitude lower than those obtained from previous short-term PFR-Cimel comparison campaigns, in which the Cimel instruments were calibrated by transferring the calibration coefficients by comparison with co-located Master instruments. This indicates the importance of good calibration and maintenance of the Cimel instruments to obtain AOD data very similar to that of GAW-PFR.~~

~~The results confirm that the AOD data outside the U95 limits due to calibration related errors is quite small and not observable for 440, 500 and 870 nm since AOD non-traceability is < 2.1 % for pristine conditions (AOD₅₀₀ₙₘ ≤ 0.03) in these channels. In addition, no dependence of the 1-minute AOD differences with the air optical mass is observed. However, for 380 nm the percentage of non-traceable values increases up to~~

Sun tracking misalignments  constitute a serious problem and a major cause of non-traceability of AOD data-pairs as demonstrated by the AOD data outside the U95 limits from the period 2005-20 as a consequence of episodic problems with the sun-tracker of the GAW-PFR radiometer. For the 2010-2015 period the percentage of traceable data-pairs improves to 93.5% (380 nm), 97.4% (440 nm), 97.2% (500 nm) and 99.1% (870 nm). However, most of these cases could be identified and excluded from the analysis.

- Cloud screening failure by both network algorithms.

~~Regarding AOD non-traceability due to the different cloud-screening algorithms of both networks, it must be said that both algorithms are very similar. GAW-PFR uses the same cloud screening as AERONET-Cimel but incorporates some additional controls. The only reason for AOD non-traceability comes from the simultaneous failure of both cloud screening algorithms because if one or both of them detect clouds, the data will not be part of the comparison.~~ According to our observations, the simultaneous failure of both cloud screening algorithms might occur only under the presence of large and stable cirrus, or altostratus (~ 6000 m a.s.l.) on the top of a heavily dust loaded Saharan air layer, hiding very wide and stable clouds. In these cases, the radiometers interpret these clouds as aerosol layers and might provide values very different from the real AOD. This effect, or the comparison at IZO, however, this effect is negligible since GAW-PFR and AERONET-Cimel cloud screening algorithms provide successful cloud identification on clear direct-sun conditions during cloudy skies (FCS < 40 %) for 99.75 % of the cases.

- ——Pressure measurements related errors.
-

 Since the accuracy of the new barometers built into new radiometers is about 3 hPa, and only errors in atmospheric pressure > 30 hPa might produce an impact on Rayleigh scattering, the AOD non-traceability due to errors in Rayleigh scattering is negligible.

- Total column ozone input uncertainty.

The largest influence of total ozone data uncertainty in ozone absorption occurs mainly at 500 nm. Total ozone needs to be determined to ±30 DU or 10 % of typical values, to ensure an uncertainty of ±0.001 ozone absorption at 500 nm. In the case of the GAW-PFR / AERONET-Cimel comparison, despite the very different methods in which both networks obtained $O_3$ values for their corresponding corrections, large ozone differences were found (> 40 DU) only on 2.4 % of the days , resulting in a

difference in the ozone optical depth slightly above ~0.001. The potential contribution to non-traceable AOD values between the two networks is negligible. However, in mid or high latitude stations where fast $O_3$ variations of several tens of DU might be registered, the correction of 1-minute AOD measurements by ozone absorption might be an issue to be considered.

- Total column $NO_2$ input uncertainty.

The differences in $NO_2$ absorption caused by differences in daily total $NO_2$ between GAW-PFR and AERONET-Cimel is of the order of $10^{-3}$ for 380 nm and 440 nm channels, while, for 500 nm channel, it is even lower, of the order of $10^{-4}$. So, differences in $NO_2$ absorption are negligible in the 1-minute AOD non-traceability of our study. However, $NO_2$ absorption might have some impact on AOD in highly polluted regions, such as in large industrial cities, where column $NO_2$ values are much larger than the climatological ones.

Taking into account the corrections for Rayleigh scattering and for the absorptions by $O_3$ and $NO_2$, we have calculated the combined effect of all of them on the non-traceability of the 1-minute AOD values. The highest impact occurs in the 380 nm channel, in which 25 % of the AOD data outside the $U$95 limits (~2% of the total compared data) are due to significant differences in pressure, and in $O_3$ and $NO_2$ absorption.  The 1-minute AOD data outside the $U$95 limits by these corrections is negligible in the 870 nm channel.

~~We have to note that the excellent results of this 11-year comparison and the small differences found under the strict $U$95 criterion cannot be linked with the relatively low AODs that can be found at IZO. This is because absolute calibration errors contribute to the AOD calculation in an absolute way so larger than 1 % calibration errors for a given period of time can lead to even negative AOD calculations for IZO site.~~

- Impact of dust forward scattering in AOD retrieval uncertainty for different instrument FOVs

 Since GAW-PFR has almost double the FOV (~2.5°°) compared to the AERONET-Cimel (1.3°°), and direct solar irradiance measurements are biased by the amount of aureole radiation that is assumed to be direct solar radiation, it is reasonable to expect that GAW-PFR is more affected by the circumsolar irradiance than AERONET-Cimel radiometer when AOD is relatively high.  Modelling the dust forward scattering we have shown ~~We have explained part of the non-traceabilities found for relatively high AOD values, by analysing the relationship between the differences in circumsolar radiation measured by both instruments with the differences observed in AOD. We have observed a clear relationship between the Cimel PFR AOD differences and the PFR Cimel circumsolar radiation differences, with the slope of the fitted line greater for shorter wavelengths (380 nm). These results show~~ that a non-negligible percentage of

the non-traceable 1-minute AOD data for AOD > 0.1, ranging from ~0.3 % at 870 nm to ~1.9 % at 380 nm is, might be caused by the different FOV. This systematic error especially affects the shorter wavelengths. . Due to this effect, the GAW-PFR provides AOD values which are ~3% lower at 380 nm, and ~2% lower at 500 nm, compared with AERONET-Cimel. However, AOD underestimation This error couldan only have only some relevance be especially important in dusty regions if radiometers with relatively large FOV are used.

A comparison of the AE provided by GAW-PFR and AERONET-Cimel has been performed using in both cases AOD data obtained from the four nearby common channels with a total of 70716 data-pairs. This is a very strict AE calculation since it is necessary that AOD be accurately measured by the four channels simultaneously. AE differences > 0.2 increase exponentially under very pristine conditions (AOD ≤ 0.03 and AE ≥ 1), reaching AE differences of up to 1.6. However, for these conditions the atmospheric aerosol load is practically zero and so, its characterization with AE does not have any importance in practice. Under non-pristine conditions or those with a high mineral dust content (associated AOD > 0.03 and AE < 1), the AE differences remain <below 0.1.

Summarizing, we have presented for the first time a long-–term (2005-2015) 1-minute AOD comparison among different types of radiometers belonging to different aerosol global networks. This comparison is a very demanding test of both GAW-PFR and AERONET-Cimel validated AOD datasets bases since aerosol scenarios correspond to extreme conditions: either very low aerosol loading, a "pristine" scenario that reveals small uncertainties in the calibration and in the cloud screening, or large dust load, which leads to a significant increase in the forward scattering aerosol with AOD, resulting in a slightly higher AOD underestimation by the GAW-PFR. From this comprehensive comparisoncomparison, analysis of the 1-minute AOD and AE data provided by the GAW-PFR and AERONET-Cimel instruments operating at the Izaña Observatory in the 2005-2015 period, we can conclude that the biases in the statistics are very small (< 0.003 in all channels) and therefore both AOD datasets bases are representative of the same AOD population, which it is a remarkable fact for the global aerosol community. It should be noted that AOD traceability at 380 nm (92.7 %) does not reach 95 % of the common data, the percentage recommended by WMO U95 criterion, so more efforts should be made to improve AOD in the UV range. In this study we have also focused much of our attention on investigated the data that are outside of the WMO U95 limits (<5% of the data at 440, 500 and 870 nm and <8% at 380nm) in order to understandknow the weak points of both GAW-PFR and AERONET-Cimel their causes and to be eventually able to correct the small inconsistencies detected in instrumental and methodological aspects in the future.

Our results suggest that, probably, WMO/CIMO traceability limits could be redefined as a function of wavelength, and the recommended radiometer FOV range of radiometers FOVs should be reconsidered.

The widely deployed AERONET-Cimel and GAW-PFR datasets play a crucial role in understanding AOD long-term AOD changes and detecting trends, so it would be desirable for both networks to be linked to the same GAW-WMO related reference. In this sense, these results will be used in future studies, not only to evaluate long term AOD trends at Izaña Observatory based on two independent instruments, but

~~also to provide additional insight on long-term AOD trend analysis, and its significance and validity, based on single instruments and their calibration and AOD processing procedure and uncertainty budget. Finally, special attention should be paid to the AERONET Cimel AOD data series used in trend detection in combination with the used data set homogeneity and their periodic calibration transfer from Master instruments.~~

*Competing interests.* The authors declare that they have no conflict of interest.

*Acknowledgements.* The authors thank Luc Blarel and Philippe Goloub (LOA, CNRS-University of Lille, France) for supervising the periodic calibrations of the Cimel Reference instruments. This study has been performed in the frame of the WMO CIMO Izaña Testbed for Aerosols and Water Vapour Remote Sensing Instruments. The work was supported by the project "The Global Atmosphere Watch Precision Filter Radiometer (GAW-PFR) Network for Aerosol Optical Depth long term measurements" funded by the Federal Office of Meteorology and Climatology MeteoSwiss International Affairs Division, Swiss GCOS Office. Part of the AERONET-Cimel radiometers have been calibrated at Izaña Observatory by AERONET- EUROPE Calibration Service, financed by the European Community specific programs for Integrating Activities: Research Infrastructure Action under the Seventh Framework Programme (FP7/2007-2013), ACTRIS grant agreement No. 262254, and Horizon 2020 Research and Innovation Program, ACTRIS-2 grant agreement No. 654109. This research has received funding from the European Union's Horizon 2020 Research and Innovation Programme under grant agreement No. 654109 (ACTRIS-2).The funding by MINECO (CTM2015-66742-R) and Junta de Castilla y León (VA100P17) is also gratefully acknowledged. We thank the staff of the Izaña Observatory for their effort and dedication in maintaining the instruments. We acknowledge the constructive comments of the anonymous referees. Our colleague Celia Milford has improved the English language of the paper. In memory of Prof Klaus Fröhlich, former director of PMOD-WRC, who initiated the AOD measurements programme at the Izaña Observatory in 1984 within the WMO Background Atmospheric Pollution Monitoring Network (BAPMoN).

---

## Author Comment (AC3) · 7 Apr 2019

**_Anonymous Referee #3 (amt-2018-438-RC3):_**

**G.1. Three major data interpretation issues discourage publication. They are about natural variability, sampling bias and instrument field of view (FOV).**

Authors:  The three "major data interpretation issues" have been addressed by the authors.

**The Referee must take into account that, according to the requirements and comments of the Referees # 3 and # 4, new relevant analysis have been incorporated in the paper what has required large additional workload, resulting in a significant improvement of the paper.**

**The most relevant new analyses have been:**

**1) Comparison of GAW-PFR with version V3 of AERONET and previous comparison between V2 and V3.**

**2) New study on modelling the impact of near-forward scattering on the AOD measured by the PFR and Cimel radiometers (with different FOVs).**

**3) Study of AOD variability in 1 minute to rule out possible differences in AOD due to a non-perfect synchronization of the PFR and Cimel sampling data.**

**4) Incorporation of case studies on the impact on AOD due to small inaccuracies in the calibrations or the clouds contamination.**

**Much more information about the above points has been included in a new Supplement material document of 24 pages.**

**G.2. First, the paper compares AOD observations between 60s averages and less-than1s averages (Page 8, line 1-2). The true AOD in the atmosphere generally changes over the 59s differentials. Yet the paper neglects the natural changes when making inferences on calibration.**

**For example, the paper shows that Cimel observes generally wider AOD differences C1 AMTD Interactive comment Printer-friendly version Discussion paper between daily minimum and maximum than does PFR (page 15, line 21-24; Figure 3). This must be at least partly because Cimel samples with 30s intervals and captures natural changes while PFR sleeps. Yet the paper interprets the wide diurnal ranges as a sign of "an imperfect calibration" of Cimel (page 15, line 24-25). The comparison is fairer if PFR data are paired up over 61s and, better still, if the second of the three consecutive Cimel samples is excluded.**

Authors:

**We do not agree with the reviewer approach concerning the synchronization issue. In any case we have performed an analysis on short-term AOD variability (see reply to G-3) in order to try to detect non exact**

**synchronization measurement issues and impact on the comparison results. We have added this new text in order to clarify this issue:**

*"GAW-PFR provides AOD values every 1 minute as an average of 10 sequential measurements of total duration less than 1 second (20 ms for each channel), then dark current is measured, going to the sleep mode until the next minute. AERONET-Cimel takes a sequence of three separate measurements (1-second per filter) in one minute interval (each one every 30 seconds). This sequence of measurements is called "triplet" and it is performed every ~15 minutes for air masses lower than 2, and with higher frequency for lower solar elevations. Therefore, AERONET-Cimel provides AOD values for each triplet, at least, every ~15 minutes. Note that AERONET-Cimel performs AOD measurements interspersed with sky radiance measurements, whose duration varies throughout the day, and therefore the AOD measurements are not necessarily provided at full minutes. We consider the 1-minute data as synchronous when GAW-PFR and AERONET-Cimel AOD data were obtained with a difference of ~30 s·"*

**Probably the Referee should take into account to properly assess this issue that the PFR takes one measure every 1 minute and the Cimel does it with a significantly lower frequency, every 10-15 minutes.**

**G.3. Misinterpretation is evident on a multi-year basis too. The paper attributes poor AOD agreements at 380 nm under pristine conditions to "insufficiently accurate calibration of AERONET-Cimel" (page 15, line 33-34). This is unsubstantiated. Because AOD is generally higher at shorter wavelengths, so is its natural variability in the absolute term. This would make the AOD discrepancies greatest at the shortest wavelength, even if calibration were perfect for both instruments.**

**Authors:**

**We have included the following subsection on "AOD natural variability". The AOD variability at 380nm is quite low (<0.005) and cannot explain the non AOD traceability. Please, see the new subsection 5.3.1:**

*"…In order to assess the relevance and quantitative impact of these causes, and estimate errors derived from a non-perfect AOD data synchronization, we first made an analysis on the natural variability of AOD in a very short time period (1 minute) shown below…"*

*"5.2.1 Short-time AOD variability*

*In order to know the variability of AOD within one minute, we have performed two independent analyses with AOD data from the PFR and Cimel for the 368/380 nm and 501/500nm channels during one year (2013). On the one hand, and taking into account that GAW-PFR provides AOD every minute, we have calculated all the AOD differences for each channel in the successive minutes. So we have the variation of AOD from one minute to the next one during a whole year. On the other hand, for AERONET-Cimel, we have taken advantage of the triplets, since each triplet consists of three successive measurements made in one minute. In this case, the strategy has been to calculate the standard deviation of the triplet AOD measurements during a whole year. We have verified that the AOD variability in 1 minute is independent of AOD (see Supplement S3). "Table 6. Percentage of AOD data with variability within 1 minute less than 0.01 and 0.005 using AOD data from GAW-PFR (at 368 and 501nm) and AERONET-Cimel (at 380 and 500 nm) for 2013. A total of ~32000 data-pairs per channel have used from GAW-PFR, and 20117 triplets (60351 individual AOD measurements) from the Cimel#244 to calculate the AOD variability."*

| GAW- PFR | | |
| --- | --- | --- |
| Percentage of data with 1-minute AOD variability (%) | | |
| | 368 nm | 501 nm |
| < 0.01 | 99.88 | 99.91 |
| < 0.005 | 99.21 | 99.35 |
| AERONET-Cimel | | |
| Percentage of data with 1-minute AOD variability (%) | | |
| | 380 nm | 500 nm |
| < 0.01 | 99.87 | 99.99 |
| < 0.005 | 99.82 | 99.42 |

*"The results obtained on the AOD variability in 1-minute from PFR data are very similar and consistent to those obtained with Cimel. Less than ~ 0.8% of the AOD data show variability higher than 0.005 in all wavelength ranges. It should be noted that the possible instrumental noise is included in this variability, so that the actual natural AOD variability would be, in any case, lower than that expressed in Table 6. The percentage of data with 1-minute AOD variability for the four GAW-PFR channels are given in Supplement S3. These results indicate that the natural AOD variability is very low thus the non-ideal measurement synchronization cannot explain the percentages of non-traceable AOD cases shown in Tables 3 and 4."*

**G.4. Second, the paper finds "a bias with positive large outliers (higher Cimel AOD)" (page 15, line 5; Figure 2). This results from the intentional exclusion of high PFR values, an attempt to assess "pristine conditions (AOD500nm <= 0.03)" (page 13, line 26, 28). It is misleading to use this assessment to suggest contaminations on Cimel (page 15, line 6, 7). It is only fair to explicitly state that many negative outliers (higher PFR AOD) exist for PFR AOD just above 0.03 (shown in Figure 7).**

**Authors:**

**There is no intentional exclusion of high PFR values at all. We have filtered data using both the daily mean AOD from PFR and Cimel with quite similar results as shown below (left: for $AOD_{PFR} \leq 0.03$; right: for $AOD_{Cimel-V2} \leq 0.03$). Cimel presents a higher number of AOD outliers and of greater magnitude. See Supplement S6.**

[Figure]

[Figure]

Anyway, we have used Cimel V3 AOD to identify AOD outliers under pristine conditions in the corrected manuscript (new Figure 3). We have rewritten subsection 5.3.2 (Calibration related errors) where a detailed analysis of the AOD outliers is given as well as illustrative examples of problems associated with the calibration of both AERONET V3 and GAW-PFR in Supplement material (see Supplements S8 and S9).

**G.5. Third, FOV is not adequately appreciated as a significant source of AOD uncertainty under dust. The forward scattering by aerosols into FOV is definitely the primary cause for the poor agreement at 380 nm away from pristine conditions, not just "might be" (page 33, line 30). A support for this statement partly comes from the theory that the forward scattering is greater at shorter wavelengths.**

**Authors:**

We fully agree that forward scattering by dust into FOV is the main cause of the poorer agreement at 380 nm in dusty conditions. Please, read the new section 5.4. "GAW PFR and AERONET-Cimel comparison under high AOD conditions: the impact of dust forward scattering for different FOVs."

We agree. We have changed "*might be*" by "*is*".

**G.6. A more definitive support is given in Figure 7. It shows that the observed AOD differences is about 3% of AOD at 380 nm. Russell et al. [2004] explain why the FOV effect increases with AOD. And their calculation predicts a ~4% error for 1.25 degree half FOV and a ~1% error for 0.60 degree half FOV, both at 380 nm (their Figure 3). The difference, ~3%, is similar to Figure 7.**

**Authors:**

First of all, note that Russell et al. (2004) and the Referee refer to half-FOV while we refer to full-FOV (FOV) in the paper.

Please, read the new section 5.4. "GAW PFR and AERONET-Cimel comparison under high AOD conditions: the impact of dust forward scattering for different FOVs.". We have cited Russell et al (2004) and previous studies on aerosol forward scattering FOV in the corrected manuscript.

**Our results on the differences in AOD between PFR and Cimel for relatively high AOD, and the corresponding explanation, are in agreement with results reported by Russel et al. (2004). Note that we have done a new modelling of dust forward scattering using a Monte Carlo RTM. We have the following text in the new Section 5.4.:**

*"….The relative error in AOD depends strongly on the particle size but it is fairly constant for each $r_e$ value considered (see Supplement S16). For re~1.5μm, the relative error in AOD at 380nm (500nm) is ~1.6% (1.0%) for Cimel, and ~5% (~3%) for PFR. These errors are in good agreement with those estimated by Russell at al. (2004),…"*

**G.7. It is incorrect to assign "an insufficient[ly] accurate calibration of AERONET-Cimel in this channel" as "the most likely cause" (page 32, line 9) of the observed differences (page 32, line 9). Such errors would not increase with AOD.**

**Authors:**

**This sentence has been removed. Anyway, this is not correct. We do not say that the error increases with AOD. We say that the AOD difference and the calibration error effect on AOD difference, increases with decreasing air mass.**

**G.8. FOV will remain as a source of uncertainty even if the adjustments are made for it. Because the FOV is much wider, the adjustments are greater for PFR than for Cimel. So is the uncertainty in them. Thus, other things being equal, the PFR is destined to be more erroneous than Cimel. It is incorrect to hold both instruments equally responsible for the resulting AOD discrepancies (page 34, line 8-9).**

**Authors:**

**The impact aerosol forward scattering on different FOVs, and hence on AOD traceability, only takes place for relatively high AOD and aerosol types with high forward scattering (e.g. dust). For example for AOD=0.5, under dust aerosol conditions, the difference between Cimel and PFR is ~0.015. The percentage of AOD data outside the U95 limits is only 1.9% for AOD>0.1 at 380nm (see Table 9), and only in this case the non AOD traceability can be mainly attributed to PFR. In former page 34, lines 8-9 we refer to the total AOD traceability at 380nm (92.7%), including no-dust conditions under which the aerosol forward scattering/FOV effect doesn't cause any measurable impact.**

**G.9. Also, the FOV-related uncertainty leads to a question as to why "WMO defines the PFR FOV as the recommended one for sun radiometers" (page 33, line 24-25). The quoted statement is not explained. Forward scattering into the FOV constitutes a deviation from the condition to which Beer's law pertains – a deviation that should be minimized, not recommended. Nor is the statement supported by a citation. To be sure, WMO (2007) recommends that the WORCC be designated the primary WMO Reference Centre for OD measurements, as referenced in page 2, line 24-26. But the WMO report does not mention a specific instrument, let alone support the PFR-specific statement.**

**Authors:**

**The WORCC has defined, designed and built the PFR instruments as standard radiometers. The PFR was build based on the specifications defined by the official WMO report entitled: "*WMO/GAW aerosol measurement procedures guidelines* and recommendations" (first in 2003, and second edition in 2016)** https://library.wmo.int/doc_num.php?explnum_id=3073 that were based on the Guide to Meteorological

Instruments and Methods of Observation (2010) **https://library.wmo.int/pmb_ged/wmo_8_en-2012.pdf. So wavelengths, FOV, and other technical specifications of the PFR are based on such CIMO and WMO recommendations.**

**The maintenance of the AOD standard by the WMO WORCC Calibration Central laboratory (CCL) is described in Kazadzis et al., 2018.**

**We have included the following information in the Introduction section:**

*"The GAW-PFR was specifically designed by WORCC for this goal following the technical specifications defined by WMO (2003; 2016)."*

**And the following two references:**

**WMO: WMO/GAW Aerosol Measurement Procedures: Guidelines and Recommendations, WMO TD No. 1178, GAW Report No 153, 67 pp, https://library.wmo.int/pmb_ged/wmo-td_1178.pdf, 2003.**

**WMO: WMO/GAW Aerosol Measurement Procedures, Guidelines and Recommendations, 2nd Edition, WMO No 1177, GAW Report No. 227, 93 pp, https://library.wmo.int/doc_num.php?explnum_id=3073, 2016.**

**G.10. Meanwhile, there are a few reasons to encourage resubmission. The topic is important, given how widely ground-based sunphotometers are used in climate science and satellite validation. The data are abundant. Careful and useful pieces of discussion are provided regarding the impact of unattended operations (page 3, line 18-21) and the implications for other measurement sites (page 24, line 4).**

**Authors:**

**We appreciate the positive feedbacks of the Referee.**

**G.11. Authors are encouraged to consider the following suggestions and comments, in addition to the major issues raised above, prior to resubmission.**

**Authors:**

**Each and every one of the Referee's comments have been addressed.**

**S.1. Page 2, line 16. Skynet with PREDE instruments is worth mentioning as another ground-based sunphotometer/sky-radiometer network. [Takamura, T., and T. Nakajima, Overview of SKYNET and its activities, Opt. Pura Apl. 37, 3303-3308, 2004., http://atmos3.cr.chiba-u.jp/skynet/index.html]. Skynet has about as many stations as GAW-PFR does.**

**Authors:**

**We agree. We have included the following text in the Introduction section:**

*There are other radiometer networks that in recent years have incorporated centralized protocols for data evaluation and databases, and performed regular intercomparisons with GAW-PFR and AERONET-Cimel.*

*These include, for example, SKYNET (SKYradiometer NETwork), and its seven associated sub-networks, that uses the Prede-POM sky radiometer to investigate aerosol-cloud-solar radiation interactions (e.g. Campanelli et al., 2004; Nakajima et al., 2007; Takamura et al., 2004).*

**and the corresponding references.**

**S.2. Page 4, line 32. Separate vice from versa.**

**Authors:**

**Done**

**S.3. Page 5, line 1. Insert something like "Dust provides a good test on the treatment of the forward scattering into the field of view."**

**Authors:**

**We have added the following sentence:**

*"The periodical presence of a dust laden SAL allows us to evaluate the impact that the dust forward scattering into the field of view has on AOD retrieval."*

**S.4. Page 5, line 14. The precise sun-tracking enabled by a quad detector should be mentioned somewhere in this paragraph for Cimel. Sun tracker is only described for PFR (page 6, line 1).**

**Authors:**

**See reply to S.10.**

**S.5. Page 5, line 27. Is there a reference on the instrument response over the field of view, especially over the 0.7 degree "slope" (Table1)?**

**Authors:**

**Yes. The reference is Wehrli (2008a), already cited in the manuscript.**

**We have included this reference in the Table 1 caption.**

**S.6. Page 5, line 28. Replace significant with significantly.**

**Authors:**

**Done**

**S.7. Page 6, line 1. Replace the second comma with a period.**

**Authors:**

**Done**

**S.8. Page 6, line 4-7. The air mass dependence of uncertainty is worth mentioning here, since radiometric calibration is the primary concern of this paper.**

**Authors:**

**We agree. We have added the following sentence:**

*" It should be taken into account that, in general, in the ultraviolet range the AOD uncertainty is higher and depends on the optical mass (Carlund et al., 2017)."*

**S.9. Page 6, line17. Make 0 subscript, as in line 16.**

**Authors:**

**Done**

**S.10. Table 1. What does "No specific Sun tracker" mean? What does "Sun tracker robot" refer to? Replace "long-term" with 6 months, as described in page 6, line 18. Replace "2-3 months" with 3-4 months as described in page 6, line 16.**

**Authors:**

**Probably the Referee would like to know that the term "robot" appears in the acronym of AERONET (AErosol RObotic NETwork). The term "robot" is widely used among the Cimel and AERONET communities.**

**We have added in Section 3 the following text in order to clarify this issue:**

*"The robot performs automatic pointing to the sun by stepping azimuth and zenith motors using ephemeris based on time, latitude and longitude. Additionally, a four-quadrant detector is used to improve the sun tracking before each scheduled measurement sequence. This sensor guides the robot to the point where the intensity of the signal channel is maximum. Diffuse-sky measurements are also performed by Cimel to infer aerosol optical and microphysical properties. Two different routines are executed: almucantar (varying the azimuth angle keeping constant the zenith angle) and principal plane (varying the zenith angle keeping constant the azimuth angle). The ability of Cimel to perform both direct and diffuse-sky measurements makes it necessary to use a specific robot rather than a simple sun tracker."*

**We have changed:**

**"***2-3 months***" by "***at least 10 good morning Langleys plots***".**

**and "***3-4months***" by "***the time necessary to perform at least 10 good morning Langley plots***"**

**S.11. Page 6, line 21. Provide references that detail the data processing protocol, preferably including Langley plots, for each instrument.**

**Authors:**

**We have included the general reference:**

*Thomason, L.W., Herman, B. M., Schotland, R.M., and Reagan, J.A.: Extraterrestrial solar flux measurement limitations due to a Beer's law assumption and uncertainty in local time, Appl. Opt., 21, 1191–1195, https://doi.org/10.1364/AO.21.001191, 1982.*

**and the following paragraph:**

*"A detailed description of how AOD is obtained and the determination of extraterrestrial constants by GAW-PFR and AERONET-Cimel are provided by Holben et al. (1994, 1998; 2001), Toledano et al. (2018), and Wehrli (2000; 2008b)."*

**S.12. Page 6, line 24. Break down the second word.**

**Authors:**

**Done**

**S.13. Page 7, line 8. Drop "s" from corresponds, since the word data is used as plural in the preceding sentence.**

**Authors:**

**Done. The corresponding sentence was removed.**

**S.14. Page 7, line 5. Replace criterium with criterion or criteria. Note that the plural, if chosen, requires changes in the rest of the sentence.**

**Authors:**

**Done**

**S.15. Page 9, line 3. Revise this sentence, as it contradicts with the fact that the agreement is achieved for less than 95% at 380 nm.**

**Authors:**

**Done**

**S.16. Page 9, line 5. Table 4 is expected to appear after Table 3. Revise numbering.**

**Authors:**

**Corrected**

**S.17. Page 9, line 6. Is the first period "2005-2010"? Table 2 has "2005-2009", as in page 12, line 1 and page 34, line 14.**

**Authors:**

**The correct period is 2005-2009. Fixed.**

**S.18. Page 9, line 11. Explain how it is determined that problems in sun pointing were "the main cause" of AOD discrepancies between PFR and Cimel. Table 4 indicates that the fail rate decreased merely by one third - from 4.2% to 2.8% - at 500 nm upon the tracker update.**

**Authors:**

**This is explained in the manuscript (Page 9; lines 7-9):**

*"This finding was confirmed with data from the four–quadrant silicon detector (Wehrli, 2008a) that showed diurnal variation of the PFR sensors position up to 0.3◦."*

**The corresponding statement has been slightly nuanced as follows:**

*"This reduced problems in sun pointing, that were the main cause of most of the AOD discrepancies between PFR and Cimel, and therefore not attributable to the instruments themselves."*

**S.19. Page 9, line 11. This sentence implies that a sun tracker is not considered a part of the PFR instrument. That is surprising to those who perceive the tracker as an essential, fully-integrated component of a sun radiometer. It is like saying the steering wheel is not part of a car. Consider dropping the comma and the subsequent eight words.**

**Authors:**

**A sun tracker is absolutely necessary for the PFR radiometer in order to take direct-sun measurements but it is not part of the PFR radiometer itself. In fact, the PFR can use any commercial sun-tracker or prototype with a resolution of at least 0.08° (included in Table 1), unlike the Cimel, a radiometer that uses an ad-hoc robot that includes the functions of sun-tracker, and specific observations in the Almucantar and Principal Plane (see reply to S.10).**

**S.20. Page 11, line 9. Replace "can be one of the main causes of part of" with "is the main cause of".**

**Authors:**

**We agree. Done**

**S.21. Page 11, line 10. Replace 5.4 with 5.3.**

**Authors:**

**Done**

**Page 11, line 16. Bring Kazadzis et al. before the first parenthesis, and drop the comma.**

**Authors:**

**Done**

**Page 12, line 3. Again it is not clear what supports the notion "the most important cause [. . .], which were associated with a poor pointing of GAW-PFR".**

**Authors:**

**Please, see reply to S.18**

**S.22. Page 12, line 5. Replace "of both periods" with "between the two periods". Page 12, line 21. Elaborate on "MB values to be within 0.01 bias".**

**Authors:**

**Done.**

**S.23. Page 12, line 23. Drop the first "and".**

**Authors:**

**Done.**

**S.24. Page 13, line 2. This paragraph feels misplaced. It deserves to be a stand-alone subsection under Section 5.2. Combine it with the sixth paragraph of section 5.1.**

**Authors:**

**Section 5 has been rewritten and restructured. This paragraph is the new subsection 5.3.2.**

**S.25. Page 15, line 3. Remove the comma.**

**Authors:**

**This sentence was rephrased.**

**S.26. Page 15, line 19. "maximum value minus minimum value of AOD in one day" is less logical a metric for evaluating the calibration than the difference between the measurements at minimum and maximum air mass factor of the day.**

**_Authors:_**

**Diurnal AOD range is a quite logical metric. It allows detecting other possible problems apart from the diurnal AOD cycle due to small inaccuracies in the calibration**

**We have include cloud-screening problems. See the new** _"5.3.3 Calibration related errors"_ **subsection.**

**S.27. Page 15, line 28. Insert a comma after "causes".**

**Authors:**

**Done.**

**S.28. Page 15, line 29. Replace "worst" with worse.**

**Authors:**

**Done.**

**S.29. Page 15, line 30. Replace the comma.**

**Authors:**

**This sentence was rephrased.**

**S.30. Page 17, line 10. Replace "a" with an.**

**Authors:**

**Done.**

**S.31. Page 24, line 17. Explain what exactly is "more clearly" shown at 500 nm.**

**Authors:**

**This was wrong. Removed.**

**S.32. Figure 7. If this figure is to remain on the paper, note in the caption that an identical data set, PFR AOD, appears in both x and y axes, a practice generally discouraged. Also state that the numbers in the legend are rounded to the significant digits. This is to forestall questions as to why the black lines do not reach exactly (x,y)=(0,0).**

**Authors:**

**This Figure has been modified completely. The new Figure 7 shows the PFR AOD in X-axis, and the Cimel-PFR AOD difference is in Y-axis only for $AOD_{PFR}$ >0.1 . The data period is 2005-2015. The linear fit equation appears in the legend.**

[Figure]

*Figure 7. Actual AOD differences between AERONET-Cimel V2 and GAW-PFR vs AOD$_{PFR}$ at (a) 380 nm (b) and 500 nm for the period 2005-2015. The fitting line has been calculated with those data points with AOD > 0.1 and Cimel-PFR AOD difference > 0. The number of data used in the plots are indicated in the legend. The percentage of non-traceable AOD data with these conditions is ~24% for 380nm, and ~8% for 500nm. Note that some traceable (black) points show larger AOD differences than non-traceable (red) points because of the air mass dependence of the WMO traceability criterion."*

**S.33. Page 26, line 7. "first" is misleading. Refer to previous studies such as Russell et al. [2004]. Replace "might play" with plays.**

**Authors:**

**Removed the word** *"first".*

**Replaced** *"might play"* **with** *"plays"*

**We agree. We have included references to Russell et al. (2004), and previous studies such as Grassl (1971).**

**We have included the following paragraph in new Section 5.4 that reads as follows:**

*"Forward aerosol scattering within the FOV of various instruments and calculated AOD was investigated some decades ago by Grassl (1971) who determined that at AOD=1 the circumsolar radiation increases by >10% the incoming radiation. Russell et al (2004), using dust and marine aerosols data, quantified the effect of diffuse light for common sun photometer FOV. They reported that the correction to AOD is negligible (<1% of AOD) for sun photometers with narrow FOV (< 2°), which is higher than the one of the Cimel and slightly smaller than the PFR FOV (2.5°). Sinyuk et al. (2012) assessed the impact of the forward scattering aerosol on the uncertainty of the AERONET AOD, concluding that only dust aerosol with high AOD and low solar elevation could cause a significant bias in AOD (> 0.01)."*

**S.34. Table 9. "(14.9)" for 500 nm, AOD>0.1 is not consistent with the "AOD>0.10" row of Table 4.**

**Authors:**

**It is consistent. Please, note that in former Table 9 we had shown the percentage of AOD data outside the U95 limits, in this case at 380 nm channel and $AOD_{500nm}$>0.1 respect to all data for each AOD interval (in brackets), which is 25%. In former Table 4 we showed the percentage of traceable AOD data at 380nm, corresponding to AOD>0.1, being 75%.**

**S.35. Figure 8. Revise the caption, yaxis labels and legends to better clarify what are plotted.**

**Authors:**

**This Figure has been removed. Please read the new Section 5.4 and corresponding information in the Supplement material (S13-S17).**

**S.36. Page 28, line 13. Replace "could lead" with leads.**

**Authors:**

**We agree. Done.**

**S.37. Table 10. Complete the right-most vertical line.**

**Authors:**

**Done.**

**S.38. Page 31, line 8. Explain in what way GAW-PFR is "the AOD reference globally" and "directly linked to WMO/CIMO".**

**Authors:**

**Please, see reply to G.9**

**S.39. Page 31, line 14, remove the first comma and "which".**

**Authors:**

**Done.**

**S.40. Page 31, line 28. Does the judgment made here with the word "excellent" hold even while the 95% criteria are not met at 380 nm?**

**Authors:**

**This sentence has been rephrased as follows:**

*"An excellent traceability of AOD from the AERONET-Cimel (V2 and V3) is found for 440, 500 and 870 nm, and fairly good results for 380nm"*

**S.41. Page 31, line 30. Remove the last five words.**

**Authors:**

**Removed.**

**S.42. Page 32, line 5. Provide more details or references regarding the Langley plots so that their quality can be verified.**

**Authors:**

**This is the conclusion section. References of the Langley plots were given in different parts of the paper, including: Holben et al. (1998), Eck et al. (1999), Toledano et al. (2018), and Kazadzis et al. (2018).**

**S.43. Page 32, line 29. Replace "in" with on.**

**Authors:**

**Done.**

**S.44. Page 32, line 33. Remove "Even so,".**

**Authors:**

**Done.**

**S.45. Page 33, line 13. Explain why the traceability metric should be redefined based on actual performance.**

**Authors:**

**Given that the AOD uncertainty is larger in the UV for several reasons, as it has been shown in this and previous studies, it seems more reasonable the traceability limits to be established in function of the spectral range in which AOD is determined.**

**S.46. Page 33, line 14. Replace "shouted".**

**Authors:**

**Done.**

**S.47. Page 33, line 15. The purpose of this paragraph is unclear. If it is a disclosure that low AOD cases are removed from the present analysis, it should be noted much earlier. If this paragraph is a suggestion to exclude negative AOD values from future data analyses, it is misguided. While such values are not physical, their exclusion would artificially bias the remaining data high.**

**Authors:**

**This paragraph had been included to clarify that the high traceability achieved is not due to the fact that a large part of the AOD values are very low and therefore their differences. However this paragraph has been removed in the present reviewed version.**

**For the referee information, the AOD values <0 in all the channels evaluated for both Cimel and PFR have been excluded from the study from the beginning. However, the number of negative AOD values is very low. For example, of 171935 CIMEL available AOD data in version V2 only 8 negative values ($5X10^{-3}$%) were excluded in 380nm, 134 in 440nm, 7 in 500nm and 1 in 870nm. For version V3, with 170749 AOD data available, only 15 values ($10^{-3}$%) were negative in 380nm, 131 in 440nm, 41 in 500nm and 11 in 870nm.**

**S.48. Page 33, line 16. Replace "absolute" with relative.**

**Authors:**

**See reply to S.47.**

**S.49. Page 34, line 7 says "both [PFR and Cimel] are representative of the same AOD population" over the 11 years, except for 380 nm. Similarly, page 12, line 7 says "[The agreement in AOD] proves the consistency and homogeneity of the long AERONETCimel AOD data series". These conclusions imply that Cimel's stability is adequate and that PFR's features for greater stability (page 5, line 27-31) are, while remarkable, not a significant advantage. It is, then, logical to favor Cimel for its much narrower FOV, a clear advantage over PFR. Arguments like this, or ones against it if any, would be a good addition.**

**Authors:**

**We don't agree. This is a long term comparison of a number of instruments of different type (Cimel and PFR). Cimel and PFR radiometers have some advantages and disadvantages compared to the other and have been designed for different goals. So based on this comparison we do not include sentences showing that one instrument is better than the other. When uncertainties and inconsistencies of one or both instruments could be reported we did so in order to address (and understand) the (small) differences observed. In addition the referee should consider that in the case of AERONET-Cimel we have worked with reference instruments ("Masters") with higher accuracy than those Cimel radiometers deployed in standard field stations.**

**S.50. Page 34, line 15. Revise the sentence. The paper does not directly address calibration transfer. Rather, the paper reveals that special attention should be paid to natural variability, sampling bias and PFR's wide FOV.**

*Authors:*

*The sentence has been removed.*

**S.51. References**

[revised manuscript text omitted]

Considering the AE criteria established by Cuevas et al. (2015) and Berjón et al. (2019), we have identified the following four main categories according to the $AE_{PFR}$ and $AE_{Cimel}$ values:

1. $AE_{PFR}$ & $AE_{Cimel} > 0.6$: Pristine conditions.

2. $0.25 < AE_{PFR}$ & $AE_{Cimel} \leq 0.6$: Hazy, mineral dust being the main aerosol component.

3. $AE_{PFR}$ & $AE_{Cimel} \leq 0.25$: Pure dust.

4. $AE_{PFR}$ and $AE_{Cimel}$ do not fit any of the previous categories.

In 94.9 % of the cases, GAW-PFR and AERONET-Cimel V2 match the AE intervals of each aerosol scenario. Similar results (93.4%) were obtained when comparing with AERONET V3. Most of the agreement (>80 %) occurs in the predominant scenario of pristine conditions despite the AE uncertainty under pristine conditions being $\geq 1$. See Supplement S18 for more details. Notice that given the special characteristics of the Izaña Observatory, and according to Cuevas et al. (2015) and Berjón et al. (2019), AE is a self-sufficient parameter to define different types of aerosol scenarios without the need to combine its information with AOD.

**6. Summary and Conclusions**

While GAW-PFR is the WMO-defined global AOD reference, being directly linked to WMO / CIMO, and was specifically designed to detect long-term AOD trends, AERONET-Cimel is the densest AOD measurement network globally, and the network most frequently used for aerosol characterization and for model and satellite observation evaluation.

An AERONET-Cimel 11-year AOD data series at IZO was obtained using a large number of radiometers. A total of 13 Reference instruments were used in the period 2005-2009 which means that every 4 and a half months, approximately, an instrument was replaced by another one to be calibrated. Their calibrations were performed during their respective measurement time periods at IZO. Therefore, these calibrations were not in any way linked with those of the instruments that preceded or replaced them, nor with GAW-PFR reference. These facts led us to investigate the homogeneity of the AERONET-Cimel AOD data series and their intercomparability with the much more homogeneous AOD data series from GAW-PFR (3 instruments in 11 years). The traceability concept for AOD suggested by WMO consists in determining whether the AOD difference of the AERONET CIMELs vs the GAW PFRs lie within the U95 limits. We have used uncertainty limits for AOD traceability established by WMO (2005) for these type of instruments with finite FOV. The acceptable traceability is when 95 % of the absolute AOD differences lie within these limits, in which case both data populations are considered equivalent. It should be clarified that "traceability" is not used in a strict metrological sense. This study has addressed the comparison of the GAW-PFR dataset with the two versions of AERONET (V2 and V3) in the period 2005-2015. An excellent agreement between V2 and V3 for the four analysed channels ($R^2 > 0.999$) has been obtained.

More than 70000 synchronous GAW-PFR (PFR) and AERONET-Cimel (Cimel) 1-minute data-pairs in each channel in the period 2005-2015 were analysed. An excellent traceability of AOD from the AERONET-Cimel (V2 and V3) is found for 440 nm, 500 nm and 870 nm, and fairly good results for 380 nm. The lowest percentage of traceable AOD data is registered in 380 nm with 92.7 % of the 1-minute data within the WMO limits, and the highest in 870 nm with 98.0 % of the data.

The different possible causes of non-traceability in AOD were investigated as follows:

- Absolute AOD measurements synchronization.

Analyzing 1-minute AOD variability we concluded that its impact on the AOD differences is negligible as only ~0.8% of the AOD data has a variability larger than 0.005 in all spectral ranges.

- Sun tracking misalignments.

Sun tracking misalignments constitute a serious problem and a major cause of non-traceability of AOD data-pairs as demonstrated by the AOD data outside the U95 limits from the period 2005-2009 as a consequence of episodic problems with the sun-tracker of the GAW-PFR radiometer. For the 2010-2015 period the percentage of traceable data-pairs improves to 93.5% (380 nm), 97.4% (440 nm), 97.2% (500 nm) and 99.1% (870 nm). However, most of these cases could be identified and excluded from the analysis.

- Cloud screening failure by both network algorithms.

According to our observations, the simultaneous failure of both cloud screening algorithms might occur only under the presence of large and stable cirrus, or altostratus ($\sim$ 6000 m a.s.l.) on the top of a heavily dust loaded Saharan air layer, hiding very wide and stable clouds. 
[revised manuscript text omitted]

Blanc, P., Espinar, B., Geuder, N., Gueymard, C., R., M., Pitz-Paal, R., Reinhardt, B., Renné, D., M., S., Wald, L., and Wilbert, S.: Direct normal irradiance related definitions and applications: The circumsolar issue, Solar Energy, 110, 561 – 577, https://doi.org/https://doi.org/10.1016/j.solener.2014.10.001, 2014.

Bodhaine, B. A., Wood, N. B., Dutton, E. G., and Slusser, J. R.: On Rayleigh optical depth calculations, Journal of Atmospheric and Oceanic Technology, 16, 1854–1861, 1999.

Böhm-Vitense, E.: Introduction to stellar astrophysics, volume 2: stellar atmospheres, Cambridge University Press, 260 pp., 1989.

Bokoye, A. I., Royer, A., O'Neill, N. T., Cliche, P., Fedosejevs, G., Teillet, P. M., and McArthur, L. J. B.: Characterization of atmospheric aerosols across Canada from a ground-based sunphotometer network:

AEROCAN, Atmosphere-Ocean, 39, 429–456, https://doi.org/10.1080/07055900.2001.9649687, 2001.

[revised manuscript text omitted]

Eissa, Y., Blanc, P., Wald, L., and Ghedira, H.: Can AERONET data be used to accurately model the monochromatic beam and circumsolar irradiances under cloud-free conditions in desert environment?, Atmospheric Measurement Techniques, 8, 5099–5112, https://doi.org/DOI = 10.5194/amt-8-5099-2015, 2015.

Eissa, Y., Blanc, P., Ghedira, H., Oumbe, A., and Wald, L.: A fast and simple model to estimate the contribution of the circumsolar irradiance to measured broadband beam irradiance under cloud-free conditions in desert environment, Solar Energy, 163, 497–509, https://doi.org/https://doi.org/10.1016/j.solener.2018.02.015, 2018.

Eskes, H. J. and Boersma, K. F.: Averaging kernels for DOAS total-column satellite retrievals, Atmospheric Chemistry and Physics, 3, 1285–1291, https://doi.org/10.5194/acp-3-1285-2003, https://www.atmos-chem-phys.net/3/1285/2003/, 2003.

Freidenreich, S. M., and Ramaswamy, V.: A new multiple-band solar radiative parameterization for general circulation models, J. Geophys. Res., 104, 31389-31409, doi: 10.1029/1999JD900456, 1999.

García, R. D., Cuevas, E., García, O. E., Cachorro, V. E., Pallé, P., Bustos, J. J., Romero-Campos, P. M., and de Frutos, A. M.: Reconstruction of global solar radiation time series from 1933 to 2013 at the Izaña Atmospheric Observatory, Atmospheric Measurement Techniques, 7, 3139–3150, https://doi.org/10.5194/amt-7-3139-2014, https://www.atmos-meas-tech.net/7/3139/2014/, 2014.

García, R. D., Barreto, A., Cuevas, E., Gröbner, J., García, O. E., Gómez-Peláez, A., Romero-Campos, P. M., Redondas, A., Cachorro, V. E., and Ramos, R.: Comparison of observed and modeled cloud-free longwave downward radiation (2010–2016) at the high mountain BSRN Izaña station, Geoscientific Model Development, 11, 2139–2152, https://doi.org/10.5194/gmd-11-2139-2018, https://www.geosci-model-dev.net/11/2139/2018/, 2018.

García, R. D., Cuevas, E., Ramos, R., Cachorro, V. E., Redondas, A., and Moreno-Ruiz, J. A.: Description of the Baseline Surface Radiation Network (BSRN) station at the Izaña Observatory (2009–2017): measurements and quality control/assurance procedures, Geosci. Instrum. Method. Data Syst., 8, 77-96, https://doi.org/10.5194/gi-8-77-2019, 2019.

[revised manuscript text omitted]

[3]Finnish Meteorological Institute, Helsinki, Finland

[4]Atmospheric Optics Group, Valladolid University, Valladolid, Spain

[5]Cimel Electronique, Paris, France

Correspondence: Emilio Cuevas (ecuevasa@aemet.es)

**Supplement S1. One-minute AOD data differences between AERONET-Cimel (V3) and GAW-PFR.**

One-minute AOD data differences between AERONET-Cimel (V3) and GAW-PFR for (a) 380 nm (75303 data-pairs), (b) 440 nm (76290 data-pairs), (c) 500 nm (75335 data-pairs) and (d) 870 nm (76307 data-pairs) for the period 2005-2015. Black dots correspond to the *U*95 limits. A small number of outliers are out of the ±0.06 AOD differences range. Black arrows indicate a change of Reference AERONET-Cimel radiometer and red arrows indicate a change of the GAW-PFR instrument.

[Figure]

**Supplement S2. Percentage of [Cimel (V3)-PFR] 1-minute AOD differences meeting the WMO criteria for the four compared channels.**

Percentage of AERONET-Cimel 1-minute AOD data (V3) meeting the WMO criteria for the four compared channels, and different AOD and AE scenarios for the period 2005-2015, number of data pairs are shown in brackets. The last row corresponds to the total percentages for the sub-period 2010-2015. AOD and AE traceability > 95% are marked in bold.This Table is equivalent to Table 4 of the manuscript for AERONET V2.

| % of data within WMO limits | 380 nm | 440 nm | 500 nm | 870 nm |
|---|---|---|---|---|
| AOD≤0.05 | 93.6 (60264) | **96.3 (62836)** | **97.1 (62545)** | **98.4 (64213)** |
| 0.05<AOD≤0.10 | 91.0 (5138) | 92.0 (5217) | 92.6 (5222) | 94.7 (5372) |
| AOD>0.10 | 77.1 (4085) | 84.1 (4537) | 81.6 (4326) | 93.3 (5034) |
| AE≤0.25 | 78.7 (2472) | 82.3 (2588) | 79.0 (2483) | 92.9 (6530) |
| 0.25<AE≤0.6 | 90.2 (5941) | 94.3 (6321) | 94.9 (6255) | **97.4 (6530)** |
| AE>0.6 | 94.1 (56952) | **96.5 (59181)** | **97.1 (58793)** | **98.7 (60514)** |
| Total 2005-2015 | 92.3 (69487) | **95.2 (72590)** | **95.7 (72093)** | **97.8 (74619)** |
| Total 2010-2015 | 92.8 (42463) | **96.8 (44328)** | **96.8 (44329)** | **98.8 (44329)** |

**Supplement S3. AOD variability 1 minute interval.**

Standard deviation values of 20117 AOD triplets measured with the Cimel#244 in 2013 for 380 and 500nm.

[Figure]

The percentage of data with 1-minute AOD variability for the four GAW-PFR channels are shown in the next figure.

[Figure]

**Supplement S4. One-minute AOD differences between AERONET-Cimel (V2 and V3) GAW- and PFR versus optical air mass (m).**

One-minute AOD differences between AERONET-Cimel (V2 and V3) and GAW-PFR versus optical air mass (m) under pristine conditions (AOD$_{500nm}$≤ 0.03) in the period 2005-2015 for (a) 380 nm, (b) 440 nm, (c) 500 nm and (d) and 870 nm.

[Figure]

**Supplement S5. Percentage of [Cimel (V3)-PFR] 1-minute AOD differences meeting the WMO criteria for each wavelength and for different optical air mass.**

Percentage of 1-minute AOD data (V3) meeting the WMO criteria for each wavelength for different optical air mass intervals under pristine conditions ($AOD_{500nm} \leq 0.03$) in the period 2005-2015. This Table is equivalent to Table 7 in the manuscript for AERONET V2.

| Percentage of AOD differences within the U95 limits $AOD500nm \leq 0.03$ | Total (%) | $1 \leq m < 2$ (%) | $2 \leq m < 3$ (%) | $3 \leq m < 4$ (%) | $4 \leq m < 5$ (%) | $5 \leq m < 6$ (%) |
|---|---|---|---|---|---|---|
| 380 nm | 94.9 | 92.9 | 95.5 | 96.7 | 96.6 | 96.6 |
| 440 nm | 97.5 | 97.2 | 97.3 | 98.0 | 97.6 | 97.7 |
| 500 nm | 98.3 | 98.2 | 98.2 | 98.5 | 98.2 | 98.3 |
| 870 nm | 99.0 | 99.1 | 99.1 | 99.1 | 98.6 | 98.7 |

**Supplement S6. AOD diurnal range corresponding to AOD outliers under pristine conditions.**

AOD diurnal range variation (maximum value minus minimum value of AOD in one day) corresponding to AOD outliers (non-traceable AOD) under pristine conditions (AOD$_{Cimel-500nm}$≤ 0.03) in the period 2005-2015 for AERONET V2 and V3 and for 440 nm, 500 nm and 870 nm: a) 440 nm V2; b) 440 nm V3; c) 500 nm V2; d) 500 nm V3; e) 870 nm V2; and f) 870 nm V3.

This Figure is equivalent to Figure 3 of the manuscript for 380 nm.

[Figure]

**Supplement S7. Percentage of AOD$_{380nm}$ outliers of GAW-PFR and AERONET Cimel (V3).**

Percentage of cases with AOD$_{380nm}$ outliers of both GAW-PFR and AERONET Cimel (V3) under pristine conditions (Cimel AOD$_{500nm} \leq 0.03$). In these cases the diurnal AOD range was higher than 25% of the daily mean AOD value for which a certain cause has been determined: calibration inaccuracies, cloud screening algorithm failures, mixture of the two previous causes, poor sun pointing, or unknown causes.

| | PFR 51 cases | Cimel 81 cases |
|---|---|---|
| Calibration inaccuracies | 7.8% | 44.4% |
| Cloud screening failures | 29.4% | 21.0% |
| Calibration+ cloud screening errors | 9.8% | 11.1% |
| Sun misalignments | 17.6% | 0% |
| Unknown | 35.3% | 33.5% |

**Supplement S8. Examples of fictitious AOD diurnal variation in both GAW-PFR and AERONET-Cimel.**

Examples of fictitious AOD diurnal variation in both GAW-PFR and AERONET-Cimel V3 due to small calibration inaccuracies in the UV channel (368 nm for GAW- PFR and 380 nm for AERONET-Cimel). The date is indicated in the x-axis. In all these cases a clear fictitious AOD diurnal cycle is observed in AERONET-Cimel V3, normally less than 0.01. In cases d), e), and f) an anomalous diurnal cycle is also observed, but in the opposite direction (convex curve), in the case of the GAW-PFR. These cases reflect a non-perfect calibration in the UV channel and are a cause of non-traceability.

[Figure]

**Supplement S9. Examples of AOD diurnal variation of all chanels from AERONET-Cimel Level 2 V3.**

The screenshots of AERONET V3 level 2 show that the fictitious diurnal cycle is accentuated, or only clearly observed, in the 340 and 380nm channels.

Screenshots from http://aeronet.gsfc.nasa.gov (last access: 1 february 2019). Izana AERONET station Level 2 Version 3.

[Figure]

**Supplement S10. Case analysis of altostratus above the Saharan Air Layer top.**

In this section of the supplement some case analyses are shown analysed in order to highlight the complexity of properly filtering some types of clouds during moderate dust intrusions.

Case analysis S10.1 (18 July 2012): The range corrected backscattering signal vertical cross section of the Micropulse lidar (MPL) (a) shows the presence of altostratus just above the top of the Saharan Ar Layer (SAL), around 6 km height. The total-sky camera images show the presence of middle clouds and dust at 09UTC (b), 11UTC (c) and 16UTC (d). The AERONET V2 AOD records are filtered correctly at those three hours (e), but the AOD values "recovered" by AERONET V3 at those times are very high unreal AOD values, greater than 1.

[Figure]

Case analysis S10.2 (8 March 2015): The range corrected backscattering signal vertical cross section of the Micropulse lidar (MPL) (a) shows the presence of thick altostratus just above the top of the Saharan Ar Layer (SAL) that increase altitude throughout the day from 5 to 8 km height to gradually disappear after 15:30 UTC. The total-sky camera image shows the presence of middle clouds and dust at 15:00 UTC (b). Both GAW-PFR and AERONET-V3 have successfully filtered contaminated data by altostratus until 15:00 UTC but do not do so after that time and until they disappear. During that time the clouds are between 7 and 8 km height and give a weaker signal. AOD contaminated by clouds (AOD500nm = 0.35) is substantially higher than at the end of the day (AOD500nm = 0.35) (c) when the SAL shows a greater thickness.

[Figure]

**S11. Case analysis of cirrus clouds.**

A second type of clouds that cause problems in AOD retrieval are the cirrus clouds, usually being present at Izaña between January and April, associated with the presence of the subtropical jet that is normally found in the vicinity of the Canary Islands at this time of year (Rodríguez-Franco and Cuevas, 2013). A constant cloud optical thickness (COT) corresponding to a cloud of a certain horizontal extension would cause the successive measurements within a minute to correspond to the same cloud stage, and therefore it would not be discernible from the extinction caused by aerosols. In the case of very thin cirrus clouds, AOD could increase up to 0.03 (Chew et al., 2011; Giannakaki et al., 2007) with small fluctuations, that cloud-screening algorithms could interpret as the presence of an aerosol layer. Huang et al. (2012) evaluated the impact on AERONET level 2.0 AOD retrievals from cirrus contamination highlighting the difficulties to remove completely their signature, mainly from those subvisual thin cirrus. According to Kinne et al. (1997), optical depth estimates from cirrus derived with sunphotometers have to include forward-scattering effects. Their results show that for cirrus, and instruments with 2.0° and 2.4° FOV, the correction factors vary between 1.6 and 2.5 depending on the crystal size. Taking into account that the FOV of the GAW-PFR is 2.5°, while that of the AERONET-Cimel is 1.3°, such cases will affect the comparison results.

Three case analyses on cirrus clouds are shown below.

Case analysis S11.1 (September 23, 2015): The range corrected backscattering signal vertical cross section of the Micropulse lidar (MPL) (a) shows scattered cirrus clouds throughout the day (a), and one in particular around 17:45UTC that affects the Cimel AOD measurements. Unfortunately, we do not have measurements for the PFR at this time. The all-sky camera confirms the presence of cirrus clouds at that time (d). The AERONET V2 snapshot registers the impact of the cirrus (b), punctually increasing the AOD values by two. AERONET V3 (c) does not filter these values.

[Figure]

Case analysis S11.2 (February 12, 2015): The range corrected backscattering signal vertical cross section of the Micropulse lidar (MPL) (a) shows the presence of cirrus clouds at around 11 km height between 17:30 and 19:00UTC (a), this is confirmed by the all-sky camera image (b). These cirrus clouds affected the AERONET V2 AOD, increasing the AOD values between 2 and 5 times, depending on the channel (c). AERONET V3 cloud screening correctly filtered these anomalous AOD values (d).

[Figure]

Case analysis S11.3 (March 27, 2010): The cirrus cloud observed by the all-sky camera around 18:30UTC affected both GAW-PFR and AERONET V3, giving AOD values about 8 times higher than those observed early in the morning. The erroneous AOD values of the GAW-PFR are slightly lower than those of AERONET V3. The cause could be a greater forward-scattering effect of the cirrus cloud on the GAW-PFR due to its higher FOV (compared with that of the Cimel).

[Figure]

**Supplement S12. Impact of low stratocumulus on AOD retrieval.**

Another cloud scenario that can affect AOD traceability is the presence of low clouds (stratocumulus) that sometimes exceed the observatory height level because the temperature inversion is around 2400 m height. Sometimes the fog can affect the radiometers as shown in the following case analysis in which the GAW-PFR cloud screening algorithm fails giving a few erroneous 1-minute AOD data around 09:00UTC.

Case analysis February 12, 2015: The range corrected backscattering signal vertical cross section of the Micropulse lidar (MPL) (a) shows the presence low stratocumulus very close to the Izaña level in the early morning as confirmed by the all-sky camera (b) and the webcam (c) images. In the all-sky camera it is possible to appreciate the presence of ice due to recent freezing fog. These clouds exceed the level of the observatory in some moments, slightly hiding the sun with mist. The result in this case are very high AOD values from GAW-PFR (one order of magnitude) (d) due to a failure of its cloud screening algortithm. The rest of the day the agreement between GAW-PFR and AERONET-Cimel AOD (V3) measurements was very good.

[Figure]

**Supplement S13. Actual AOD differences between AERONET-Cimel V3 and GAW-PFR vs PFR AOD**

Actual AOD differences between AERONET-Cimel V3 and GAW-PFR vs PFR AOD at (a) 380 nm (b) and 500 nm for the period 2005-2015. The fitting line has been calculated with AOD data > 0.1 and Cimel-PFR AOD difference > 0. Number of data used in the plots are indicated in the legend. The percentage of non-traceable AOD data with these conditions is ~22% for 380nm, and ~13% for 500nm. Note that some traceable (black) points show larger AOD differences than non-traceable (red) points because of air mass dependence of the WMO traceability criterion.

[Figure]

**Supplement S14. Percentage of AERONET V3 AOD data outside the U95 limits for high AOD conditions**

Percentage of AERONET V3 AOD data outside the U95 limits at 380 nm, 440 nm, 500 nm and 870 nm channels and for three $AOD_{500nm}$ thresholds with respect to all data and with respect to all data for each AOD interval (in brackets).

| | Percentage of AOD data outside the $U95$ limits (%) | | |
|---|---|---|---|
| | $AOD_{500nm} > 0.1$ | $AOD_{500nm} > 0.2$ | $AOD_{500nm} > 0.3$ |
| 380 nm | 1.6 (22.9) | 1.1 (42.0) | 0.4 (54.4) |
| 440 nm | 1.1 (15.9) | 0.9 (32.5) | 0.4 (49.0) |
| 500 nm | 1.3 (18.4) | 1.0 (37.6) | 0.5 (55.7) |
| 870 nm | 0.5 (6.7) | 0.4 (13.4) | 0.2 (19.0) |

Comparing versions V2 and V3, we can see that, except for the 380 nm channel, in V3 the non-AOD traceability increases with respect to that found in V2.

**Supplement S15. Simulations of scattered to direct radiation simulations.**

Scattered to direct radiation simulations made with a forward Monte Carlo model (Barker 1992, Barker 1996, Räisänen et al. 2003) for FOVs of 2.5° and 1.2° for seven values of effective radius ($r_e$=0.2, 0.5, 1.0, 1.5, 2.0, 2.5, and 3.0 µm), for five AOD values (AOD= 0.1, 0.2, 0.3, 0.4, and 0.5), and for five solar zenith angles ($\theta$ = 20°, 30°, 45°, 60° and 80°).

[Figure]

**Supplement S16. Relative error in AOD for PFR and Cimel.**

Relative error in AOD for PFR (x-axis) and Cimel (y-axis) for seven values of effective radius ($r_e$=0.2, 0.5, 1.0, 1.5, 2.0, 2.5 and 3.0 μm), for five AOD values  (AOD= 0.1, 0.2, 0.3, 0.4, and 0.5), and for five solar zenith angles (θ = 20°, 30°, 45°, 60° and 80°).

[Figure]

**Supplement S17. Actual AOD differences between AERONET-Cimel V3 and GAW-PFR vs PFR AOD after AOD$_{PFR}$ correction.**

The same as the Figure of Supplement S13 (AERONET V3) after the PFR AOD data were "corrected" by adding + 3.3% at 380nm and + 2.2.% at 500 nm to the 1-minute AOD PFR data > 0.1.

[Figure]

**Supplement S18. Ångström exponent comparison**

Percentage of cases in which GAW PFR and AERONET V2 (a) and V3 (b) coincide in each AE scenario (period 2005-2015).

[Figure]

[Figure]

*Table 1. Main features of the GAW-PFR (PFR (Wehrli, 2000, 2005, 2008a, b) and AERONET-Cimel (Holben et al., 1994, 1998; Torres et al., 2013) radiometers used in this study.*

| | GAW-PFR | AERONET-Cimel |
|---|---|---|
| Type of instrument | Standard version
 | Standard version
Reference instrument |
| Type of observation | Automatic continuous direct sun irradiance | Automatic sun-sky tracking |
| Available standard channels | 368, 412, 500, 862 nm | 340, 380, 440, 500, 675, 870, 1020, nm |
| FWHM | 5 nm | 2 nm (340 nm), 4 nm (380 nm), 10 nm(VIS-NIR) |
| AOD uncertainty | ± 0.01 | ± 0.005 (Reference instruments) |
| FOV (FWHM) | 2.5°
(1.2° plateau, 0.7° slope angle) | 1.3° (slope angle unknown) |
| Sun tracker | Any sun tracker with a resolution of at least 0.08° | Robot specifically designed by Cimel and controlled in conjunction with the radiometer |
| Temperature control | Temperature controlled 20°C ± 0.5°C | Temperature correction is applied in V2. Corrections from filter-specific temperature characterization in V3 (Giles et al., 2019) |
| Power | Grid | Solar panels/grid |

| Data transmission | Local PC /
 | Local PC / FTP
Satellite transmission
 |
|---|---|---|
| Calibration | Comparison with reference triad. Additional in situ long-term Langleys
 | At least 10 good morning
Langleys plots |

**4.2 Cloud filtering**

The data matching in our comparison analysis was performed with synchronous 1-minute AOD values of both networks labelled with quality control (QC flags that guarantee proven quality data not affected by the presence of clouds.  In the case of the AERONET-Cimel network, the selected AOD data are Level 2 data from both V2 and V3 AERONET databases, which have been cloud filtered by the Smirnov algorithm (Smirnov et al., 2000), based on the triplet method, with a second-order temporal derivative constraint (McArthur et al., 2003), and visually screened in V2. The cloud screening in AERONET V3 has been completely automated, and notably improved, especially by refining the triplet variability and cirrus cloud detection and removal (Giles et al., 2019). GAW-PFR cloud screening algorithms also use the Smirnov triplet measurement, and the second-order derivative check, but add a test for optically thick clouds with $AOD_{500nm} > 2$ (Kazadzis et al., 2018b). In the case of the GAW-PFR network (Wehrli, 2008a) the flags take the value 0 (cloudless conditions, no wavelength crossings and sun pointing within certain limits, more details in Kazadzis et al. (2018a)) for all those selected records.

*Table 2. GAW-PFR and AERONET-Cimel instrument numbers used in this study in the period 2005-2015. Data from Reference Cimel #398 was not upgraded to Level 2 in V3 during the period 12 July 2008 - 15 September 2008.*

| Instruments used in this study | Period 2005-2009 | Period 2010-2015 |
|---|---|---|
| GAW-PFR | 2 instruments: #6,#25 | 2 instruments: #6,#21 |
| AERONET-Cimel | 13 instruments: #25,#44,#45,#79,#117,#140 #244,#245,#380,#382,#383,#398,#421 | 5 instruments: #244, #347, #380 #421, #548 |

GAW-PFR provides AOD values every 1 minute as an average of 10 sequential measurements of total duration less than 1 second, while AERONET-Cimel provides AOD values every 15 minutes (from 3 measurements separated by 30 seconds). We consider synchronous 1-minute data when GAW-PFR and AERONET-Cimel AOD data were obtained with a difference of ~ 30 s.

**4.3. WMO traceability criteria.**

The criterionum for traceability used in this study follows the recommendation of the WMO (WMO, 2005) which states that 95% of the AOD measurements fall within the specified acceptance limits, taking the GAW PFR as a reference:

$$U_{95} = \pm(0.005 + 0.010/m) \tag{2}$$

where $m$ is the optical air mass. Note that the U95 range is larger for smaller optical mass.

The acceptance limits proposed by WMO take into account, on the one hand, the uncertainty inherent in the calculations of the AOD, and on the other hand, the uncertainty associated with the calibration of the instrument. The latter, for the case of instruments with finite field of view direct transmissions, such as the PFR and the Cimel, is dominated by the influence of the top-of-the-atmosphere signal determined by Langley plot measurementsd, divided by the optical air mass.

The first term of Eq. 2 (0.005) represents the maximum desirable tolerance for the uncertainty due to the atmospheric parameters used for the AOD calculation (additional atmospheric trace gas corrections, and Rayleigh scattering). The second term describes the calibration related relative uncertainties which scale therefore with the inverse of air mass. The WMO recommends an upper limit for the calibration uncertainty of 1%.

A first simple approach to calculate the circumsolar radiation of each radiometer taking into account their respective FOVs, AOD, total $O_3$ and pressure values, has been performed with the SMARTS (Simple Model of the Atmospheric Radiative Transfer of Sunshine) model version 2.9.5 (Gueymard, 1995). This spectral model, that covers the UVA, UVB, Visible and Near-Infrared bands, can be used to simulate the spectral irradiance that would be measured by a spectroradiometer (Gueymard, 2001). This model, which has been used and compared with LibRadtran for determining circumsolar radiation (Eissa et al., 2015) is used in

this study to estimate, in a first approximation, the differences in AOD caused by the different FOV of PFR and Cimel radiometers.

*4.4. Modelling the impact of near-forward scattering on the AOD measured by the PFR and Cimel radiometers*

In order to study the impact of near-forward scattering on the irradiance measured by the PFR and CimelIMEL instruments, a forward Monte Carlo model (Barker 1992, Barker 1996, Räisänen et al. 2003) was employed. For the present work, the model was updated to account for the finite width of the solar disk. The starting point of each photon was selected randomly within the solar disk, assuming a disk half-width of 0.267° and the impact of limb darkening on the intensity distribution was included following Böhm-Vitense (1989). Some diagnostics were also added to keep track of the distribution of downwelling photons at the surface with respect to the angular distance from the centre of the sun. Gaseous absorption was accounted for following Freidenreich and Ramaswamy (1999), while the Rayleigh scattering optical depth was computed using Bodhaine et al. (1999) below..

**5. Results.**

**5.1. CPreliminary comparison of long-term AERONET V2 and V3 data basesdatasets at Izaña site.**

Since V3 has been released recently (Giles et al., 2019), we present a comparison between V2 and V3 for the Cimel channels 380 nm, 440 nm, 500 nm and 870 nm for the period 2005-2015.

The results indicate that for the Izaña site the agreement and consistency between the two AERONET versions is very high for the four channels ($R^2$>0.999) in full agreement with the results of the V2-V3 comparison reported by Giles et al. (2019). So, we can advanceIt follows that the results of the comparison between GAW-PFR and the two versions of AERONET are very similar as shown throughout this work. A detailed description ofn AERONET V3 and its improvements with respect to V2 is given in Giles et al. (2019). A detailed description of the improvements introduced in V3 are given in Giles et al. (2019). As such Logically, these improvements depend on aerosol type, are not homogeneous in terms of theirfor the different types of aerosols and their nature and variable impact at a global level. A according to the changes introduced in V3, for thea high mountain site such as Izaña characterized by low background AOD values or, alternatively, by the presence of dust presence of dust (, but with no pollution or biomass burning aerosols), the expected AOD differences between V2 and V3 are expected to be minimum as it is confirmed in this studyhas been shown. (Figure 1).
the

However, it should be noted  that  AERONET V3 does not restrict the calculation of AOD to optical masses less than 5.0 (Giles et al., 2019), as V2 does. This results in an increase in the number of solar measurements occurring in the early morning and the late evening. Consequently, the GAW-PFR comparisons with AERONET V3 consisted of ~ 5000 more data pairs than  the GAW-PFR comparison with V2 (see Supplement S.1.1.).

[Figure]

*Figure 1. AERONET Version 3 (V3) vs Version 2 (V2) AOD 1-minute data scatterplot at Izaña Observatory  for the period 2005-2015: a) 380 nm; b) 440 nm; c) 500 nm and d) 870 nm. The corresponding equations of the linear fits, the coefficients of determination (R2), Mean Bias (MB), Root Mean Square Error (RMSE) and the number of data pairs (N) used are included in each legend.*

*5.2.- AERONET-Cimel AOD comparison with GAW-PFR data.*

The  shows that the AOD from AERONET-Cimel radiometers meet the WMO traceability criteria ("traceable AOD data" from now on) in 440 nm, 500 nm and 870 nm  channels. The lowest agreement is found in the UV channel (380 nm) with 92.7 % of the  data, and  the highest in the infrared channel (870 nm) with 98.0 % for V2 (Figure 2; Table 3). Almost identical results are obtained for V3 (Supplement S1 and S2).

However, in the first half of the comparison period (2005-2009) there were some  mechanical problems in the solar tracker where the GAW-PFR was mounted on, which caused sporadic problems of sun pointing. This finding was confirmed with data from the four–quadrant silicon detector (Wehrli, 2008a) that showed diurnal variation of the PFR sensors position up to 0.3°. From 2010 onwards, the PFR was mounted on an upgraded solar tracker of higher performance and precision. This reduced problems in sun pointing, that were the main cause of  most of the AOD discrepancies between PFR and Cimel, and therefore not attributable to the instruments themselves.

In addition, since 2010, Cimel #244 has been in continuous operation for most of the time at the Izaña Observatory, greatly simplifying calibration procedures and the corresponding data evaluation, and minimizing errors of calibration uncertainties introduced by the use of a high number of radiometers in the intercomparison. During the 2010-2015 period, the fraction of traceable AOD measurements of the total between the AERONET-Cimel radiometer and the GAW-PFR improves to 93.46 % in the 380 nm channel, and this percentage rises to 99.07 % for the 870 nm channel.

Despite the technical differences between both radiometers, described above, and the different calibration protocols, cloud screening  and data processing algorithms, the data series of both instruments, can be considered as equivalent, except for 380 nm, according to the WMO traceability criteria defined previously (Eq. 2). This explains the excellent agreement in the long-term AOD climatology shown for GAW-PFR and AERONET-Cimel in Toledano et al. (2018).

We have compared the percentages of AERONET-Cimel AOD V2 data meeting the WMO criteria for the four interpolated GAW-PFR channels with those of AERONET V3 (Table 3).

~~For shorter wavelengths, the percentage of data within the WMO limits decreases when the original GAW-PFR channels are used as a reference (not shown here), mainly, and as expected, in the 412 nm channel as this differs considerably from the nominal value of the corresponding AERONET-Cimel channel (440 nm). For 500 nm and 870/862 nm there are no significant differences. Hereinafter, in this study the interpolated GAW-PFR channels are used.~~

A more detailed statistical evaluation for different scenarios of aerosol loading (three ranges of AOD) and aerosol size (three ranges of AE) for each compared channel has been performed (see Table 4). We

observe that the poorest agreement is obtained at the shorter wavelength channels (440 nm, and especially 380 nm).

Kazadzis et al. (2018b) also found a decrease in the percentage of AOD meeting the WMO criteria for 368 nm and 412 nm spectral bands during the Fourth WMO Filter Radiometer Comparison for aerosol optical depth measurements. As these authors pointed out, the shorter the wavelength, the poorer the agreement because of several reasons: AOD in the UV suffers from out--of-band or at least different blocking of the filters, small differences in central wavelength or FWHM have a larger impact, the Rayleigh correction is more critical, and $NO_2$ absorptions are treated differently. Regarding the effect of the aerosol load and particle size on the AOD differences, our results confirm the decrease of agreement between the two instruments for very large particles coincident with almost pure dust (AE $\leq$ 0.3), and high turbidity conditions (AOD > 0.1). However, it should be noted that the percentage of data pairs in these situations is relatively low (e.g., 6% for AOD > 0.1, and 3.2% for AE > 0.25 at 380nm) with respect to the total data (Table 4). A similar result was reported by Kim et al. (2008), who attributed these discrepancies to the possible spatial and temporal variability of aerosols under larger optical depths in addition to the effect of the different FOV of both radiometers. In our case, and according to previous studies on AOD climatology at IZO (Barreto et al., 2014), the presence of high mineral dust burden when the station is within the SAL, does not necessarily imply lower atmospheric stability conditions resulting in daily AOD means with greater standard deviation. For these reasons, we assumed that the different FOV of these instruments is  the main cause of part of the AOD 1-minute differences outside the U95 limits, under high AOD conditions. This issue is specifically addressed in Section 5.3.

[Figure]

*Figure 2. One-minute AOD data differences between AERONET-Cimel (V2) and GAW-PFR for (a) 380 nm (70838 data-pairs), (b) 440 nm (71645 data-pairs), (c) 500 nm (70833 data-pairs) and (d) 870 nm (71660 data-pairs) for the period 2005-2015. Black dots correspond to the U95 limits. A small number of  outliers are out of the ±0.06 AOD differences range. Black arrows  indicate a change of Reference AERONET-Cimel radiometer and red arrows indicate a change of the GAW-PFR instrument.*

*Table 3. Percentage of AERONET-Cimel (V2 and V3) 1-minute AOD data meeting the WMO criteria for the four interpolated GAW-PFR channels for the period 2005-2015.*

| Interpolated GAW-PFR channel (%) | Original GAW-PFR channel (%) |
|---|---|
|  |  |
|  |  |
|  |  |
|  |  |

| Channel | V2 (%) | V3 (%) |
|---|---|---|

| | | |
|---|---|---|
| 380 nm | 92.7 | 92.3 |
| 440 nm | 95.7 | 95.2 |
| 500 nm | 95.8 | 95.7 |
| 870 nm | 98.0 | 97.8 |

*Table 4. Percentage of AERONET-Cimel 1-minute AOD data (V2) meeting the WMO criteria for the four compared channels, and different AOD and AE scenarios for the period 2005-2015, number of data pairs are shown in brackets. The lLast row corresponds to the total percentages for the sub-period 2010-2015. In bold, AOD and AE traceability is > 95% are marked in bold. Number of data pairs are in brackets.*

| % of data within WMO limits | 380 nm | 440 nm | 500 nm | 870 nm |
|---|---|---|---|---|
| AOD ≤ 0.05 | 94.4 (57008) | **96.8 (59130)** | **97.0 (58572)** | **98.5 (60191)** |
| 0.05 < AOD ≤ 0.10 | 91.0 (4723) | 93.1 (4850) | 92.8 (4817) | 94.2 (4908) |
| AOD > 0.10 | 75.0 (3938) | 86.5 (4615) | 85.1 (4466) | **95.9 (5118)** |
| AE ≤ 0.25 | 73.1 (2145) | 82.3 (2417) | 80.1 (2351) | **96.2 (2824)** |
| 0.25 < AE ≤ 0.6 | 91.2 (5407) | **96.2 (5810)** | **96.0 (5691)** | **97.9 (5911)** |
| AE > 0.6 | 94.6 (55114) | **96.9 (57089)** | **97.0 (56504)** | **98.7 (58146)** |
| Total 2005-2015 | 92.7 (65669) | **95.7 (68595)** | **95.8 (67855)** | **98.0 (70217)** |
| Total 2010-2015 | 93.5 (41977) | **97.4 (43745)** | **97.2 (43627)** | **99.1 (44498)** |

In general, the agreement obtained with the 1-minute AOD data is slightly lower than that obtained during short campaigns, such as those reported by (Kazadzis et al. (, 2014) at Athens observatory (4685 data-pairs), and Barreto et al. (2016) at Izaña ObservatoryIZO (5566 data-pairs), with agreements > 99 % for $AOD_{870nm}$ and $AOD_{500nm}$ in case of Barreto et al. (2016). However, our results for $AOD_{500nm}$ (> 95 % of 70833 data-pairs) areis significantly better that that observed by Kazadzis et al. (2014) (~ 48 % of 4685 data-pairs) covering a relatively narrowshort range of AOD. The probable cause for the poor agreement found by (Kazadzis et al., 2014) was a poor calibration in the 500 nm channel in at least one of the instruments operating at Athens.

In addition, short-term campaigns usually cover a small range of AOD, normally with low AOD, and instruments are carefully and frequently supervised. On the contrary, during our intercomparison over a period of 11 years, the operation of the instruments can be described much moreconsidered as the normal operation of such a system. for a long term period of measurements, 20 than that of intensively attended instrumentation during short period intercomparison campaigns.

An additional interesting aspect of this study is that it is not a simple intercomparison exercise between two instruments but a comparison of a number of instruments that acted as reference instruments s for the AERONET/Europe Network.

*Table 5. Basic skill-scores from the AOD intercomparison between GAW-PFR and AERONET-Cimel V2 for the period 2005-2015. The skill scores definitions are found in Huijnen and Eskes (2012).*

| Period | 2005-2015 | | | |
|---|---|---|---|---|
| Wavelengths (nm) | 380 | 440 | 500 | 870 |
| Mean Bias (MB) | -0.0026 | -0.0018 | -0.0021 | -0.0001 |
| Modified Normalized Mean Bias (MNMB) | -0.1301 | -0.1046 | -0.1474 | 0.0129 |
| Fractional Gross Error (FGE) | 0.1727 | 0.1546 | 0.1918 | 0.1837 |
| Root Mean Square Error (RMSE) | 0.0081 | 0.0070 | 0.0064 | 0.0049 |
| Pearson's correlation coefficient (r) | 0.9910 | 0.9925 | 0.9939 | 0.9949 |
| Number of data-pairs | 70838 | 71645 | 70833 | 71660 |

In the first period (2005-2009), a total of 13 Cimel radiometers were used, while in the second period (2010-2015), five Cimel radiometers have participated, and for much of this period, the Cimel #244 was operating as the permanent AERONET reference instrument at IZO. Once the most important causes of non-traceability in the first period, which were associated with a poor pointing of GAW-PFR due to problems in the sun-tracker, were discounted ruled out, we can conclude that there are no significant differences in the percentages of traceable data between the two of both periods. This means that the continuous change of Master Reference Cimel instruments used in the 2005-2009 10 period did not have a significant impact on AOD data comparison differences. This provides proof of proves the consistency and homogeneity of the long AERONET-Cimel AOD data series, and their comparability with the GAW-PFR AOD data series, regardless of the number of instruments used to generate these data series.

In our study, with a number of comparison data-pairs one or two orders of magnitude higher than those used in short campaigns, the results shown in Table 4 can be considered excellent as fairly good.

In addition to the traceability scores, we have introduced some basic skill scores corresponding to the AOD intercomparison between GAW-PFR and AERONET-Cimel for the period 2005-2015 (Table 5) to be in line with previous studies that have performed short-term comparisons between these two instruments. The definitions of the used skill scores can be found in Huijnen and Eskes (2012).

The Pearson's correlation coefficient (r) values of the PFR-Cimel 1-minute AOD data-pairs, are higher than 0.99 in all channels. Concerning Mean Bias (MB) and Root Mean Square Error (RMSE) associated with to AOD differences, our results show quite similar skill scores to those found at Mauna Loa, USA for $AOD_{500nm}$ (Kim et al., 2008), although the number of data pairs used at Izaña Observatory IZO (~71000) is

much higher ~~than that of Mauna Loa (~9700)Master or R~~reference instruments. These results show MB values to be within 0.01 bias, one order of magnitude lower than in Mauna Loa and Izaña Observatories, highlighting the importance of having well calibrated instruments to carry out these type of comparisons.

For the period 2010-2015 (not shown here),  as expected, the RMSE and the Pearson's correlation improve slightly compared with the whole period 2005-2015.
* * *
**5.3. Non-traceability assessment**

As presented in the able 3, data outside the WMO traceability criteria vary from 2% for 870 nm up to 7.3% for 380 nm. In this section, the different possible causes of non-traceability in AOD are evaluated and, if possible, quantitatively estimated. In order to assess the relevance and quantitative impact of these causes, and estimate errors derived from a non-perfect AOD data synchronization, we first made an analysis on the natural variability of AOD in a very short time period (1 minute) shown below.

**5.3.1. Short-time AOD variability**

In order to determine the variability of AOD within one minute, we have performed two independent analyses with AOD data from the PFR and Cimel for the  368/380 nm and 501/500nm channels during one year (2013). On the one hand, and taking into account that GAW-PFR provides AOD every minute, we have calculated all the AOD differences for each channel in the successive minutes. So we have the variation of AOD from one minute to the next one during a whole year. On the other hand, for AERONET-Cimel, we have taken advantage of the triplets, since each triplet consists of three successive measurements made in one minute . In this case, the strategy has been to calculate the standard deviation of the triplet AOD measurements during a whole year. We have verified that the AOD variability in 1 minute is independent of AOD (see Supplement S3).

*Table 6. Percentage of AOD data with variability within 1 minute less than 0.01 and 0.005, respectively, using AOD data from GAW-PFR (at 368 and 501nm) and AERONET-Cimel (at 380 and 500 nm) for 2013. A total of ~32000 d data-pairs per channel have been used from GAW-PFR, and 20117 triplets (60351 individual AOD measurements) from the Cimel#244 to calculate the AOD variability.*

| GAW- PFR |
|---|

| Percentage of data with 1-minute AOD variability (%) | | |
|---|---|---|
| | 368 nm | 501 nm |
| < 0.01 | 99.88 | 99.91 |
| < 0.005 | 99.21 | 99.35 |
| AERONET-Cimel | | |
| Percentage of data with 1-minute AOD variability (%) | | |
| | 380 nm | 500 nm |
| < 0.01 | 99.87 | 99.99 |
| < 0.005 | 99.82 | 99.42 |

The results obtained on the AOD variability in 1-minute from PFR data are very similar and consistent to those obtained with Cimel. Less than ~ 0.8% of the AOD data show variability higher than 0.005 in all wavelength ranges. It should be noted that the possible instrumental noise is included in this variability, so that the actual natural AOD variability would be, in any case, lower than that expressed in Table 6. The percentage of data with 1-minute AOD variability for allthe four GAW-PFR channels are given in Supplement S3. These results indicate that the natural AOD variability is very low thus the non-ideal measurement synchronization cannot explain the percentages of non-traceable AOD cases shown in Tables 3 and 4.

**5.3.2. Uncertainties ofby GAW-PFR channel interpolation to AERONET-Cimel channels**

The interpolation of the CIMEL AODs to the PFR AOD wavelengths can be one of the sources of uncertainty in this comparison assessment. The greatest uncertainty arises in the extrapolation of the $AOD_{412\ nm}$ of the PFR to the Cimel wavelength 440 nmretrieve AODs at the CimelCIMEL $AOD_{440nm}$.

Using the Angström formula we have calculated that for an uncertainty of ±0.5 in the Angström exponentAE and for AOD of 0.1 at 412 nm, the introduced uncertainty in the AOD extrapolation from 412 nm to 440 nm is ~5% (i.e., 0.005 for $AOD_{412nm}$=0.1). The introduced uncertainty in AOD extrapolation is reduced to ~2% for an uncertainty of ±0.3 in AE. of the order of ±0.003, while for an $AOD_{412nm}$ of 0.5 and an AE uncertainty of ±0.3, the introduced uncertainty is ±0.008. For all other AOD interpolations the errors are smaller.

**5.32.31 Calibration related errors**

[revised manuscript text omitted]

The correct cause attribution of each outlier would require manual inspection and additional specific information on instrumental checking and maintenance information that is not always available. We have investigated in more  detail the origin of the outliers and whether one of the two instruments predominantly caused them.

Thus, we have calculated for the non-traceable AOD data the diurnal range of AOD variation (maximum value minus minimum value of AOD in one day) at 380 nm for each instrument under pristine conditions (Figure 3) using Cimel AOD$_{500nm}$ daily mean < 0.03 to select the pristine days. According to this approach, the instrument that shows the highest daytime AOD range is the one that is responsible for the outlier. As the wavelength increases both the number of outliers and the magnitude thereof decreases significantly. (Supplement S6). Then, we identified those outliers with a diurnal AOD range higher  than 25% of the mean daily AOD value and investigated their possible causes. A total of 51 cases for GAW-PFR and 81 cases for AERONET Cimel V3 were obtained and  analysed in detail, using  auxiliary information, such as 1-minute in-situ meteorological data, 5-minute all-sky images, 1-minute BSRN data, and satellite imagery (not shown here). We obtained the percentage of AOD outliers of GAW-PFR and AERONET Cimel (V3) for which a certain cause has been identified, such as calibration uncertainties, cloud screening algorithm failures, mixture of the two previous causes, poor sun pointing, or not well-defined  causes (electronic problems, humidity inside the lenses, filter dirtiness, obstruction of the lenses collimators, insects on the optics outside, etc.) (see Supplement S7).

From the analysis of these cases, under the conditions described above, it should be noted that ~ 44% of the cases with fictitious AOD diurnal cycles were due to small uncertainties in the calibration of AERONET-Cimel (V3), while for this same cause  ~ 8% of cases were identified in GAW-PFR.

Some examples of  AOD non-traceability for both AERONET-Cimel and GAW-PFR in the ~380 nm channel are shown in Supplement S8. The fictitious diurnal AOD cycle is mainly visible in the UV channels as shown in the examples reported in  Supplement S9.  that the fictitious diurnal AOD  can be more easily identified  under very low AOD conditions.

[Figure]

*Figure 3. AOD diurnal range  variation (maximum value minus minimum value of AOD in one day) at 380nm corresponding to AOD outliers (non-traceable AOD)  under pristine conditions (AOD$_{Cimel-500nm}$ ≤ 0.03) in the period 2005-2015 for AERONET V2 (a) and V3 (b).*

*5.2.2 Differences in cloud-screening and sun tracking*

We have examined the effect that the presence of clouds might have on AOD differences and the  percentage of cases outside the U95 limits. The impact of clouds on AOD differences only occurs when both GAW-PFR and AERONET-Cimel cloud screening algorithms fail to identify clouds in the direct sun path. ~~AERONET-Cimel Version 2 data uses the so-called "triplet check" cloud screening algorithm developed by Smirnov et al. (2000) and a second order temporal derivative constraint (McArthur et al., 2003) to rule out AOD measurements potentially contaminated by clouds. GAW-PFR algorithms also use the Smirnov triplet measurement, and the second-order derivative check, but add a test for optically thick clouds with> 2 (Kazadzis et al., 2018b). This algorithm, used by both networks with certain variants, assumes a transitory character in the presence of a cloud, which causes a sudden change of AOD. This sharp change would be detected by measuring the stability of three successive optical depth measurements, so that, when a cloud totally or partially blocks the sun, the standard deviation associated with the average of the triplets increases enormously. Note that if either one or both cloud screening algorithms (GAW-PFR and AERONET-Cimel) are flagged as cloudy, then the corresponding AOD data pair does not take part in the comparison.However, in the case of stratiform and very stable clouds or in the case of very thin clouds such as cirrus clouds, the algorithm could erroneously interpret that there are no clouds since there would be no appreciable changes in the stability of the triplets. A hint that cloud flagging failure could lead to large ADO calculated differences is coming from an analysis of AOD differences for days with different cloudy sky fractions.We do not have precise ancillary information to verify in each 1-minute data the influence that a certain cloud could cause in the non-traceability found. As a first approach forassessingIzaña Observatory~~IZO (García et al., 2014). FCS represents the percentage of observed sunshine hours in a day with respect to the maximum possible sunshine hours in that day. The higher the daily FCS, the higher the clear sky percentage we have on that day.

The percentages of traceable and non-traceable AOD data versus FCS values grouped into five intervals are shown in Table 8.  It should be emphasized that the number of cases linked with FCS between 0% and 60% are less than 2% of the total cases. ~~There, ~8% (870nm) to 24% (380nm) of the data are outside the WMO limits (maximum of 0.5% of the total data for 380nm outside the WMO limits).~~ As the fraction of clear

sky increases, the percentage of traceable AOD data significantly exceeds the number of non-traceable AOD data. The percentage of traceable data is especially large (> 90 %) when FCS > 80 % (almost clear skies).

This is the FCS range in which a significant percentage of days with cases presenting scattered clouds are recorded, which qualitatively confirms that V3 has introduced more efficient cloud screening than V2. However, the real impact of clouds on AOD traceability at IZO is very low due to its special characteristics of a high mountain station with very little cloudiness.  Therefore, in practice, the possible impact of clouds on the non-traceability of AOD data-pairs is insignificant at IZO .  GAW-PFR and AERONET-Cimel cloud screening algorithms provide successful identification on clear direct-sun conditions during cloudy skies (FCS < 40 %) for  99.75 % of the cases, excluding those with very thin clouds.  of AERONET V3 (Giles et al., 2018).

However, in the case of stratiform and very stable clouds or in the case of very thin clouds such as cirrus clouds, the algorithm could erroneously interpret that there are no clouds since there would be no appreciable changes in the stability of the triplets.

In the particular case of Izaña there are some very specific cloud scenarios in which cloud screening algorithms could fail resulting in non-AOD traceability: 1) Altostratus above the top of the SAL, at ~6 Km altitude (see Supplement S10); 2) Cirrus clouds (see Supplement S11); and 3) low clouds (stratocumulus) that sometimes exceed the observatory height level (see Supplement S11).

~~A more detailed analysis of more rare atmospheric conditions, such as those of strati-form and homogeneous cirrus clouds, or when altostratus are present above the Saharan Air Layer (SAL), around 6 Km altitude, and thus masked by a heavy mineral dust layer below needs further investigation. A constant cloud optical thickness (COT) corresponding to a cloud of a certain horizontal extension would cause the successive measurements within a minute to correspond to the same cloud stage, and therefore it would not be discernible from the extinction caused by aerosols. In the case of very thin cirrus clouds, the fluctuations in AOD would be very small and could be interpreted as the presence of a light layer of aerosols. Another factor that must be taken into account is that the FOV of the instruments is different. Thus, GAW-PFR (FOV = 2.5°) could detect the entry of a constant COT cloud in part of its~~

field of view in a different way than AERONET-Cimel (FOV = 1.2°). In all these cases, the cloud-screening algorithms may fail simultaneously in both GAW-PFR and AERONET-Cimel, resulting in a different AOD measurement derived by the two instruments. shown reported1-minute

As can be deduced from the analysis of these cloud cases, the impact of the different types of clouds oin AOD retrieval is very complex and further specific investigations are required in order to understand, the reasons behind failures in the GAW-PFR and AERONET-Cimel cloud screening algorithms.

These type of rare situations should be the subject of future studies through measurement campaigns using ancillary observation systems (e.g. lidar, all sky camera).

*Table 7Table 8. Percentage of traceable (T) data and percentage of AOD data outside within the U95 limits (NT) for each channel and 5 
[revised manuscript text omitted]

climatology. Concerning AERONET-Cimel Version 2, a NASA TOMS 1°_x_1.25° resolution $O_3$ climatology is used. From Eq. 10, the differences in $O_3$ optical depth $\triangle 4\tau_{O3}$ can be derived:

$$\triangle \tau_{O3} = \sigma_{O3}(\lambda) \frac{1}{1000} \frac{m_{O3}(O_{3PFR} - O_{3Cimel})}{m_a}$$

(12)

The largest influence of total ozone data uncertainty in $\tau_{O3}$ occurs , by far, at 500 nm (Figure 5). According to Wehrli (2008b) and Kazadzis et al. (2018b), total ozone needs to be determined to ± 30 DU or 10 % of typical values, to ensure an uncertainty of ± 0.001 in $\tau_{O3}$ at 500 nm. In the case of the GAW-PFR / AERONET-Cimel comparison, and due to the very different method in which both networks obtained $O_3$ values for their corresponding corrections, the ozone differences found on some days (1761 out of 71965 days; 2.4 %) are very large (> 40 DU), exceeding a difference in the ozone optical depth of 0.001. Even so, the potential contribution toof AOD differences outside the $U95$ limits between the two networks is negligible. Total $O_3$ over the Izaña ObservatoryIZO is quite stable, showingshows a relatively small amplitude throughout the year, but both surface ozone concentrations and column ozone amount could sharply increase under the influence of cut-off lows injecting air from the high-mid troposphere into the lower subtropical troposphere, which is not uncommon in spring and the first half of summer (Cuevas et al., 2015; Kentarchos et al., 2000). In addition through exchange processes in the Upper Troposphere Lower Stratosphere (UTLS) due to the presence of the subtropical jet (mainly from February to April) (Rodríguez-Franco and Cuevas, 2013). However, if we wanted to repeat this traceability study of 1-minute AOD data in mid or high latitude stations where sharp $O_3$ variations (several tens of DU) could be registered in a few hours, the correction of 1-minute AOD measurements by $\tau_{O3}$ might be a challenging issue.

[Figure]

*Figure 5. (a) Total O₃ used by GAW-PFR (measured Brewer O₃ values from IZO, OMI O₃ overpass or Brewer O₃ climatology) and AERONET-Cimel (TOMS O₃ climatology), and (b) $\Delta\tau_{O3}$ (λ) caused by differences in daily total O₃ between the two instruments in the period 2005-2015.*

-5.3.54.3 Differences in NO₂ absorption

AERONET-Cimel applies a correction by absorption of NO₂, but GAW-PFR does not include this correction. AERONET--Cimel obtains daily total NO₂ data from a 0.25° ᵒx 0.2°ᵒ resolution NO₂ monthly climatology obtained from the ESA Scanning Imaging Absorption SpectroMeter for Atmospheric CHartographY (SCIAMACHY) (Eskes and Boersma, 2003). In order to assess the contribution toin AERONET-Cimel 1-minute AOD data non-traceability by NO₂ absorption what we havereally to estimate is the NO₂ optical depth ($\tau_{NO2}(\lambda)$) of AERONET-Cimel since GAW-PFR does not perform this correction. Analogously to $\Delta\tau_{O3}$, the differences in nitrogen dioxide optical depth $\Delta4\tau_{NO2}$ can be obtained from:

$$\triangle \tau_{NO2} = \sigma_{NO2}(\lambda)\frac{1}{1000}\frac{m_{NO2}}{m_a}(-NO_{2Cimel})$$

(13)

Where $m_a$ is given by Eq. 7, $NO_{2Cimel}$ (DU) is the daily total NO₂ used by AERONET-Cimel, $\sigma_{NO2}(\lambda)$ is the NO₂ absorption  (Gueymard, 1995)  weighted by the specific filter response: 15.6 cm$^{-1}$ (380 nm), 12.3 cm$^{-1}$ (440 nm), and 4.62 cm$^{-1}$ (500 nm). Finally,  $m_{NO2}$ has the following expression (Gueymard, 1995):

$$m_{NO2} = \frac{1}{sin\theta + 602.30(90-\theta)^{0.5}(27.96+\theta)^{-3.4536}}$$

(14)

[Figure]

*Figure 5. (a) Total O₃ used by GAW-PFR (OMI O₃ overpass or Brewer O₃ climatology) and AERONET-Cimel (TOMS O₃ climatology), and (b) ΔτO3(λ) caused by differences in daily total O₃ between the two instruments in the period 2005-2015.*

[Figure]

*Figure 6. (a)*  *NO₂ monthly climatology obtained from the ESA SCanning Imaging Absorption SpectroMeter for Atmospheric CHartographY (SCIAMACHY), used by AERONET-Cimel at* *IZO, and (b)* *τ* *(λ) caused by differences in daily total NO₂ between GAW-PFR and AERONET-Cimel in the period 2005-2015. Note that GAW-PFR does not take into account the correction for the NO₂ absorption.*

*Table 9. Percentage (%) of additional traceable AERONET-Cimel AOD 1-minute data (V2 and V3)*  *after correcting by pressure, and total column $O_3$ and $NO_2$ for the period 2005-2015.*

| Channel | Increment (%) of traceable data after P, O₃ and NO₂ corrections | |
| --- | --- | --- |
|  |  | |
|  |  | |
|  |  | |
|  |  | |
|  | ~ | |

| Channel | Increment (%) of traceable AOD data after P, $O_3$ and $NO_2$ corrections | |
| --- | --- | --- |
| | V2 | V3 |
| 380 nm | 1.3 | 1.7 |
| 440 nm | 0.2 | 0.3 |
| 500 nm | 0.3 | 0.1 |
| 870 | ~0.0 | ~0.1 |

In Figure 6a the total $NO_2$ used by AERONET-Cimel to evaluate $\tau_{NO2}(\lambda)$ is depicted. Figure 6b shows the $\Delta\tau_{NO2}(\lambda)$ caused by differences in daily total $NO_2$ between GAW-PFR and AERONET-Cimel. $\Delta\tau_{NO2}$ is of the order of $10^{-3}$ for 380 and 440 nm channels, while, for 500 nm channel, it is of the order of $10^{-4}$.  However, it should  be noted  that an impact on AOD calculation is expected when replicating similar analysis in  highly $NO_2$ polluted regions  . Such cases include  large industrial cities from East Asia and Central and Eastern Europe  (e.g., Chubarova et al., 2016).

Taking into account the corrections for Rayleigh scattering and for the absorptions by $O_3$ and $NO_2$, we have calculated the additional traceable AOD   that lie within the U95 AOD limits  the non-traceability of the 1-minute AOD values (Figure 6;  Table 9). This percentage is maximum at 380 nm with 1.3% (V2) and 1.7% (V3) of the whole dataset.  The 870 nm channel is only affected by the Rayleigh correction component and therefore the increment of traceable data after the mentioned corrections is

minimalmum.The 1minute AOD data outside the *U*95 limits by these corrections is negligible in the 870 nm channel.

*————5.4.3 —GAW PFR and AERONET-Cimel comparison under high AOD conditions: the impact of dust forward scattering role of the foron different FOVs.*

When we represent the AOD differences between AERONET-Cimel and GAW-PFR versus AOD (GAW-PFR) for AOD > 0.1, we observe a positive slope that increases when the AOD fitted data are > 0.05 (dusty non-pristine conditions), we noteing that AERONET-Cimel shows slightly higher AOD values than GAW-PFR, being higher than +0.01 for AOD > 0.15 (Figure 7). , and more clearly at 500 nm. The AOD data outside the *U*95 limits (in red) increases notably from AOD > 0.1.

In fact, the percentage of data outside the U95 limitsnon traceable AOD data increases as AOD increases (Table 9Table 10), so that for dust-related aerosol conditions ($AOD_{500nm}$ > 0.3) the percentage of AOD data outside the *U*95 limits is > 50 % for 380 nm and 440 nm (all channels except for 870 nm (Table 9Table 10, percentages in brackets). Similar results are found when using AERONET V3 (see Supplement S13). The increase in the percentage of AOD data outside the *U*95 limits is especially significant at the 380 nm channel. Taking into account the number of data compared with the total cases, Tthese results show a small but non-negligible percentage of AOD differences outside the *U*95 limits for AOD > 0.1, ranging from ~0.3 % at 870 nm to ~ 1.9 % at 380 nm. This especially affects the shorter wavelengths (Table 9Table 10).

[Figure]

*Figure 7. Actual AOD differences between AERONET-Cimel V2 and GAW-PFR vs  AOD_{PFR} at (a) 380 nm (b) and 500 nm for the period 2010-2015. The fitting line has been calculated with those data points with AOD  > 0.1 and  Cimel-PFR AOD difference > 0. The number of data used in the plots are indicated in the legend. The percentage of non-traceable AOD data with these conditions is ~24% for 380 nm, and ~8% for 500 nm. Note that some traceable (black) points show larger AOD differences than non-traceable (red) points because of the air mass dependence of the WMO traceability criterion.*

*Table 10. Percentage of  AOD data outside the U95 limits at 380, 440, 500 and 870 nm channels and for three $AOD_{500nm}$ thresholds respect to all data and respect to all data for each AOD interval (in brackets).*

|  | Percentage of AOD data outside the U95 limits (%) | | |
|---|---|---|---|
|  | $AOD_{500nm}$>0.1 | $AOD_{500nm}$>0.2 | $AOD_{500nm}$>0.3 |
| 380 nm | 1.9 (25.0) | 1.2 (47.2) | 0.5 (59.8) |
| 440 nm | 1.0 (13.5) | 0.8 (32.0) | 0.5 (57.6) |
| 500 nm | 0.6 (8.0) | 0.5 (18.7) | 0.3 (39.3) |
| 870 nm | 0.3 (4.1) | 0.2 (6.4) | 0.1 (14.0) |

aerosol forward scattering within the FOV of various instruments and calculated AOD was investigated some decades ago by Grassl (1971) who determined that at AOD=1 the circumsolar radiation increases by >10% the incoming radiation. Russell et al (2004), using dust and marine aerosols data, quantified the effect of diffuse light for common sun photometer FOV. They reported that the correction to AOD is negligible (<1% of AOD) for sun photometers with narrow FOV (< 2°), which is greater than  the Cimel FOV and slightly smaller than the PFR FOV  (2.5°). Sinyuk et al. (2012) assessed the impact of the forward scattering aerosol on the uncertainty of the AERONET AOD, concluding that only dust aerosol  with high AOD and low solar elevation could cause a significant bias in AOD (> 0.01). ~~Torres et al. (2013) investigated the uncertainty of the FOV in the AERONET-Cimel measurements indicating that direct solar irradiance measurements are biased by the amount of aureole radiation that is assumed to be direct solar radiation. The solar aureole, also known as the circumsolar region, is the bright region surrounding the solar disc, which becomes especially visible when there is a burden of moderate-high aerosols in the atmosphere.~~

[revised manuscript text omitted]
 completelyis fairly good but it is not complet, e mainly for two reasons. The first one it is the inherent limitation of data correction using the percentage difference in AOD obtained by model simulation for a fixed effective radius.

We have assumed an effective radius of 1.5 μm but, in reality, the radius of dust particles varies. A reasonable range of dust particle radiussize is between 0.1 and 3 μm (Balkanski et al., 1996; Denjean et al., 2016; Mahowald et al., 2014). So, depending on the distance from the dust source to Izaña ObservatoryIZO and the size of the emitted dust, the effective radius could vary slightly between dust episodes. As can be seen in Figures 8a and 8b, the percentage differences in AOD between Cimel and PFR for a 1-2[1,2 μum] effective radius interval, the PFR-Cimel AOD relative difference at 380 nm (500 nm) might change between (around) ~-1.8% (-1.1%) to -4.9% (3.3%).

The second reason is a possible cloud contamination in AOD retrieval when altostratus are present above the SAL, as discussed in Section 5.3.4.

A similar analysis has been carried out for AERONET V3 (see Supplement S17), where we observe that the corrections obtained are not as good as those obtained for V2. This may be due to the very high AOD data retention in V3 which could include more cases in which altostratus clouds and dust are present. The effect of FOV on AOD retrieval should be taken into account for those radiometers with a relatively high FOV (>3°) measuring in regions with relatively high AOD (> 0.2) for most of the year, as is the case in many sites of Northern Africa, the Middle East and East Asia (Basart et al., 2009; Cuevas et al., 2015; Eck et al., 1999; Kim et al., 2007). This effect leads to AOD underestimation, and the variable number of high AOD episodes in each season of the year might affect the AOD long-term trends. AOD measurements under these conditions would be especially affected for optical air mass < 3.

[Figure]

*Figure 9. The same as Figure 7 after "correcting" the PFR AOD data by adding + 3.3% at 380nm and + 2.2.% at 500 nm to the 1-minute PFR AOD data > 0.1.*

Furthermore, it should be taken into account that, as discussed in section 5.2.2., under relatively high AOD conditions, the presence of altostratus above SAL is not infrequent, and they could also cause non-traceability in AOD when the cloud screening algorithms

~~fail. Note that a graphic equivalent to Figure 9 is shown in Supplement S17 but for AERONET V3, observing that the corrections obtained are not as good as those obtained for V2. This may be due to the very high AOD data retention in V3 which could include more cases in which altostratus clouds and dust are present. Therefore, the FOV study should be done once the dust events with presence of clouds over the SAL have been ruled out.~~

   sites of Northern Africa, the Middle East and East Asia (Basart et al., 2009; Cuevas et al., 2015; Eck et al., 1999; Kim et al., 2007).

*5.4 Angström exponent comparison*

We have performed a comparison of the AE provided by GAW-PFR and AERONET-Cimel using in both cases the AOD data obtained from the four common channels (380 nm, 440 nm, 500 nm and 870 nm) with a total of 70716 data-pairs. The PFR-AOD values have been ordered from lowest to highest by grouping them in intervals of 500 values for which the averages (and corresponding standard deviations) of the PFR-Cimel AE differences have been calculated ( Figure 10a). In a similar way we proceeded with the PFR-AE values (Figure 10b).

[revised manuscript text omitted]

The different possible causes of non-traceability in AOD were investigated as follows:

- Absolute AOD measurements synchronization.

  Analyzing 1-minute AOD variability we concluded that its impact on the AOD differences is negligible as only ~0.8% of the AOD data has a variability larger than 0.005 in all spectral ranges.

- Sun tracking misalignments.

~~The 1-minute AOD differences Mean Bias of this study is 0.001, an order of magnitude lower than those obtained from previous short-term PFR-Cimel comparison campaigns, in which the Cimel instruments were calibrated by transferring the calibration coefficients by comparison with co-located Master instruments. This indicates the importance of good calibration and maintenance of the Cimel instruments to obtain AOD data very similar to that of GAW-PFR.~~

$_{500nm}$

Sun tracking misalignments  constitute a serious problem and a major cause of non-traceability of AOD datapairs as demonstrated by the AOD data outside the U95 limits from the period 2005-20 as a consequence of episodic problems with the sun-tracker of the GAW-PFR radiometer. For the 2010-2015 period the percentage of traceable data-pairs improves to 93.5% (380 nm), 97.4% (440 nm), 97.2% (500 nm) and 99.1% (870 nm). However, most of these cases could be identified and excluded from the analysis.

- Cloud screening failure by both network algorithms.

~~Regarding AOD non-traceability due to the different cloud-screening algorithms of both networks, it must be said that both algorithms are very similar. GAW-PFR uses the same cloud screening as AERONET-Cimel but incorporates some additional controls. The only reason for AOD non-traceability comes from the simultaneous failure of both cloud screening algorithms because if one or both of them detect clouds, the data will not be part of the comparison.~~ According to our observations, the simultaneous failure of both cloud screening algorithms might occur only under the presence of large and stable cirrus, or altostratus (~ 6000 m a.s.l.) on the top of a heavily dust loaded Saharan air layer, hiding very wide and stable clouds. In these cases, the radiometers interpret these clouds as aerosol layers and might provide values very different from the real AOD. This effect, or the comparison at IZO, however, this effect is negligible since GAW-PFR and AERONET-Cimel cloud screening algorithms provide successful cloud identification on clear direct-sun conditions during cloudy skies (FCS < 40 %) for 99.75 % of the cases.

- Pressure measurements related errors.

-

 Since the accuracy of the new barometers built into new radiometers is about 3 hPa, and only errors in atmospheric pressure > 30 hPa might produce an impact on Rayleigh scattering, the AOD non-traceability due to errors in Rayleigh scattering is negligible.

- Total column ozone input uncertainty.

The largest influence of total ozone data uncertainty in ozone absorption occurs mainly at 500 nm. Total ozone needs to be determined to ±30 DU or 10 % of typical values, to ensure an uncertainty of ±0.001 ozone absorption at 500 nm. In the case of the GAW-PFR / AERONET-Cimel comparison, despite the very different methods in which both networks obtained $O_3$ values for their corresponding corrections, large ozone differences were found (> 40 DU) only on 2.4 % of the days , resulting in a

difference in the ozone optical depth slightly above ~0.001. The potential contribution to non-traceable AOD values between the two networks is negligible. However, in mid or high latitude stations where fast $O_3$ variations of several tens of DU might be registered, the correction of 1-minute AOD measurements by ozone absorption might be an issue to be considered.

- **Total column $NO_2$ input uncertainty.**

The differences in $NO_2$ absorption caused by differences in daily total $NO_2$ between GAW-PFR and AERONET-Cimel is of the order of $10^{-3}$ for 380 nm and 440 nm channels, while, for 500 nm channel, it is even lower, of the order of $10^{-4}$. So, differences in $NO_2$ absorption are negligible in the 1-minute AOD non-traceability of our study. However, $NO_2$ absorption might have some impact on AOD in highly polluted regions, such as in large industrial cities, where column $NO_2$ values are much larger than the climatological ones.

Taking into account the corrections for Rayleigh scattering and for the absorptions by $O_3$ and $NO_2$, we have calculated the combined effect of all of them on the non-traceability of the 1-minute AOD values. The highest impact occurs in the 380 nm channel, in which 25 % of the AOD data outside the $U$95 limits (~2% of the total compared data) are due to significant differences in pressure, and in $O_3$ and $NO_2$ absorption.  The 1-minute AOD data outside the $U$95 limits by these corrections is negligible in the 870 nm channel.

~~We have to note that the excellent results of this 11-year comparison and the small differences found under the strict $U$95 criterion cannot be linked with the relatively low AODs that can be found at IZO. This is because absolute calibration errors contribute to the AOD calculation in an absolute way so larger than 1 % calibration errors for a given period of time can lead to even negative AOD calculations for IZO site.~~

- **Impact of dust forward scattering in AOD retrieval uncertainty for different instrument FOVs**

 Since GAW-PFR has almost double the FOV (~2.5°°) compared to the AERONET-Cimel (~1.3°°), and direct solar irradiance measurements are biased by the amount of aureole radiation that is assumed to be direct solar radiation, it is reasonable to expect that GAW-PFR is more affected by the circumsolar irradiance than AERONET-Cimel radiometer when AOD is relatively high.  Modelling the dust forward scattering we have shown ~~We have explained part of the non-traceabilities found for relatively high AOD values, by analysing the relationship between the differences in circumsolar radiation measured by both instruments with the differences observed in AOD. We have observed a clear relationship between the Cimel-PFR AOD differences and the PFR-Cimel circumsolar radiation differences, with the slope of the fitted line greater for shorter wavelengths (380 nm). These results show~~ that a non-negligible percentage of

the non-traceable 1-minute AOD data for AOD > 0.1, ranging from ~0.3 % at 870 nm to ~1.9 % at 380 nm is, might be caused by the different FOV. This systematic error especially affects the shorter wavelengths. . Due to this effect, the GAW-PFR provides AOD values which are ~3% lower at 380 nm, and ~2% lower at 500 nm, compared with AERONET-Cimel. However, AOD underestimation This error couldan only have only some relevance be especially important in dusty regions if radiometers with relatively large FOV are used.

A comparison of the AE provided by GAW-PFR and AERONET-Cimel has been performed using in both cases AOD data obtained from the four nearby common channels with a total of 70716 data-pairs. This is a very strict AE calculation since it is necessary that AOD be accurately measured by the four channels simultaneously. AE differences > 0.2 increase exponentially under very pristine conditions (AOD ≤ 0.03 and AE ≥ 1), reaching AE differences of up to 1.6. However, for these conditions the atmospheric aerosol load is practically zero and so, its characterization with AE does not have any importance in practice. Under non-pristine conditions or those with a high mineral dust content (associated AOD > 0.03 and AE < 1), the AE differences remain <below 0.1.

Summarizing, we have presented for the first time a long-term (2005-2015) 1-minute AOD comparison among different types of radiometers belonging to different aerosol global networks. This comparison is a very demanding test of both GAW-PFR and AERONET-Cimel validated AOD datasets bases since aerosol scenarios correspond to extreme conditions: either very low aerosol loading, a "pristine" scenario that reveals small uncertainties in the calibration and in the cloud screening, or large dust load, which leads to a significant increase in the forward scattering aerosol with AOD, resulting in a slightly higher AOD underestimation by the GAW-PFR. From this comprehensive comparisoncomparison, analysis of the 1-minute AOD and AE data provided by the GAW-PFR and AERONET-Cimel instruments operating at the Izaña Observatory in the 2005-2015 period, we can conclude that the biases in the statistics are very small (< 0.003 in all channels) and therefore both AOD datasets bases are representative of the same AOD population, which it is a remarkable fact for the global aerosol community. It should be noted that AOD traceability at 380 nm (92.7 %) does not reach 95 % of the common data, the percentage recommended by WMO U95 criterion, so more efforts should be made to improve AOD in the UV range. In this study we have also focused much of our attention on investigated the data that are outside of the WMO U95 limits (<5% of the data at 440, 500 and 870 nm and <8% at 380nm) in order to understandknow the weak points of both GAW-PFR and AERONET-Cimel their causes and to be eventually able to correct the small inconsistencies detected in instrumental and methodological aspects in the future.

Our results suggest that, probably, WMO/CIMO traceability limits could be redefined as a function of wavelength, and the recommended radiometer FOV range of radiometers FOVs should be reconsidered.

The widely deployed AERONET-Cimel and GAW-PFR datasets play a crucial role in understanding AOD long-term AOD changes and detecting trends, so it would be desirable for both networks to be linked to the same GAW-WMO related reference. In this sense, these results will be used in future studies, not only to evaluate long term AOD trends at Izaña Observatory based on two independent instruments, but

also to provide additional insight on long-term AOD trend analysis, and its significance and validity, based on single instruments and their calibration and AOD processing procedure and uncertainty budget. Finally, special attention should be paid to the AERONET Cimel AOD data series used in trend detection in combination with the used data set homogeneity and their periodic calibration transfer from Master instruments.

*Competing interests.* The authors declare that they have no conflict of interest.

*Acknowledgements.* The authors thank Luc Blarel and Philippe Goloub (LOA, CNRS-University of Lille, France) for supervising the periodic calibrations of the Cimel MastersReference instruments. This study has been performed in the frame of the WMO CIMO Izaña Testbed for Aerosols and Water Vapour Remote Sensing Instruments. The work was supported by the project "The Global Atmosphere Watch Precision Filter Radiometer (GAW-PFR) Network for Aerosol Optical Depth long term measurements" funded by the Federal Office of Meteorology and Climatology MeteoSwiss International Affairs Division, Swiss GCOS Office. Part of the AERONET-Cimel radiometers have been calibrated at Izaña Observatory by AERONET- EUROPE Calibration Service, financed by the European Community specific programs for Integrating Activities: Research Infrastructure Action under the Seventh Framework Programme (FP7/2007-2013), ACTRIS grant agreement No. 262254, and Horizon 2020 Research and Innovation Program, ACTRIS-2 grant agreement No. 654109. This research has received funding from the European Union's Horizon 2020 Research and Innovation Programme under grant agreement No. 654109 (ACTRIS-2).The funding by MINECO (CTM2015-66742-R) and Junta de Castilla y León (VA100P17) is also gratefully acknowledged. We thank the staff of the Izaña Oobservatory for their effort and dedication in maintaining the instruments. We acknowledge the constructive comments of the anonymous referees. Our colleague Celia Milford has improved the English language of the paper. In memory of Prof Klaus Fröhlich, former director of PMOD-WRC, who initiated the AOD measurements programme at the Izaña Observatory in 1984 within the WMO Background Atmospheric Pollution Monitoring Network (BAPMoN).

---

## Author Response (AR1)

**Anonymous Referee #1 (amt-2018-438-RC1):**

**G.1. This is a very thorough and overall quite clearly written manuscript of great use to anyone relying on PFR or Aeronet derived AODs. It establishes how well the master instruments used in both networks compare over the long term and where and why minute differences arise. It does not address how well these calibrations are transferred across the networks and what the resulting uncertainties are for various stations. However, such questions have been addressed by many others as evidenced by citations in this manuscript. This reviewer only has a very small number of relatively minor comments**

**Authors:**

**We appreciate the positive comments of the Referee, and his/her specific corrections and comments that are addressed below.**

**The Referee must take into account that, according to the requirements and comments of the Referees # 3 and # 4, new relevant analysis have been incorporated in the paper what has required large additional workload, resulting in a significant improvement of the paper.**

**The most relevant new analyses have been:**

**1) Comparison of GAW-PFR with version V3 of AERONET and previous comparison between V2 and V3.**

**2) New study on modelling the impact of near-forward scattering on the AOD measured by the PFR and Cimel radiometers (with different FOVs).**

**3) Study of AOD variability in 1 minute to rule out possible differences in AOD due to a non-perfect synchronization of the PFR and Cimel sampling data.**

**4) Incorporation of case studies on the impact on AOD due to small inaccuracies in the calibrations or the clouds contamination.**

**Much more information about the above points has been included in a new Supplement material document of 24 pages.**

**S.1. Page 1, Line 4. Suggest to say "wavelength near" instead of "nearby wavelength"**

**Authors:**

**Rephrased.**

**S.2. Page 2, Line 15. Suggest mentioning some other sunphotometer networks, PHOTON, Japanese network etc.**

**Authors:**

**References to other networks have been requested by the four Referees. Although we referred to global networks with centralized data processing and databases, as well as standard calibration procedures in each of the networks, we fully agree to include a reference to other similar sunphotometer networks of global regional scope.**

**The new paragraphs introduced in the manuscript are as follows:**

*"…These are GAW-PFR (Global Atmosphere Watch - Precision Filter Radiometer; http://www.pmodwrc.ch/worcc/; last access: 05 September 2018) and AERONET-Cimel (AErosol RObotic NETwork - Cimel Electronique radiometer; http://aeronet.gsfc.nasa.gov; last access: 01 September 2018) networks. AERONET is, in fact, a federation of ground-based remote sensing aerosol networks established by NASA (National Aeronautics and Space Administration) and PHOTONS (PHOtométrie pour le Traitement Opérationnel de Normalisation Satellitaire; University of Lille- Service d'Observation de l'INSU, France; Goloub et al., 2007), being complemented by other sub-networks, such as, AEROCAN (Canadian sunphotometry network; Bokoye et al., 2001), AeroSibnet (Siberian system for Aerosol monitoring; Sakerin et al., 2005), AeroSpan (Aerosol characterisation via Sun photometry: Australia Network; Mitchell et al., 2017), CARSNET (China Aerosol Remote Sensing NETwork; Che et al., 2015), and RIMA (The Iberian network for aerosol measurements; Toledano et al., 2011).*

*There are other radiometer networks that in recent years have incorporated centralized protocols for data evaluation and databases, and performed regular intercomparisons with GAW-PFR and AERONET-Cimel. These include, for example, SKYNET (SKYradiometer NETwork), and its seven associated sub-networks, that uses the Prede-POM sky radiometer to investigate aerosol-cloud-solar radiation interactions (e.g. Campanelli et al., 2004; Nakajima et al., 2007; Takamura et al., 2004)."*

**The corresponding references have been added.**

**S.3. Page 2, Line 30. Suggest inserting "NASA" before "Goddard"**

**Authors:  Done**

**S.4. Page 3, Line 6. Numerous is an understatement.**

**Authors:  Replaced "numerous" by "many".**

**S.5. Page 4, Line 5. The use of "absolute" is misleading here. A Langley calibration alone is never absolute (i.e. the calibration value is just a signal in an engineering unit not W/m$^2$. The beauty is that an absolute calibration is not needed to derive AOD.**

**Authors:  We agree. "Absolute" has been removed.**

**S.6. Page 5, Line 5. Suggest replacing "constant" with "signal" here and everywhere else in the manuscript. Or maybe just explain once that in sunphotometry the calibration "constant" is the signal the instrument would read outside the atmosphere (extraterrestrial) at a normalized earth-sun distance?**

**Authors:  We have added:**

*"Note, that the extraterrestrial constant (calibration constant) is the signal the instrument would measure outside the atmosphere at a normalized earth-sun distance."*

**S.7. Page 6, Line 24. Font issue.**

**Authors:  Corrected**

**S.8. Page 6, Line 6. Incorrect grammar: "were" not "was".**

**Authors:  We have not found this mistake. The paper has now been corrected throughout for English.**

**S.9. Page 14: Line 14. This is misleading as the error in AOD due to error in the calibration "constant" is independent of AOD.**

**Authors:**

**We agree, but we guess the misleading statement was in Page 13, Line 14. So, the paragraph:**

*"A not sufficiently accurate determination of the calibration constant results in a fictitious AOD diurnal evolution presenting a concave or convex characteristic curve due to the calibration error dependence on solar air mass. The largest error occurs in the central part of the day (or lower air masses), mainly, in clean days with very low aerosol load (< 0.02 in 500 nm), as reported by Romero and Cuevas (2002) and Cachorro et al. (2004)."*

**has been replaced by the following one:**

*"A not sufficiently accurate determination of the calibration constant results in a fictitious AOD diurnal evolution presenting a concave or convex characteristic curve due to the calibration error dependence on solar air mass. The largest error occurs in the central part of the day (lower air masses), mainly, in clean days with very low aerosol load (< 0.02 in 500 nm) as reported by Romero and Cuevas (2002) and Cachorro et al. (2004), and as can be derived from Equation 2."*

*amt-2018-438*
**G.1. The paper "Aerosol optical Depth comparison between GAW-PFR and AERONET Cimel radiometers from long term (2005-2015) 1-minute synchronous measurements" is very interesting for the scientific community working on photometry. It is really well written, very accurate in the analysis of all the aspects affecting the comparison and it is pleasing to read. I recommend the publication on this journal also because homogenization of international networks of photometers is an important issue at this stage of research.**

Authors:  Thank you for the positive assessment of the paper. Specific comments and suggestions are addressed below.

The Referee must take into account that, according to the requirements and comments of the Referees # 3 and # 4, new relevant analysis have been incorporated in the paper what has required large additional workload, resulting in a significant improvement of the paper.

The most relevant new analyses have been:

1) Comparison of GAW-PFR with version V3 of AERONET and previous comparison between V2 and V3.

2) New study on modelling the impact of near-forward scattering on the AOD measured by the PFR and Cimel radiometers (with different FOVs).

3) Study of AOD variability in 1 minute to rule out possible differences in AOD due to a non-perfect synchronization of the PFR and Cimel sampling data.

4) Incorporation of case studies on the impact on AOD due to small inaccuracies in the calibrations or the clouds contamination.

Much more information about the above points has been included in a new Supplement material document of 24 pages.

 **I have few specific comments for the authors:**

**S.1.  1. Introduction, lines 12-15. It paper is not true that only two global ground-based radiometer networks exist, that is AERONET and GAW. SKYNET (https://www.skynet-isdc.org/) and its regional subnetwork ESR (www.eroroskyrad.net) provide centralized AOD and other optical and physical aerosol parameters, on a daily bases and downloadable from the website. SKYNET is also attending several intercomparison campaigns against PFR as example www.eroroskyrad.net/quatram.html, and the Fourth WMO Filter Radiometer Comparison for aerosol optical depth measurements. So I'd suggest the authors to mention their existence.**

**Authors:** References to other networks have been requested by the four Referees. Although we referred to global networks with centralized data processing and databases, as well as standard calibration procedures in each of the networks, we fully agree to include a reference to other similar sunphotometer networks of global and regional scope.

**The new paragraphs introduced in the manuscript are as follows:**

*"…These are GAW-PFR (Global Atmosphere Watch - Precision Filter Radiometer; http://www.pmodwrc.ch/worcc/; last access: 05 September 2018) and AERONET-Cimel (AErosol RObotic NETwork - Cimel Electronique radiometer; http://aeronet.gsfc.nasa.gov; last access: 01 September 2018) networks. AERONET is, in fact, a federation of ground-based remote sensing aerosol networks established by NASA (National Aeronautics and Space Administration) and PHOTONS (PHOtométrie pour le Traitement Opérationnel de Normalisation Satellitaire; University of Lille- Service d'Observation de l'INSU, France; Goloub et al., 2007), being complemented by other sub-networks, such as, AEROCAN (Canadian sunphotometry network; Bokoye et al., 2001), AeroSibnet (Siberian system for Aerosol monitoring; Sakerin et al., 2005), AeroSpan (Aerosol characterisation via Sun photometry: Australia Network; Mitchell et al., 2017), CARSNET (China Aerosol Remote Sensing NETwork; Che et al., 2015), and RIMA (The Iberian network for aerosol measurements; Toledano et al., 2011).*

*There are other radiometer networks that in recent years have incorporated centralized protocols for data evaluation and databases, and performed regular intercomparisons with GAW-PFR and AERONET-Cimel. These include, for example, SKYNET (SKYradiometer NETwork), and its seven associated sub-networks, that uses the Prede-POM sky radiometer to investigate aerosol-cloud-solar radiation interactions (e.g. Campanelli et al., 2004; Nakajima et al., 2007; Takamura et al., 2004)."*

**The corresponding references have been added.**

**S.2. 3.GAW-PFR and AERONET-Cimel radiometers, line 21: specify if these Cimel models have one sensor for both direct and diffuse solar radiation ( new models) or two different sensors ( old version). Lines 21-24, the deterioration of the filter is however "not minimized" by the absence of a system for keeping the temperature constant inside the optics. Page 6 line 1-3, it is better stating also here that the final measurement is an average of the 10-s measurements, even if declared later. Line 12, is not the triad of PFR calibrated by lamps, but using Langley plot?**

**Authors:** The Cimel radiometers used in this study are CE318-N, the classical standard version of AERONET before accepting the triple CE318-T. This information appears already in the manuscript.

**Concerning specific information on the sensors, the Cimel CE318-N models used in this study and classified by number of sensors, for both direct and diffuse solar radiation, are listed below:**

- **One sensor for both direct and diffuse solar radiation: Cimel instruments #25, #44, #45, #140, #244, #347, #421, #548.**
- **Two different sensors for direct and diffuse solar radiation: Cimel instruments #79, #117, #245, #380, #382, #383, #398.**

However, we have not included this detailed information in the paper since we consider it is irrelevant in terms of AERONET level 2 V2 and V3 quality data.

Concerning filter deterioration, we agree. We have replaced "minimized" by "reduced". The corresponding sentences were slightly modified as follows:

*"The possible deterioration of the interference filters is reduced since they are only sun-exposed during three consecutive 1-second direct-sun measurements per channel, this cycle being scheduled every ~15 minutes. The rest of the time the Cimel is taking sky radiance measurements, or at rest position, looking downwards."*

We have completed the Cimel measurement schedule in section 4.1 as follows:

*"AERONET-Cimel takes a sequence of three separate measurements (1-second per filter) in one minute interval (each one every 30 seconds). This sequence of measurements is called "triplet" and it is performed every ~15 minutes for air masses lower than 2, and with higher frequency for lower solar elevations."*

The GAW-PFR is calibrated routinely onsite with regular Langley plots, and additionally is periodically compared with the PMOD GAW-PFR reference triad.

**S.3.  4. Data and methodology used in this study Line 24, format problem Line 33, state here that in the section 5.4 AE will be compared**

Authors:  Corrected the format problem in line 24.

In section 4.1 we have added the following text::

*"Synchronous AE data provided by both instruments have also been compared (see section 5.5). GAW-PFR determines AE using all four PFR wavelengths (Nyeki et al., 2015), while AERONET-Cimel uses different wavelength ranges (340-440 nm, 380-500 nm, 440-675 nm, 440-870 nm, 500-870 nm) (Eck et al., 1999). As a consequence, we have calculated a new AE for the Cimel radiometer using the four channels equivalent to those of the PFR."*

**S.4.  5.1 AERONET-Cimel AOD traceability Line 5, I don't know if it is easy to represent, but in Figure1 it would be interesting to highlight the changes of equipment during the time.**

Authors:  We agree. We have included the change of instruments for both GAW-PFR (red arrows pointing down) and AERONET-Cimel (black arrows pointing up). See the following example for 870 nm.

[Figure]

**S.5.**   **5.2.1 Calibration related errors line 16: are all the involved Cimel "Maters"?; "in-situ absolute calibration": with absolute do you mean by lamp? Lines 20-21: It is not clear why you transfer calibration among Cimels if they are all masters, and therefore calibrated separately. Figure 2: why there is a hole of data for optical mass at about 4.2?**

**Authors:  The term "absolute" is not correct. We refer to Langely plot calibrations. Each Cimel master is individually calibrated at Izana using the Langley plot calibration. Calibration factors are not transferred from Master to Master instruments, but to field instruments, and this is done in other labs (LOA-Lille, GOA-Valladolid, CARSNET-Beijing).**

**Regarding the gap around 4.2 masses of air is, indeed, caused by the Teide volcano mountain since the sunset and previous moments take place behind the Teide during a part of the year around winter.**

**S.6.  5.2.3 Did you consider a possible influence of WV absorption at 500 nm? Small differences came out during the Fourth WMO Filter Radiometer Comparison. Pag 19 line 11 Cimel doesn't measure pressure, so better saying " from the different way the two instruments assume the atmospheric pressure".**

**Authors:   Neither GAW-PFR nor AERONET Cimel consider the absorption of PWV at 500 nm. It has not been taken into account in this work. In any case, and according to Kazadzis et al. (2018) (their Figure 9), differences in column optical thickness (when compared with the PFR triad) due to PWV is of the order of $10^{-4}$, very small.**

**S.7. Pag 24 line 9: Figure 6 is not about the combined effect of the 3 components, but about NO2.**

**Authors:**  **The reference to Figure 6 has been removed.**

**AMT-2018-438**
**G.1. Three major data interpretation issues discourage publication. They are about natural variability, sampling bias and instrument field of view (FOV).**

Authors:  The three "major data interpretation issues" have been addressed by the authors.

The Referee must take into account that, according to the requirements and comments of the Referees # 3 and # 4, new relevant analysis have been incorporated in the paper what has required large additional workload, resulting in a significant improvement of the paper.

The most relevant new analyses have been:

1) Comparison of GAW-PFR with version V3 of AERONET and previous comparison between V2 and V3.

2) New study on modelling the impact of near-forward scattering on the AOD measured by the PFR and Cimel radiometers (with different FOVs).

3) Study of AOD variability in 1 minute to rule out possible differences in AOD due to a non-perfect synchronization of the PFR and Cimel sampling data.

4) Incorporation of case studies on the impact on AOD due to small inaccuracies in the calibrations or the clouds contamination.

Much more information about the above points has been included in a new Supplement material document of 24 pages.

**G.2. First, the paper compares AOD observations between 60s averages and less-than1s averages (Page 8, line 1-2). The true AOD in the atmosphere generally changes over the 59s differentials. Yet the paper neglects the natural changes when making inferences on calibration.**

For example, the paper shows that Cimel observes generally wider AOD differences C1 AMTD Interactive comment Printer-friendly version Discussion paper between daily minimum and maximum than does PFR (page 15, line 21-24; Figure 3). This must be at least partly because Cimel samples with 30s intervals and captures natural changes while PFR sleeps. Yet the paper interprets the wide diurnal ranges as a sign of "an imperfect calibration" of Cimel (page 15, line 24-25). The comparison is fairer if PFR data are paired up over 61s and, better still, if the second of the three consecutive Cimel samples is excluded.

Authors:

We do not agree with the reviewer approach concerning the synchronization issue. In any case we have performed an analysis on short-term AOD variability (see reply to G-3) in order to try to detect non exact

**synchronization measurement issues and impact on the comparison results. We have added this new text in order to clarify this issue:**

*"GAW-PFR provides AOD values every 1 minute as an average of 10 sequential measurements of total duration less than 1 second (20 ms for each channel), then dark current is measured, going to the sleep mode until the next minute. AERONET-Cimel takes a sequence of three separate measurements (1-second per filter) in one minute interval (each one every 30 seconds). This sequence of measurements is called "triplet" and it is performed every ~15 minutes for air masses lower than 2, and with higher frequency for lower solar elevations. Therefore, AERONET-Cimel provides AOD values for each triplet, at least, every ~15 minutes. Note that AERONET-Cimel performs AOD measurements interspersed with sky radiance measurements, whose duration varies throughout the day, and therefore the AOD measurements are not necessarily provided at full minutes. We consider the 1-minute data as synchronous when GAW-PFR and AERONET-Cimel AOD data were obtained with a difference of ~30 s·"*

**Probably the Referee should take into account to properly assess this issue that the PFR takes one measure every 1 minute and the Cimel does it with a significantly lower frequency, every 10-15 minutes.**

**G.3. Misinterpretation is evident on a multi-year basis too. The paper attributes poor AOD agreements at 380 nm under pristine conditions to "insufficiently accurate calibration of AERONET-Cimel" (page 15, line 33-34). This is unsubstantiated. Because AOD is generally higher at shorter wavelengths, so is its natural variability in the absolute term. This would make the AOD discrepancies greatest at the shortest wavelength, even if calibration were perfect for both instruments.**

**Authors:**

**We have included the following subsection on "AOD natural variability". The AOD variability at 380nm is quite low (<0.005) and cannot explain the non AOD traceability. Please, see the new subsection 5.3.1:**

*"…In order to assess the relevance and quantitative impact of these causes, and estimate errors derived from a non-perfect AOD data synchronization, we first made an analysis on the natural variability of AOD in a very short time period (1 minute) shown below…"*

*"5.2.1 Short-time AOD variability*

*In order to know the variability of AOD within one minute, we have performed two independent analyses with AOD data from the PFR and Cimel for the 368/380 nm and 501/500nm channels during one year (2013). On the one hand, and taking into account that GAW-PFR provides AOD every minute, we have calculated all the AOD differences for each channel in the successive minutes. So we have the variation of AOD from one minute to the next one during a whole year. On the other hand, for AERONET-Cimel, we have taken advantage of the triplets, since each triplet consists of three successive measurements made in one minute. In this case, the strategy has been to calculate the standard deviation of the triplet AOD measurements during a whole year. We have verified that the AOD variability in 1 minute is independent of AOD (see Supplement S3). "Table 6. Percentage of AOD data with variability within 1 minute less than 0.01 and 0.005 using AOD data from GAW-PFR (at 368 and 501nm) and AERONET-Cimel (at 380 and 500 nm) for 2013. A total of ~32000 data-pairs per channel have used from GAW-PFR, and 20117 triplets (60351 individual AOD measurements) from the Cimel#244 to calculate the AOD variability."*

| GAW- PFR | | |
|---|---|---|
| *Percentage of data with 1-minute AOD variability (%)* | | |
| | *368 nm* | *501 nm* |
| *< 0.01* | *99.88* | *99.91* |
| *< 0.005* | *99.21* | *99.35* |
| AERONET-Cimel | | |
| *Percentage of data with 1-minute AOD variability (%)* | | |
| | *380 nm* | *500 nm* |
| *< 0.01* | *99.87* | *99.99* |
| *< 0.005* | *99.82* | *99.42* |

*"The results obtained on the AOD variability in 1-minute from PFR data are very similar and consistent to those obtained with Cimel. Less than ~ 0.8% of the AOD data show variability higher than 0.005 in all wavelength ranges. It should be noted that the possible instrumental noise is included in this variability, so that the actual natural AOD variability would be, in any case, lower than that expressed in Table 6. The percentage of data with 1-minute AOD variability for the four GAW-PFR channels are given in Supplement S3. These results indicate that the natural AOD variability is very low thus the non-ideal measurement synchronization cannot explain the percentages of non-traceable AOD cases shown in Tables 3 and 4."*

**G.4. Second, the paper finds "a bias with positive large outliers (higher Cimel AOD)" (page 15, line 5; Figure 2). This results from the intentional exclusion of high PFR values, an attempt to assess "pristine conditions (AOD500nm <= 0.03)" (page 13, line 26, 28). It is misleading to use this assessment to suggest contaminations on Cimel (page 15, line 6, 7). It is only fair to explicitly state that many negative outliers (higher PFR AOD) exist for PFR AOD just above 0.03 (shown in Figure 7).**

**Authors:**

**There is no intentional exclusion of high PFR values at all. We have filtered data using both the daily mean AOD from PFR and Cimel with quite similar results as shown below (left: for AOD$_{PFR}$≤0.03; right: for AOD$_{Cimel-V2}$≤0.03). Cimel presents a higher number of AOD outliers and of greater magnitude. See Supplement S6.**

[Figure]

[Figure]

Anyway, we have used Cimel V3 AOD to identify AOD outliers under pristine conditions in the corrected manuscript (new Figure 3). We have rewritten subsection 5.3.2 (Calibration related errors) where a detailed analysis of the AOD outliers is given as well as illustrative examples of problems associated with the calibration of both AERONET V3 and GAW-PFR in Supplement material (see Supplements S8 and S9).

**G.5. Third, FOV is not adequately appreciated as a significant source of AOD uncertainty under dust. The forward scattering by aerosols into FOV is definitely the primary cause for the poor agreement at 380 nm away from pristine conditions, not just "might be" (page 33, line 30). A support for this statement partly comes from the theory that the forward scattering is greater at shorter wavelengths.**

**Authors:**

We fully agree that forward scattering by dust into FOV is the main cause of the poorer agreement at 380 nm in dusty conditions. Please, read the new section 5.4. "GAW PFR and AERONET-Cimel comparison under high AOD conditions: the impact of dust forward scattering for different FOVs."

We agree. We have changed "*might be*" by "*is*".

**G.6. A more definitive support is given in Figure 7. It shows that the observed AOD differences is about 3% of AOD at 380 nm. Russell et al. [2004] explain why the FOV effect increases with AOD. And their calculation predicts a ~4% error for 1.25 degree half FOV and a ~1% error for 0.60 degree half FOV, both at 380 nm (their Figure 3). The difference, ~3%, is similar to Figure 7.**

**Authors:**

First of all, note that Russell et al. (2004) and the Referee refer to half-FOV while we refer to full-FOV (FOV) in the paper.

Please, read the new section 5.4. "GAW PFR and AERONET-Cimel comparison under high AOD conditions: the impact of dust forward scattering for different FOVs.". We have cited Russell et al (2004) and previous studies on aerosol forward scattering FOV in the corrected manuscript.

**Our results on the differences in AOD between PFR and Cimel for relatively high AOD, and the corresponding explanation, are in agreement with results reported by Russel et al. (2004). Note that we have done a new modelling of dust forward scattering using a Monte Carlo RTM. We have the following text in the new Section 5.4.:**

*"….The relative error in AOD depends strongly on the particle size but it is fairly constant for each $r_e$ value considered (see Supplement S16). For re~1.5μm, the relative error in AOD at 380nm (500nm) is ~1.6% (1.0%) for Cimel, and ~5% (~3%) for PFR. These errors are in good agreement with those estimated by Russell at al. (2004),…"*

**G.7.** It is incorrect to assign "an insufficient[ly] accurate calibration of AERONET-Cimel in this channel" as "the most likely cause" (page 32, line 9) of the observed differences (page 32, line 9). Such errors would not increase with AOD.

**Authors:**

**This sentence has been removed. Anyway, this is not correct. We do not say that the error increases with AOD. We say that the AOD difference and the calibration error effect on AOD difference, increases with decreasing air mass.**

**G.8.** FOV will remain as a source of uncertainty even if the adjustments are made for it. Because the FOV is much wider, the adjustments are greater for PFR than for Cimel. So is the uncertainty in them. Thus, other things being equal, the PFR is destined to be more erroneous than Cimel. It is incorrect to hold both instruments equally responsible for the resulting AOD discrepancies (page 34, line 8-9).

**Authors:**

**The impact aerosol forward scattering on different FOVs, and hence on AOD traceability, only takes place for relatively high AOD and aerosol types with high forward scattering (e.g. dust). For example for AOD=0.5, under dust aerosol conditions, the difference between Cimel and PFR is ~0.015. The percentage of AOD data outside the U95 limits is only 1.9% for AOD>0.1 at 380nm (see Table 9), and only in this case the non AOD traceability can be mainly attributed to PFR. In former page 34, lines 8-9 we refer to the total AOD traceability at 380nm (92.7%), including no-dust conditions under which the aerosol forward scattering/FOV effect doesn't cause any measurable impact.**

**G.9.** Also, the FOV-related uncertainty leads to a question as to why "WMO defines the PFR FOV as the recommended one for sun radiometers" (page 33, line 24-25). The quoted statement is not explained. Forward scattering into the FOV constitutes a deviation from the condition to which Beer's law pertains – a deviation that should be minimized, not recommended. Nor is the statement supported by a citation. To be sure, WMO (2007) recommends that the WORCC be designated the primary WMO Reference Centre for OD measurements, as referenced in page 2, line 24-26. But the WMO report does not mention a specific instrument, let alone support the PFR-specific statement.

**Authors:**

**The WORCC has defined, designed and built the PFR instruments as standard radiometers. The PFR was build based on the specifications defined by the official WMO report entitled: "*WMO/GAW aerosol measurement procedures guidelines* and recommendations" (first in 2003, and second edition in 2016)** https://library.wmo.int/doc_num.php?explnum_id=3073 that were based on the Guide to Meteorological

Instruments and Methods of Observation (2010) **https://library.wmo.int/pmb_ged/wmo_8_en-2012.pdf. So wavelengths, FOV, and other technical specifications of the PFR are based on such CIMO and WMO recommendations.**

**The maintenance of the AOD standard by the WMO WORCC Calibration Central laboratory (CCL) is described in Kazadzis et al., 2018.**

**We have included the following information in the Introduction section:**

*"The GAW-PFR was specifically designed by WORCC for this goal following the technical specifications defined by WMO (2003; 2016)."*

**And the following two references:**

**WMO: WMO/GAW Aerosol Measurement Procedures: Guidelines and Recommendations, WMO TD No. 1178, GAW Report No 153, 67 pp, https://library.wmo.int/pmb_ged/wmo-td_1178.pdf, 2003.**

**WMO: WMO/GAW Aerosol Measurement Procedures, Guidelines and Recommendations, 2nd Edition, WMO No 1177, GAW Report No. 227, 93 pp, https://library.wmo.int/doc_num.php?explnum_id=3073, 2016.**

**G.10. Meanwhile, there are a few reasons to encourage resubmission. The topic is important, given how widely ground-based sunphotometers are used in climate science and satellite validation. The data are abundant. Careful and useful pieces of discussion are provided regarding the impact of unattended operations (page 3, line 18-21) and the implications for other measurement sites (page 24, line 4).**

**Authors:**

**We appreciate the positive feedbacks of the Referee.**

**G.11. Authors are encouraged to consider the following suggestions and comments, in addition to the major issues raised above, prior to resubmission.**

**Authors:**

**Each and every one of the Referee's comments have been addressed.**

**S.1. Page 2, line 16. Skynet with PREDE instruments is worth mentioning as another ground-based sunphotometer/sky-radiometer network. [Takamura, T., and T. Nakajima, Overview of SKYNET and its activities, Opt. Pura Apl. 37, 3303-3308, 2004., http://atmos3.cr.chiba-u.jp/skynet/index.html]. Skynet has about as many stations as GAW-PFR does.**

**Authors:**

**We agree. We have included the following text in the Introduction section:**

*There are other radiometer networks that in recent years have incorporated centralized protocols for data evaluation and databases, and performed regular intercomparisons with GAW-PFR and AERONET-Cimel.*

*These include, for example, SKYNET (SKYradiometer NETwork), and its seven associated sub-networks, that uses the Prede-POM sky radiometer to investigate aerosol-cloud-solar radiation interactions (e.g. Campanelli et al., 2004; Nakajima et al., 2007; Takamura et al., 2004).*

**and the corresponding references.**

**S.2. Page 4, line 32. Separate vice from versa.**

**Authors:**

**Done**

**S.3. Page 5, line 1. Insert something like "Dust provides a good test on the treatment of the forward scattering into the field of view."**

**Authors:**

**We have added the following sentence:**

*"The periodical presence of a dust laden SAL allows us to evaluate the impact that the dust forward scattering into the field of view has on AOD retrieval."*

**S.4. Page 5, line 14. The precise sun-tracking enabled by a quad detector should be mentioned somewhere in this paragraph for Cimel. Sun tracker is only described for PFR (page 6, line 1).**

**Authors:**

**See reply to S.10.**

**S.5. Page 5, line 27. Is there a reference on the instrument response over the field of view, especially over the 0.7 degree "slope" (Table1)?**

**Authors:**

**Yes. The reference is Wehrli (2008a), already cited in the manuscript.**

**We have included this reference in the Table 1 caption.**

**S.6. Page 5, line 28. Replace significant with significantly.**

**Authors:**

**Done**

**S.7. Page 6, line 1. Replace the second comma with a period.**

**Authors:**

**Done**

**S.8. Page 6, line 4-7. The air mass dependence of uncertainty is worth mentioning here, since radiometric calibration is the primary concern of this paper.**

**Authors:**

**We agree. We have added the following sentence:**

*" It should be taken into account that, in general, in the ultraviolet range the AOD uncertainty is higher and depends on the optical mass (Carlund et al., 2017)."*

**S.9. Page 6, line17. Make 0 subscript, as in line 16.**

**Authors:**

**Done**

**S.10. Table 1. What does "No specific Sun tracker" mean? What does "Sun tracker robot" refer to? Replace "long-term" with 6 months, as described in page 6, line 18. Replace "2-3 months" with 3-4 months as described in page 6, line 16.**

**Authors:**

**Probably the Referee would like to know that the term "robot" appears in the acronym of AERONET (AErosol RObotic NETwork). The term "robot" is widely used among the Cimel and AERONET communities.**

**We have added in Section 3 the following text in order to clarify this issue:**

*"The robot performs automatic pointing to the sun by stepping azimuth and zenith motors using ephemeris based on time, latitude and longitude. Additionally, a four-quadrant detector is used to improve the sun tracking before each scheduled measurement sequence. This sensor guides the robot to the point where the intensity of the signal channel is maximum. Diffuse-sky measurements are also performed by Cimel to infer aerosol optical and microphysical properties. Two different routines are executed: almucantar (varying the azimuth angle keeping constant the zenith angle) and principal plane (varying the zenith angle keeping constant the azimuth angle). The ability of Cimel to perform both direct and diffuse-sky measurements makes it necessary to use a specific robot rather than a simple sun tracker."*

**We have changed:**

**"***2-3 months***" by "***at least 10 good morning Langleys plots***".**

**and "***3-4months***" by "***the time necessary to perform at least 10 good morning Langley plots***"**

**S.11. Page 6, line 21. Provide references that detail the data processing protocol, preferably including Langley plots, for each instrument.**

**Authors:**

**We have included the general reference:**

*Thomason, L.W., Herman, B. M., Schotland, R.M., and Reagan, J.A.: Extraterrestrial solar flux measurement limitations due to a Beer's law assumption and uncertainty in local time, Appl. Opt., 21, 1191–1195, https://doi.org/10.1364/AO.21.001191, 1982.*

**and the following paragraph:**

*"A detailed description of how AOD is obtained and the determination of extraterrestrial constants by GAW-PFR and AERONET-Cimel are provided by Holben et al. (1994, 1998; 2001), Toledano et al. (2018), and Wehrli (2000; 2008b)."*

**S.12. Page 6, line 24. Break down the second word.**

**Authors:**

**Done**

**S.13. Page 7, line 8. Drop "s" from corresponds, since the word data is used as plural in the preceding sentence.**

**Authors:**

**Done. The corresponding sentence was removed.**

**S.14. Page 7, line 5. Replace criterium with criterion or criteria. Note that the plural, if chosen, requires changes in the rest of the sentence.**

**Authors:**

**Done**

**S.15. Page 9, line 3. Revise this sentence, as it contradicts with the fact that the agreement is achieved for less than 95% at 380 nm.**

**Authors:**

**Done**

**S.16. Page 9, line 5. Table 4 is expected to appear after Table 3. Revise numbering.**

**Authors:**

**Corrected**

**S.17. Page 9, line 6. Is the first period "2005-2010"? Table 2 has "2005-2009", as in page 12, line 1 and page 34, line 14.**

**Authors:**

**The correct period is 2005-2009. Fixed.**

**S.18. Page 9, line 11. Explain how it is determined that problems in sun pointing were "the main cause" of AOD discrepancies between PFR and Cimel. Table 4 indicates that the fail rate decreased merely by one third - from 4.2% to 2.8% - at 500 nm upon the tracker update.**

**Authors:**

**This is explained in the manuscript (Page 9; lines 7-9):**

*"This finding was confirmed with data from the four–quadrant silicon detector (Wehrli, 2008a) that showed diurnal variation of the PFR sensors position up to 0.3∘."*

**The corresponding statement has been slightly nuanced as follows:**

*"This reduced problems in sun pointing, that were the main cause of most of the AOD discrepancies between PFR and Cimel, and therefore not attributable to the instruments themselves."*

**S.19. Page 9, line 11. This sentence implies that a sun tracker is not considered a part of the PFR instrument. That is surprising to those who perceive the tracker as an essential, fully-integrated component of a sun radiometer. It is like saying the steering wheel is not part of a car. Consider dropping the comma and the subsequent eight words.**

**Authors:**

**A sun tracker is absolutely necessary for the PFR radiometer in order to take direct-sun measurements but it is not part of the PFR radiometer itself. In fact, the PFR can use any commercial sun-tracker or prototype with a resolution of at least 0.08° (included in Table 1), unlike the Cimel, a radiometer that uses an ad-hoc robot that includes the functions of sun-tracker, and specific observations in the Almucantar and Principal Plane (see reply to S.10).**

**S.20. Page 11, line 9. Replace "can be one of the main causes of part of" with "is the main cause of".**

**Authors:**

**We agree. Done**

**S.21. Page 11, line 10. Replace 5.4 with 5.3.**

**Authors:**

**Done**

**Page 11, line 16. Bring Kazadzis et al. before the first parenthesis, and drop the comma.**

**Authors:**

**Done**

**Page 12, line 3. Again it is not clear what supports the notion "the most important cause [. . .], which were associated with a poor pointing of GAW-PFR".**

**Authors:**

**Please, see reply to S.18**

**S.22. Page 12, line 5. Replace "of both periods" with "between the two periods". Page 12, line 21. Elaborate on "MB values to be within 0.01 bias".**

**Authors:**

**Done.**

**S.23. Page 12, line 23. Drop the first "and".**

**Authors:**

**Done.**

**S.24. Page 13, line 2. This paragraph feels misplaced. It deserves to be a stand-alone subsection under Section 5.2. Combine it with the sixth paragraph of section 5.1.**

**Authors:**

**Section 5 has been rewritten and restructured. This paragraph is the new subsection 5.3.2.**

**S.25. Page 15, line 3. Remove the comma.**

**Authors:**

**This sentence was rephrased.**

**S.26. Page 15, line 19. "maximum value minus minimum value of AOD in one day" is less logical a metric for evaluating the calibration than the difference between the measurements at minimum and maximum air mass factor of the day.**

*Authors:*

**Diurnal AOD range is a quite logical metric. It allows detecting other possible problems apart from the diurnal AOD cycle due to small inaccuracies in the calibration**

**We have include cloud-screening problems. See the new** *"5.3.3 Calibration related errors"* **subsection.**

**S.27. Page 15, line 28. Insert a comma after "causes".**

**Authors:**

**Done.**

**S.28. Page 15, line 29. Replace "worst" with worse.**

**Authors:**

**Done.**

**S.29. Page 15, line 30. Replace the comma.**

**Authors:**

**This sentence was rephrased.**

**S.30. Page 17, line 10. Replace "a" with an.**

**Authors:**

**Done.**

**S.31. Page 24, line 17. Explain what exactly is "more clearly" shown at 500 nm.**

**Authors:**

**This was wrong. Removed.**

**S.32. Figure 7. If this figure is to remain on the paper, note in the caption that an identical data set, PFR AOD, appears in both x and y axes, a practice generally discouraged. Also state that the numbers in the legend are rounded to the significant digits. This is to forestall questions as to why the black lines do not reach exactly (x,y)=(0,0).**

**Authors:**

**This Figure has been modified completely. The new Figure 7 shows the PFR AOD in X-axis, and the Cimel-PFR AOD difference is in Y-axis only for $AOD_{PFR}$ >0.1 . The data period is 2005-2015. The linear fit equation appears in the legend.**

[Figure]

*Figure 7. Actual AOD differences between AERONET-Cimel V2 and GAW-PFR vs AOD$_{PFR}$ at (a) 380 nm (b) and 500 nm for the period 2005-2015. The fitting line has been calculated with those data points with AOD > 0.1 and Cimel-PFR AOD difference > 0. The number of data used in the plots are indicated in the legend. The percentage of non-traceable AOD data with these conditions is ~24% for 380nm, and ~8% for 500nm. Note that some traceable (black) points show larger AOD differences than non-traceable (red) points because of the air mass dependence of the WMO traceability criterion."*

**S.33. Page 26, line 7. "first" is misleading. Refer to previous studies such as Russell et al. [2004]. Replace "might play" with plays.**

**Authors:**

**Removed the word** *"first".*

**Replaced** *"might play"* **with** *"plays"*

**We agree. We have included references to Russell et al. (2004), and previous studies such as Grassl (1971).**

**We have included the following paragraph in new Section 5.4 that reads as follows:**

*"Forward aerosol scattering within the FOV of various instruments and calculated AOD was investigated some decades ago by Grassl (1971) who determined that at AOD=1 the circumsolar radiation increases by >10% the incoming radiation. Russell et al (2004), using dust and marine aerosols data, quantified the effect of diffuse light for common sun photometer FOV. They reported that the correction to AOD is negligible (<1% of AOD) for sun photometers with narrow FOV (< 2°), which is higher than the one of the Cimel and slightly smaller than the PFR FOV (2.5°). Sinyuk et al. (2012) assessed the impact of the forward scattering aerosol on the uncertainty of the AERONET AOD, concluding that only dust aerosol with high AOD and low solar elevation could cause a significant bias in AOD (> 0.01)."*

**S.34. Table 9. "(14.9)" for 500 nm, AOD>0.1 is not consistent with the "AOD>0.10" row of Table 4.**

**Authors:**

**It is consistent. Please, note that in former Table 9 we had shown the percentage of AOD data outside the U95 limits, in this case at 380 nm channel and AOD$_{500nm}$>0.1 respect to all data for each AOD interval (in brackets), which is 25%. In former Table 4 we showed the percentage of traceable AOD data at 380nm, corresponding to AOD>0.1, being 75%.**

**S.35. Figure 8. Revise the caption, yaxis labels and legends to better clarify what are plotted.**

**Authors:**

**This Figure has been removed. Please read the new Section 5.4 and corresponding information in the Supplement material (S13-S17).**

**S.36. Page 28, line 13. Replace "could lead" with leads.**

**Authors:**

**We agree. Done.**

**S.37. Table 10. Complete the right-most vertical line.**

**Authors:**

**Done.**

**S.38. Page 31, line 8. Explain in what way GAW-PFR is "the AOD reference globally" and "directly linked to WMO/CIMO".**

**Authors:**

**Please, see reply to G.9**

**S.39. Page 31, line 14, remove the first comma and "which".**

**Authors:**

**Done.**

**S.40. Page 31, line 28. Does the judgment made here with the word "excellent" hold even while the 95% criteria are not met at 380 nm?**

**Authors:**

**This sentence has been rephrased as follows:**

*"An excellent traceability of AOD from the AERONET-Cimel (V2 and V3) is found for 440, 500 and 870 nm, and fairly good results for 380nm"*

**S.41. Page 31, line 30. Remove the last five words.**

**Authors:**

**Removed.**

**S.42. Page 32, line 5. Provide more details or references regarding the Langley plots so that their quality can be verified.**

**Authors:**

**This is the conclusion section. References of the Langley plots were given in different parts of the paper, including: Holben et al. (1998), Eck et al. (1999), Toledano et al. (2018), and Kazadzis et al. (2018).**

**S.43. Page 32, line 29. Replace "in" with on.**

**Authors:**

**Done.**

**S.44. Page 32, line 33. Remove "Even so,".**

**Authors:**

**Done.**

**S.45. Page 33, line 13. Explain why the traceability metric should be redefined based on actual performance.**

**Authors:**

**Given that the AOD uncertainty is larger in the UV for several reasons, as it has been shown in this and previous studies, it seems more reasonable the traceability limits to be established in function of the spectral range in which AOD is determined.**

**S.46. Page 33, line 14. Replace "shouted".**

**Authors:**

**Done.**

**S.47. Page 33, line 15. The purpose of this paragraph is unclear. If it is a disclosure that low AOD cases are removed from the present analysis, it should be noted much earlier. If this paragraph is a suggestion to exclude negative AOD values from future data analyses, it is misguided. While such values are not physical, their exclusion would artificially bias the remaining data high.**

**Authors:**

**This paragraph had been included to clarify that the high traceability achieved is not due to the fact that a large part of the AOD values are very low and therefore their differences. However this paragraph has been removed in the present reviewed version.**

**For the referee information, the AOD values <0 in all the channels evaluated for both Cimel and PFR have been excluded from the study from the beginning. However, the number of negative AOD values is very low. For example, of 171935 CIMEL available AOD data in version V2 only 8 negative values ($5X10^{-3}$%) were excluded in 380nm, 134 in 440nm, 7 in 500nm and 1 in 870nm. For version V3, with 170749 AOD data available, only 15 values ($10^{-3}$%) were negative in 380nm, 131 in 440nm, 41 in 500nm and 11 in 870nm.**

**S.48. Page 33, line 16. Replace "absolute" with relative.**

**Authors:**

**See reply to S.47.**

**S.49. Page 34, line 7 says "both [PFR and Cimel] are representative of the same AOD population" over the 11 years, except for 380 nm. Similarly, page 12, line 7 says "[The agreement in AOD] proves the consistency and homogeneity of the long AERONETCimel AOD data series". These conclusions imply that Cimel's stability is adequate and that PFR's features for greater stability (page 5, line 27-31) are, while remarkable, not a significant advantage. It is, then, logical to favor Cimel for its much narrower FOV, a clear advantage over PFR. Arguments like this, or ones against it if any, would be a good addition.**

**Authors:**

**We don't agree. This is a long term comparison of a number of instruments of different type (Cimel and PFR). Cimel and PFR radiometers have some advantages and disadvantages compared to the other and have been designed for different goals. So based on this comparison we do not include sentences showing that one instrument is better than the other. When uncertainties and inconsistencies of one or both instruments could be reported we did so in order to address (and understand) the (small) differences observed. In addition the referee should consider that in the case of AERONET-Cimel we have worked with reference instruments ("Masters") with higher accuracy than those Cimel radiometers deployed in standard field stations.**

**S.50. Page 34, line 15. Revise the sentence. The paper does not directly address calibration transfer. Rather, the paper reveals that special attention should be paid to natural variability, sampling bias and PFR's wide FOV.**

*Authors:*

*The sentence has been removed.*

**S.51. References**

Russell, P. B., Livingston, J. M., Dubovik, O., Ramirez, S. A., Wang, J., Redemann, J., Schmid, B., Box, M., and Holben, B. N.: Sunlight transmission through desert dust and marine aerosols: Diffuse light corrections to Sun photometry and pyrheliometry, J. Geophys. Res., 109, D08207, 10.1029/2003JD004292, 2004.

**Authors:**

**Included.**

*AMT-2018-438*
**G.1. Three major data interpretation issues discourage publication. They are about natural variability, sampling bias and instrument field of view (FOV).**

Authors:  The three "major data interpretation issues" have been addressed by the authors.

The Referee must take into account that, according to the requirements and comments of the Referees # 3 and # 4, new relevant analysis have been incorporated in the paper what has required large additional workload, resulting in a significant improvement of the paper.

The most relevant new analyses have been:

1) Comparison of GAW-PFR with version V3 of AERONET and previous comparison between V2 and V3.

2) New study on modelling the impact of near-forward scattering on the AOD measured by the PFR and Cimel radiometers (with different FOVs).

3) Study of AOD variability in 1 minute to rule out possible differences in AOD due to a non-perfect synchronization of the PFR and Cimel sampling data.

4) Incorporation of case studies on the impact on AOD due to small inaccuracies in the calibrations or the clouds contamination.

Much more information about the above points has been included in a new Supplement material document of 24 pages.

**G.2. First, the paper compares AOD observations between 60s averages and less-than1s averages (Page 8, line 1-2). The true AOD in the atmosphere generally changes over the 59s differentials. Yet the paper neglects the natural changes when making inferences on calibration.**

**For example, the paper shows that Cimel observes generally wider AOD differences C1 AMTD Interactive comment Printer-friendly version Discussion paper between daily minimum and maximum than does PFR (page 15, line 21-24; Figure 3). This must be at least partly because Cimel samples with 30s intervals and captures natural changes while PFR sleeps. Yet the paper interprets the wide diurnal ranges as a sign of "an imperfect calibration" of Cimel (page 15, line 24-25). The comparison is fairer if PFR data are paired up over 61s and, better still, if the second of the three consecutive Cimel samples is excluded.**

Authors:

We do not agree with the reviewer approach concerning the synchronization issue. In any case we have performed an analysis on short-term AOD variability (see reply to G-3) in order to try to detect non exact

**synchronization measurement issues and impact on the comparison results. We have added this new text in order to clarify this issue:**

*"GAW-PFR provides AOD values every 1 minute as an average of 10 sequential measurements of total duration less than 1 second (20 ms for each channel), then dark current is measured, going to the sleep mode until the next minute. AERONET-Cimel takes a sequence of three separate measurements (1-second per filter) in one minute interval (each one every 30 seconds). This sequence of measurements is called "triplet" and it is performed every ~15 minutes for air masses lower than 2, and with higher frequency for lower solar elevations. Therefore, AERONET-Cimel provides AOD values for each triplet, at least, every ~15 minutes. Note that AERONET-Cimel performs AOD measurements interspersed with sky radiance measurements, whose duration varies throughout the day, and therefore the AOD measurements are not necessarily provided at full minutes. We consider the 1-minute data as synchronous when GAW-PFR and AERONET-Cimel AOD data were obtained with a difference of ~30 s·"*

**Probably the Referee should take into account to properly assess this issue that the PFR takes one measure every 1 minute and the Cimel does it with a significantly lower frequency, every 10-15 minutes.**

**G.3. Misinterpretation is evident on a multi-year basis too. The paper attributes poor AOD agreements at 380 nm under pristine conditions to "insufficiently accurate calibration of AERONET-Cimel" (page 15, line 33-34). This is unsubstantiated. Because AOD is generally higher at shorter wavelengths, so is its natural variability in the absolute term. This would make the AOD discrepancies greatest at the shortest wavelength, even if calibration were perfect for both instruments.**

Authors:

**We have included the following subsection on "AOD natural variability". The AOD variability at 380nm is quite low (<0.005) and cannot explain the non AOD traceability. Please, see the new subsection 5.3.1:**

*"…In order to assess the relevance and quantitative impact of these causes, and estimate errors derived from a non-perfect AOD data synchronization, we first made an analysis on the natural variability of AOD in a very short time period (1 minute) shown below…"*

*"5.2.1 Short-time AOD variability*

*In order to know the variability of AOD within one minute, we have performed two independent analyses with AOD data from the PFR and Cimel for the 368/380 nm and 501/500nm channels during one year (2013). On the one hand, and taking into account that GAW-PFR provides AOD every minute, we have calculated all the AOD differences for each channel in the successive minutes. So we have the variation of AOD from one minute to the next one during a whole year. On the other hand, for AERONET-Cimel, we have taken advantage of the triplets, since each triplet consists of three successive measurements made in one minute. In this case, the strategy has been to calculate the standard deviation of the triplet AOD measurements during a whole year. We have verified that the AOD variability in 1 minute is independent of AOD (see Supplement S3). "Table 6. Percentage of AOD data with variability within 1 minute less than 0.01 and 0.005 using AOD data from GAW-PFR (at 368 and 501nm) and AERONET-Cimel (at 380 and 500 nm) for 2013. A total of ~32000 data-pairs per channel have used from GAW-PFR, and 20117 triplets (60351 individual AOD measurements) from the Cimel#244 to calculate the AOD variability."*

| GAW- PFR | | |
|---|---|---|
| *Percentage of data with 1-minute AOD variability (%)* | | |
| | *368 nm* | *501 nm* |
| *< 0.01* | *99.88* | *99.91* |
| *< 0.005* | *99.21* | *99.35* |
| *AERONET-Cimel* | | |
| *Percentage of data with 1-minute AOD variability (%)* | | |
| | *380 nm* | *500 nm* |
| *< 0.01* | *99.87* | *99.99* |
| *< 0.005* | *99.82* | *99.42* |

*"The results obtained on the AOD variability in 1-minute from PFR data are very similar and consistent to those obtained with Cimel. Less than ~ 0.8% of the AOD data show variability higher than 0.005 in all wavelength ranges. It should be noted that the possible instrumental noise is included in this variability, so that the actual natural AOD variability would be, in any case, lower than that expressed in Table 6. The percentage of data with 1-minute AOD variability for the four GAW-PFR channels are given in Supplement S3. These results indicate that the natural AOD variability is very low thus the non-ideal measurement synchronization cannot explain the percentages of non-traceable AOD cases shown in Tables 3 and 4."*

**G.4. Second, the paper finds "a bias with positive large outliers (higher Cimel AOD)" (page 15, line 5; Figure 2). This results from the intentional exclusion of high PFR values, an attempt to assess "pristine conditions (AOD500nm <= 0.03)" (page 13, line 26, 28). It is misleading to use this assessment to suggest contaminations on Cimel (page 15, line 6, 7). It is only fair to explicitly state that many negative outliers (higher PFR AOD) exist for PFR AOD just above 0.03 (shown in Figure 7).**

**Authors:**

**There is no intentional exclusion of high PFR values at all. We have filtered data using both the daily mean AOD from PFR and Cimel with quite similar results as shown below (left: for $AOD_{PFR} \leq 0.03$; right: for $AOD_{Cimel-V2} \leq 0.03$). Cimel presents a higher number of AOD outliers and of greater magnitude. See Supplement S6.**

[Figure]

[Figure]

Anyway, we have used Cimel V3 AOD to identify AOD outliers under pristine conditions in the corrected manuscript (new Figure 3). We have rewritten subsection 5.3.2 (Calibration related errors) where a detailed analysis of the AOD outliers is given as well as illustrative examples of problems associated with the calibration of both AERONET V3 and GAW-PFR in Supplement material (see Supplements S8 and S9).

**G.5. Third, FOV is not adequately appreciated as a significant source of AOD uncertainty under dust. The forward scattering by aerosols into FOV is definitely the primary cause for the poor agreement at 380 nm away from pristine conditions, not just "might be" (page 33, line 30). A support for this statement partly comes from the theory that the forward scattering is greater at shorter wavelengths.**

**Authors:**

We fully agree that forward scattering by dust into FOV is the main cause of the poorer agreement at 380 nm in dusty conditions. Please, read the new section 5.4. "GAW PFR and AERONET-Cimel comparison under high AOD conditions: the impact of dust forward scattering for different FOVs."

We agree. We have changed "*might be*" by "*is*".

**G.6. A more definitive support is given in Figure 7. It shows that the observed AOD differences is about 3% of AOD at 380 nm. Russell et al. [2004] explain why the FOV effect increases with AOD. And their calculation predicts a ~4% error for 1.25 degree half FOV and a ~1% error for 0.60 degree half FOV, both at 380 nm (their Figure 3). The difference, ~3%, is similar to Figure 7.**

**Authors:**

First of all, note that Russell et al. (2004) and the Referee refer to half-FOV while we refer to full-FOV (FOV) in the paper.

Please, read the new section 5.4. "GAW PFR and AERONET-Cimel comparison under high AOD conditions: the impact of dust forward scattering for different FOVs.". We have cited Russell et al (2004) and previous studies on aerosol forward scattering FOV in the corrected manuscript.

Our results on the differences in AOD between PFR and Cimel for relatively high AOD, and the corresponding explanation, are in agreement with results reported by Russel et al. (2004). Note that we have done a new modelling of dust forward scattering using a Monte Carlo RTM. We have the following text in the new Section 5.4.:

*"….The relative error in AOD depends strongly on the particle size but it is fairly constant for each $r_e$ value considered (see Supplement S16). For re~1.5μm, the relative error in AOD at 380nm (500nm) is ~1.6% (1.0%) for Cimel, and ~5% (~3%) for PFR. These errors are in good agreement with those estimated by Russell at al. (2004),…"*

**G.7. It is incorrect to assign "an insufficient[ly] accurate calibration of AERONET-Cimel in this channel" as "the most likely cause" (page 32, line 9) of the observed differences (page 32, line 9). Such errors would not increase with AOD.**

**Authors:**

This sentence has been removed. Anyway, this is not correct. We do not say that the error increases with AOD. We say that the AOD difference and the calibration error effect on AOD difference, increases with decreasing air mass.

**G.8. FOV will remain as a source of uncertainty even if the adjustments are made for it. Because the FOV is much wider, the adjustments are greater for PFR than for Cimel. So is the uncertainty in them. Thus, other things being equal, the PFR is destined to be more erroneous than Cimel. It is incorrect to hold both instruments equally responsible for the resulting AOD discrepancies (page 34, line 8-9).**

**Authors:**

The impact aerosol forward scattering on different FOVs, and hence on AOD traceability, only takes place for relatively high AOD and aerosol types with high forward scattering (e.g. dust). For example for AOD=0.5, under dust aerosol conditions, the difference between Cimel and PFR is ~0.015. The percentage of AOD data outside the U95 limits is only 1.9% for AOD>0.1 at 380nm (see Table 9), and only in this case the non AOD traceability can be mainly attributed to PFR. In former page 34, lines 8-9 we refer to the total AOD traceability at 380nm (92.7%), including no-dust conditions under which the aerosol forward scattering/FOV effect doesn't cause any measurable impact.

**G.9. Also, the FOV-related uncertainty leads to a question as to why "WMO defines the PFR FOV as the recommended one for sun radiometers" (page 33, line 24-25). The quoted statement is not explained. Forward scattering into the FOV constitutes a deviation from the condition to which Beer's law pertains – a deviation that should be minimized, not recommended. Nor is the statement supported by a citation. To be sure, WMO (2007) recommends that the WORCC be designated the primary WMO Reference Centre for OD measurements, as referenced in page 2, line 24-26. But the WMO report does not mention a specific instrument, let alone support the PFR-specific statement.**

**Authors:**

**The WORCC has defined, designed and built the PFR instruments as standard radiometers. The PFR was build based on the specifications defined by the official WMO report entitled: "*WMO/GAW aerosol measurement procedures guidelines* and recommendations" (first in 2003, and second edition in 2016)** https://library.wmo.int/doc_num.php?explnum_id=3073 that were based on the Guide to Meteorological

Instruments and Methods of Observation (2010) **https://library.wmo.int/pmb_ged/wmo_8_en-2012.pdf. So wavelengths, FOV, and other technical specifications of the PFR are based on such CIMO and WMO recommendations.**

**The maintenance of the AOD standard by the WMO WORCC Calibration Central laboratory (CCL) is described in Kazadzis et al., 2018.**

**We have included the following information in the Introduction section:**

*"The GAW-PFR was specifically designed by WORCC for this goal following the technical specifications defined by WMO (2003; 2016)."*

**And the following two references:**

**WMO: WMO/GAW Aerosol Measurement Procedures: Guidelines and Recommendations, WMO TD No. 1178, GAW Report No 153, 67 pp, https://library.wmo.int/pmb_ged/wmo-td_1178.pdf, 2003.**

**WMO: WMO/GAW Aerosol Measurement Procedures, Guidelines and Recommendations, 2nd Edition, WMO No 1177, GAW Report No. 227, 93 pp, https://library.wmo.int/doc_num.php?explnum_id=3073, 2016.**

**G.10. Meanwhile, there are a few reasons to encourage resubmission. The topic is important, given how widely ground-based sunphotometers are used in climate science and satellite validation. The data are abundant. Careful and useful pieces of discussion are provided regarding the impact of unattended operations (page 3, line 18-21) and the implications for other measurement sites (page 24, line 4).**

**Authors:**

**We appreciate the positive feedbacks of the Referee.**

**G.11. Authors are encouraged to consider the following suggestions and comments, in addition to the major issues raised above, prior to resubmission.**

**Authors:**

**Each and every one of the Referee's comments have been addressed.**

**S.1. Page 2, line 16. Skynet with PREDE instruments is worth mentioning as another ground-based sunphotometer/sky-radiometer network. [Takamura, T., and T. Nakajima, Overview of SKYNET and its activities, Opt. Pura Apl. 37, 3303-3308, 2004., http://atmos3.cr.chiba-u.jp/skynet/index.html]. Skynet has about as many stations as GAW-PFR does.**

**Authors:**

**We agree. We have included the following text in the Introduction section:**

*There are other radiometer networks that in recent years have incorporated centralized protocols for data evaluation and databases, and performed regular intercomparisons with GAW-PFR and AERONET-Cimel.*

*These include, for example, SKYNET (SKYradiometer NETwork), and its seven associated sub-networks, that uses the Prede-POM sky radiometer to investigate aerosol-cloud-solar radiation interactions (e.g. Campanelli et al., 2004; Nakajima et al., 2007; Takamura et al., 2004).*

**and the corresponding references.**

**S.2. Page 4, line 32. Separate vice from versa.**

**Authors:**

**Done**

**S.3. Page 5, line 1. Insert something like "Dust provides a good test on the treatment of the forward scattering into the field of view."**

**Authors:**

**We have added the following sentence:**

*"The periodical presence of a dust laden SAL allows us to evaluate the impact that the dust forward scattering into the field of view has on AOD retrieval."*

**S.4. Page 5, line 14. The precise sun-tracking enabled by a quad detector should be mentioned somewhere in this paragraph for Cimel. Sun tracker is only described for PFR (page 6, line 1).**

**Authors:**

**See reply to S.10.**

**S.5. Page 5, line 27. Is there a reference on the instrument response over the field of view, especially over the 0.7 degree "slope" (Table1)?**

**Authors:**

**Yes. The reference is Wehrli (2008a), already cited in the manuscript.**

**We have included this reference in the Table 1 caption.**

**S.6. Page 5, line 28. Replace significant with significantly.**

**Authors:**

**Done**

**S.7. Page 6, line 1. Replace the second comma with a period.**

**Authors:**

**Done**

**S.8. Page 6, line 4-7. The air mass dependence of uncertainty is worth mentioning here, since radiometric calibration is the primary concern of this paper.**

**Authors:**

**We agree. We have added the following sentence:**

*" It should be taken into account that, in general, in the ultraviolet range the AOD uncertainty is higher and depends on the optical mass (Carlund et al., 2017)."*

**S.9. Page 6, line17. Make 0 subscript, as in line 16.**

**Authors:**

**Done**

**S.10. Table 1. What does "No specific Sun tracker" mean? What does "Sun tracker robot" refer to? Replace "long-term" with 6 months, as described in page 6, line 18. Replace "2-3 months" with 3-4 months as described in page 6, line 16.**

**Authors:**

**Probably the Referee would like to know that the term "robot" appears in the acronym of AERONET (AErosol RObotic NETwork). The term "robot" is widely used among the Cimel and AERONET communities.**

**We have added in Section 3 the following text in order to clarify this issue:**

*"The robot performs automatic pointing to the sun by stepping azimuth and zenith motors using ephemeris based on time, latitude and longitude. Additionally, a four-quadrant detector is used to improve the sun tracking before each scheduled measurement sequence. This sensor guides the robot to the point where the intensity of the signal channel is maximum. Diffuse-sky measurements are also performed by Cimel to infer aerosol optical and microphysical properties. Two different routines are executed: almucantar (varying the azimuth angle keeping constant the zenith angle) and principal plane (varying the zenith angle keeping constant the azimuth angle). The ability of Cimel to perform both direct and diffuse-sky measurements makes it necessary to use a specific robot rather than a simple sun tracker."*

**We have changed:**

**"***2-3 months***" by "***at least 10 good morning Langleys plots***".**

**and "***3-4months***" by "***the time necessary to perform at least 10 good morning Langley plots***"**

**S.11. Page 6, line 21. Provide references that detail the data processing protocol, preferably including Langley plots, for each instrument.**

**Authors:**

**We have included the general reference:**

*Thomason, L.W., Herman, B. M., Schotland, R.M., and Reagan, J.A.: Extraterrestrial solar flux measurement limitations due to a Beer's law assumption and uncertainty in local time, Appl. Opt., 21, 1191–1195, https://doi.org/10.1364/AO.21.001191, 1982.*

**and the following paragraph:**

*"A detailed description of how AOD is obtained and the determination of extraterrestrial constants by GAW-PFR and AERONET-Cimel are provided by Holben et al. (1994, 1998; 2001), Toledano et al. (2018), and Wehrli (2000; 2008b)."*

**S.12. Page 6, line 24. Break down the second word.**

**Authors:**

**Done**

**S.13. Page 7, line 8. Drop "s" from corresponds, since the word data is used as plural in the preceding sentence.**

**Authors:**

**Done. The corresponding sentence was removed.**

**S.14. Page 7, line 5. Replace criterium with criterion or criteria. Note that the plural, if chosen, requires changes in the rest of the sentence.**

**Authors:**

**Done**

**S.15. Page 9, line 3. Revise this sentence, as it contradicts with the fact that the agreement is achieved for less than 95% at 380 nm.**

**Authors:**

**Done**

**S.16. Page 9, line 5. Table 4 is expected to appear after Table 3. Revise numbering.**

**Authors:**

**Corrected**

**S.17. Page 9, line 6. Is the first period "2005-2010"? Table 2 has "2005-2009", as in page 12, line 1 and page 34, line 14.**

**Authors:**

**The correct period is 2005-2009. Fixed.**

**S.18. Page 9, line 11. Explain how it is determined that problems in sun pointing were "the main cause" of AOD discrepancies between PFR and Cimel. Table 4 indicates that the fail rate decreased merely by one third - from 4.2% to 2.8% - at 500 nm upon the tracker update.**

**Authors:**

**This is explained in the manuscript (Page 9; lines 7-9):**

*"This finding was confirmed with data from the four–quadrant silicon detector (Wehrli, 2008a) that showed diurnal variation of the PFR sensors position up to 0.3◦."*

**The corresponding statement has been slightly nuanced as follows:**

*"This reduced problems in sun pointing, that were the main cause of most of the AOD discrepancies between PFR and Cimel, and therefore not attributable to the instruments themselves."*

**S.19. Page 9, line 11. This sentence implies that a sun tracker is not considered a part of the PFR instrument. That is surprising to those who perceive the tracker as an essential, fully-integrated component of a sun radiometer. It is like saying the steering wheel is not part of a car. Consider dropping the comma and the subsequent eight words.**

**Authors:**

**A sun tracker is absolutely necessary for the PFR radiometer in order to take direct-sun measurements but it is not part of the PFR radiometer itself. In fact, the PFR can use any commercial sun-tracker or prototype with a resolution of at least 0.08° (included in Table 1), unlike the Cimel, a radiometer that uses an ad-hoc robot that includes the functions of sun-tracker, and specific observations in the Almucantar and Principal Plane (see reply to S.10).**

**S.20. Page 11, line 9. Replace "can be one of the main causes of part of" with "is the main cause of".**

**Authors:**

**We agree. Done**

**S.21. Page 11, line 10. Replace 5.4 with 5.3.**

**Authors:**

**Done**

**Page 11, line 16. Bring Kazadzis et al. before the first parenthesis, and drop the comma.**

**Authors:**

**Done**

**Page 12, line 3. Again it is not clear what supports the notion "the most important cause [. . .], which were associated with a poor pointing of GAW-PFR".**

**Authors:**

**Please, see reply to S.18**

**S.22. Page 12, line 5. Replace "of both periods" with "between the two periods". Page 12, line 21. Elaborate on "MB values to be within 0.01 bias".**

**Authors:**

**Done.**

**S.23. Page 12, line 23. Drop the first "and".**

**Authors:**

**Done.**

**S.24. Page 13, line 2. This paragraph feels misplaced. It deserves to be a stand-alone subsection under Section 5.2. Combine it with the sixth paragraph of section 5.1.**

**Authors:**

**Section 5 has been rewritten and restructured. This paragraph is the new subsection 5.3.2.**

**S.25. Page 15, line 3. Remove the comma.**

**Authors:**

**This sentence was rephrased.**

**S.26. Page 15, line 19. "maximum value minus minimum value of AOD in one day" is less logical a metric for evaluating the calibration than the difference between the measurements at minimum and maximum air mass factor of the day.**

*Authors:*

**Diurnal AOD range is a quite logical metric. It allows detecting other possible problems apart from the diurnal AOD cycle due to small inaccuracies in the calibration**

**We have include cloud-screening problems. See the new** *"5.3.3 Calibration related errors"* **subsection.**

**S.27. Page 15, line 28. Insert a comma after "causes".**

**Authors:**

**Done.**

**S.28. Page 15, line 29. Replace "worst" with worse.**

**Authors:**

**Done.**

**S.29. Page 15, line 30. Replace the comma.**

**Authors:**

**This sentence was rephrased.**

**S.30. Page 17, line 10. Replace "a" with an.**

**Authors:**

**Done.**

**S.31. Page 24, line 17. Explain what exactly is "more clearly" shown at 500 nm.**

**Authors:**

**This was wrong. Removed.**

**S.32. Figure 7. If this figure is to remain on the paper, note in the caption that an identical data set, PFR AOD, appears in both x and y axes, a practice generally discouraged. Also state that the numbers in the legend are rounded to the significant digits. This is to forestall questions as to why the black lines do not reach exactly (x,y)=(0,0).**

**Authors:**

**This Figure has been modified completely. The new Figure 7 shows the PFR AOD in X-axis, and the Cimel-PFR AOD difference is in Y-axis only for $AOD_{PFR}$ >0.1 . The data period is 2005-2015. The linear fit equation appears in the legend.**

[Figure]

*Figure 7. Actual AOD differences between AERONET-Cimel V2 and GAW-PFR vs AOD$_{PFR}$ at (a) 380 nm (b) and 500 nm for the period 2005-2015. The fitting line has been calculated with those data points with AOD > 0.1 and Cimel-PFR AOD difference > 0. The number of data used in the plots are indicated in the legend. The percentage of non-traceable AOD data with these conditions is ~24% for 380nm, and ~8% for 500nm. Note that some traceable (black) points show larger AOD differences than non-traceable (red) points because of the air mass dependence of the WMO traceability criterion."*

**S.33. Page 26, line 7. "first" is misleading. Refer to previous studies such as Russell et al. [2004]. Replace "might play" with plays.**

**Authors:**

**Removed the word** *"first".*

**Replaced** *"might play"* **with** *"plays"*

**We agree. We have included references to Russell et al. (2004), and previous studies such as Grassl (1971).**

**We have included the following paragraph in new Section 5.4 that reads as follows:**

*"Forward aerosol scattering within the FOV of various instruments and calculated AOD was investigated some decades ago by Grassl (1971) who determined that at AOD=1 the circumsolar radiation increases by >10% the incoming radiation. Russell et al (2004), using dust and marine aerosols data, quantified the effect of diffuse light for common sun photometer FOV. They reported that the correction to AOD is negligible (<1% of AOD) for sun photometers with narrow FOV (< 2°), which is higher than the one of the Cimel and slightly smaller than the PFR FOV (2.5°). Sinyuk et al. (2012) assessed the impact of the forward scattering aerosol on the uncertainty of the AERONET AOD, concluding that only dust aerosol with high AOD and low solar elevation could cause a significant bias in AOD (> 0.01)."*

**S.34. Table 9. "(14.9)" for 500 nm, AOD>0.1 is not consistent with the "AOD>0.10" row of Table 4.**

**Authors:**

**It is consistent. Please, note that in former Table 9 we had shown the percentage of AOD data outside the U95 limits, in this case at 380 nm channel and AOD$_{500nm}$>0.1 respect to all data for each AOD interval (in brackets), which is 25%. In former Table 4 we showed the percentage of traceable AOD data at 380nm, corresponding to AOD>0.1, being 75%.**

**S.35. Figure 8. Revise the caption, yaxis labels and legends to better clarify what are plotted.**

**Authors:**

**This Figure has been removed. Please read the new Section 5.4 and corresponding information in the Supplement material (S13-S17).**

**S.36. Page 28, line 13. Replace "could lead" with leads.**

**Authors:**

**We agree. Done.**

**S.37. Table 10. Complete the right-most vertical line.**

**Authors:**

**Done.**

**S.38. Page 31, line 8. Explain in what way GAW-PFR is "the AOD reference globally" and "directly linked to WMO/CIMO".**

**Authors:**

**Please, see reply to G.9**

**S.39. Page 31, line 14, remove the first comma and "which".**

**Authors:**

**Done.**

**S.40. Page 31, line 28. Does the judgment made here with the word "excellent" hold even while the 95% criteria are not met at 380 nm?**

**Authors:**

**This sentence has been rephrased as follows:**

*"An excellent traceability of AOD from the AERONET-Cimel (V2 and V3) is found for 440, 500 and 870 nm, and fairly good results for 380nm"*

**S.41. Page 31, line 30. Remove the last five words.**

**Authors:**

**Removed.**

**S.42. Page 32, line 5. Provide more details or references regarding the Langley plots so that their quality can be verified.**

**Authors:**

**This is the conclusion section. References of the Langley plots were given in different parts of the paper, including: Holben et al. (1998), Eck et al. (1999), Toledano et al. (2018), and Kazadzis et al. (2018).**

**S.43. Page 32, line 29. Replace "in" with on.**

**Authors:**

**Done.**

**S.44. Page 32, line 33. Remove "Even so,".**

**Authors:**

**Done.**

**S.45. Page 33, line 13. Explain why the traceability metric should be redefined based on actual performance.**

**Authors:**

Given that the AOD uncertainty is larger in the UV for several reasons, as it has been shown in this and previous studies, it seems more reasonable the traceability limits to be established in function of the spectral range in which AOD is determined.

**S.46. Page 33, line 14. Replace "shouted".**

**Authors:**

Done.

**S.47. Page 33, line 15. The purpose of this paragraph is unclear. If it is a disclosure that low AOD cases are removed from the present analysis, it should be noted much earlier. If this paragraph is a suggestion to exclude negative AOD values from future data analyses, it is misguided. While such values are not physical, their exclusion would artificially bias the remaining data high.**

**Authors:**

This paragraph had been included to clarify that the high traceability achieved is not due to the fact that a large part of the AOD values are very low and therefore their differences. However this paragraph has been removed in the present reviewed version.

For the referee information, the AOD values <0 in all the channels evaluated for both Cimel and PFR have been excluded from the study from the beginning. However, the number of negative AOD values is very low. For example, of 171935 CIMEL available AOD data in version V2 only 8 negative values ($5 \times 10^{-3}$%) were excluded in 380nm, 134 in 440nm, 7 in 500nm and 1 in 870nm. For version V3, with 170749 AOD data available, only 15 values ($10^{-3}$%) were negative in 380nm, 131 in 440nm, 41 in 500nm and 11 in 870nm.

**S.48. Page 33, line 16. Replace "absolute" with relative.**

**Authors:**

See reply to S.47.

**S.49. Page 34, line 7 says "both [PFR and Cimel] are representative of the same AOD population" over the 11 years, except for 380 nm. Similarly, page 12, line 7 says "[The agreement in AOD] proves the consistency and homogeneity of the long AERONETCimel AOD data series". These conclusions imply that Cimel's stability is adequate and that PFR's features for greater stability (page 5, line 27-31) are, while remarkable, not a significant advantage. It is, then, logical to favor Cimel for its much narrower FOV, a clear advantage over PFR. Arguments like this, or ones against it if any, would be a good addition.**

**Authors:**

**We don't agree. This is a long term comparison of a number of instruments of different type (Cimel and PFR). Cimel and PFR radiometers have some advantages and disadvantages compared to the other and have been designed for different goals. So based on this comparison we do not include sentences showing that one instrument is better than the other. When uncertainties and inconsistencies of one or both instruments could be reported we did so in order to address (and understand) the (small) differences observed. In addition the referee should consider that in the case of AERONET-Cimel we have worked with reference instruments ("Masters") with higher accuracy than those Cimel radiometers deployed in standard field stations.**

**S.50. Page 34, line 15. Revise the sentence. The paper does not directly address calibration transfer. Rather, the paper reveals that special attention should be paid to natural variability, sampling bias and PFR's wide FOV.**

*Authors:*

*The sentence has been removed.*

**S.51. References**

Russell, P. B., Livingston, J. M., Dubovik, O., Ramirez, S. A., Wang, J., Redemann, J., Schmid, B., Box, M., and Holben, B. N.: Sunlight transmission through desert dust and marine aerosols: Diffuse light corrections to Sun photometry and pyrheliometry, J. Geophys. Res., 109, D08207, 10.1029/2003JD004292, 2004.

**Authors:**

**Included.**

**AMT-2018-438**
**Review of Aerosol Optical Depth comparison between GAW-PFR and AERONET Cimel radiometers from long term (2005-2015) 1-minute synchronous measurements**

**General Comments**

**G.1.** The manuscript provides a comparison of AERONET-Cimel and GAW-PFR at IZO. The work is scientifically relevant given the analyzed data volume compared to previous studies. The manuscript identifies issues in the comparison of the AOD from the two different radiometers that are independently calibrated and processed in different networks.

**Authors:**

We thank the referee for this comment.

The Referee must take into account that, according to the requirements and comments of the Referees # 3 and # 4, new relevant analysis have been incorporated in the paper what has required large additional workload, resulting in a significant improvement of the paper.

The most relevant new analyses have been:

1) Comparison of GAW-PFR with version V3 of AERONET and previous comparison between V2 and V3.

2) New study on modelling the impact of near-forward scattering on the AOD measured by the PFR and Cimel radiometers (with different FOVs).

3) Study of AOD variability in 1 minute to rule out possible differences in AOD due to a non-perfect synchronization of the PFR and Cimel sampling data.

4) Incorporation of case studies on the impact on AOD due to small inaccuracies in the calibrations or the clouds contamination.

Much more information about the above points has been included in a new Supplement material document of 24 pages.

**G.2.** One major issue is the comparison of AERONET-Cimel to potentially suspect GAW-PFR field instrument (not reference) data (e.g., 2005-2010).

**Authors:**

We have clarified in the paper and in several replies below that the GAW-PFR field instrument is traceable to the World aerosol optical depth research and calibration center (WORCC) AOD reference and cannot be considered as a standard station instrument.

In practise IZO instruments are part of the calibration scheme of WORCC. Their long term stability is checked with frequent signal comparisons (6 months to 1 year) with the WORCC reference PFR triad and their short term stability by the calculation of in situ (IZO) Langley based ETCs.

Maintaining the stability of the PFR triad includes criteria:

- A continuous (minute and signal based) comparison of each of the triad PFRs with the other two.
- A 6-12 month check with the IZO (and Mauna Loa) PFRs

So the instrument is considered by WORCC as a reference based on the above short summary of the WORCC protocols.

We have added the following text in Section 3 in order to clarify this issue:

*"In case of the GAW-PFR, the calibration system is more complex in order to ensure traceability with the WORCC world reference. The maintenance of the AOD standard by the WORCC Calibration Central laboratory is described in Kazadzis et al. (2018). It consists of a triad of instruments that measure continuously, and three additional portable transfer standard radiometers located at Mauna Loa (one instrument) and Izaña (two instruments) observatories. Every six months, one of the portable transfer standard radiometers visits the reference triad based at PMOD/WRC (Davos) and compares the calibration constants defined by the 6-month Langley calibrations in the two high mountain stations (Table 1 of Kazadzis et al. 2018) with the one defined by the triad. The comparison is based on the signals (voltages) and not on AOD values. The differences between the Izaña GAW-PFR radiometers and the reference triad have been always lower than 0.5%, being within the uncertainty of the Langley method plus the small possible instrument degradations that can be detected in a 6-12 month period. Such degradations are quite small and are accounted for in the calibration analysis since extraterrestrial constants are linearly interpolated between two triad visits or every 6-month periods. Additionally, the Izaña GAW-PFR "field" radiometers are calibrated on a routine basis using the Langley-plot technique for double checking quality assurance. Therefore, these radiometers cannot be considered as simple "field" instruments, but as regularly calibrated radiometers with assured traceability with the WORCC triad reference."*

**G.3. The authors present results in the abstract and conclusion with percentage agreement of 92.7% to 98.0% spectrally; however, these values are not from the apparently more optimal data set comparison for GAW-PFR (2010-2015), which show 1% to 2% improvement.**

Authors:

This is exactly what is stated from the beginning (in the abstract). We do not wish to present a study in which only good results are shown. We have identified and analyzed causes of lack of agreement introduced by both GAW-PFR and AERONET-Cimel and that appear in the databases of the highest quality level of both networks. Logically if the small percentage of erroneous data of both instruments are removed, the agreement skill scores improve. However, we emphasize the fact that, even considering all supposedly good data available in the official data bases of both networks, the agreement between GAW-PFR and AERONET-Cimel is excellent according to the WMO criteria in 440, 500 and 870nm. We could also consider as very good the results in the most complex 380 nm channel since the percentage of intercompared data within the WMO criteria is > 92%.

**G.4. Further, description of the causes of anomalies (i.e., not meeting the WMO standards) tend to be difficult to follow (e.g. reasoning for calibration and FOV) and it is not clear any corrections are actually applied to a final GAW-PFR data set.**

**Authors:**

**See specific replies to the different issues the Referee consider are difficult to follow. We consider these difficulties have been properly addressed. However, we have to make it very clear that we have not made any correction to the data of any instrument to match that of the other. The paper is focused on determining the degree of agreement in AOD data provided by three GAW-PFR radiometers, each one traceable to the world AOD reference, with the data provided by 15 reference AERONET-Cimel that have been individually calibrated by the Langely plot technique at Izaña in the framework of the AERONET calibration system. All this has allowed us to estimate the traceability of the AERONET-Cimel AOD data series with the WORCC world AOD reference, and to determine and quantify, when possible, the causes of non-traceability.**

**Please, read the new Section 5.4 on dust forward scattering and the instrument FOV.**

**G.5. Also, the time matching criteria of 30 seconds is quite large for an instrument that performs measurements every minute and the AOD could change up to 0.01 per minute or potentially higher for dust.**

**Authors:**

**The time matching criteria of 30 seconds is the criteria normally used in short intensive AOD radiometers intercomparison campaigns that last less 2-3 weeks and have been specifically designed to compare instrument performances (see, e.g., Kazadzis et al., 2018b). The Referee should keep in mind that although a minute value is available in the GAW-PFR, the AERONET-Cimel not only measures AOD but also performs sky radiance measurements (Principal Plane and Almucantar) with periods of measurement times that vary throughout the day, resulting in AOD measurements every 10 or 15 minutes that do not necessarily coincide in the full minute. The 30 seconds of matching criteria are absolutely acceptable ... and technically necessary.**

**Concerning the statement of the Referee that AOD might change 0.01 per minute caused by atmospheric aerosol variability, we have never seen that. This is really an "uncommon" situation at Izaña. We don't know if this value provided by the Referee comes from observations of a severe dust event (e.g. haboob). Anyway, and in order to elucidate this point, we have made a 1-year analysis of the AOD variability within 1 minute performed independently with GAW-PFR and AERONET-Cimel data (see new Section 5.3.1).In this Section the Referee can verify that the natural variability of AOD within one minute, including instrumental noise, is really low, even for high AOD values, and that the 30 seconds matching criteria is adequate and reasonable.**

**G.6. Some comparison results appear to be repeated in the tables. For example, the same "traceability" results (i.e., 92.7%, 95.6%, 95.8%, and 98.0%) appear to be repeated in three different tables (Tables 3, 4, and 8).**

**Authors:**

We repeated some data in order to facilitate the access of information to the reader avoiding the query of several tables at the same time. However, we have removed this information from Table 8, but not in Table 4 where it seems appropriate to continue appearing as a reference to the results obtained in different ranges of AOD and AE.

G.7. Another major issue is that the study focuses more on the long-term and does not present any specific cases in pristine and dust events to specifically show examples of the differences in each instrument AOD measurements with subsequent analysis.

Authors:

Yes. It is clear that the approach of this paper is long-term as it is explicitly stated in its title. However, we have included different types of case analyses as Supplement material (S8-S12) as suggested by the Referee.

G.8. A further issue is that the AERONET version 3 data are not included in the analysis. While these data are referenced as being available, the new product has some significant changes in regards to cloud screening and corrections made to the data that may impact agreement between the AERONET-Cimel and GAW-PFR instruments. Utilizing these AERONET V3 data would provide an added element of importance in AERONET/GAW-PFR comparisons.

Authors:

We fully agree. Although we started this study long before Version 3 (V3) was available and published (Giles et al., 2019), and considering that most of the scientific community that has used AERONET data is very interested in the comparison of the Version 2 (V2) (hundreds of scientific papers), we think it's a good idea to do the evaluation of V3. This new version is the official AERONET one from now on. A comparison of V2 and V3 has been included in the new Section 5.1. All results of the GAW-PFR comparison with AERONET-Cimel made for V2 have been replicated for V3: Figures 1 and 3, and Tables 3 and 8 in the manuscript, and S1-S14, and S17-S18 in the Supplement material.

G.9. Last, the presentation of the document was difficult to follow at times. For example, the study objective statement is first encountered in the summary and conclusions section and this is not easy to follow.

Authors:

We have included a detailed description of the objectives of the paper at the very beginning. Removed from conclusions.

G.10. The "traceability" criteria tend to indicate that AERONET-Cimel is compared to the GAW-PFR but, in this case, GAW-PFR is not a reference instrument but a field instrument, which has higher uncertainty.

Authors:

See reply to G.4.

**G.11. Also, description in the text of the Figures and Tables needs further elaboration. Some specific comments are provided below on organization and other issues. The authors should take care to correct and address the issues here and below before resubmission.**

Authors:

All the specific comments have been addressed below.

**Specific Comments**

**S.1. Abstract, Page 1, Line 6-9: "Traceability" as described only relates the precision of these two instruments. The result of the measurement may not be accurate and but both measurements may be precise. What do you mean by "WMO standard?" In the abstract, the authors need to state that GAW-PFR is considered a ground-based WMO standard for AOD measurements and the field GAW-PFR and AERONET Cimel tend to have strong agreement. The use of "traceable" is an ambiguous term in defining more specifically the "agreement" between the two instruments. For example, the field PFR has a "traceable" calibration to the PFR triad. Please consider changing instances of "traceable" and "traceability" through the document.**

Authors:

This is a long term comparison of a number of instruments of different type (Cimel and PFR). Of course, the results of the measurements of both instruments may not be accurate. So based on this comparison we have not included sentences showing that one instrument is better than the other. When uncertainties and inconsistencies of one or both instruments could be reported we did so in order to address the (small) differences observed.

On the other hand, and according to the Commission of Instruments and Methods of Observations (CIMO) that is the WMO related body "*responsible for matters relating to international standardization, compatibility and sustainability of instruments and methods of observation of meteorological, climatological, hydrological, marine, and related geophysical and environmental variables*", the  World aerosol optical depth research and calibration center (WORCC) has been defined as a designate primary WMO Reference Center for OD measurements.

In the related report https://www.wmo.int/pages/prog/www/CIMO/CIMO14-WMO1019/1019_E.pdf: "*The Commission recognized the need for establishing a primary reference Aerosol  Optical Depth Centre to satisfy the need for traceability of Optical Depth (OD) measurements, conducting international intercomparisons guaranteeing data quality needed in climate studies. This Commission recommended that the World Optical Depth Research and Calibration Centre (WORCC) at  PMOD/WRC Davos be designated the primary WMO Reference Centre for OD measurements as  part of the World Radiation Centre (WRC) activities and adopted Recommendation 6 (CIMO-XIV)*"

The Recommendation 6 reads like this: "*CIMO recommends that the WORCC at PMOD-WRC be recognized as the primary WMO reference center for aerosol optical depth measurements as part of the World Radiation Center facilities*"

The WORCC has defined, designed and built the PFR instruments as the ones that will serve as this standard. The PFR was build based on the specifications that were defined by the official

WMO report entitled: "*WMO/GAW aerosol measurement procedures guidelines and recommendations*" (first in 2003 and second edition in 2016) https://library.wmo.int/doc_num.php?explnum_id=3073. So wavelengths, FOV, and other technical specifications of the PFR are based on such CIMO and WMO recommendations.

The maintenance of the AOD standard by the WMO WORCC Calibration Central laboratory (CCL) is described in Kazadzis et al., 2018.

So, we have included the following information in the Introduction section:

*"The GAW-PFR was specifically designed by WORCC for this goal following the technical specifications defined by WMO (2003; 2016)."*

And the following two references:

WMO: WMO/GAW Aerosol Measurement Procedures: Guidelines and Recommendations, WMO TD No. 1178, GAW Report No 153, 67 pp, https://library.wmo.int/pmb_ged/wmo-td_1178.pdf, 2003.

WMO: WMO/GAW Aerosol Measurement Procedures, Guidelines and Recommendations, 2nd Edition, WMO No 1177, GAW Report No. 227, 93 pp, https://library.wmo.int/doc_num.php?explnum_id=3073, 2016.

We have added the following text in section 3:

*"In case of the GAW-PFR, the calibration system is more complex in order to ensure traceability with the WORCC world reference. The maintenance of the AOD standard by the WORCC Calibration Central laboratory is described in Kazadzis et al. (2018). It consists of a triad of instruments that measure continuously, and three additional portable transfer standard radiometers located at Mauna Loa (one instrument) and Izaña (two instruments) observatories. Every six months, one of the portable transfer standard radiometers visits the reference triad based at PMOD/WRC (Davos) and compares the calibration constants defined by the 6-month Langley calibrations in the two high mountain stations (Table 1 of Kazadzis et al. 2018) with the one defined by the triad. The comparison is based on the signals (voltages) and not on AOD values. The differences between the Izaña GAW-PFR radiometers and the reference triad have been always lower than 0.5%, being within the uncertainty of the Langley method plus the small possible instrument degradations that can be detected in a 6-12 month period. Such degradations are quite small and are accounted for in the calibration analysis since extraterrestrial constants are linearly interpolated between two triad visits or every 6-month periods. Additionally, the Izaña GAW-PFR "field" radiometers are calibrated on a routine basis using the Langley-plot technique for double checking quality assurance. Therefore, these radiometers cannot be considered as simple "field" instruments, but as regularly calibrated radiometers with assured traceability with the WORCC triad reference."*

So, assuming that there is no "better" and "worse" instruments, and since WORCC, defined as the WMO OD standard, has defined the above calibration procedure, we can scientifically state that the 15 reference Cimel radiometers operated at Izana over the 2005-2015 period are traceable to

the WMO standards. Of course, the WORCC "standard' includes uncertainties that are defined by the WMO criteria (U95% limit) described in the text.

Concerning the abstract, it has been rewritten, clarifying the WMO reference and traceability issues, and incorporating the new results that appear in the new version of the paper.

**S.2. Page 2, Lines 28-29: How did you determine these totals? The AERONET web site provides partitioning of sites by equivalent data year and not the actual years (which can be greater). >10 years (data equivalent) is 84 sites (https://aeronet.gsfc.nasa.gov: last accessed 1/8/2019) >5 years (data equivalent) is 242 sites (https://aeronet.gsfc.nasa.gov last accessed 1/8/2019).**

**Authors:**

**Corrected**

**S.3. Page 3: Line 9: Please define "Triad" for PFR; perhaps the mean of three master PFR instruments?**

**Authors:**

**Yes. This is exactly the meaning: three Master GAW-PFR radiometers that make up the world reference. This has been introduced in the text.**

**S.4. Page 3, Line 13: What is a portable transfer standard radiometer? Is it a "reference" PFR?**

**Authors:**

**Please, see reply to S.1.**

**S.5. Page 3, Lines 26-29: Additional reference is needed (e.g., Holben 1998, Eck 1999, Toledano 2018)**

**Authors:**

**Added.**

**S.6. Page 4, Line 6: Change "thanks" to "has to be"**

**Authors:**

**The term "…Thanks to…" has been replaced by "…because…"**

**S.7. Page 4, Line 18: Need to spell-out "IZO" since it starts the sentence.**

**Authors:**

**Done**

**S.8. Page 5, Lines 10-11: Briefly discuss the differences between the two Cimels that affect the optical characteristics (i.e., why is this important?).**

**Authors:**

The technical differences between PFR and Cimel radiometers are remarkable in some aspects such as the use of filters with different FWHM and centred in different wavelengths in some channels (around 380 and 440 nm), their different FOV, or the control or correction of the temperature. All these differences must be taken into account in the final assessment of the AOD comparison between both instruments. A detailed description of the technical characteristics of both AERONET-Cimel and GAW-PFR instruments, and their main differences, are shown in Section 3.

**S.9. Page 5: Line 32: What type of filters? Page 5: Line 33: Need citation and further explanation.**

**Authors:**

**The type of filters are described in Page 5 Lines 27-29:**

*"Dielectric interference filters manufactured by the ion-assisted-deposition technique are used to assure significantly larger stability in comparison to the one manufactured by classic soft-coatings.*

**S.10. Page 6, Line 1: Place period after "position"**

**Authors:**

**Done:** *"position"* **has been replaced by** *"sun-pointing".*

**S.11. Page 6, Lines 26-27: How is the time matching performed between the AERONET Cimel and the PFR? Is the closest PFR value used or an average of the two PFR values?**

**Authors:**

**We have added this new text in order to clarify this issue:**

*"GAW-PFR provides AOD values every 1 minute as an average of 10 sequential measurements of total duration less than 1 second (20 ms for each channel), then dark current is measured, going to the sleep mode until the next minute. AERONET-Cimel takes a sequence of three separate measurements (1-second per filter) in one minute interval (each one every 30 seconds). This sequence of measurements is called "triplet" and it is performed every ~15 minutes for air masses lower than 2, and with higher frequency for lower solar elevations. Therefore, AERONET-Cimel provides AOD values for each triplet, at least, every ~15 minutes. Note that AERONET-Cimel performs AOD measurements interspersed with sky radiance measurements, whose duration varies throughout the day, and therefore the AOD measurements are not necessarily provided at full minutes. We consider the 1-minute data as synchronous when GAW-PFR and AERONET-Cimel AOD data were obtained with a difference of ~30 s."*

**S.12. Page 7, Table 1: What is a GAW-PFR "field instrument?" What other can there be if it is the "world standard?" Also, AERONET-Cimel temperature control is different between AERONET Version 2 and Version 3.**

**Authors:**

"Field instrument" **term has been deleted.**

"Corrections from filter-specific temperature characterization in V3 (Giles et al., 2019)" **has been added in Table 1.**

**S.13. Page 7, Lines 9-10: The results of the present study should be using AERONET Version 3, which has been available since early 2018 as discussed in Giles et al 2018.**

**Authors:**

**AERONET Version 3 has also been evaluated. See the new results in different Sections and in the Supplement. See reply to G-8.**

**S.14. Page 8, Line 1-4: Most instruments should be collecting data every 3 minutes. Why was the Cimel instrument at IZO collecting every 15 minutes? Why is 30 second difference used for time matching (the farthest away the PFR can be from the Cimel measurement? Should not the PFR be within say 10 seconds of the Cimel measurement to be matched?**

**Authors:**

**See reply to S.11.**

**S.15. Page 8, Line 19: AERONET data provides wavelength pairs and also computation of 440-870nm using all of the wavelengths in the range.**

**Authors:**

**Yes. It is said in the text, but this computation does not include 380 nm. This is the reason why we calculated Cimel AE with the four common wavelengths to GAW-PFR (380, 440, 500 and 870nm).**

**S.16. Page 9, Lines 1-31: The paragraphs are fragmented and they should be reorganization. Please revise and condense.**

**Authors:**

**This section has been shortened and reorganized.**

**S.17. Page 9, Lines 10-12: As a result, it seems the PFR data from 2005-2010 should not be used in the study?**

**Authors:**

**We don't agree because:**

1) **The very small percentage of data with deficient sun-pointing was identified and consequently removed from the official GAW-PFR database.**

2) **The high percentage of AERONET-Cimel AOD data meeting the WMO criteria for the four compared channels for the entire period 2005-2015 (below) can really raise this question?**
**92.7% (380nm), 95.7% (440nm), 95.8% (500nm), 98.0% (870nm) (Table 4)**

**S.18. Page 10, Figure 1 Caption, Line 3: Please discuss here and in the text providing at least one specific case to analyze what is causing the large outliers.**

Authors:

**The causes of the outliers are discussed in new Section 5.3.3. Some case analyses have been included and shown as Supplement S8**

**S.19. Page 10, Table 3: The "Original GAW-PFR channel (%)" column is not correct if since you are expecting the AERONET values at different wavelengths to meet the AOD at PFR wavelengths. This column should be removed unless you interpolate AERONET AOD to PFR wavelengths.**

Authors:

**This information has been removed. Table 3 has been modified in order to show the percentage of both V2 and V3 AERONET AOD data meeting the WMO criteria.**

**S.20. Page 11, Table 4: What are the number of measurements for each cell used to compute the "traceability?"**

Authors:

**The percentage and number of measurements (in brackets) are the following:**

**For AERONET V2:**

|  | 380nm | 440nm | 500nm | 870nm |
|---|---|---|---|---|
| AOD<=0.05 | 94.4 (57008) | 96.8 (59130) | 97.0 (58572) | 98.5 (60191) |
| 0.05<AOD<=0.10 | 91.0 (4723) | 93.1 (4850) | 92.8 (4817) | 94.2 (4908) |
| AOD>0.10 | 75.0 (3938) | 86.5 (4615) | 85.1 (4466) | 95.9 (5118) |
| AE<=0.25 | 73.1 (2145) | 82.3 (2417) | 80.1 (2351) | 96.2 (2824) |
| 0.25<AE<=0.6 | 91.2 (5407) | 96.2 (5810) | 96.0 (5691) | 97.9 (5911) |
| AE>0.6 | 94.6 (55114) | 96.9 (57089) | 97.0 (56504) | 98.7 (58146) |
| Total 2005-2015 | 92.7 (65669) | 95.7 (68595) | 95.8 (67855) | 98.0 (70217) |
| Total 2010-2015 | 93.5 (41977) | 97.4 (43745) | 97.2 (43627) | 99.1 (44498) |

**For AERONET V3:**

|  | 380nm | 440nm | 500nm | 870nm |
|---|---|---|---|---|

| | | | | |
|---|---|---|---|---|
| AOD<=0.05 | 93.6 (60264) | 96.3 (62836) | 97.1 (62545) | 98.4 (64213) |
| 0.05<AOD<=0.10 | 91.0 (5138) | 92.0 (5217) | 92.6 (5222) | 94.7 (5372) |
| AOD>0.10 | 77.1 (4085) | 84.1 (4537) | 81.6 (4326) | 93.3 (5034) |
| AE<=0.25 | 78.7 (2472) | 82.3 (2588) | 79.0 (2483) | 92.9 (2921) |
| 0.25<AE<=0.6 | 90.2 (5941) | 94.3 (6321) | 94.9 (6255) | 97.4 (6530) |
| AE>0.6 | 94.1 (56952) | 96.5 (59181) | 97.1 (58793) | 98.7 (60514) |
| Total 2005-2015 | 92.3 (69487) | 95.2 (72590) | 95.7 (72093) | 97.8 (74619) |
| Total 2010-2015 | 92.8 (42463) | 96.8 (44328) | 96.8 (44329) | 98.8 (45212) |

Notice that the AE thresholds have been slightly modified to be consistent with new thresholds in section 5.5.

**S.21. Page 11, Line 15: Data-pairs should be 468 for AOD500nm as stated in Kazadzis et al (2014) plots.**

**Authors:**

**Corrected**

**S.22. Page 11, Line 16: Which instrument had poor calibration in the 500nm channel during the 9-day analysis period? Were both instruments field PFR and field Cimel instrument and not reference instruments or triad? It is important to draw this distinction in relation to the present study.**

**Authors:**

**This sentence has been removed because we were speculating about the results of another study.**

**S.23. Page 11, Line 21: Change "among" to "between"**

**Authors:**

**Done.**

**S.24. Page 12, Lines 3-5: It should be made clear that "non-traceability" is referring to PFR instrument and not the AERONET Cimel reference.**

**Authors:  That is clearly stated.**

**S.25. Page 12, Lines 9-13: More analysis and interpretation of Table 5 is needed with respect to the statistics presented in Table 5.**

**Authors:  The statistics of Table 5 have been used to compare with statistics from previous intercomparison studies, and this is done in the next paragraph (former lines 16-22).**

**S.26. Page 12, Line 20: Better to use "reference" instead of "Master**

**" throughout manuscript.**

**Authors:**

**Done.**

**S.27. Page 13, Lines 8-33: The paragraphs are fragmented and they should be reorganized. Please revise and condense paragraphs.**

**Authors:**

**Done. The text has been shortened and reorganized.**

**S. 28. Page 13, Lines 8-11: AERONET reference instrument should obtain Langley calibration coefficients every 3 to 8 months. (e.g., Giles et al., 2019). Please check with AERONET calibration center on the calibration interval.**

**Authors:**

**The Referee should take into account that many co-authors of this paper are closely involved in AERONET calibration activities and specifically in AERONET-Europe/ACTRIS calibration and quality assurance responsibilities. The calibrations performed by us are directly incorporated into the NASA AERONET calibration system.**

**According to Giles et al. (2019): "The reference instruments obtain a calibration based on the Langley method morning-only analyses based on typically 4 to 20 days of data performed at a mountaintop calibration site. The primary mountaintop calibration sites in AERONET are located at Mauna Loa Observatory (latitude 19.536, longitude -155.576, 3402 m) on the island of Hawaii and Izana Observatory (latitude 28.309, longitude -16.499, 2401 m) on the island of Tenerife in the Canary Islands (Toledano et al., 2018). These reference instruments are routinely monitored for stability and typically recalibrated every 3 to 8 months".**

**Anyway, we require a minimum of 10 high quality Langley plots, independently of the number of days needed for that.**

**Note, that during the second period, between 2010 and 2015, when the Cimel # 244 has been a permanent reference instrument, it was calibrated 21 times with Langley calibrations, so the average frequency is somewhat higher than 3 calibrations per year (3.5 calibrations / year).**

**S.29. Page 13, Lines 24-27; Page 14, Figure 2 Caption: Is the PFR AOD500nm used for the limitation "AOD-500nm <= 0.03" shown in Figure 2? The outliers in Figure 2 appear to be independent of air mass. Are these due to the PFR or Cimel?**

**Authors:**

**AOD-500nm ≤0.03 corresponds to GAW-PFR.**

**In former page 13, lines 30-31 is said:** "There is no dependence on 1-minute AOD differences with optical air mass for 440, 500 and 870 nm, and a slight dependence for 380 nm (Table 6) with lower traceability at lower optical air masses."

**The outliers are mainly due to Cimel as is explained with Figure 4. We have included some case analysis, as the Referee suggests, in Supplement S8 and S9.**

**S.30. Page 13, Lines 30-31: Uncertainty of field PFR (0.01) and uncertainty of reference AERONET-Cimel (0.005) at 380nm are maximum at low optical air mass and therefore the agreement between the two instruments will be inherently lower.**

**Authors:**

**The uncertainty of field PFR is not 0.01 as the Referee can understand from reply to S1.**

**In former page 15, Lines 10-13 it is said:**

*"However, for 380 nm the percentage of non-traceable values increases by 1.4 % from $1 \leq m < 2$ to $4 \leq m < 5$. This result is consistent with the fact that the greatest uncertainty in the determination of the calibration constants is observed in the UV range, and the lowest uncertainty in the near-infrared channel (Toledano et al., 2018). We also have to consider that other AOD retrieval input uncertainties are air mass dependent, such as Rayleigh related optical thickness."*

**This and other arguments discussed in this section might be considered by the Referee.**

**S.31. Page 14, Figure 2: Please change in the y-axis ticks to show zero. What is the optical air mass limitation for PFR measurements? The AERONET Version 3 processing is available up to optical air mass of up to 7 and AERONET Version 2 is up to 5.**

**Authors:  The y-axis ticks in figures 2 have been changed to show zero.**

[Figure]

There is no optical air mass limitation for PFR measurements (m >7).

We have also compared GAW-PFR with AERONET V3 for optical air mass >5 showing quite small differences with the corresponding comparison of V2. (See Supplement S4).

**S.32. Page 14, Table 6: What are the number of matched measurements in each optical air mass interval?**

Authors:

The number of matched measurements in each optical air mass interval are:

For AERONET V2:

| Total | 1 ≤ m <2 | 2 ≤ m <3 | 3 ≤ m <4 | 4 ≤ m <5 | |
|---|---|---|---|---|---|
| 49326 | 21347 | 12947 | 10995 | 4036 | |

For AERONET V3:

| Total | 1 ≤ m <2 | 2 ≤ m <3 | 3 ≤ m <4 | 4 ≤ m <5 | 5 ≤m <6 |
|---|---|---|---|---|---|
| 52299 | 20257 | 12028 | 9975 | 3775 | 3572 |

**S. 33. Page 15: Line 12: Also, include reference to Eck et al. 1999.**

Authors:

**Added.**

**S. 34. Page 15: Lines 29-32: AERONET-Cimel reference Langley is performed more frequently than once a year?**

**Authors:  Yes. It is. Please, see reply to S.28.**

**S. 35. Page 16, Figure 3: It is difficult to quantify the relative significant difference between the Cimel and PFR in the logarithmic scale? Can you show a plot of the relative difference between the two instruments? Some significant variability in the differences exist and it could be due to differences in cloud screening, for example.**

**Authors:**

**We have plotted relative differences for both V2 and V3. See new Figure 3 and Supplement S6.**

**Yes. Some differences are attributable to different cloud screening as discussed in new Section 5.3.4, and shown in detail in the case studies shown in Supplement S10-S12.**

[Figure]

V2                                                 V3

**S.36. Page 16, Lines 3-12; Page 17 Lines 20-22: Authors should also utilize AERONET Version 3 with improved cloud screening techniques (Giles et al., 2019). It is not clear why the authors do not investigate AERONET Version 3 for this study since these data are available.**

**Authors:  We have also compared GAW-PFR with AERONET V3 in the reviewed paper.  All results of the GAW-PFR comparison with AERONET-Cimel made for V2 have been replicated for V3: Figures 1 and 3, and Tables 3 and 8 in the manuscript, and S1-S14, and S17-S18 in the Supplement material. However, note that the results are quite similar to those using V2.**

**S.37. Page 17, Lines 10-22: The "traceability" here is difficult to interpret since the instruments do not use the same cloud screening which is evidenced by the lower "T(%)" numbers in the Table 7. More importantly, perhaps, are the percentage numbers in the column with FCS ranges in Table 7; however, although the solar radiation data may indicate a cloud, the sun photometers have the ability to find gaps in the clouds to perform measurements.**

**Authors:** Precisely the impact on non-traceability of AOD by the use of different cloud-screening is what we wish to assess. However, we admit that there are two problems in this study on this issue:

1) Izaña is not an ideal site to compare cloud-screening algorithms given the very low cloud frequency.

2) The tools we use, which are the only ones available automatically, have quite a few limitations.

Please consider what we say in the following three paragraphs addressing these comments:

*"However, the real impact of clouds on AOD traceability at IZO is very low due to its special characteristics of high mountain station with very little cloudiness. Therefore, in practice, the possible impact of clouds on the non-traceability of AOD data-pairs is insignificant at the IZO. GAW-PFR and AERONET-Cimel cloud screening algorithms provide successful identification on clear direct-sun conditions during cloudy skies (FCS < 40 %) for99.75 % of the cases, excluding those with very thin clouds."*

**S.38. Page 17, Lines 29-35: A discussion of the COT effect on apparent optical depth due to different instrument FOV should be discussed (Kinne et al., 1997).**

**Authors:** We agree. We have included the following text in Supplement S11:

*A second type of clouds that cause problems in AOD retrieval are the cirrus clouds, usually being present at Izaña between January and April, associated with the presence of the subtropical jet that is normally found in the vicinity of the Canary Islands at this time of year (Rodríguez-Franco and Cuevas, 2013).A constant cloud optical thickness (COT) corresponding to a cloud of a certain horizontal extension would cause the successive measurements within a minute to correspond to the same cloud stage, and therefore it would not be discernible from the extinction caused by aerosols. In the case of very thin cirrus clouds, AOD could increase up to 0.03 (Chew et al., 2011; Giannakaki et al., 2007) with small fluctuations, that cloud-screening algorithms could interpret as the presence of an aerosol layer. Huang et al. (2012) evaluated the impact on AERONET level 2.0 AOD retrievals from cirrus contamination highlighting the difficulties to remove completely their signature, mainly from those subvisual thin cirrus. According to Kinne et al. (1997), optical depth estimates from cirrus derived with sunphotometers have to include forward-scattering effects. Their results show that for cirrus, and instruments with 2.0° and 2.4° FOV, the correction factors vary between 1.6 and 2.5 depending on the crystal size. Taking into account that the FOV of the GAW-PFR is 2.5°, while that of the AERONET-Cimel is 1.3°, such cases will affect the comparison results.*

**S.39. Page 18: Table 7: Define "FCS" in the caption. What are the number of measurements in each interval?**

**Authors:**

**"FCS" has been defined in the caption.**

**The number of measurements used in each interval for each AERONET version were the following:**

1) **For AERONET V2**

| FCS interval | Number of data |
|---|---|
| 0%<=FCS<20% | 23 |
| 20%<=FCS<40% | 146 |
| 40%<=FCS<60% | 713 |
| 60%<=FCS<80% | 4703 |
| FCS>=80% | 60648 |
| TOTAL FCS | 66233 |

**Number of traceable data, non-traceable data, and total data for each FCS interval and channel.**

| 380nm | | | 440nm | | | 500nm | | | 870nm | | |
|---|---|---|---|---|---|---|---|---|---|---|---|
| Tr | NTr | TT | Tr | NTr | TT | Tr | NTr | TT | Tr | NTr | TT |
| 10 | 11 | 21 | 10 | 13 | 23 | 10 | 11 | 21 | 20 | 3 | 23 |
| 97 | 43 | 140 | 107 | 39 | 146 | 103 | 37 | 140 | 126 | 20 | 146 |
| 551 | 146 | 697 | 625 | 87 | 712 | 619 | 78 | 697 | 654 | 58 | 712 |
| 4112 | 541 | 4653 | 4410 | 288 | 4698 | 4346 | 307 | 4653 | 4593 | 105 | 4698 |
| 55735 | 3994 | 59729 | 58079 | 2310 | 60389 | 57479 | 2245 | 59724 | 59399 | 1005 | 60404 |

2) **For AERONET V3**

| FCS interval | Number of data |
|---|---|
| 0%≤FCS<20% | 27 |
| 20%≤FCS<40% | 152 |
| 40%≤FCS<60% | 764 |
| 60%≤FCS<80% | 5021 |
| FCS≥80% | 64097 |
| TOTAL FCS | 70061 |

**Number of traceable data, non-traceable data, and total data for each FCS interval and channel.**

| 380nm | | | 440nm | | | 500nm | | | 870nm | | |
|---|---|---|---|---|---|---|---|---|---|---|---|
| Tr | NTr | TT | Tr | NTr | TT | Tr | NTr | TT | Tr | NTr | TT |
| 12 | 15 | 27 | 12 | 15 | 27 | 12 | 15 | 27 | 25 | 2 | 27 |
| 111 | 34 | 145 | 125 | 27 | 152 | 118 | 28 | 146 | 143 | 9 | 152 |
| 576 | 167 | 743 | 647 | 116 | 763 | 649 | 95 | 744 | 702 | 61 | 763 |
| 4433 | 517 | 4950 | 4709 | 306 | 5015 | 4677 | 276 | 4953 | 4895 | 120 | 5015 |
| 58407 | 4542 | 62949 | 60974 | 2793 | 63767 | 60535 | 2454 | 62989 | 62599 | 1187 | 63786 |

**Note that the number of matching data in V3 is higher than in V2 because V3 provides AD for m>5.**

**S.40. Page 19, Lines 22-25: Is this the PFR instrument with erroneous pressure reading in late 2014 at shown in Figure 4? Are the values in Table 4 computed without the malfunctioned barometer?**

**Authors:**

**All data with have been used in Table 4, including those affected by erroneous pressure readings.**

**We have added the following sentence in the manuscript:**

*"However, it should be noted that only 99 AOD data pairs have been registered for which the pressure difference between PFR and Cimel is greater than 20hPa at 870nm and 440nm, and only one AOD data pair at 500nm and 380nm channels."*

**S.41. Page 20, Figure 4: What do you mean by 1-minute pressure data for AERONET-Cimel? Did Cimel have a pressure sensor in 2005? Please clarify.**

**Authors:** **This is a mistake. The Cimel does not have a pressure sensor. Caption of Figure 4 reads now as follows:**

*"Figure 4. (a) 1-minute pressure data (hPa) from GAW-PFR and 6-hour pressure data at Izaña Observatory altitude from the National Centers for Environmental Prediction (NCEP) and the National Center for Atmospheric Research (NCAR) reanalysis for the case of AERONET-Cimel, and (b) corresponding 1-minute ΔτR caused by pressure differences in the period 2005-2015."*

**S.42. Page 21, Lines 7-8: However, OMI O3 data are problematic due to sampling issues (McPeters et al, 2015). Page 21, Lines 14-17: Are the discrepancies when using the OMI O3 for PFR?**

**Authors:**

**The procedure used by GAW-PFR to obtain total O3 is partly described in Kazadzis et al., 2018. However, more detailed explanation specific for Izaña Observatory has been added in the text as follows:**

*"In the case of Izaña, if the OMI overpass fails, GAW-PFR uses the Brewer O3 climatology."*

**Total column ozone values are mainly needed to correct optical depth at 500 nm for ozone absorption. As the absorption coefficient at 500 nm is low, total ozone needs to be known to ±30 Dobson units or 10% of typical values, for an uncertainty of ±0.001 optical depths at 500 nm. The above procedure ensures the achievement of the uncertainty of ±0.001.**

**S.43. Page 22, Figure 5: Need to state GAW-PFR uses OMI O3 and AERONET-Cimel uses climatological monthly average of TOMS O3.**

**Authors:**

**Done. Caption of Figure 5 reads now as follows:**

*"Figure 5. (a) Total O3 used by GAW-PFR (measured Brewer O3 values from IZO, OMI O3 overpass or Brewer O3 climatology) and AERONET-Cimel (TOMS O3 climatology), and (b) ΔτO3 (λ) caused by differences in daily total O3 between the two instruments in the period 2005-2015."*

**S. 44. Page 23, Figure 6: What is NO2 "annual course" in caption?**

Authors:

**"Annual course" has been removed.**

**S.45. Page 24, Table 8: What do these data represent? What is the date period? Please provide the total number of match measurements for each wavelength.**

Authors:

**For sake of clarity we have changed the Caption of Table 8 (New table 9) as follows:**

*"Table 9. Percentage (%) of additional traceable AERONET-Cimel AOD 1-minute data (V2 and V3) after correcting by pressure, and total column O3 and NO2 for the period 2005-2015."*

**And the text has been rewritten as follows:**

*"This percentage is maximum at 380 nm with 1.3% (V2) and 1.7% (V3) of the whole dataset."*

**S.46. Page 24, Lines 8-11: Where is it shown that 25% of the data outside U95 are from P, O3, and NO2?**

Authors:

**See reply to S.45.**

**S.47. Page 24, Lines 11-13: What corrections were applied and to which instrument?**

Authors:

**See reply to S.45.**

**S.48. Page 24, Lines 19-21: The AOD 870nm is only affected by Rayleigh component and therefore has the highest agreement as well as the lowest midday uncertainty of the four wavelengths analyzed.**

Authors:

**We agree. We have rephrased the corresponding sentence as follows:**

*"The 870 nm channel is only affected by the Rayleigh correction component and therefore the increment of traceable data after the mentioned corrections is minimal."*

**S.49. Page 25, Figure 7: Are these data now all PFR (or AERONET-Cimel) corrected values to remove anomlies due to Rayleigh (due to pressure sensor issues) and trace gas corrections? I notice now that 2010-2015 is only presented. What are the total number of measurements in these plots? The limit of AOD>0.1 seems arbitrary as a threshold to use a fit of the non-traceable values. Note that many points appear to be very close to the "traceability" boundary limit. Also,**

black "traceable" points have larger difference than red "untraceable" points with a smaller difference (somewhat counterintuitive). The reason (e.g., air mass dependence impact on the traceability criterion) for this affect should be stated in the caption for clarity.

**Authors:**

**In the new Figure 7 we have included data for the entire period 2005-2015 and for AOD>0.1 since we are interested in this section in relatively high dust conditions. According to Basart et al. (2009) AOD> 0.1 corresponds to representative situations of almost-pure dust. The total number of measurements used in the plots have been included in the legend. We have incorporated the suggestions of the Referee in the new Figure 7 caption as follows:**

*"Figure 7. Actual AOD differences between AERONET-Cimel V2 and GAW-PFR vs AODPFR at (a) 380 nm (b) and 500 nm for the period 2005-2015. The fitting line has been calculated with those data points with AOD > 0.1 and Cimel-PFR AOD difference > 0. The number of data used in the plots are indicated in the legend. The percentage of non-traceable AOD data with these conditions is ~24% for 380 nm, and ~8% for 500 nm. Note that some traceable (black) points show larger AOD differences than non-traceable (red) points because of the air mass dependence of the WMO traceability criterion."*

**These are the new Figures:**

[Figure]

Equivalent figures are shown in Supplement S17 for AERONET V3.

**S.50. Page 25, Table 9: Provide the number of measurements for each wavelength and AOD range.**

**Authors:**

**The number of measurements for each wavelength and AOD range, and AERONET version are the following:**

**For AERONET V2:**

|         | TT    | TT_A>0.1 | TT_A>0.2 | TT_A>0.3 |
|---------|-------|----------|----------|----------|
| 380nm   | 70838 | 5250     | 1845     | 620      |
| 440nm   | 71645 | 5336     | 1886     | 620      |
| 500nm   | 70833 | 5250     | 1845     | 620      |
| 870nm   | 71660 | 5336     | 1886     | 620      |

For AERONET V3:

|         | TT    | TT_A>0.1 | TT_A>0.2 | TT_A>0.3 |
|---------|-------|----------|----------|----------|
| 380nm   | 75303 | 5300     | 1956     | 612      |
| 440nm   | 76290 | 5395     | 2003     | 612      |
| 500nm   | 75335 | 5299     | 1956     | 612      |
| 870nm   | 76307 | 5395     | 2003     | 612      |

**S.51. Page 26, Line 12-13: Do you vary the surface albedo spectrally?**

**Authors:**

**Note we have completely rewritten this Section. Please see new Section 5.4 "GAW PFR and AERONET-Cimel comparison under high AOD conditions: the impact of dust forward scattering for different FOVs". We have replaced the analysis performed by SMARTS RTM with a more accurate analysis using Monte Carlo RTM.**

**S.52. Page 26, Lines 20-24: AERONET-Cimel and PFR are simulated? Please state.**

*Authors:*

**S.53. Page 27, Figure 8: What are the outliers between 1% and 2% PFR Circumsolar/Direct radiation in (a) and (b)? I cannot see the blue dots very well on panels (c) and (d); perhaps you can include on another plot? Please state that these results are simulated from the radiometers.**

Authors:  **Please, see reply to S51.**

**S.54. Page 28, Lines 17-22: Include the analyzed wavelength range for AE in the text, Figure 9, and Figure 9 caption.**

*Authors:*

**The introduction of this Section has been completed clarifying some aspects:**

"We have performed a comparison of the AE provided by GAW-PFR and AERONET-Cimel using in both cases the AOD data obtained from the four common channels (380 nm, 440 nm, 500 nm and

870 nm) with a total of 70716 data-pairs. The PFR-AOD values have been ordered from lowest to highest by grouping them in intervals of 500 values for which the averages (and corresponding standard deviations) of the PFR-Cimel AE differences have been calculated (Figure 10a). In a similar way we proceeded with the PFR-AE values (Figure 10b)."

**S.55. Page 29, Table 10: Provide additional context on how these values were determined in the text.**

Authors:

**We have rewritten this paragraph as follows_**
*"Following the methodology shown by Wagner and Silva (2008), the AE uncertainty estimations have been calculated using AOD measurements at four wavelengths and AOD uncertainty error propagation (Table 11)."*

**S.56. Page 29, Lines 21-25: Please provide more background on how these values were determined. For example, it is not clear why AE is provided differently to AE PFR and AE Cimel in the number list.**

Authors:

**The new threshold values of AE are referenced (Cuevas et al., 2015 and Berjón et al, 2019).**

**The new text clarifies the $AE_{PFR}$ and $AE_{Cimel}$. It reads as follows:**

*"Considering the AE criteria established by Cuevas et al. (2015) and Berjón et al. (2019), we have identified the following four main categories according to the $AE_{PFR}$ and $AE_{Cimel}$ values:*

*1.    $AE_{PFR}$ & $AE_{Cimel}$ > 0.6: Pristine conditions.*

*2.    0.25 < $AE_{PFR}$ & $AE_{Cimel}$ ≤ 0.6: Hazy, mineral dust being the main aerosol component.*

*3.    $AE_{PFR}$ & $AE_{Cimel}$ ≤ 0.25: Pure dust.*

*4.    $AE_{PFR}$ and $AE_{Cimel}$ do not fit any of the previous categories.*

*In 94.9 % of the cases, GAW-PFR and AERONET-Cimel V2 match the AE intervals of each aerosol scenario. Similar results (93.4%) were obtained when comparing with AERONET V3. Most of the agreement (>80 %) occurs in the predominant scenario of pristine conditions despite the AE uncertainty under pristine conditions being ≥ 1. See Supplement S18 for more details. Notice that given the special characteristics of the Izaña Observatory, and according to Cuevas et al. (2015) and Berjón et al. (2019), AE is a self-sufficient parameter to define different types of aerosol scenarios without the need to combine its information with AOD."*

**The following Figures have been included in Supplement S18:**

[Figure]

For V2        For V3

**S.57. Page 30, Figure 9: Please correct formatting problem with x-axis label of panel (a). Legend state "500-data intervals and standard deviations"; what does this mean? How do the plots change with "relative" difference since all differences are taken as absolute value in this plot? Does the relative difference show any trend in AE for Cimel lower than PFR or vice versa?**

**Authors:**

**The X-label panel a) formatting problem has been fixed.**

**The legend has been removed. It is now explained in the text (see Reply to S.54)**

**The new caption of former figure 9 (new Figure 10) reads as follows:**

*"Figure 10. (a) PFR-Cimel AE mean absolute differences (and corresponding standard deviations) vs PFR mean AOD500nm in 500 data intervals (b) and vs PFR mean AE in 500 data intervals. AE has been computed for both PFR and Cimel using the four common channels (380, 440, 500 and 870nm)."*

**The Referee can see below how the plots change with "relative" differences.**

**No significant trends are observed in any of the plots in the relative differences of AE, although a few points of higher Cimel AE (or lower PFR AE) are observed for AE <0.5, and for AOD values around 0.05 and between 0.4 and 0.5 .**

[Figure]

[Figure]

**S.58. Page 31, Lines 3-4: This comment needs to be substantiated.**

**Authors:**

*We agree. The sentence reads now as follows:*

 *"Notice that given the special characteristics of the Izaña Observatory, and according to Cuevas et al. (2015) and Berjón et al. (2019), AE is a self-sufficient parameter to define different types of aerosol scenarios without the need to combine its information with AOD."*

**S.59. Page 31, Line 26: "traceability" is a confusing term and should be changed to agreement or something similar.**

**Authors:**

**In this case, we think the clarification on the term** *"traceability"* **is adequate**

**S.60. Page 31, Lines 17-18: What is the "concern?"**

**Authors:**

**We agree. This sentence has been rephrased as follows:**

 *"These facts led us to investigate the homogeneity of the AERONET-Cimel AOD data series and their intercomparability with the much more homogeneous AOD data series from GAW-PFR (3 instruments in 11 years)."*

**S.61. Page 31, Lines 21-26: What are these "limits?"**

**Authors:**

**U95 limits. It has been included in the text.**

**S.62. Page 31, Line 30: Fragmented sentence.**

**Authors:**

**Corrected**

**S.63. Page 32, Lines 8-10: The fact that 380nm channel is more divergent is not a new finding since it is known to have higher uncertainty than other channels. Please state relevant citations.**

**Authors:**

**We agree. The higher uncertainty of the 380 nm channel, and corresponding references, has been highlighted in the "Calibration related errors" Section (Eck et al., 1999; Jarosyawski et al. 2003; Toledano et al., 2018).**

**S.64. Page 32, Lines 18-20: Also, optically thin cirrus clouds can also produce a difference in the measured values (Chew et al. 2011, Huang et al., 2011).**

**Authors:**

**Included Chew et al. (2011) and Huang et al. (2012) in former Section 5.2.2 (moved to Supplement S11.**

**S.65. Page 32, Lines 23-28: Which instrument? PFR or Cimel?**

**Authors:**

**PFR is the only radiometer using barometer. We have clarified this sentence.**

**S.66. Page 32, Lines 30-33: What causes large change in O3 concentration at Izana high altitude site?**

**Authors:**

**This paragraph has been included in former Section 5.2.3 (Differences in $O_3$ absorption)**

*"Total $O_3$ over IZO shows a relatively small amplitude throughout the year, but both surface ozone concentrations and column ozone amount could sharply increase under the influence of cut-off lows injecting air from the high-mid troposphere into the lower subtropical troposphere, which is not uncommon in spring and the first half of summer (Cuevas et al., 2015; Kentarchos et al., 2000). In addition through exchange processes in the Upper Troposphere Lower Stratosphere (UTLS) due to the presence of the subtropical jet (mainly from February to April) (Rodríguez-Franco and Cuevas, 2013)."*

*Table 1. Main features of the GAW-PFR (PFR (Wehrli, 2000, 2005, 2008a, b) and AERONET-Cimel (Holben et al., 1994, 1998; Torres et al., 2013) -radiometers used in this study.*

|  | GAW-PFR | AERONET-Cimel |
|---|---|---|
| Type of instrument | Standard version
 Field instrument | Standard version
 MasterReference instrument |
| Type of observation | Automatic continuous direct sun irradiance | Automatic sun-sky tracking |
| Available standard channels | 368, 412, 500, 862 nm | 340, 380, 440, 500, 675, 870, 1020,1640 nm |
| FWHM | 5 nm | 2 nm (340 nm), 4 nm (380 nm), 10 nm(VIS-NIR), 25nm(1640nm) |
| AOD uncertainty | ± 0.01 | ± 0.005 (MasterReference instruments) |
| FOV (FWHM) | 2.5°
 (1.2° plateau, 0.7° slope angle) | 1.21.3° (slope angle unknown) |
| Sun tracker | No specificAny Sun trackersun tracker with a resolution of at least 0.08° | Sun tracker Robot specifically designed by Cimel and controlled in conjunction with the radiometer |
| Temperature control | Temperature controlled 20°C ± 0.5°C | Temperature correction is applied in V2. Corrections from filter-specific temperature characterization in V3 (Giles et al., 2019) |
| Power | Grid | Solar panels/grid |

| Data transmission | Local PC / FTP  | Local PC / FTP
 Satellite transmission
  |
|---|---|---|
| Calibration | Comparison with reference triad. Additional in situ long-term Langleys  | At least 10 good morning  Langleys plots |

*4.2. Cloud filtering.*

The data matching in our comparison analysis was performed with synchronous 1-minute AOD values of both networks labelled with quality control (QC) flags that guarantee proven quality data not affected by the presence of clouds.  In the case of the AERONET-Cimel network, the selected AOD data are Level 2 data from both V2 and V3 AERONET databases, which have been cloud filtered by the Smirnov algorithm (Smirnov et al., 2000), based on the triplet method, with a second-order temporal derivative constraint (McArthur et al., 2003), and visually screened in V2. The cloud screening in AERONET V3 has been completely automated, and notably improved, especially by refining the triplet variability and cirrus cloud detection and removal (Giles et al., 2019). GAW-PFR cloud screening algorithms also use the Smirnov triplet measurement, and the second-order derivative check, but add a test for optically thick clouds with $AOD_{500nm} > 2$ (Kazadzis et al., 2018b). In the case of the GAW-PFR network (Wehrli, 2008a) the flags take the value 0 (cloudless conditions, no wavelength crossings and sun pointing within certain limits, more details in Kazadzis et al. (2018a)) for all those selected records.

*Table 2. GAW-PFR and AERONET-Cimel instrument numbers used in this study in the period 2005-2015. Data from Reference Cimel #398 was not upgraded to Level 2 in V3 during the period 12 July 2008 - 15 September 2008.*

| Instruments used in this study | Period 2005-2009 | Period 2010-2015 |
|---|---|---|
| GAW-PFR | 2 instruments: #6,#25 | 2 instruments: #6,#21 |
| AERONET-Cimel | 13 instruments: #25,#44,#45,#79,#117,#140 | 5 instruments: #244, #347, #380 |
| | #244,#245,#380,#382,#383,#398,#421 | #421, #548 |

~~GAW-PFR provides AOD values every 1 minute as an average of 10 sequential measurements of total duration less than 1 second, while AERONET-Cimel provides AOD values every 15 minutes (from 3 measurements separated by 30 seconds). We consider synchronous 1-minute data when GAW-PFR and AERONET-Cimel AOD data were obtained with a difference of ~ 30 s.~~

**4.3. WMO traceability criteria.**

The criterion for traceability used in this study follows the recommendation of the WMO (WMO, 2005) which states that 95% of the AOD measurements fall within the specified acceptance limits, taking the  PFR as a reference:

$$U_{95} = \pm(0.005 + 0.010/m) \tag{2}$$

where $m$ is the optical air mass. Note that the U95 range is larger for smaller optical mass.

The acceptance limits proposed by WMO take into account, on the one hand, the uncertainty inherent in the calculations of the AOD, and on the other hand, the uncertainty associated with the calibration of the instrument. The latter, for the case of instruments with finite field of view direct transmissions, such as the PFR and the Cimel, is dominated by the influence of the top-of-the-atmosphere signal determined by Langley plot measurements, divided by the optical air mass.

The first term of Eq. 2 (0.005) represents the maximum  tolerance for the uncertainty due to the atmospheric parameters used for the AOD calculation (additional atmospheric trace gas corrections, and Rayleigh scattering). The second term describes the calibration related relative uncertainties . The WMO recommends an upper limit for the calibration uncertainty of 1%.

~~A first simple approach to calculate the circumsolar radiation of each radiometer taking into account their respective FOVs, AOD, total O₃ and pressure values, has been performed with the SMARTS (Simple Model of the Atmospheric Radiative Transfer of Sunshine) model version 2.9.5 (Gueymard, 1995). This spectral model, that covers the UVA, UVB, Visible and Near-Infrared bands, can be used to simulate the spectral irradiance that would be measured by a spectroradiometer (Gueymard, 2001). This model, which has been used and compared with LibRadtran for determining circumsolar radiation (Eissa et al., 2015) is used in~~

this study to estimate, in a first approximation, the differences in AOD caused by the different FOV of PFR and Cimel radiometers.

**4.4. Modelling the impact of near-forward scattering on the AOD measured by the PFR and Cimel radiometers**

In order to study the impact of near-forward scattering on the irradiance measured by the PFR and CimelIMEL instruments, a forward Monte Carlo model (Barker 1992, Barker 1996, Räisänen et al. 2003) was employed. For the present work, the model was updated to account for the finite width of the solar disk. The starting point of each photon was selected randomly within the solar disk, assuming a disk half-width of 0.267° and the impact of limb darkening on the intensity distribution was included following Böhm-Vitense (1989). Some diagnostics were also added to keep track of the distribution of downwelling photons at the surface with respect to the angular distance from the centre of the sun. Gaseous absorption was accounted for following Freidenreich and Ramaswamy (1999), while the Rayleigh scattering optical depth was computed using Bodhaine et al. (1999) below..

**5. Results.**

**5.1. CPreliminary comparison of long-term AERONET V2 and V3 data basesdatasets at Izaña site.**

Since V3 has been released recently (Giles et al., 2019), we present a comparison between V2 and V3 for the Cimel channels 380 nm, 440 nm, 500 nm and 870 nm for the period 2005-2015.

The results indicate that for the Izaña site the agreement and consistency between the two AERONET versions is very high for the four channels ($R^2$>0.999) in full agreement with the results of the V2-V3 comparison reported by Giles et al. (2019). So, we can advanceIt follows that the results of the comparison between GAW-PFR and the two versions of AERONET are very similar as shown throughout this work. A detailed description ofn AERONET V3 and its improvements with respect to V2 is given in Giles et al. (2019). A detailed description of the improvements introduced in V3 are given in Giles et al. (2019). As such Logically, these improvements depend on aerosol type, are not homogeneous in terms of theirfor the different types of aerosols and their nature and variable impact at a global level. A according to the changes introduced in V3, for thea high mountain site such as Izaña characterized by low background AOD values or, alternatively, by the presence of dust presence of dust (, but with no pollution or biomass burning aerosols), the expected AOD differences between V2 and V3 are expected to be minimum as it is confirmed in this studyhas been shown. (Figure 1).

the

However, it should be noted  that  AERONET V3 does not restrict the calculation of AOD to optical masses less than 5.0 (Giles et al., 2019), as V2 does. This results in an increase in the number of solar measurements occurring in the early morning and the late evening. Consequently, the GAW-PFR comparisons with AERONET V3 consisted of ~ 5000 more data pairs than  the GAW-PFR comparison with V2 (see Supplement S.1.1.).

[Figure]

*Figure 1. AERONET Version 3 (V3) vs Version 2 (V2) AOD 1-minute data scatterplot at Izaña Observatory  for the period 2005-2015: a) 380 nm; b) 440 nm; c) 500 nm and d) 870 nm. The corresponding equations of the linear fits, the coefficients of determination (R2), Mean Bias (MB), Root Mean Square Error (RMSE) and the number of data pairs (N) used are included in each legend.*

*5.2.- AERONET-Cimel AOD comparison with GAW-PFR data.*

The  shows that the AOD from AERONET-Cimel radiometers meet the WMO traceability criteria ("traceable AOD data" from now on) at 440 nm, 500 nm and 870 nm  channels. The lowest agreement is found in the UV channel (380 nm) with 92.7 % of the  data, and  the highest in the infrared channel (870 nm) with 98.0 % for V2 (Figure 2; Table 3). Almost identical results are obtained for V3 (Supplement S1 and S2).

However, in the first half of the comparison period (2005-2010) there was a some  mechanical problems in the solar tracker where the GAW-PFR was mounted on, which caused sporadic problems of sun pointing. This finding was confirmed with data from the four-quadrant silicon detector (Wehrli, 2008a) that showed diurnal variation of the PFR sensors position up to 0.3°. From 2010 onwards, the PFR was mounted on an upgraded solar tracker of higher performance and precision. This reduced problems in sun pointing, that were the main cause of  most of the AOD discrepancies between PFR and Cimel, and therefore not attributable to the instruments themselves.

In addition, since 2010, Cimel #244 has been in continuous operation for most of the time at the Izaña Observatory, greatly simplifying calibration procedures and the corresponding data evaluation, and minimizing errors of calibration uncertainties introduced by the use of a high number of radiometers in the intercomparison. During the 2010-2015 period, the fraction of traceable AOD measurements of the total between the AERONET-Cimel radiometer and the GAW-PFR improves to 93.46 % in the 380 nm channel, and this percentage rises to 99.07 % for the 870 nm channel.

Despite the technical differences between both radiometers, described above, and the different calibration protocols, cloud screening  and data processing procedures, the data series of both instruments, can be considered as equivalent, except for 380 nm, according to the WMO traceability criteria defined previously (Eq. 2). This explains the excellent agreement in the long-term AOD climatology shown for GAW-PFR and AERONET-Cimel in Toledano et al. (2018).

We have compared the percentages of AERONET-Cimel AOD V2 data meeting the WMO criteria for the four interpolated GAW-PFR channels with those of AERONET V3 (Table 3).

~~For shorter wavelengths, the percentage of data within the WMO limits decreases when the original GAW-PFR channels are used as a reference (not shown here), mainly, and as expected, in the 412 nm channel as this differs considerably from the nominal value of the corresponding AERONET-Cimel channel (440 nm). For 500 nm and 870/862 nm there are no significant differences. Hereinafter, in this study the interpolated GAW-PFR channels are used.~~

A more detailed statistical evaluation for different scenarios of aerosol loading (three ranges of AOD) and aerosol size (three ranges of AE) for each compared channel has been performed (see Table 4). We

observe that the poorest agreement is obtained at the shorter wavelength channels (440 nm, and especially 380 nm).

Kazadzis et al. (2018b) also found a decrease in the percentage of AOD meeting the WMO criteria for 368 nm and 412 nm spectral bands during the Fourth WMO Filter Radiometer Comparison for aerosol optical depth measurements. As these authors pointed out, the shorter the wavelength, the poorer the agreement because of several reasons: AOD in the UV suffers from out-of-band or at least different blocking of the filters, small differences in central wavelength or FWHM have a larger impact, the Rayleigh correction is more critical, and $NO_2$ absorptions are treated differently. Regarding the effect of the aerosol load and particle size on the AOD differences, our results confirm the decrease of agreement between the two instruments for very large particles coincident with almost pure dust (AE $\leq$ 0.3), and high turbidity conditions (AOD > 0.1). However, it should be noted that the percentage of data pairs in these situations is relatively low (e.g., 6% for AOD > 0.1, and 3.2% for AE > 0.25 at 380nm) with respect to the total data (Table 4). A similar result was reported by Kim et al. (2008), who attributed these discrepancies to the possible spatial and temporal variability of aerosols under larger optical depths in addition to the effect of the different FOV of both radiometers. In our case, and according to previous studies on AOD climatology at IZO (Barreto et al., 2014), the presence of high mineral dust burden when the station is within the SAL, does not necessarily imply lower atmospheric stability conditions resulting in daily AOD means with greater standard deviation. For these reasons, we assumed that the different FOV of these instruments is  the main cause of part of the AOD 1-minute differences outside the U95 limits, under high AOD conditions. This issue is specifically addressed in Section 5.3.

[Figure]

*Figure 2. One-minute AOD data differences between AERONET-Cimel (V2) and GAW-PFR for (a) 380 nm (70838 data-pairs), (b) 440 nm (71645 data-pairs), (c) 500 nm (70833 data-pairs) and (d) 870 nm (71660 data-pairs) for the period 2005-2015. Black dots correspond to the U95 limits. A small number of  outliers are out of the ~±0.06 AOD differences range. Black arrows  indicate a change of Reference AERONET-Cimel radiometer and red arrows indicate a change of the GAW-PFR instrument.*

*Table 3. Percentage of AERONET-Cimel (V2 and V3) 1-minute AOD data meeting the WMO criteria for the four interpolated GAW-PFR channels for the period 2005-2015.*

| Interpolated GAW-PFR channel (%) | Original GAW-PFR channel (%) |
|---|---|
|  |  |
|  |  |
|  |  |
|  |  |

| Channel | V2 (%) | V3 (%) |
|---|---|---|

| | | |
|---|---|---|
| 380 nm | 92.7 | 92.3 |
| 440 nm | 95.7 | 95.2 |
| 500 nm | 95.8 | 95.7 |
| 870 nm | 98.0 | 97.8 |

*Table 4. Percentage of AERONET-Cimel 1-minute AOD data (V2) meeting the WMO criteria for the four compared channels, and different AOD and AE scenarios for the period 2005-2015, . number of data pairs are shown in brackets. The lLast row corresponds to the total percentages for the sub-period 2010-2015. In bold, AOD and AE traceability is > 95% are marked in bold. Number of data pairs are in brackets.*

| % of data within WMO limits | 380 nm | 440 nm | 500 nm | 870 nm |
|---|---|---|---|---|
| AOD ≤ 0.05 | 94.4 (57008) | **96.8 (59130)** | **97.0 (58572)** | **98.5 (60191)** |
| 0.05 < AOD ≤ 0.10 | 91.0 (4723) | 93.1 (4850) | 92.8 (4817) | 94.2 (4908) |
| AOD > 0.10 | 75.0 (3938) | 86.5 (4615) | 85.1 (4466) | **95.9 (5118)** |
| AE ≤ 0.25 | 73.1 (2145) | 82.3 (2417) | 80.1 (2351) | **96.2 (2824)** |
| 0.25 < AE ≤ 0.6 | 91.2 (5407) | **96.2 (5810)** | **96.0 (5691)** | **97.9 (5911)** |
| AE > 0.6 | 94.6 (55114) | **96.9 (57089)** | **97.0 (56504)** | **98.7 (58146)** |
| Total 2005-2015 | 92.7 (65669) | **95.7 (68595)** | **95.8 (67855)** | **98.0 (70217)** |
| Total 2010-2015 | 93.5 (41977) | **97.4 (43745)** | **97.2 (43627)** | **99.1 (44498)** |

In general, the agreement obtained with the 1-minute AOD data is slightly lower than that obtained during short campaigns, such as those reported by (Kazadzis et al. (, 2014) at Athens observatory (4685 data-pairs), and Barreto et al. (2016) at Izaña ObservatoryIZO (5566 data-pairs), with agreements > 99 % for $AOD_{870nm}$ and $AOD_{500nm}$ in case of Barreto et al. (2016). However, our results for $AOD_{500nm}$ (> 95 % of 70833 data-pairs) areis significantly better that that observed by Kazadzis et al. (2014) (~ 48 % of 4685 data-pairs) covering a relatively narrowshort range of AOD. The probable cause for the poor agreement found by (Kazadzis et al., 2014) was a poor calibration in the 500 nm channel in at least one of the instruments operating at Athens.

In addition, short-term campaigns usually cover a small range of AOD, normally with low AOD, and instruments are carefully and frequently supervised. On the contrary, during our intercomparison over a period of 11 years, the operation of the instruments can be described much moreconsidered as the normal operation of such a system. for a long term period of measurements, 20 than that of intensively attended instrumentation during short period intercomparison campaigns.

An additional interesting aspect of this study is that it is not a simple intercomparison exercise between two instruments but a comparison of a number of instruments that acted as reference instruments s for the AERONET/Europe Network.

*Table 5. Basic skill-scores from the AOD intercomparison between GAW-PFR and AERONET-Cimel V2 for the period 2005-2015. The skill scores definitions are found in Huijnen and Eskes (2012).*

| Period | 2005-2015 | | | |
|---|---|---|---|---|
| Wavelengths (nm) | 380 | 440 | 500 | 870 |
| Mean Bias (MB) | -0.0026 | -0.0018 | -0.0021 | -0.0001 |
| Modified Normalized Mean Bias (MNMB) | -0.1301 | -0.1046 | -0.1474 | 0.0129 |
| Fractional Gross Error (FGE) | 0.1727 | 0.1546 | 0.1918 | 0.1837 |
| Root Mean Square Error (RMSE) | 0.0081 | 0.0070 | 0.0064 | 0.0049 |
| Pearson's correlation coefficient (r) | 0.9910 | 0.9925 | 0.9939 | 0.9949 |
| Number of data-pairs | 70838 | 71645 | 70833 | 71660 |

In the first period (2005-2009), a total of 13 Cimel radiometers were used, while in the second period (2010-2015), five Cimel radiometers have participated, and for much of this period, the Cimel #244 was operating as the permanent AERONET reference instrument at IZO. Once the most important causes of non-traceability in the first period, which were associated with a poor pointing of GAW-PFR due to problems in the sun-tracker, were discountedruled out, we can conclude that there are no significant differences in the percentages of traceable data between the two of both periods. This means that the continuous change of MasterReference Cimel instruments used in the 2005-20091period did not have a significant impact on AOD data comparison differences. This provides proof of proves the consistency and homogeneity of the long AERONET-Cimel AOD data series, and their comparability with the GAW-PFR AOD data series, regardless of the number of instruments used to generate these data series.

In our study, with a number of comparison data-pairs one or two orders of magnitude higher than those used in short campaigns, the results shown in Table 4 can be considered excellentas fairly good.

In addition to the traceability scores, we have introduced some basic skill scores corresponding to the AOD intercomparison between GAW-PFR and AERONET-Cimel for the period 2005-2015 (Table 5) to be in line with previous studies that have performed short-term comparisons between these two instruments. The definitions of the used skill scores can be found in Huijnen and Eskes (2012).

The Pearson's correlation coefficient (r) values of the PFR-Cimel 1-minute AOD data-pairs, are higher than 0.99 in all channels. Concerning Mean Bias (MB) and Root Mean Square Error (RMSE) associated withto AOD differences, our results show quite similar skill scores to those found at Mauna Loa, USA for $AOD_{500nm}$ (Kim et al., 2008), although the number of data pairs used at Izaña ObservatoryIZO (~71000) is

much higher ~~than that of Mauna Loa (~9700)Master or R~~reference instruments. These results show MB values to be within 0.01 bias, one order of magnitude lower than in Mauna Loa and Izaña Observatories, highlighting the importance of having well calibrated instruments to carry out these type of comparisons.

For the period 2010-2015 (not shown here),  as expected, the RMSE and the Pearson's correlation improve slightly compared with the whole period 2005-2015.
* * *
**5.3. Non-traceability assessment**

As presented in the able 3, data outside the WMO traceability criteria vary from 2% for 870 nm up to 7.3% for 380 nm. In this section, the different possible causes of non-traceability in AOD are evaluated and, if possible, quantitatively estimated. In order to assess the relevance and quantitative impact of these causes, and estimate errors derived from a non-perfect AOD data synchronization, we first made an analysis on the natural variability of AOD in a very short time period (1 minute) shown below.

**5.3.1. Short-time AOD variability**

In order to determine the variability of AOD within one minute, we have performed two independent analyses with AOD data from the PFR and Cimel for the  368/380 nm and 501/500nm channels during one year (2013). On the one hand, and taking into account that GAW-PFR provides AOD every minute, we have calculated all the AOD differences for each channel in the successive minutes. So we have the variation of AOD from one minute to the next one during a whole year. On the other hand, for AERONET-Cimel, we have taken advantage of the triplets, since each triplet consists of three successive measurements made in one minute . In this case, the strategy has been to calculate the standard deviation of the triplet AOD measurements during a whole year. We have verified that the AOD variability in 1 minute is independent of AOD (see Supplement S3).

*Table 6. Percentage of AOD data with variability within 1 minute less than 0.01 and 0.005, respectively, using AOD data from GAW-PFR (at 368 and 501nm) and AERONET-Cimel (at 380 and 500 nm) for 2013. A total of ~32000 d data-pairs per channel have been used from GAW-PFR, and 20117 triplets (60351 individual AOD measurements) from the Cimel#244 to calculate the AOD variability.*

| GAW- PFR |
|---|

| Percentage of data with 1-minute AOD variability (%) | | |
|---|---|---|
| | 368 nm | 501 nm |
| ≤ 0.01 | 99.88 | 99.91 |
| ≤ 0.005 | 99.21 | 99.35 |
| AERONET-Cimel | | |
| Percentage of data with 1-minute AOD variability (%) | | |
| | 380 nm | 500 nm |
| ≤ 0.01 | 99.87 | 99.99 |
| ≤ 0.005 | 99.82 | 99.42 |

The results obtained on the AOD variability in 1-minute from PFR data are very similar and consistent to those obtained with Cimel. Less than ~ 0.8% of the AOD data show variability higher than 0.005 in all wavelength ranges. It should be noted that the possible instrumental noise is included in this variability, so that the actual natural AOD variability would be, in any case, lower than that expressed in Table 6. The percentage of data with 1-minute AOD variability for allthe four GAW-PFR channels are given in Supplement S3. These results indicate that the natural AOD variability is very low thus the non-ideal measurement synchronization cannot explain the percentages of non-traceable AOD cases shown in Tables 3 and 4.

*5.3.2. Uncertainties ofby GAW-PFR channel interpolation to AERONET-Cimel channels*

The interpolation of the CIMEL AODs to the PFR AOD wavelengths can be one of the sources of uncertainty in this comparison assessment. The greatest uncertainty arises in the extrapolation of the $AOD_{412\ nm}$ of the PFR to the Cimel wavelength 440 nmretrieve AODs at the CimelCIMEL $AOD_{440nm}$.

Using the Angström formula we have calculated that for an uncertainty of ±0.5 in the AngströmexponentAE and for AOD of 0.1 at 412 nm, the introduced uncertainty in the AOD extrapolation from 412 nm to 440 nm is ~5% (i.e., 0.005 for $AOD_{412nm}$=0.1). The introduced uncertainty in AOD extrapolation is reduced to ~2% for an uncertainty of ±0.3 in AE. of the order of ±0.003, while for an $AOD_{412nm}$ of 0.5 and an AE uncertainty of ±0.3, the introduced uncertainty is ±0.008. For all other AOD interpolations the errors are smaller.

*5.3.32.31  Calibration related errors*

[revised manuscript text omitted]

The correct cause attribution of each outlier would require manual inspection and additional specific information on instrumental checking and maintenance information that is not always available. We have investigated in more  detail the origin of the outliers and whether one of the two instruments predominantly caused them.

Thus, we have calculated for the non-traceable AOD data the diurnal range of AOD variation (maximum value minus minimum value of AOD in one day) at 380 nm for each instrument under pristine conditions (Figure 3) using Cimel AOD$_{500nm}$ daily mean < 0.03 to select the pristine days. According to this approach, the instrument that shows the highest daytime AOD range is the one that is responsible for the outlier. As the wavelength increases both the number of outliers and the magnitude thereof decreases significantly (Supplement S6). Then, we identified those outliers with a diurnal AOD range higher  than 25% of the mean daily AOD value and investigated their possible causes. A total of 51 cases for GAW-PFR and 81 cases for AERONET Cimel V3 were obtained and  analysed in detail, using  auxiliary information, such as 1-minute in-situ meteorological data, 5-minute all-sky images, 1-minute BSRN data, and satellite imagery (not shown here). We obtained the percentage of AOD outliers of GAW-PFR and AERONET Cimel (V3) for which a certain cause has been identified, such as calibration uncertainties, cloud screening algorithm failures, mixture of the two previous causes, poor sun pointing, or not well-defined  causes (electronic problems, humidity inside the lenses, filter dirtiness, obstruction of the lenses collimators, insects on the optics outside, etc.) (see Supplement S7).

From the analysis of these cases, under the conditions described above, it should be noted that ~ 44% of the cases with fictitious AOD diurnal cycles were due to small uncertainties in the calibration of AERONET-Cimel (V3), while for this same cause  ~ 8% of cases were identified in GAW-PFR.

Some examples of  AOD non-traceability for both AERONET-Cimel and GAW-PFR in the ~380 nm channel are shown in Supplement S8. The fictitious diurnal AOD cycle is mainly visible in the UV channels as shown in the examples reported in  Supplement S9.  that the fictitious diurnal AOD  can be more easily identified  under very low AOD conditions.

[Figure]

*Figure 3.* AOD diurnal range  variation (maximum value minus minimum value of AOD in one day) at 380nm corresponding to AOD outliers (non-traceable AOD)  under pristine conditions ($AOD_{Cimel\text{-}500nm} \leq 0.03$) in the period 2005-2015 for AERONET V2 (a) and V3 (b).

**2.2 _Differences in cloud-screening and sun tracking_**

We have examined the effect that the presence of clouds might have on AOD differences and the  percentage of cases outside the U95 limits. The impact of clouds on AOD differences only occurs when both GAW-PFR and AERONET-Cimel cloud screening algorithms fail to identify clouds in the direct sun path. ~~AERONET-Cimel Version 2 data uses the so-called "triplet check" cloud screening algorithm developed by Smirnov et al. (2000) and a second order temporal derivative constraint (McArthur et al., 2003) to rule out AOD measurements potentially contaminated by clouds. GAW-PFR algorithms also use the Smirnov triplet measurement, and the second-order derivative check, but add a test for optically thick clouds with $AOD_{500nm} > 2$ (Kazadzis et al., 2018b). This algorithm, used by both networks with certain variants, assumes a transitory character in the presence of a cloud, which causes a sudden change of AOD. This sharp change would be detected by measuring the stability of three successive optical depth measurements, so that, when a cloud totally or partially blocks the sun, the standard deviation associated with the average of the triplets increases enormously. Note that if either one or both cloud screening algorithms (GAW-PFR and AERONET-Cimel) are flagged as cloudy, then the corresponding AOD data pair does not take part in the comparison.However, in the case of stratiform and very stable clouds or in the case of very thin clouds such as cirrus clouds, the algorithm could erroneously interpret that there are no clouds since there would be no appreciable changes in the stability of the triplets.A hint that cloud flagging failure could lead to large ADO calculated differences is coming from an analysis of AOD differences for days with different cloudy sky fractions.We do not have precise ancillary information to verify in each 1-minute data the influence that a certain cloud could cause in the non-traceability found. As a first approach forassessingIzaña Observatory~~IZO (García et al., 2014). FCS represents the percentage of observed sunshine hours in a day with respect to the maximum possible sunshine hours in that day. The higher the daily FCS, the higher the clear sky percentage we have on that day.

The percentages of traceable and non-traceable AOD data versus FCS values grouped into five intervals are shown in Table 8.  It should be emphasized that the number of cases linked with FCS between 0% and 60% are less than 2% of the total cases. ~~There, ~8% (870nm) to 24% (380nm) of the data are outside the WMO limits (maximum of 0.5% of the total data for 380nm outside the WMO limits).~~ As the fraction of clear

sky increases, the percentage of traceable AOD data significantly exceeds the number of non-traceable AOD data. The percentage of traceable data is especially large (> 90 %) when FCS > 80 % (almost clear skies).

This is the FCS range in which a significant percentage of days with cases presenting scattered clouds are recorded, which qualitatively confirms that V3 has introduced more efficient cloud screening than V2. However, the real impact of clouds on AOD traceability at IZO is very low due to its special characteristics of a high mountain station with very little cloudiness.  Therefore, in practice, the possible impact of clouds on the non-traceability of AOD data-pairs is insignificant at IZO .  GAW-PFR and AERONET-Cimel cloud screening algorithms provide successful identification on clear direct-sun conditions during cloudy skies (FCS < 40 %) for  99.75 % of the cases, excluding those with very thin clouds.  of AERONET V3 (Giles et al., 2018).

However, in the case of stratiform and very stable clouds or in the case of very thin clouds such as cirrus clouds, the algorithm could erroneously interpret that there are no clouds since there would be no appreciable changes in the stability of the triplets.

In the particular case of Izaña there are some very specific cloud scenarios in which cloud screening algorithms could fail resulting in non-AOD traceability: 1) Altostratus above the top of the SAL, at ~6 Km altitude (see Supplement S10); 2) Cirrus clouds (see Supplement S11); and 3) low clouds (stratocumulus) that sometimes exceed the observatory height level (see Supplement S11).

~~A more detailed analysis of more rare atmospheric conditions, such as those of strati-form and homogeneous cirrus clouds, or when altostratus are present above the Saharan Air Layer (SAL), around 6 Km altitude, and thus masked by a heavy mineral dust layer below needs further investigation. A constant cloud optical thickness (COT) corresponding to a cloud of a certain horizontal extension would cause the successive measurements within a minute to correspond to the same cloud stage, and therefore it would not be discernible from the extinction caused by aerosols. In the case of very thin cirrus clouds, the fluctuations in AOD would be very small and could be interpreted as the presence of a light layer of aerosols. Another factor that must be taken into account is that the FOV of the instruments is different. Thus, GAW-PFR (FOV = 2.5°) could detect the entry of a constant COT cloud in part of its~~

field of view in a different way than AERONET-Cimel (FOV = 1.2°). In all these cases, the cloud-screening algorithms may fail simultaneously in both GAW-PFR and AERONET-Cimel, resulting in a different AOD measurement derived by the two instruments. shown reported1-minute

As can be deduced from the analysis of these cloud cases, the impact of the different types of clouds oin AOD retrieval is very complex and further specific investigations are required in order to understand, the reasons behind failures in the GAW-PFR and AERONET-Cimel cloud screening algorithms.

These type of rare situations should be the subject of future studies through measurement campaigns using ancillary observation systems (e.g. lidar, all sky camera).

*Table 7Table 8. Percentage of traceable (T) data and percentage of AOD data outside within the U95 limits (NT) for each channel and 5 
[revised manuscript text omitted]

climatology. Concerning AERONET-Cimel Version 2, a NASA TOMS 1°_x_1.25° resolution $O_3$ climatology is used. From Eq. 10, the differences in $O_3$ optical depth $\Delta 4\tau_{O3}$ can be derived:

$$\triangle\tau_{O3} = \sigma_{O3}(\lambda)\frac{1}{1000}\frac{m_{O3}(O_{3PFR} - O_{3Cimel})}{m_a}$$

(12)

The largest influence of total ozone data uncertainty in $\tau_{O3}$ occurs , by far, at 500 nm (Figure 5). According to Wehrli (2008b) and Kazadzis et al. (2018b), total ozone needs to be determined to ± 30 DU or 10 % of typical values, to ensure an uncertainty of ± 0.001 in $\tau_{O3}$ at 500 nm. In the case of the GAW-PFR / AERONET-Cimel comparison, and due to the very different method in which both networks obtained $O_3$ values for their corresponding corrections, the ozone differences found on some days (1761 out of 71965 days; 2.4 %) are very large (> 40 DU), exceeding a difference in the ozone optical depth of 0.001. Even so, the potential contribution toof AOD differences outside the $U$95 limits between the two networks is negligible. Total $O_3$ over the Izaña ObservatoryIZO is quite stable, showingshows a relatively small amplitude throughout the year, but both surface ozone concentrations and column ozone amount could sharply increase under the influence of cut-off lows injecting air from the high-mid troposphere into the lower subtropical troposphere, which is not uncommon in spring and the first half of summer (Cuevas et al., 2015; Kentarchos et al., 2000). In addition through exchange processes in the Upper Troposphere Lower Stratosphere (UTLS) due to the presence of the subtropical jet (mainly from February to April) (Rodríguez-Franco and Cuevas, 2013). However, if we wanted to repeat this traceability study of 1-minute AOD data in mid or high latitude stations where sharp $O_3$ variations (several tens of DU) could be registered in a few hours, the correction of 1-minute AOD measurements by $\tau_{O3}$ might be a challenging issue.

[Figure]

*Figure 5. (a) Total $O_3$ used by GAW-PFR (measured Brewer $O_3$ values from IZO, OMI $O_3$ overpass or Brewer $O_3$ climatology) and AERONET-Cimel (TOMS $O_3$ climatology), and (b) $\Delta\tau_{O3}$ (λ) caused by differences in daily total $O_3$ between the two instruments in the period 2005-2015.*

-5.3.54.3 Differences in $NO_2$ absorption

AERONET-Cimel applies a correction by absorption of $NO_2$, but GAW-PFR does not include this correction. AERONET--Cimel obtains daily total $NO_2$ data from a 0.25° ⁑x 0.2°⁑ resolution $NO_2$ monthly climatology obtained from the ESA Scanning Imaging Absorption SpectroMeter for Atmospheric CHartographY (SCIAMACHY) (Eskes and Boersma, 2003). In order to assess the contribution toin AERONET-Cimel 1-minute AOD data non-traceability by $NO_2$ absorption what we havereally to estimate is the $NO_2$ optical depth ($\tau_{NO2}$(λ)) of AERONET-Cimel since GAW-PFR does not perform this correction. Analogously to $\Delta\tau_{O3}$, the differences in nitrogen dioxide optical depth $\Delta$4$\tau_{NO2}$ can be obtained from:

$$\triangle\tau_{NO2} = \sigma_{NO2}(\lambda)\frac{1}{1000}\frac{m_{NO2}}{m_a}(-NO_{2Cimel})$$

(13)

Where $m_a$ is given by Eq. 7, $NO_{2Cimel}$ (DU) is the daily total $NO_2$ used by AERONET-Cimel, $\sigma_{NO2}(\lambda)$ is the $NO_2$ absorption  (Gueymard, 1995)  weighted by the specific filter response: 15.6 cm$^{-1}$ (380 nm), 12.3 cm$^{-1}$ (440 nm), and 4.62 cm$^{-1}$ (500 nm). Finally,  $m_{NO2}$ has the following expression (Gueymard, 1995):

$$m_{NO2} = \frac{1}{sin\theta + 602.30(90-\theta)^{0.5}(27.96+\theta)^{-3.4536}}$$

(14)

[Figure]

*Figure 5. (a) Total O₃ used by GAW-PFR (OMI O₃ overpass or Brewer O₃ climatology) and AERONET-Cimel (TOMS O₃ climatology), and (b) Δτ_O₃ (λ) caused by differences in daily total O₃ between the two instruments in the period 2005-2015.*

[Figure]

*Figure 6. (a)*  *NO₂ monthly climatology obtained from the ESA SCanning Imaging Absorption SpectroMeter for Atmospheric CHartographY (SCIAMACHY), used by AERONET-Cimel at* *IZO, and (b) Δ4τ₆NO2 (λ) caused by differences in daily total NO₂ between GAW-PFR and AERONET-Cimel in the period 2005-2015. Note that GAW-PFR does not take into account the correction for the NO₂ absorption.*

*Table 9. Percentage (%) of* additional *traceable AERONET-Cimel AOD 1-minute data* (V2 and V3)  *after correcting by pressure, and total column O₃ and NO₂* for the period 2005-2015.

|  |  |
|---|---|
|  |  |
|  |  |
|  |  |
|  | ~ |

| Channel | Increment (%) of traceable AOD data after P, $O_3$ and $NO_2$ corrections | |
|---|---|---|
| | V2 | V3 |
| 380 nm | 1.3 | 1.7 |
| 440 nm | 0.2 | 0.3 |
| 500 nm | 0.3 | 0.1 |
| 870 | ~0.0 | ~0.1 |

In Figure 6a the total $NO_2$ used by AERONET-Cimel to evaluate $\tau_{NO2}(\lambda)$ is depicted. Figure 6b shows the $\Delta\tau_{NO2}(\lambda)$ caused by differences in daily total $NO_2$ between GAW-PFR and AERONET-Cimel. $\Delta\tau_{NO2}$ is of the order of $10^{-3}$ for 380 and 440 nm channels, while, for 500 nm channel, it is of the order of $10^{-4}$.  However, it should  be noted that an impact on AOD calculation is expected when replicating similar analysis in  highly $NO_2$ polluted regions  . Such cases include  large industrial cities from East Asia and Central and Eastern Europe  (e.g., Chubarova et al., 2016).

Taking into account the corrections for Rayleigh scattering and for the absorptions by $O_3$ and $NO_2$, we have calculated the additional traceable AOD  that lie within the U95 AOD limits  the non-traceability of the 1-minute AOD values (Figure 6; Table 9). This percentage is maximum at 380 nm with 1.3% (V2) and 1.7% (V3) of the whole dataset.  The 870 nm channel is only affected by the Rayleigh correction component and therefore the increment of traceable data after the mentioned corrections is

minimalmum.The 1minute AOD data outside the *U*95 limits by these corrections is negligible in the 870 nm channel.

*——————5.4.3 —GAW PFR and AERONET-Cimel comparison under high AOD conditions: the impact of dust forward scattering role of the foron different FOVs.*

When we represent the AOD differences between AERONET-Cimel and GAW-PFR versus AOD (GAW-PFR) for AOD > 0.1, we observe a positive slope that increases when the AOD fitted data are > 0.05 (dusty non-pristine conditions), we noteing that AERONET-Cimel shows slightly higher AOD values than GAW-PFR, being higher than +0.01 for AOD > 0.15 (Figure 7). , and more clearly at 500 nm. The AOD data outside the *U*95 limits (in red) increases notably from AOD > 0.1.

In fact, the percentage of data outside the U95 limitsnon traceable AOD data increases as AOD increases (Table 9Table 10), so that for dust-related aerosol conditions (AOD$_{500nm}$ > 0.3) the percentage of AOD data outside the *U*95 limits is > 50 % for 380 nm and 440 nm (all channels except for 870 nm (Table 9Table 10, percentages in brackets). Similar results are found when using AERONET V3 (see Supplement S13). The increase in the percentage of AOD data outside the *U*95 limits is especially significant at the 380 nm channel. Taking into account the number of data compared with the total cases, Tthese results show a small but non-negligible percentage of AOD differences outside the *U*95 limits for AOD > 0.1, ranging from ~0.3 % at 870 nm to ~ 1.9 % at 380 nm. This especially affects the shorter wavelengths (Table 9Table 10).

[Figure]

*Figure 7. Actual AOD differences between AERONET-Cimel V2 and GAW-PFR vs  AOD_{PFR} at (a) 380 nm (b) and 500 nm for the period 2010-2015. The fitting line has been calculated with those data points with AOD  > 0.1 and  Cimel-PFR AOD difference > 0. The number of data used in the plots are indicated in the legend. The percentage of non-traceable AOD data with these conditions is ~24% for 380 nm, and ~8% for 500 nm. Note that some traceable (black) points show larger AOD differences than non-traceable (red) points because of the air mass dependence of the WMO traceability criterion.*

*Table 10. Percentage of  AOD data outside the U95 limits at 380, 440, 500 and 870 nm channels and for three AOD$_{500nm}$ thresholds respect to all data and respect to all data for each AOD interval (in brackets).*

|  | Percentage of AOD data outside the $U$95 limits (%) | | |
|---|---|---|---|
|  | AOD$_{500nm}$>0.1 | AOD$_{500nm}$>0.2 | AOD$_{500nm}$>0.3 |
| 380 nm | 1.9 (25.0) | 1.2 (47.2) | 0.5 (59.8) |
| 440 nm | 1.0 (13.5) | 0.8 (32.0) | 0.5 (57.6) |
| 500 nm | 0.6 (8.0) | 0.5 (18.7) | 0.3 (39.3) |
| 870 nm | 0.3 (4.1) | 0.2 (6.4) | 0.1 (14.0) |

aerosol forward scattering within the FOV of various instruments and calculated AOD was investigated some decades ago by Grassl (1971) who determined that at AOD=1 the circumsolar radiation increases by >10% the incoming radiation. Russell et al (2004), using dust and marine aerosols data, quantified the effect of diffuse light for common sun photometer FOV. They reported that the correction to AOD is negligible (<1% of AOD) for sun photometers with narrow FOV (< 2°), which is greater than  the Cimel FOV and slightly smaller than the PFR FOV  (2.5°). Sinyuk et al. (2012) assessed the impact of the forward scattering aerosol on the uncertainty of the AERONET AOD, concluding that only dust aerosol  with high AOD and low solar elevation could cause a significant bias in AOD (> 0.01). ~~Torres et al. (2013) investigated the uncertainty of the FOV in the AERONET-Cimel measurements indicating that direct solar irradiance measurements are biased by the amount of aureole radiation that is assumed to be direct solar radiation. The solar aureole, also known as the circumsolar region, is the bright region surrounding the solar disc, which becomes especially visible when there is a burden of moderate-high aerosols in the atmosphere.~~

GAW-PFR has double the FOV (2.5°_; Wehrli (2000)) compared to the AERONET-Cimel (1.3° ± 4.8 %; Torres et al.,_ (2013), so it is reasonable to expect that it is more affected by the circumsolar radiation than the AERONET-Cimel radiometer.

Taking advantage of the fact that Saharan dust intrusions regularly affect IZO, we provide a detailed analysis on the impact  dust forward scattering causes in the AOD retrieval of the two radiometers with different FOV,  the AOD differences    high dust load (AOD > 0.1) conditions. For this purpose we have used a forward Monte Carlo model (see section 4.4) with which we perform simulations that include accurate dust aerosol near-forward scattering effects.

Dust aerosol single-scattering properties were computed using Mie theory, assuming a refractive index of 1.47+0.0025i at the wavelengths 380 nm, 440 nm and 500 nm and 1.46+0.012i at 870 nm, based on AERONET measurements at IZOzañnaa. Seven values of aAerosol effective radiius ($r_e$) in the range 0.2 to 3.0 μm were considered, and a lognormal size distribution with a geometric standard deviation of 2 was assumed. A mid-latitude summer atmospheric profile starting from the Izaña altitude (2.4 km a.s.l.) was assumed, withbeing the aerosol layer located at 5-6 km a.s.l. (typical of summertime). A spectrally uniform surface albedo of 0.11 was employed. Computations were performed for nine AOD values (AOD= 0, 0.1, 0.2, 0.3, 0.4, 0.5, 0.6, 0.8, and 1.0) and for fiveour solar elevation angles (θ=80°, 60°, 45°, 340° and 20°). Ten million photons were used for each case and wavelength .-The Monte Carlo model assumes a plane-parallel atmosphere, so the air mass factor is m=1/sinθ. Ten million photons were used for each case and wavelength.

Supplement S15 shows the ratio of scattered to direct radiation for cases with AOD up to 0.5. We have performed scattered to direct radiation simulations for FOVs of 2.5° and 1.2° for six values of effective radius ($r_e$=0.2, 0.5, 1.0, 1.5, 2.0 and 3.0 μm), for five AOD values (AOD= 0.1, 0.2, 0.3, 0.4, and 0.5), and for five solar zenith angles (θ = 10°, 30°, 45°, 60°, and 70°) (see Supplement S15). The ratio increases with increasing $r_e$, as the aerosol forward-scattering peak grows stronger. In the case of Saharan dust intrusions at IZOzañna Observatory, the median $r_e$ median determined from both AERONET data inversion and the in-situ aerodynamic particle sizer (APS) analyzer is ~1.5 μm. This value agrees with the dust size distribution found during SAMUM-2 during long-range transport regime (Weinzierl et al., 2011). For this particle size, the ratio of scattered to direct radiation is ~3 times larger for FOV of 2.5° than FOV of 1.3°.

The error in the retrieved AOD due to scattered radiation within the instrument FOV was evaluated by comparing the apparent AODs, defined as:

$$AOD_{app,PFR} = -\frac{1}{m} ln \frac{F_{PFR}}{F_{PFR}(AOD=0)}$$

—(15)

$$AOD_{app,Cimel} = -\frac{1}{m} ln \frac{F_{Cimel}}{F_{Cimel}(AOD=0)}$$

—(16)

with the true AOD

$$AOD_{true} = -\frac{1}{m} ln \frac{F_{dir}}{F_{dir}(AOD=0)}.$$ (17)

HWhere, $F_{dir}$ is the irradiance due to direct (i.e., non-scattered) radiation, and $F_{PFR}$ ($F_{Cimel}$) is the total irradiance that would be measured by the PFR (Cimel) radiometer, considering the instrument FOV and the FOV angular function. The relative error in AOD depends strongly on the particle size but it is fairly constant for each $r_e$ value considered (see Supplement S16). For $r_e$ ~1.5 μm, the relative error in AOD at

380 nm (500 nm) is ~1.6% (1.0%) for Cimel, and ~5% (~3%) for PFR. These errors are in good agreement with those estimated by Russell at al. (2004), and slightly higher than the relative AOD error of 0.7% due to coarse dust  aerosol forward scattering reported by Eck et al. (1999).

The Monte-Carlo-simulated relative differences in retrieved AOD (in %) that would result from the scattered radiation within the FOV of the PFR and Cimel instruments, and the difference in retrieved AOD between PFR and Cimel as a function of the AOD retrieved with PFR, for 380 nm and 500 nm, are shown in Figure 8.  The main results of these simulations are: 1) the higher FOV of the PFR, compared to that of the Cimel, results in lower AOD values for the PFR ; 2) the fractional AOD difference related to the different FOVs of PFR and Cimel is fairly constant for any aerosol effective radius, but increases with increasing the effective radius; and 3) this fact might explain at least some of the systematic differences seen in Fig. 7. Note that, as  lower AOD values derived from the PFR are expected based on its larger FOV, the linear fit in Fig. 7 has been calculated for those data points with  the Cimel-PFR AOD differences> 0. In this way, we discard those pairs of AOD data whose difference is not only due to the different FOV between both instruments, obtaining in this way a better approximation to quantify this effect.

The slopes of the fitting lines of the Cimel-PFR AOD differences vs. PFR AOD for AOD> 0.1 (dusty conditions) are 2.7% for 380 nm and 2.3% for 500 nm (Figure 7), which are quite consistent with the percentage differences of AOD between Cimel and PFR for an effective radius of 1.5 μm (Figures 8a and 8b). These percentages correspond to absolute AOD differences of 0.016 at 380 nm, and 0.011 at 500 nm for AOD=0.5 (Figures 8c and 8d), that are of sufficient magnitude to cause an appreciable number of 1-minute AOD data outside the U95 limits, as indicated in Table 10.

[Figure]

*Figure 8. Panels a) and b): the simulated relative differences in retrieved AOD (in %) that would result from the scattered radiation within the FOV of the PRFR and Cimel instruments. The red (blue) dots show the differences between the AOD that would be retrieved using PRFR (Cimel) and the actual AOD, and the grey dots the difference between PRFR and Cimel, at wavelengths (a) 380 nm and (b) 500 nm. Panels c) and d): the difference in retrieved AOD between PRFR and CimelIMEL, plotted as a function of the AOD retrieved with PFR, for seven values of aerosol effective radius between 0.2 and 3.0 μm, at (c) 380 nm and (d) 500 nm.*

If we apply the corresponding corrections to the 1-minute AOD PFR data > 0.1 assuming an effective radius of 1.5μm, + 3.3% at 380nm and + 2.2.% at 500 nm, it turns out that the slopes of the fitting lines of the CimelIMEL-PFR AOD differences vs. PFR AOD they become practically zero (Figure 9). Moreover, the number of AOD data outside the U95 limits is reduced by approximately 53% for 380 nm and by 13% for 500 nm. It must be taken into account that the percentage of AOD data for AOD> 0.1 outside the U95 limits, before the corrections, is only 8% at 500 nm, while at 380 nm it is a significant value (24%).

Thise AOD "correction" reduces the Cimel-PFR AOD differences substantially but does not eliminate them completelyis fairly good but it is not complet, e mainly for two reasons. The first one it is the inherent limitation of data correction using the percentage difference in AOD obtained by model simulation for a fixed effective radius.

We have assumed an effective radius of 1.5 µm but, in reality, the radius of dust particles varies. A reasonable range of dust particle radiussize is between 0.1 and 3 µm (Balkanski et al., 1996; Denjean et al., 2016; Mahowald et al., 2014). So, depending on the distance from the dust source to Izaña ObservatoryIZO and the size of the emitted dust, the effective radius could vary slightly between dust episodes. As can be seen in Figures 8a and 8b, the percentage differences in AOD between Cimel and PFR for a 1-2[1,2 µum] effective radius interval, the PFR-Cimel AOD relative difference at 380 nm (500 nm) might change between (around) ~-1.8% (-1.1%) to -4.9% (3.3%).

The second reason is a possible cloud contamination in AOD retrieval when altostratus are present above the SAL, as discussed in Section 5.3.4.

A similar analysis has been carried out for AERONET V3 (see Supplement S17), where we observe that the corrections obtained are not as good as those obtained for V2. This may be due to the very high AOD data retention in V3 which could include more cases in which altostratus clouds and dust are present. The effect of FOV on AOD retrieval should be taken into account for those radiometers with a relatively high FOV (>3°) measuring in regions with relatively high AOD (> 0.2) for most of the year, as is the case in many sites of Northern Africa, the Middle East and East Asia (Basart et al., 2009; Cuevas et al., 2015; Eck et al., 1999; Kim et al., 2007). This effect leads to AOD underestimation, and the variable number of high AOD episodes in each season of the year might affect the AOD long-term trends. AOD measurements under these conditions would be especially affected for optical air mass < 3.

[Figure]

*Figure 9. The same as Figure 7 after "correcting" the PFR AOD data by adding + 3.3% at 380nm and + 2.2.% at 500 nm to the 1-minute PFR AOD data > 0.1.*

Furthermore, it should be taken into account that, as discussed in section 5.2.2., under relatively high AOD conditions, the presence of altostratus above SAL is not infrequent, and they could also cause non-traceability in AOD when the cloud screening algorithms

fail. ~~Note that a graphic equivalent to Figure 9 is shown in Supplement S17 but for AERONET V3, observing that the corrections obtained are not as good as those obtained for V2. This may be due to the very high AOD data retention in V3 which could include more cases in which altostratus clouds and dust are present. Therefore, the FOV study should be done once the dust events with presence of clouds over the SAL have been ruled out.~~

   sites of Northern Africa, the Middle East and East Asia (Basart et al., 2009; Cuevas et al., 2015; Eck et al., 1999; Kim et al., 2007).

*5.4 Angström exponent comparison*

We have performed a comparison of the AE provided by GAW-PFR and AERONET-Cimel using in both cases the AOD data obtained from the four common channels (380 nm, 440 nm, 500 nm and 870 nm) with a total of 70716 data-pairs. The PFR-AOD values have been ordered from lowest to highest by grouping them in intervals of 500 values for which the averages (and corresponding standard deviations) of the PFR-Cimel AE differences have been calculated ( Figure 10a). In a similar way we proceeded with the PFR-AE values (Figure 10b).

[revised manuscript text omitted]

Considering the AE criteria established by Cuevas et al. (2015) and Berjón et al. (2019),  we have  identified the following  four main  categories according to  the $AE_{PFR}$ and $AE_{Cimel}$ values :

1. $AE_{PFR}$ & $AE_{Cimel}$ > 0.6: Pristine conditions.
2. 0.25 < $AE_{PFR}$ & $AE_{Cimel}$ ≤ 0.6: Hazy,  mineral dust being the main aerosol component.
3.  $AE_{PFR}$ & $AE_{Cimel}$ ≤ 0.25: Pure dust.

$AE_{PFR}$ and $AE_{Cimel}$ do not fit any of the previous categories.

[Figure]

*Figure 109. (a) PFR Cimel AE mean absolute differences (and corresponding standard deviations) vs PFR mean AOD₅₀₀ₙₘ in 500 data intervals (b) and vs PFR mean AE in 500 data intervals. AE has been computed for both PFR and Cimel using the four common channels (380, 440, 500 and 870nm).*

4.

In 94.93.8 % of the cases, GAW-PFR and AERONET-Cimel V2 match the AE intervals of each aerosol scenario. Similar results (93.4%) were obtained when comparing with AERONET V3. Most of the agreement (>80 79 %) occurs in the predominant scenario of pristine conditions despite the AE uncertainty under pristine conditions being ≥ 1. See Supplement S18 for more details. Notice that given the special characteristics of the Izaña Observatory, and according to Cuevas et al. (2015) and Berjón et al. (2019), AE is a self-sufficient parameter to define different types of aerosol scenarios without the need to combine its information with AOD.

**6. Summary and Conclusions**

While GAW-PFR is the WMO-defined global AOD reference , being directly linked to WMO / CIMO, and was specifically designed to detect long-term AOD trends, AERONET-Cimel is the densest AOD measurement network globally, and the network most frequently used for aerosol characterization and for model and satellite observation evaluation.

An AERONET-Cimel 11-year AOD data series at IZO was obtained using a large number of radiometers. A total of 13 Reference instruments were used  in the period 2005-2009 which means that every 4 and a half months, approximately, an instrument was replaced by another one to be calibrated. Their calibrations were performed during their respective measurement time periods at IZO. Therefore, these calibrations were not in any way linked with those of the instruments that preceded or replaced them, nor with GAW-PFR reference. These facts led us to investigate the homogeneity of the AERONET-Cimel AOD data series and their intercomparability with the much more homogeneous AOD data series from GAW-PFR (3 instruments in 11 years).

 The traceability concept for AOD suggested by WMO consists in determining whether the AOD difference of the AERONET CIMELs vs the GAW PFRs lie within the U95 limits.

We have used uncertainty limits for AOD traceability established by WMO (2005) for these type of instruments with finite FOV. The acceptable traceability is when 95 % of the absolute AOD differences lie within these limits, in which case both data populations are considered equivalent. It should be clarified that "traceability" is not used in a strict metrological sense.

This study has addressed the comparison of the GAW-PFR data with the two versions of AERONET (V2 and V3) in the period 2005-2015. An excellent agreement between V2 and V3 for the four analysed channels ($R^2 > 0.999$) has been obtained.

More than 70000 synchronous GAW-PFR (PFR) and AERONET-Cimel (Cimel) 1-minute data-pairs in each channel in the period 2005-2015 were analysed. An excellent traceability of AOD from the AERONET-Cimel (V2 and V3) is found for 440 nm, 500 nm and 870 nm, and fairly good results for 380 nm. The lowest percentage of traceable AOD data is registered in 380 nm with 92.7 % of the 1-minute data within the WMO limits, and the highest in 870 nm with 98.0 % of the data.

The different possible causes of non-traceability in AOD were investigated as follows:

- Absolute AOD measurements synchronization.

  Analyzing 1-minute AOD variability we concluded that its impact on the AOD differences is negligible as only ~0.8% of the AOD data has a variability larger than 0.005 in all spectral ranges.

-

~~The 1-minute AOD differences Mean Bias of this study is 0.001, an order of magnitude lower than those obtained from previous short-term PFR-Cimel comparison campaigns, in which the Cimel instruments were calibrated by transferring the calibration coefficients by comparison with co-located Master instruments. This indicates the importance of good calibration and maintenance of the Cimel instruments to obtain AOD data very similar to that of GAW-PFR.~~

$_{500nm}$

5.5 % for 1 ≤ m < 2, being the most likely cause an insufficient 10 accurate calibration of AERONET-Cimel in this channel.

Sun tracking misalignments  constitutes a serious problem and a major cause of non-traceability of AOD data-pairs as demonstrated by the AOD data outside the U95 limits from the period 2005-2010 as a consequence of episodic problems with the sun-tracker of the GAW-PFR radiometer. For the 2010-2015 period the percentage of traceable data-pairs improves to 93.5% (380 nm), 97.4% (440 nm), 97.2% (500 nm) and 99.1% (870 nm). However, most of these cases could be identified and excluded from the analysis.

- Cloud screening failure by both network algorithms.

~~Regarding AOD non-traceability due to the different cloud-screening algorithms of both networks, it must be said that both algorithms are very similar. GAW-PFR uses the same cloud screening as AERONET-Cimel but incorporates some additional controls. The only reason for AOD non-traceability comes from the simultaneous failure of both cloud screening algorithms because if one or both of them detect clouds, the data will not be part of the comparison.~~ According to our observations, the simultaneous failure of both cloud screening algorithms might occur only under the presence of large and stable cirrus, or altostratus (~ 6000 m a.s.l.) on the top of a heavily dust loaded Saharan air layer, hiding very wide and stable clouds. In these cases, the radiometers interpret these clouds as aerosol layer and might provide values very different from the real AOD. This effect, or the comparison at IZO, however, this effect is negligible since GAW-PFR and AERONET-Cimel cloud screening algorithms provide successful cloud identification on clear direct-sun conditions during cloudy skies (FCS < 40 %) for 99.75 % of the cases.

- Pressure measurements related errors.

-

 Since the accuracy of the new barometers built into new radiometers is about 3 hPa, and only errors in atmospheric pressure > 30 hPa might produce an impact on Rayleigh scattering, the AOD non-traceability due to errors in Rayleigh scattering is negligible.

- Total column ozone input uncertainty.

The largest influence of total ozone data uncertainty in ozone absorption occurs mainly at 500 nm. Total ozone needs to be determined to ±30 DU or 10 % of typical values, to ensure an uncertainty of ±0.001 ozone absorption at 500 nm. In the case of the GAW-PFR / AERONET-Cimel comparison, despite the very different methods in which both networks obtained $O_3$ values for their corresponding corrections, large ozone differences were found (> 40 DU) only on 2.4 % of the days , resulting in a

difference in the ozone optical depth slightly above ~0.001. The potential contribution to non-traceable AOD values between the two networks is negligible. However, in mid or high latitude stations where fast $O_3$ variations of several tens of DU might be registered, the correction of 1-minute AOD measurements by ozone absorption might be an issue to be considered.

- Total column $NO_2$ input uncertainty.

The differences in $NO_2$ absorption caused by differences in daily total $NO_2$ between GAW-PFR and AERONET-Cimel is of the order of $10^{-3}$ for 380 nm and 440 nm channels, while, for 500 nm channel, it is even lower, of the order of $10^{-4}$. So, differences in $NO_2$ absorption are negligible in the 1-minute AOD non-traceability of our study. However, $NO_2$ absorption might have some impact on AOD in highly polluted regions, such as in large industrial cities, where column $NO_2$ values are much larger than the climatological ones.

Taking into account the corrections for Rayleigh scattering and for the absorptions by $O_3$ and $NO_2$, we have calculated the combined effect of all of them on the non-traceability of the 1-minute AOD values. The highest impact occurs in the 380 nm channel, in which 25 % of the AOD data outside the $U$95 limits (~2% of the total compared data) are due to significant differences in pressure, and in $O_3$ and $NO_2$ absorption.  The 1-minute AOD data outside the $U$95 limits by these corrections is negligible in the 870 nm channel.

~~We have to note that the excellent results of this 11-year comparison and the small differences found under the strict $U$95 criterion cannot be linked with the relatively low AODs that can be found at IZO. This is because absolute calibration errors contribute to the AOD calculation in an absolute way so larger than 1 % calibration errors for a given period of time can lead to even negative AOD calculations for IZO site.~~

- Impact of dust forward scattering in AOD retrieval uncertainty for different instrument FOVs

 Since GAW-PFR has almost double the FOV (~2.5°) compared to the AERONET-Cimel (1.3°), and direct solar irradiance measurements are biased by the amount of aureole radiation that is assumed to be direct solar radiation, it is reasonable to expect that GAW-PFR is more affected by the circumsolar irradiance than AERONET-Cimel radiometer when AOD is relatively high.  Modelling the dust forward scattering we have shown ~~We have explained part of the non-traceabilities found for relatively high AOD values, by analysing the relationship between the differences in circumsolar radiation measured by both instruments with the differences observed in AOD. We have observed a clear relationship between the Cimel-PFR AOD differences and the PFR-Cimel circumsolar radiation differences, with the slope of the fitted line greater for shorter wavelengths (380 nm). These results show~~ that a non-negligible percentage of

the non-traceable 1-minute AOD data for AOD > 0.1, ranging from ~0.3 % at 870 nm to ~1.9 % at 380 nm is, might be caused by the different FOV. This systematic error especially affects the shorter wavelengths. . Due to this effect, the GAW-PFR provides AOD values which are ~3% lower at 380 nm, and ~2% lower at 500 nm, compared with AERONET-Cimel. However, AOD underestimation This error couldan only have only some relevance be especially important in dusty regions if radiometers with relatively large FOV are used.

A comparison of the AE provided by GAW-PFR and AERONET-Cimel has been performed using in both cases AOD data obtained from the four nearby common channels with a total of 70716 data-pairs. This is a very strict AE calculation since it is necessary that AOD be accurately measured by the four channels simultaneously. AE differences > 0.2 increase exponentially under very pristine conditions (AOD ≤ 0.03 and AE ≥ 1), reaching AE differences of up to 1.6. However, for these conditions the atmospheric aerosol load is practically zero and so, its characterization with AE does not have any importance in practice. Under non-pristine conditions or those with a high mineral dust content (associated AOD > 0.03 and AE < 1), the AE differences remain <below 0.1.

[remaining 4,573 characters of this post omitted]

---

## Referee Report (RR1)

Review of Aerosol Optical Depth comparison between GAW-PFR and AERONET-1 Cimel radiometers from long-term (2005-2015) 1-minute synchronous measurements

General comments:

The structure and content of the main manuscript has improved significantly from the first submission. However, several new portions of the manuscript and supplement need correction and further clarification. After these revisions are implemented, then the manuscript should be publication ready.

Manuscript:

Abstract: "fairly good agreement at 380nm"; please provide a phrase or sentence discussing improving UV traceability as discussed in conclusions.

Page 2, Line 10: "https"?

Page 2, Lines 19-23: Is this a separate paragraph or does it belong to the previous one?

Page 3, Line 37: "is based on:", why does the sentence end with a colon? The following paragraphs are not enumerated.

Page 4, Line 25: Perhaps the Supplement should be described in one or two sentences.

Page 4, Line 27: Check formatting of degree symbols

Page 6, Lines 29-30: "and depends on the optical mass"; The optical air mass has a dependence for all channels and not only UV. Please use "optical air mass" and not "optical mass"

Page 7: Line 20: change "lineal" to "linear"

Page 8: Lines 29-31: This reference (Holben et al., 2006) is for the sky inversion QA (Dubovik and King 2000, Dubovik et al. 2000, 2002, 2006) and not for Version 2 AOD.

Page 9, Table 1:

"AOD uncertainty": Do these values vary by wavelength for PFR and Cimel?

"Temperature control": Is this applied to all wavelengths for both instruments and AERONET versions?

Page 11, Line 20: How do 5000 more points for Cimel compare to PFR at air mass greater than 5?

Page 11, Line 21: Only Supplement S.1 exists (not S.1.1)

Page 12, Line 10: 870nm AOD has better agreement due to less trace gas contribution or is it related to the Cimel staying the same since 2011 as shown in Figure 2? Please make a statement in regards to the Cimel/PFR agreement at 870nm.

Page 21, Lines 22-28: These lines need to be revised based on Supplement comments below.

Page 26, Line 9: Is the NO2 spatial resolution "0.25 x 0.2"?  Is this the same NO2 for both AERONET versions?

Page 26, Line 12: The optical depth of O3 and NO2 for each measurement should be available from AERONET.  Please state why they are not used.

Page 28, Table 9: "870" should be "870 nm"

Page 28, Lines 12-13: PFR is corrected or Cimel corrections are removed?

Page 33, Line 3: What is the total traceability percentage for 380nm and 500nm with the adjustment factor applied?  Does it bring 380nm into compliance with PFR standard?

Page 33, Lines 13-14:  These lines need to be revised based on Supplement comments below.

Page 33, Line 17:  "very high AOD retention" is based on Angstrom Exponent >1.2 (675-1020nm); it seems highly unlikely that measurements would be contaminated by altostratus and dust with such a high AE required for the retention.

Page 37: Lines 4-16:  The AE can be greater than 0.6 and not be "Pristine" conditions.  This conditional logic conflicts with supplement case 10.1 where locally generated smoke affects the measurements.  Also, it is not impossible that smoke can be transported from sub-Saharan Africa.  These AE conditions were applied in Cuevas et al. 2015 for comparing satellite with ground-based observations as a "first approach to discriminate when mineral dust is the main aerosol component."  However, this technique is not very appropriate in considering synchronous 1-minute data from ground based AERONET and PFR, which are measuring the same aerosols (unlike satellite, which uses a spatially distributed algorithm) at much higher temporal resolution than satellite.  Only using AE for characterizing the aerosol condition is very problematic.

Page 38, Lines 17-20: The cases presented do not support these conclusions; see supplement comments.

Page 39, Lines 13-21:  Discuss how the total traceability changed (not just the increment change) when applying the correction for FOV.

Page 39, Lines 22-29:  Is this a separate conclusion paragraph?

Supplement Comments:

Are captions needed for these figures in the supplement?  It would be useful to have them.

S1:  Why is it that the high anomalies stop for the AOD 870nm after the last change of Cimel in 2011?

S4: The y-axis title is missing in plot V3 d).  "Optical mass" should be "optical air mass"

S6. For example, why are data missing for V3 AOD 870nm (f) compared to plot (e) of V2 in the first part of 2007 and last part of 2009 but these periods are available in Figure 3 of the manuscript?

S10.1: This case appears to be mostly if not all dust and smoke.  The sky camera images look inconclusive for clouds. MODIS satellite visible imagery indicates much more aerosol than clouds. MODIS hot spots (in red) indicate fires near Guía de Isora and Teide National Park to the southwest of Izana Observatory on Tenerífe.  Also, MPL suggests aerosol rather than cloud due to weak backscatter (not

strong like for clouds).  Further, the Angstrom Exponent is very high for Izana (see plot from https://aeronet.gsfc.nasa.gov; last accessed 20 May 2019) indicating smoke and dust aerosols.

Aqua MODIS 18 July 2012 at 14:55 UTC

[Figure]

[Figure]

S10.2: MPL shows altostratus ending around 15:50UTC and Cimel is not pointing to the same portion of the sky as the lidar (zenith). Sky camera does not show stratus on the solar disk. The impact of Altostratus appears inconclusive in this case.

S11.1: In this case, the cloud appears to be present in the mpl but the backscatter does show an apparent increase between 2km and 4 km. The sun photometers can measure between clouds so it is possible it found a gap. The AE does tend to decrease increasing the chance of cloud contamination but this distinction becomes more difficult in a dust transport region.

[Figure]

S11.3: It is difficult to "see" cirrus in the sky camera visible image. Do you have a processed whole sky cloud image indicating clouds? AE from 14UTC to 18UTC looks like dust. MODIS visible images show dust plume transported from the Sahara desert.

Aqua MODIS at 14:30 UTC on 27 March 2010

[Figure]

[Figure]

[Figure]

Angstrom Parameter data from MAR 27 of 2010

S12, Case analysis 12 February 2015:

Note again the photometers and Lidar do not point in the same direction. Why is PFR affected and not AERONET, it is not clear why they have such different AOD? Did ice freeze on PFR external lens like whole sky imager?

S18. More explanation is needed here or in the manuscript text.

---

## Author Response (AR2)

**Anonymous Referee #3 (amt-2018-438-RC3)- Second Round**

**G.1. I appreciate the authors' responses and revision, especially regarding the FOV. Unfortunately, two of the three major data interpretation issues remain.**

**Authors:**

**These two major data interpretation issues, for the Referee's consideration, are addressed below.**

**G2. First, as the authors confirm, the duration of each measurement is less than 1s for PFR and a little over one minute for the Cimel triplets. The 59s differences make it more likely for Cimel to capture the atmospheric changes than for PFR. Thus, the wider diurnal ranges of Cimel AOD shown in Figure 3 and S7 may be attributable to the natural variability rather than calibration inaccuracies and other instrument- and algorithm-related causes. The author response G.3 does not refute this hypothesis. It points out that the natural variability over one minute exceeds 0.005 for 0.8% of the records, taking the year 2013 as an example. The authors seem to dismiss this fraction as negligible. But the data highlighted in Figure 3 and S7 – presented in the discussion of instrument and algorithm artifacts in Page 19 – may well overlap with that small and special subset of data. These evaluations are centered on the daily max AOD–the most extreme measurement of each day (see also the review S.26). That single measurement represents only a small fraction of the entire records. It is 3.3% if 30 data points are available on average for the days shown in Figure 3. Of these, approximately a third (i.e, quite possibly around 1% of the entire records) has the max-min difference exceeding 0.005 (shown in the original Figure 3). That is comparable to the 0.8% identified as the high natural variability cases.**

**This general comment includes different aspects. So it has been split into several comments of the Referee dealing with different issues.**

**G2.1. First, as the authors confirm, the duration of each measurement is less than 1s for PFR and a little over one minute for the Cimel triplets. The 59s differences make it more likely for Cimel to capture the atmospheric changes than for PFR.**

**Authors:**

**The actual timing difference of CIMEL and PFR is not 59 seconds because the 1-minute Cimel measurement time is really an average of t-30'', t, and t+30'' (triplets), and PFR AOD measurements are taken at t', being t-t'<30''seconds. So, statistically the impact of 1-minute aerosol natural variability (ANV) on AOD traceability is negligible.**

**G2.2. Thus, the wider diurnal ranges of Cimel AOD shown in Figure 3 and S7 may be attributable to the natural variability rather than calibration inaccuracies and other instrument- and algorithm-related causes. The author response G.3 does not refute this hypothesis. It points out that the natural variability over one minute exceeds 0.005 for 0.8% of the records, taking the year 2013 as an example. The authors seem to**

dismiss this fraction as negligible. But the data highlighted in Figure 3 and S7 – presented in the discussion of instrument and algorithm artifacts in Page 19 – may well overlap with that small and special subset of data.

The authors don't find evidence supporting the hypothesis of the Referee that the 0.8% of AOD differences >0.005 due to AVN can affect the results shown in Figure 3 (pristine conditions). The Referee should take into account that all the days plotted in Figure 3 with a diurnal AOD range higher than 25% of the mean daily AOD value were analysed, one by one, and their possible causes investigated (S7). A total of 51 cases for GAW-PFR and 81 cases for AERONET Cimel V3 were obtained and analysed in detail in order to identify calibration and cloud screening issues for both instruments.

In order to rule out any doubt, the authors have determined for several AOD intervals the 1-minute AOD data percentage from the Cimel triplets (year 2013) whose range of variation $AOD_{max}$-$AOD_{min}$ > 0.015. This value is half of the WMO traceability interval when m = 1 (maximum possible interval) (see Eq.2 of the manuscript). We must emphasize that the ANV may include any instrumental noise. The results shown in the following two tables demonstrate that, even using the strict WMO criterion, the AVN is responsible for only 106 (64) 1-minute Cimel AOD values outside the WMO limits for 380 nm (500 nm) for the total 0-1 AOD range.

380nm: 114 points outside WMO limits

| AOD range | # cases outside WMO limit | Total # cases | % in AOD range | % in total # cases |
|---|---|---|---|---|
| ≤ 0.03 | 13 | 11800 | 0.11 | $6.5 \times 10^{-4}$ |
| (0.03-0.05] | 14 | 3712 | 0.38 | $7.0 \times 10^{-4}$ |
| (0.05-0.1] | 18 | 1932 | 0.93 | $9.0 \times 10^{-4}$ |
| (0.1-1.0] | 61 | 2637 | 2.31 | 0.30 |
| Total [0.0-1.0] | 106 | 20081 | 0.52 | 0.52 |

500nm: 64 points outside WMO limits

| AOD range | # cases outside WMO limit | Total # cases | % in AOD range | % in total # cases |
|---|---|---|---|---|
| ≤ 0.03 | 2 | 13629 | 0.01 | $9.9 \times 10^{-5}$ |
| (0.03-0.05] | 11 | 2401 | 0.46 | $5.4 \times 10^{-4}$ |
| (0.05-0.1] | 9 | 1600 | 0.56 | $4.5 \times 10^{-4}$ |
| (0.1-1.0] | 42 | 2484 | 1.69 | 0.20 |
| Total [0.0-1.0] | 64 | 20114 | 0.32 | 0.32 |

The 1-minute AVN is responsible for only 0.11% (0.01%) of 1-minute Cimel AOD values outside the WMO limits in the [0-0.03] AOD range (pristine conditions) for 380 nm (500 nm). This is the AOD range to which Figure 3, S6, and S7 refer. Therefore, we confirm that the impact of AVN is negligible in AOD non-traceability in pristine conditions. For higher AOD ranges, the AVN impact is slightly higher, though very small in any case. These cases correspond, as expected, to air mass changes such as transitions from pristine to dusty conditions, and vice versa, or to sharp onset and disappearance of very sporadic biomass burning plumes. Moreover, the 1-minute aerosol variability > 0.02 (V2) or > 0.01 (V3) are filtered by the triplets-based AERONET cloud screening.

We have included the previous Tables in Supplement (S3.2), and the following paragraph at the end of Section 5.3.1 in Page 17:

*"We have also determined the percentage of 1-minute AOD data from the Cimel triplets (year 2013) whose range of variation $AOD_{max}$-$AOD_{min}$ > 0.015, for several AOD intervals. Note that this value is half of the WMO traceability interval when m = 1 (minimum possible interval) (see Eq.2). The results shown in S3.2 indicate that the 1-minute AOD variability is responsible for only 0.11% (0.01%) of 1-minute Cimel AOD values outside the WMO limits in the [0-0.03] AOD range (pristine conditions) for 380 nm (500 nm). The AOD variability maximizes in the 0.1-1 AOD range causing 2.31% and 1.69% of the AOD data outside WMO limits for 380 and 500 nm, respectively. This last scenario corresponds, as expected, to changes of air masses, such as transitions from pristine to dusty conditions, and vice versa, or to sharp onset and disappearance of very sporadic biomass burning plumes. In any case, the AOD data with 1-minute variability exceeding 0.02 (V2) or 0.01 (V3) are filtered by AERONET (see Section 4.2) and therefore are not included in the GAW-PFR and AERONET-Cimel comparison."*

**G2.3. These evaluations are centered on the daily max AOD–the most extreme measurement of each day (see also the review S.26). That single measurement represents only a small fraction of the entire records.**

Calibration issues are not solely defined by the difference between daily max and min AOD but also by the convex or concave curvature symmetrical at noon that provides a hint of calibration inaccuracies as it is very well known (see S8). The AOD diurnal evolutions displayed in S8 are quite different for the PFR and the Cimel, sometimes antisymmetric, typical of calibration issues of one or both instruments. These do not correspond to natural diurnal AOD variations.

In general, the comparison of the two instruments have provided very good results as, out of ~70K data points, only a very small percent are outside the WMO limits. These limits are quite strict, especially comparing long-term periods including different instruments. For the remaining few data points outside the limits, we assessed the sources of such deviations. It is not easy to isolate all sources as a number of the deviations are affected by a combination of different sources.

**G2.4. It is 3.3% if 30 data points are available on average for the days shown in Figure 3. Of these, approximately a third (i.e, quite possibly around 1% of the entire records) has the max-min difference exceeding 0.005 (shown in the original Figure 3). That is comparable to the 0.8% identified as the high natural variability cases.**

Please see reply to G2.2.

**G3. Second, Figure 7 of the original discussion paper shows that PFR AOD is significantly higher than Cimel in many cases with PFR AOD just above 0.03. It is misleading to be quiet about these cases while discussing the high Cimel cases in relation to the artifacts. The author response G.4 is not adequate, as it does not present the PFR-Cimel differences. Just as lamentably, the new Figure 7 hides the negative outliers by starting its abscissa from 0.1. It does not adequately respond to the review S.32 either.**

**Authors:**

**This general comment includes different aspects. This paragraph has been split into several sentences comments of the Referee dealing with different issues.**

**G3.1. Second, Figure 7 of the original discussion paper shows that PFR AOD is significantly higher than Cimel in many cases with PFR AOD just above 0.03. It is misleading to be quiet about these cases while discussing the high Cimel cases in relation to the artifacts. The author response G.4 is not adequate, as it does not present the PFR-Cimel differences.**

**Authors:**

**Fig. 7 of the first version of the paper was replaced by the current Fig. 7, in which the AOD in the abscissa axis starts at 0.1 because the corresponding section addresses the subject that is described in the name of the section:** *"GAW PFR and AERONET-Cimel comparison under high AOD conditions: the impact of dust forward scattering for different FOVs".* **The purpose of this section is exactly introduced in order to justify the differences for AOD>0.1, PFR measuring lower than Cimel.**

**The G4 Referee's comment in his/her report said:** *"G.4. Second, the paper finds "a bias with positive large outliers (higher Cimel AOD)" (page 15, line 5; Figure 2). This results from the intentional exclusion of high PFR values, an attempt to assess "pristine conditions (AOD500nm <= 0.03)" (page 13, line 26, 28). It is misleading to use this assessment to suggest contaminations on Cimel (page 15, line 6, 7). It is only fair to explicitly state that many negative outliers (higher PFR AOD) exist for PFR AOD just above 0.03 (shown in Figure 7)".*

**Addressing the Referee's comment, and G4 comment of the first Referee Report, we have calculated the number and percentage of AERONET V3 AOD diurnal range variation (maximum value minus minimum value of AOD in one day) data (dAOD) at 380nm and 500nm above +0.03 and below -0.03 for AOD <0.1 in the period 2005-2015. The results are shown in the following Table:**

| Non-Traceability AERONET-V3 | 380nm | | 500nm | |
|---|---|---|---|---|
| | # data | % data | # data | % data |
| AOD < 0.1 & dAOD > 0.03 | 300 | 0.40% | 214 | 0.28% |
| AOD < 0.1 & dAOD < -0.03 | 174 | 0.23% | 108 | 0.14% |

**In these cases we cannot speak of "many cases" (≤300 out of more than 70,000; ≤0.40%) finding almost double the data in which Cimel measures more than PFR (dAOD > 0.03), which is consistent with the results described in Page 19 Lines 25-27. Of course, we have also detected high dAOD cases in the PFR that have also been reported. Case analyses shown in S7 and S8 describe calibration inaccuracies, cloud screening algorithm failures, mixture of the two previous causes, poor sun pointing, etc., for both PFR and Cimel.**

**The Referee should especially consider Table in S7, which shows the percentage of cases with AOD$_{380nm}$ outliers of both GAW-PFR and AERONET Cimel (V3) under pristine conditions (Cimel AOD$_{500nm}$≤0.03). As stated, there was a manual inspection of these cases.**

| | PFR 51 cases | Cimel 81 cases |
|---|---|---|
| Calibration inaccuracies | 7.8% | 44.4% |
| Cloud screening failures | 29.4% | 21.0% |

| | | |
|---|---|---|
| **Calibration+ cloud screening errors** | **9.8%** | **11.1%** |
| **Sun misalignments** | **17.6%** | **0%** |
| **Unknown** | **35.3%** | **33.5%** |

**We would like to emphasize that the rare finding of small calibration inaccuracies in a high mountain site with pristine skies and stable atmosphere does not detract from the quality of any instrument as they often measure near or below the detection limit. Showing these deficits that affect a small percentage of data is fair, demonstrating the limitations of the technique.**

**We have modified and extended the following paragraph at the end of Section 5.3.3. (Page 20) as follows:**

*"The fictitious diurnal AOD cycle is mainly visible in the UV channels as shown in the examples reported in Supplement S9, where the convex or concave diurnal AOD curvature symmetrical at noon provides a hint of calibration inaccuracies. Note that the fictitious diurnal AOD can be more easily identified under very low AOD conditions. We should emphasize that the rare finding of small calibration inaccuracies in a high mountain site with pristine skies and stable atmosphere does not detract from the quality of any instrument as they often measure near or below the detection limit. Simply, these small inaccuracies are the result of limitations in the photometric measurement technique"*

**G3.2. Just as lamentably, the new Figure 7 hides the negative outliers by starting its abscissa from 0.1.**

**Since the instrument to instrument comparison and statistics have been extensively presented in the previous sections, the change of X-axis interval in Figure 7 was solely to focus on the FOV impact on AOD retrieval as the reviewers correctly pointed out. Figure 7 clearly shows that a radiometer with a higher FOV underestimates AOD, to a greater extent, than does a radiometer with a lower FOV because the higher impact of dust forward scattering. In our case, AOD derived from PFR is lower than that obtained with Cimel for AOD relatively high.**

**If we had used the negative AOD differences (CIMEL AOD< PFR AOD), which clearly are not caused by FOV issues, we would have obtained erroneous results from the linear regression. Therefore, in Figure 7 only the points with positive AOD differences have been included in order to isolate the FOV effect. All other differences (including the few negative values for low AODs) are included in the statistical analysis presented in the paper.**

**G3.3. It does not adequately respond to the review S.32 either.**

**The Referee's comment S32 in his/her first report said:**
"S.32. Figure 7. If this figure is to remain on the paper, note in the caption that an identical data set, PFR AOD, appears in both x and y axes, a practice generally discouraged. Also state that the numbers in the legend are rounded to the significant digits. This is to forestall questions as to why the black lines do not reach exactly (x,y)=(0,0)."

**In order to show the impact of increasing AOD in the PFR-Cimel AOD difference, we had to include AOD (from either one of the two instruments) in X-axis. Using Cimel AOD or PFR-AOD in the X-axis is irrelevant because the linear fit constants change slightly. The linear fits are just to show semi-quantitatively the FOV effect. A thorough analysis of this effect is now performed with the new radiative model sensitivity analysis.**

**In the new Figure 7, the slope and the intercept in the legend appear with four decimals.**

**AMT-2018-438**
The structure and content of the main manuscript has improved significantly from the first submission. However, several new portions of the manuscript and supplement need correction and further clarification. After these revisions are implemented, then the manuscript should be publication ready.

**Authors:**

**The authors appreciate the positive assessment of the Referee. The minor comments are addressed below.**

**Manuscript:**

**M1. Abstract: "fairly good agreement at 380nm"; please provide a phrase or sentence discussing improving UV traceability as discussed in conclusions.**

**Authors:**
**The following sentence has been included at the end of the abstract:**

*"…, although, AOD should be improved in the UV range."*

**M2 Page 2, Line 10: "https"?**

**Authors:**
**Done.**

**M3 Page 2, Lines 19-23: Is this a separate paragraph or does it belong to the previous one?**

**Authors:**
**It belongs to the previous paragraph. Fixed.**

**M4 Page 3, Line 37: "is based on:", why does the sentence end with a colon? The following paragraphs are not enumerated.**

**Authors:**
**It was an error. Fixed.**

**M5 Page 4, Line 25: Perhaps the Supplement should be described in one or two sentences.**

**Authors:**
**We agree. We have added the following sentence:**

*"The Supplement contains case analyses of inaccurate calibration and cloud contamination, results of the comparison between PFR and Cimel with AERONET V3, and additional information of the very short natural AOD variability, and the simulations performed with the Monte Carlo model to evaluate the impact of dust forward scattering radiation on AOD determination."*

**M6 Page 4, Line 27: Check formatting of degree symbols**

**Authors:**

**Done.**

**M7 Page 6, Lines 29-30: "and depends on the optical mass"; The optical air mass has a dependence for all channels and not only UV. Please use "optical air mass" and not "optical mass"**

**Authors:**

**Done.**

**M8 Page 7: Line 20: change "lineal" to "linear"**

**Authors:**

**Done.**

**M9 Page 8: Lines 29-31: This reference (Holben et al., 2006) is for the sky inversion QA (Dubovik and King 2000, Dubovik et al. 2000, 2002, 2006) and not for Version 2 AOD.**

**Authors:**
**Corrected.**

**M10 Page 9, Table 1: "AOD uncertainty": Do these values vary by wavelength for PFR and Cimel? "Temperature control": Is this applied to all wavelengths for both instruments and AERONET versions?**

**Authors:**
**These data have been corrected/completed in Table 1 as follows:**

| | | |
|---|---|---|
| *AOD uncertainty* | *± 0.01* | *0.002-0.009 spectrally dependent with the higher errors in the UV (Reference instruments) (Eck et al., 1999)* |
| *Temperature control/correction* | *Temperature controlled 20°C ± 0.5°C* | *Temperature correction is applied to 1020 nm in V2. Corrections from filter-specific temperature characterization in V3 for VIS and NIR spectral bands (Giles et al., 2019)* |

**M11 Page 11, Line 20: How do 5000 more points for Cimel compare to PFR at air mass greater than 5?**

**Authors:**
**This is a good point.**

**First of all, we have corrected in Page 11 Line 20, the additional number of AOD available in V3 for optical air mass >5. In fact, it is more than 9,000 data instead of 5,000 data.**

**We have included the following AOD comparison between GAW-PFR and AERONET-Cimel V3 for optical air mass > 5.0 as Supplement S2.2:**

"S2.2 Percentage of AERONET-Cimel 1-minute AOD data (V3) meeting the WMO criteria for optical air mass > 5.0 for the period 2005-2015. The number of data pairs are shown in brackets.

| % of data within WMO limits | 380 nm | 440 nm | 500 nm | 870 nm |
|---|---|---|---|---|
| m > 5.0 | 90.9 (9328) | 93.4 (9474) | 94.7 (9412) | 96.7 (9475) |

**We have also included the following paragraph in the manuscript at the end of Section 5.2 (page 16, after Line 28). We believe that it is relevant and novel information.**

*"In relation to the comparison between GAW-PFR and AERONET-Cimel V3, we have calculated the percentage of AERONET-Cimel 1-minute AOD data (V3) meeting the WMO criteria for optical air mass> 5.0 for the period 2005-2015 (Supplement S2.1). The results are somewhat poorer than for optical air mass <5.0, since the solar elevation is very low. Only for the 870 nm channel 95% of the data meet with the WMO criteria, although the percentages of data in the 440 nm and 500 nm channels are close to this value. This would be the main reason to find slightly poorer traceability results with all V3 data compared to those found with V2 data limited to optical air mass < 5.0"*

**M12 Page 11, Line 21: Only Supplement S.1 exists (not S.1.1)**

**Authors:**
**Fixed.**

**M13 Page 12, Line 10: 870nm AOD has better agreement due to less trace gas contribution or is it related to the Cimel staying the same since 2011 as shown in Figure 2? Please make a statement in regards to the Cimel/PFR agreement at 870nm.**

**Authors:**
**It is because the 870nm channel is less affected by trace gases contribution. We have included the following sentence:**

"…because this channel is less affected by trace gases absorption".

**The fact that since 2011 only one Cimel is issued benefits all channels.**

**M14 Page 21, Lines 22-28: These lines need to be revised based on Supplement comments below.**

**Authors:**

The authors think the text of lines 22-28 is valid. Only a case analysis has been removed because potential contamination by fire smoke. See M30.

**M15 Page 26, Line 9: Is the NO2 spatial resolution "0.25 x 0.2"? Is this the same NO2 for both AERONET versions?**

Authors:

This is an error. It should say NO2 spatial resolution "0.25 x 0.25".

AERONET-Cimel V3 uses a geographic and temporally dependent multiyear monthly climatology from the Ozone Monitoring Instrument (OMI) NO2 concentration (Giles et al., 2019).

These corrections have been incorporated into the manuscript.

**M16 Page 26, Line 12: The optical depth of O3 and NO2 for each measurement should be available from AERONET. Please state why they are not used.**

Authors:

The original Cimel-AERONET AOD values have been used as such from the corresponding versions V2 and V3, and no modification have been made to them.

What we did in section 5.3.5 of the manuscript was a theoretical study to examine the change that would occur in PFR AOD if the PFR had used the Cimel-AERONET O3, NO2 and pressure data.

**M17 Page 28, Table 9: "870" should be "870 nm"**

Authors:
Done

**M18 Page 28, Lines 12-13: PFR is corrected or Cimel corrections are removed?**

Authors:
This correction is used to assess the impact of the O3 and NO2 absorptions on the traceability of Cimel AOD data to PFR AOD, but data from either instrument is not corrected.

**M19 Page 33, Line 3: What is the total traceability percentage for 380nm and 500nm with the adjustment factor applied? Does it bring 380nm into compliance with PFR standard?**

Authors:
The correction raised by the Referee is not applicable. The impact of dust forward scattering for different FOVs has been modelled with the Monte Carlo model for AOD> 0.1 assuming that the particles have an effective radius of 1.5um (based on observations), a scenario that does not apply to air with little or no dust , where fine particles dominate.

**M20 Page 33, Lines 13-14: These lines need to be revised based on Supplement comments below.**

Authors:

Done

**M21 Page 33, Line 17:** "very high AOD retention" is based on Angstrom Exponent >1.2 (675-1020nm); it seems highly unlikely that measurements would be contaminated by altostratus and dust with such a high AE required for the retention.

**Authors:**

**Thank you. This speculation has been removed from the manuscript**

**M22 Page 37: Lines 4-16:** The AE can be greater than 0.6 and not be "Pristine" conditions. This conditional logic conflicts with supplement case 10.1 where locally generated smoke affects the measurements. Also, it is not impossible that smoke can be transported from sub-Saharan Africa. These AE conditions were applied in Cuevas et al. 2015 for comparing satellite with ground-based observations as a "first approach to discriminate when mineral dust is the main aerosol component." However, this technique is not very appropriate in considering synchronous 1-minute data from ground based AERONET and PFR, which are measuring the same aerosols (unlike satellite, which uses a spatially distributed algorithm) at much higher temporal resolution than satellite. Only using AE for characterizing the aerosol condition is very problematic.

**Authors:**

**We totally agree with the Referee. The objective of this study is not to perform a characterization of the aerosol measured at the Izaña Observatory, but to perform a comparison of 1-minute values of the AE.**

**We have selected those 4 categories, which would be close enough to what would be used in a categorization of the aerosols but adding AOD in order to assess the degree of agreement in the long-term AE comparison. Introducing the AOD of one of the two instruments would have introduced a bias in the comparison of AE. That is why we have not used the necessary AOD for a scientific aerosol characterization. In fact, the chosen AE categories are irrelevant and only serve to examine the degree of agreement of AE under four different aerosols scenarios.**

**In order to clarify this issue, we have replaced the last sentence of this Section by the following sentence:**

*"Note that the choice of this category is not relevant since this is only used to examine the long-term agreement in AE between GAW-PFR and AERONET-Cimel in different atmospheric conditions."*

**M23 Page 38, Lines 17-20:** The cases presented do not support these conclusions; see supplement comments.

**Authors:**

**See the responses to the Referee's comments on the Supplement. Only the two case analyses dealing with the impact of stratocumulus over the Saharan Air Layer have been discarded. The corresponding references have been removed in the manuscript. These changes do not alter the rest of the conclusions.**

**M24 Page 39, Lines 13-21:** Discuss how the total traceability changed (not just the increment change) when applying the correction for FOV.

**Authors:**

**See reply to M19.**

**M25 Page 39, Lines 22-29: Is this a separate conclusion paragraph?**

**Authors:**

**Yes. Corrected.**

**Supplement Comments:**

**M26 Are captions needed for these figures in the supplement? It would be useful to have them.**

**Authors:**

**Done.**

**M27 S1: Why is it that the high anomalies stop for the AOD 870nm after the last change of Cimel in 2011?**

**Authors:**

**This is explained in Page 12 (Lines 18 and 19) and page 13 (lines 1 and 2):** *"In addition, since 2010, Cimel #244 has been in continuous operation for most of the time at the Izaña Observatory, greatly simplifying calibration procedures and the corresponding data evaluation, and minimizing errors of calibration uncertainties introduced by the use of a high number of radiometers in the intercomparison".*
**This affects to all channels.**

**M28 S4: The y-axis title is missing in plot V3 d). "Optical mass" should be "optical air mass"**

**Authors:**
**Corrected.**

**M29 S6. For example, why are data missing for V3 AOD 870nm (f) compared to plot (e) of V2 in the first part of 2007 and last part of 2009 but these periods are available in Figure 3 of the manuscript?**

**Authors:**

**In Figure 3 of the manuscript the AOD diurnal range variation corresponding to AOD outliers at 380nm have been plotted, while in S6 the information corresponds to 440nm, 500nm, 870nm. Data filtering for V2 and V3 is not the same, and within the same AERONET version (V2 or V3) data quality procedures might filter data from one channel and not from others. This explains these differences.**

**M30 S10.1: This case appears to be mostly if not all dust and smoke. The sky camera images look inconclusive for clouds. MODIS satellite visible imagery indicates much more aerosol than clouds. MODIS hot spots (in red) indicate fires near Guía de Isora and Teide National Park to the southwest of Izana Observatory on Tenerífe. Also, MPL suggests aerosol rather than cloud due to weak backscatter (not strong like for clouds). Further, the Angstrom Exponent is very high for Izana (see plot from https://aeronet.gsfc.nasa.gov; last accessed 20 May 2019) indicating smoke and dust aerosols. Aqua MODIS 18 July 2012 at 14:55 UTC**

**Authors:**

**We agree to remove this case analysis because of potential contamination by wildfire smoke at Guía de Isora, at least for part of the day.**

**M31 S10.2: MPL shows altostratus ending around 15:50UTC and Cimel is not pointing to the same portion of the sky as the LIDAR (zenith). Sky camera does not show stratus on the solar disk. The impact of Altostratus appears inconclusive in this case.**

**Authors:**
**We did not find any case analysis, with the available information, that would unambiguously confirm the AOD contamination by Altostratus located above the Saharan Air layer. So we removed S10 and all references in the manuscript to the possible impact of Altostratus on AOD retrieval. In the future, we will conduct specific experiments with additional tools to examine the possible impact of Altostratus on AOD retrieval.**

**M32 S11.1: In this case, the cloud appears to be present in the mpl but the backscatter does show an apparent increase between 2km and 4 km. The sun photometers can measure between clouds so it is possible it found a gap. The AE does tend to decrease increasing the chance of cloud contamination but this distinction becomes more difficult in a dust transport region.**

**Authors:**
**The September 23, 2015 case analysis shows high AOD caused by cloud contamination. In order to confirm this, we show below, Direct Normal Irradiance (DNI) and Global Horizontal Irradiance (GHI) records from the Baseline Surface radiation Network (BSRN) program that clearly indicates a high attenuation in DNI from just before 14UTC and during the rest of the day. We have added this result to S10.1. Note that former S11.1 is now S10.1**

[Figure]

**M33 S11.3: It is difficult to "see" cirrus in the sky camera visible image. Do you have a processed whole sky cloud image indicating clouds? AE from 14UTC to 18UTC looks like dust. MODIS visible images show dust plume transported from the Sahara desert. Aqua MODIS at 14:30 UTC on 27 March 2010**

**Authors:**

**Yes. We have software to identify clouds from whole-sky imagery.**

**Modis visible image shows a NE-SW axis band of dust south of the Canary Islands but this does not affect Tenerife Island as can be seen well in the image itself. In addition, the Saharan Air Layer had a small thickness, typical of the season, remaining below the height of the Izaña Observatory.**

[Figure]

**However, the clearest proof of the cirrus impact is obtained from the Direct Normal Irradiance (DNI) record of our Baseline Surface Radiation Network (BSRN) Program (image below). The noisy DNI started around 11:30UTC ending at around 19UTC (see DNI standard deviation). This noisy DNI record is typical of attenuation by cirrus. Daylight cloud top pressure from Aqua/Modis confirm the presence of cirrus near Tenerife.**

**We have added the DNI plot and part of the information above to the new S10.4.**

[Figure]

*DNI record at Izaña Observatory on 27 March 2010.*

**Note that former S11.3 is now S10.3.**

**M34 S12, Case analysis 12 February 2015: Note again the photometers and Lidar do not point in the same direction. Why is PFR affected and not AERONET, it is not clear why they have such different AOD? Did ice freeze on PFR external lens like whole sky imager?**

**Authors:**

**This is really an interesting case analysis. We have included the following information in the corresponding Supplement section (former S12, now S.11):**

*Another cloud scenario that can affect AOD traceability is the presence of low clouds (stratocumulus) that sometimes exceed the observatory height level because the temperature inversion is around 2400 m height.*

*The selected case analysis is really interesting. Moreover, it is representative of a relatively frequent situation in winter, when the temperature inversion is very close to the altitude of the Izaña Observatory.*

*The Modis image of that day (S11.1a) shows a large part of the island of Tenerife covered with stratocumulus except in its central part corresponding to the summits of the island (2400 m a.s.l. plateau) in whose NE limit the Izaña Observatory is located. This is confirmed by the range corrected backscattering signal vertical cross section of the Micropulse lidar (MPL) (S11.1b) indicating a quasi-permanent stratocumulus layer above 2000 m a.s.l. throughout the day. In these cases, the appearance of intermittent fog banks in the Observatory or on the horizon (S11.1c), in its vicinity, is very common.*

*The AOD outliers measured by the PFR around 08:00 UTC (S11.1d) are due precisely to these intermittent fog blanks on the horizon and/or above Izaña Observatory. In S11.2 we can see a sequence of all-sky images from 07:30 UTC. Although the all-sky camera records some frozen ice on its dome, it does not appear in the sunlit part. The PFR external lens were free of frozen ice all the*

*time. From the all-sky camera imagery, the presence of fog from around 08:00 UTC to 09:00 UTC is observed. This is confirmed with DNI and other radiation-components measurements (S11.3). Early morning fog veils caused erroneous AOD values from PFR but not from Cimel. The explanation is in the measurement mode. As the sky conditions changes are very fast under intermittent fog blanks, the 1-second measurements (at 1-minute intervals) of the PFR may not capture this AOD variation, while the triplets of the Cimel (3 consecutive 1-second measurements) might do so, correctly functioning the cloud screening in this case.*

[Figure]

*S11.1. February 12, a) 2015: Modis visible image; b) the range corrected backscattering signal vertical cross section of the Micropulse lidar (MPL); c) East facing webcam picture around 09:00UTC; d) PFR and Cimel AOD.*

[Figure]

All-sky images at Izaña Observatory on February 12, 2015

*S11.2. All-sky images sequence on February 12, 2015.*

[Figure]

*S11.3. Global Horizontal (GHI), Direct Normal Irradiance (DNI), Difuse Horizontal Irradiance (DHI) and UVB radiation from the Surface Baseline Radiation Network (BSRN) program at Izaña Observatory on February 12, 2015.*

**M35 S18. More explanation is needed here or in the manuscript text**

Authors:
**Note that former S18 is now S17.**

**We have included the following text in S17:**

*"This basic statistic indicates the degree of agreement between GAW-PFR and AERONET-Cimel in the aerosol "characterization" in the long term using only AE. Therefore, we did not include AOD as it would have been strictly necessary to carry out a proper characterization of the aerosol types present in a specific site. The four chosen categories are very close to the real ones at Izaña but without including AOD. What is relevant here in is not the chosen categories, but the degree of agreement that both radiometers have to provide the same aerosol category according to the AE. This requires a very high simultaneous agreement in AOD in the four channels."*

[revised manuscript text omitted]
 -4.9% (3.3%). The second reason is a possible cloud contamination in AOD retrieval when altostratus are present above the SAL, as discussed in Section 5.3.4.

A similar analysis has been carried out for AERONET V3 (see Supplement S167), where we observe that the corrections obtained are not as good as those obtained for V2. This may be due to the very high AOD data retention in V3 which could include more cases in which altostratus clouds and dust are present. The effect of FOV on AOD retrieval should be taken into account for those radiometers with a relatively high FOV (>3°) measuring in regions with relatively high AOD (> 0.2) for most of the year, as is the case in many sites of Northern Africa, the Middle East and East Asia (Basart et al., 2009; Cuevas et al., 2015; Eck et al., 1999; Kim et al., 2007). This effect leads to AOD underestimation, and the variable number of high AOD episodes in each season of the year might affect the AOD long-term trends. AOD measurements under these conditions would be especially affected for optical air mass < 3.

[Figure]

Figure 9. The same as Figure 7 after "correcting" the PFR AOD data by adding + 3.3% at 380nm and
+ 2.2.% at 500 nm to the 1-minute PFR AOD data > 0.1.

**5.5. Angström exponent comparison**

We have performed a comparison of the AE provided by GAW-PFR and AERONET-Cimel using in
both cases the AOD data obtained from the four common channels (380 nm, 440 nm, 500 nm and 870 nm)
with a total of 70716 data-pairs. The PFR-AOD values have been ordered from lowest to highest by grouping them in intervals of 500 values for which the averages (and corresponding standard deviations)

of the PFR-Cimel AE differences have been calculated (Figure 10a). In a similar way we proceeded with the PFR-AE values (Figure 10b).

[Figure]

*Figure 10. (a) PFR-Cimel AE mean absolute differences (and corresponding standard deviations) vs*

*PFR mean AOD$_{500nm}$ in 500 data intervals (b) and vs PFR mean AE in 500 data intervals. AE has been*

*computed for both PFR and Cimel using the four common channels (380, 440, 500 and 870nm).*

AE differences > 0.2 increase exponentially for AOD < 0.02, reaching AE differences of up to 1.6

under pristine conditions (Figure 10a). For very low AOD the provided instruments uncertainty is the source of the sharp increase in AE, and at the same time AE becomes very sensitive to slight AOD changes.

However, for AOD < 0.02 the atmospheric aerosol load is practically zero and so, its characterization with

AE have in practice relatively minor importance.

In addition, the AE differences remain < 0.1 when AE$_{PFR}$ values are < 1 (Figure 10b), which shows that these differences are small in most of the possible atmospheric scenarios. For 1 < AE$_{PFR}$ < 1.2 the AE

differences increase slightly to values < 0.2, and for AE$_{PFR}$ > 1.2 (very fine particles or pristine conditions)

the AE differences increase sharply to reach values of ~ 1.2. In our case, the non-pristine conditions, or those with a high content of mineral dust, have associated AOD > 0.03 and AE < 1, where the AE

differences remain < 0.1. In case of pristine conditions AOD ≤ 0.03 and AE ≥ 1 the AE differences can reach a maximum of 1.6. Wagner and Silva (2008) estimated the usual maximum AE error by error propagation using a pair of spectral channels in which AOD is measured. Their results show that for clean optical conditions ($AOD_{440nm}$= 0.06) the maximum AE error is 1.17, and for hazy conditions ($AOD_{440nm}$=0.17) the error is 0.17, assuming an underlying AE of 1.5. These values decrease to 0.73 and 0.11, respectively, if AE=0. The AE differences found between GAW-PFR and AERONET-Cimel lie within the estimated errors reported by Wagner and Silva (2008).

*Table 11. Uncertainty in AE determination for three typical atmospheric situations.*

|  | Uncertainty in AE |
|---|---|
| Normal pristine conditions

$AOD_{500nm}$= 0.03 and AE = 1.4 | ≥ 1 |
| Hazy conditions
$AOD_{500nm}$= 0.14 and AE = 1.15 | ≥ 0.2 |
| Strong dust intrusion
$AOD_{500nm}$= 0.3 and AE = 0.3 | ~ 0 |

[revised manuscript text omitted]

Bokoye, A. I., Royer, A., O'Neill, N. T., Cliche, P., Fedosejevs, G., Teillet, P. M., and McArthur, L. J. B.: Characterization of atmospheric aerosols across Canada from a ground-based sunphotometer network:

AEROCAN, Atmosphere-Ocean, 39, 429–456, https://doi.org/10.1080/07055900.2001.9649687, 2001.

[revised manuscript text omitted]

Eissa, Y., Blanc, P., Wald, L., and Ghedira, H.: Can AERONET data be used to accurately model the monochromatic beam and circumsolar irradiances under cloud-free conditions in desert environment?, Atmospheric Measurement Techniques, 8, 5099–5112, https://doi.org/DOI = 10.5194/amt-8-5099-2015, 2015.

Eissa, Y., Blanc, P., Ghedira, H., Oumbe, A., and Wald, L.: A fast and simple model to estimate the contribution of the circumsolar irradiance to measured broadband beam irradiance under cloud-free conditions in desert environment, Solar Energy, 163, 497–509, https://doi.org/https://doi.org/10.1016/j.solener.2018.02.015, 2018.

Eskes, H. J. and Boersma, K. F.: Averaging kernels for DOAS total-column satellite retrievals, Atmospheric Chemistry and Physics, 3, 1285–1291, https://doi.org/10.5194/acp-3-1285-2003, https://www.atmos-chem-phys.net/3/1285/2003/, 2003.

Freidenreich, S. M., and Ramaswamy, V.: A new multiple-band solar radiative parameterization for general circulation models, J. Geophys. Res., 104, 31389-31409, doi: 10.1029/1999JD900456, 1999.

García, R. D., Cuevas, E., García, O. E., Cachorro, V. E., Pallé, P., Bustos, J. J., Romero-Campos, P. M., and de Frutos, A. M.: Reconstruction of global solar radiation time series from 1933 to 2013 at the Izaña Atmospheric Observatory, Atmospheric Measurement Techniques, 7, 3139–3150, https://doi.org/10.5194/amt-7-3139-2014, https://www.atmos-meas-tech.net/7/3139/2014/, 2014.

García, R. D., Barreto, A., Cuevas, E., Gröbner, J., García, O. E., Gómez-Peláez, A., Romero-Campos, P. M., Redondas, A., Cachorro, V. E., and Ramos, R.: Comparison of observed and modeled cloud-free longwave downward radiation (2010–2016) at the high mountain BSRN Izaña station, Geoscientific Model Development, 11, 2139–2152, https://doi.org/10.5194/gmd-11-2139-2018, https://www.geosci-model-dev.net/11/2139/2018/, 2018.

García, R. D., Cuevas, E., Ramos, R., Cachorro, V. E., Redondas, A., and Moreno-Ruiz, J. A.: Description of the Baseline Surface Radiation Network (BSRN) station at the Izaña Observatory (2009–2017): measurements and quality control/assurance procedures, Geosci. Instrum. Method. Data Syst., 8, 77-96, https://doi.org/10.5194/gi-8-77-2019, 2019.

[revised manuscript text omitted]

WMO: Abridged final report with resolutions and recommendations, GAW Report WMO TD No. 1019, WMO-CIMO Fourteenth session Geneva 7–14 December 2006, 2007. WMO: WMO/GAW Aerosol Measurement Procedures, Guidelines and Recommendations, 2nd Edition, WMO No 1177, GAW Report No. 227, 93 pp, https://library.wmo.int/doc_num.php?explnum_id=3073, 2016.

Young, A. T.: Revised depolarization corrections for atmospheric extinction, Appl. Opt., 19, 3427–3428,https://doi.org/10.1364/AO.19.003427, 1980.°